# ◉ PHYSBENCH: BENCHMARKING AND ENHANCING VISION-LANGUAGE MODELS FOR PHYSICAL WORLD UNDERSTANDING

**Wei Chow**[*1], **Jiageng Mao**[*1], **Boyi Li**[2], **Daniel Seita**[1], **Vitor Guizilini**[3], **Yue Wang**[1]
[1]University of Southern California, [2]UC Berkeley, [3]Toyota Research Institute

## ABSTRACT

Understanding the physical world is a fundamental challenge in embodied AI, critical for enabling agents to perform complex tasks and operate safely in real-world environments. While Vision-Language Models (VLMs) have shown great promise in reasoning and task planning for embodied agents, their ability to comprehend physical phenomena remains extremely limited. To close this gap, we introduce PhysBench, a comprehensive benchmark designed to evaluate VLMs' physical world understanding capability across a diverse set of tasks. PhysBench contains 10,002 entries of interleaved video-image-text data, categorized into four major domains: physical object properties, physical object relationships, physical scene understanding, and physics-based dynamics, further divided into 19 subclasses and 8 distinct capability dimensions. Our extensive experiments, conducted on 75 representative VLMs, reveal that while these models excel in common-sense reasoning, they struggle with understanding the physical world—likely due to the absence of physical knowledge in their training data and the lack of embedded physical priors. To tackle the shortfall, we introduce PhysAgent, a novel framework that combines the generalization strengths of VLMs with the specialized expertise of vision models, significantly enhancing VLMs' physical understanding across a variety of tasks, including an 18.4% improvement on GPT-4o. Furthermore, our results demonstrate that enhancing VLMs' physical world understanding capabilities can help embodied agents such as MOKA. We believe that PhysBench and PhysAgent offer valuable insights and contribute to bridging the gap between VLMs and physical world understanding. **Project Page is here**

## 1 INTRODUCTION

Understanding the physical world is a fundamental challenge in embodied AI (Gupta et al., 2021; Srivastava et al., 2021). Embodied agents are required to understand the physical properties of objects (e.g., mass, stiffness) to accurately interact with these objects. They also need to understand the relationships of physical objects to operate efficiently in cluttered environments, understand the structure of physical scenes for safe navigation and manipulation, and anticipate the outcomes of interactions and physics-based dynamics for better planning and preventing accidents. These capabilities of intuitive physics (McCloskey et al., 1983; Carey, 2000) are innate to humans and can also greatly benefit embodied agents, allowing them to perform complex tasks and operate safely in real-world scenarios (Kill & Kim, 2020).

Vision-language models (VLMs) (Liu et al., 2024c; Achiam et al., 2023; Team et al., 2023) have emerged as promising solutions for building embodied agents (Liu et al., 2024a; Nasiriany et al., 2024; Huang et al., 2023a). Trained on large amounts of human knowledge, these models have developed strong capabilities in reasoning and task planning (Yue et al., 2024; Lu et al., 2024b; Kim et al., 2024; Niu et al., 2024; Zhen et al., 2024). However, relying solely on these capabilities is insufficient for developing generalist embodied agents. A series of studies have highlighted a gap in understanding the physical world, leading to operational errors (Liu et al., 2024a), such as mishandling fragile objects (Wang et al., 2023c) or failing to recognize appropriate grasping affordances (Guo et al., 2024). **Since these agents operate in and interact with the real world, VLMs must possess a comprehensive understanding of the physical world—a critical yet underexplored domain**. This deficiency in physical world understanding limits the effective deployment of VLMs in embodied applications (Liu et al., 2024a; Guo et al., 2024; Gao et al., 2024a).

---

[*]Equal contribution.

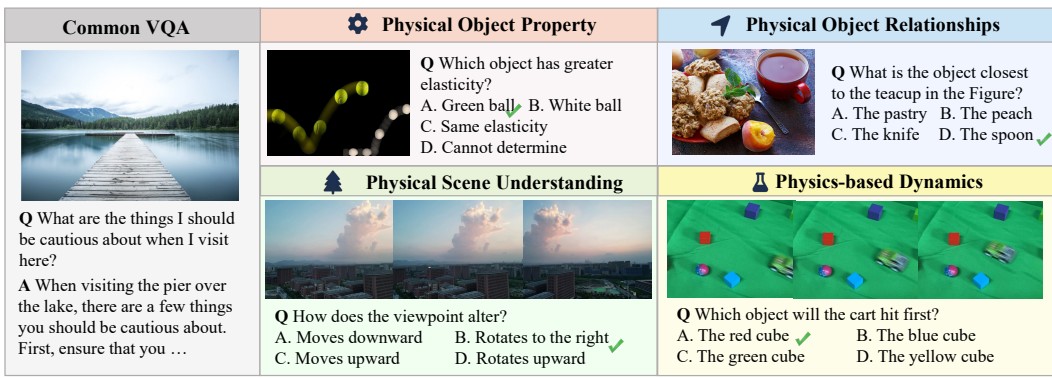

Figure 1: **Common VQA** tasks typically involve questions about visual content and general knowledge. **PhysBench** emphasizes understanding the physical world, encompassing 4 dimensions.

To further investigate this issue, we pose **two fundamental questions**: *(1) Do VLMs possess an understanding of the physical world, and if not, what factors contribute to this limitation? (2) How can we enhance VLMs' physical world understanding capabilities and facilitate the effective deployment of embodied agents like MOKA?*

To answer the above questions and comprehensively assess the extent of the gap between VLMs and physical world understanding, we introduce PhysBench, a dataset comprising 10,002 interleaved video-image-text entries. Given the difficulty of acquiring such data, where expressing specific properties often requires multiple images, we undertook a five-step process, spending a total of 4,000 hours on annotation. We systematically evaluate 75 representative VLM across four domains—physical object properties, physical object relationships, physical scene understanding, and physics-based dynamics—encompassing 19 sub-tasks, as shown in Figure 1. Our extensive experiments reveal that (1) *most current VLMs exhibit poor understanding of the physical world,* particularly in physical scene understanding and physics-based dynamics, with closed-source models significantly outperforming open-source ones; and (2) *the training data for VLMs is likely a major factor contributing to their subpar performance*, as it often lacks the necessary physical knowledge. Notably, when VLMs were fine-tuned on our physically grounded data, their performance improved.

To further improve VLM's physical world understanding capabilities, we propose PhysAgent, a unified framework that incorporates vision foundation models and a physics knowledge memory. By analyzing the sources of errors for VLMs on PhysBench, we identified perceptual inaccuracies and insufficient knowledge as the primary causes of mistakes. To address these issues, we incorporated vision foundation models to enhance perceptual capabilities and assist VLMs in handling tasks they typically struggle with, such as depth estimation and numerical distance calculation. Additionally, we integrated a knowledge memory module to embed essential knowledge about the physical world, which can be selectively invoked by PhysAgent. Unlike previous methods designed for physical reasoning (Zheng et al., 2024b; Tung et al., 2023), PhysAgent retains the strong generalization abilities of VLMs and their capacity to solve open-ended problems, without relying on manually predefined processing logic or being limited to specific tasks. Experimental results demonstrate that PhysAgent improves GPT-4o's zero-shot performance on PhysBench by 18.4%. Furthermore, we investigate how physical world understanding helps the deployment of embodied agents through extensive robotic manipulation experiments on MOKA (Liu et al., 2024a). Specifically, we employ two approaches: fine-tuning the VLM with PhysBench and utilizing PhysAgent for zero-shot inference across five representative manipulation tasks. The improvement in those tasks further validates that PhysBench and PhysAgent can facilitate the deployment of embodied agents like MOKA.

We hope this work offers valuable insights and contributes to bridging the gap between VLMs and physical world understanding, ultimately advancing embodied AI toward human-level capabilities. In summary, this paper has two technical contributions: (1) We present PhysBench, a large-scale benchmark for evaluating the performances of vision-language models in physical world understanding. We identify the key challenges through extensive studies and provide insights into why the existing VLMs have insufficient physical world understanding capabilities. (2) We propose PhysAgent, a unified approach that improves VLMs' physical world understanding abilities. Through extensive experiments, we demonstrate that enhancing VLMs' comprehension of physical environments can significantly facilitate the deployment of embodied agents.

## 2    RELATED WORK

**Physical Comprehension Datasets**. Early benchmarks (Riochet et al., 2018; Rajani et al., 2020) were developed primarily for vision-only models, while more recent efforts (Yi et al., 2019; Chen et al., 2022; Wang et al., 2024g) have predominantly focused on simple visual primitives, such as spheres, cubes, and rigid object collision events, often restricted to a limited set of simulated scenarios (Zheng et al., 2024b; Tung et al., 2023). We summarize the key features of these various benchmarks and compare them against our benchmark in Table 1. However, existing VQA datasets assessing physical knowledge (Lu et al., 2022; He et al., 2024) mainly focus on commonsense reasoning rather than physical world perception. Spatial VQA benchmarks (Chen et al., 2024a; Lyu et al., 2024; Bonnen et al., 2024; Wang et al., 2024d) emphasize geometric relationships in 3D sence, which represent only a part of the physical. In contrast, PhysBench is the first comprehensive dataset designed to evaluate models' understanding of the physical world, encompassing a wide variety of scenarios and tasks not covered by previous benchmarks.

**Physical Reasoning Models**. Models for understanding the physical world generally fall into two categories. The first comprises physics-specialized models (Guen & Thome, 2020; Duan et al., 2022), which are typically limited to predicting the next state and are not applicable to other tasks. The second includes physical oracle models (Zheng et al., 2024b; Tung et al., 2023), which are suitable for only a narrow range of tasks due to their reliance on predefined rules. These models often require training additional modules like R-CNN, and their probabilistic outputs restrict them to classification tasks, limiting their ability to handle open-ended questions. In contrast, PhysAgent offers greater flexibility and adaptability across a broader spectrum of problems without these limitations.

**Vision-Language Models**. Vision-Language Models (VLMs) are large-scale models that integrate visual modalities with language understanding (Wu et al., 2023b; Zhan et al., 2024; Dai et al., 2024). In recent years, there has been a surge of work leveraging VLMs as agents for embodied AI (Liu et al., 2024a; Nasiriany et al., 2024). Although these approaches are generalizable, they face challenges due to weak physical world understanding capabilities (Liu et al., 2024a; Guo et al., 2024). By employing PhysBench and PhysAgent, these shortcomings can be mitigated, enhancing the physical world understanding capabilities of VLMs and enabling more reliable robotic control. Additionally, spatial VLMs (Bonnen et al., 2024) have identified that most VLMs lack 3D spatial reasoning capabilities due to insufficient data. However, since spatial reasoning represents only a subset of physical world understanding, our work aims to provide a more comprehensive evaluation and improvement of VLMs' physical world understanding abilities. For additional related work, see Appendix G.

Table 1: A comparison between PhysBench and other physical understanding question-answering benchmarks. PhysBench is a comprehensive dataset, covering a wide range of tasks related to physical world understanding.

| | Property | Attribute | Location | Motion | Temperature | Viewpoint | Light | Collision | Manipulation | Fluid | Interleaved | Size | More than cube |
|---|---|---|---|---|---|---|---|---|---|---|---|---|---|
| CLEVRER (Yi et al., 2019) | ✓ | ✗ | ✗ | ✗ | ✗ | ✗ | ✗ | ✓ | ✗ | ✗ | ✗ | 300,000 | ✗ |
| Cater (Girdhar & Ramanan, 2019) | ✓ | ✗ | ✗ | ✗ | ✗ | ✗ | ✗ | ✓ | ✗ | ✗ | ✗ | 5,500 | ✗ |
| CRIPP-VQA (Patel et al., 2022) | ✓ | ✗ | ✗ | ✗ | ✗ | ✗ | ✗ | ✓ | ✗ | ✗ | ✗ | 5,000 | ✗ |
| ComPhy (Chen et al., 2022) | ✓ | ✗ | ✗ | ✗ | ✗ | ✗ | ✗ | ✓ | ✗ | ✗ | ✗ | 99,844 | ✗ |
| EmbSpatial (Du et al., 2024) | ✗ | ✗ | ✓ | ✗ | ✗ | ✗ | ✗ | ✗ | ✗ | ✗ | ✗ | 3,600 | ✓ |
| Physion (Bear et al., 2021) | ✓ | ✗ | ✗ | ✗ | ✗ | ✗ | ✗ | ✓ | ✓ | ✗ | ✗ | 17,200 | ✓ |
| Physion++ (Tung et al., 2023) | ✓ | ✓ | ✗ | ✗ | ✗ | ✗ | ✗ | ✓ | ✗ | ✗ | ✗ | 2,000 | ✓ |
| ContPhy (Zheng et al., 2024b) | ✓ | ✗ | ✗ | ✗ | ✗ | ✗ | ✗ | ✓ | ✗ | ✓ | ✗ | 6,500 | ✓ |
| SuperCLEVR (Wang et al., 2024g) | ✓ | ✗ | ✗ | ✓ | ✗ | ✗ | ✗ | ✗ | ✗ | ✗ | ✗ | 1,200 | ✗ |
| PhysBench | ✓ | ✓ | ✓ | ✓ | ✓ | ✓ | ✓ | ✓ | ✓ | ✓ | ✓ | 10,002 | ✓ |

## 3    PHYSBENCH

To assess VLMs' physical world understanding ability, we first define the concept of physical world understanding and introduce PhysBench in Section 3.1. Next, we provide a detailed description of the data collection process in Section 3.2. Utilizing PhysBench, we conduct experiments to determine whether VLMs can effectively comprehend the physical world in Section 3.3. Finally, in Section 3.4, we discuss the potential reasons for poor performance.

### 3.1    OVERVIEW OF PHYSBENCH

Understanding the physical world is essential yet fundamentally challenging for embodied AI, as systems must perceive, interpret, and predict the properties and dynamics of objects and environments. This involves comprehending object properties and relationships, interpreting environmental scenes, and anticipating interaction outcomes based on visual cues and core physical principles to ensure safe and effective operation.

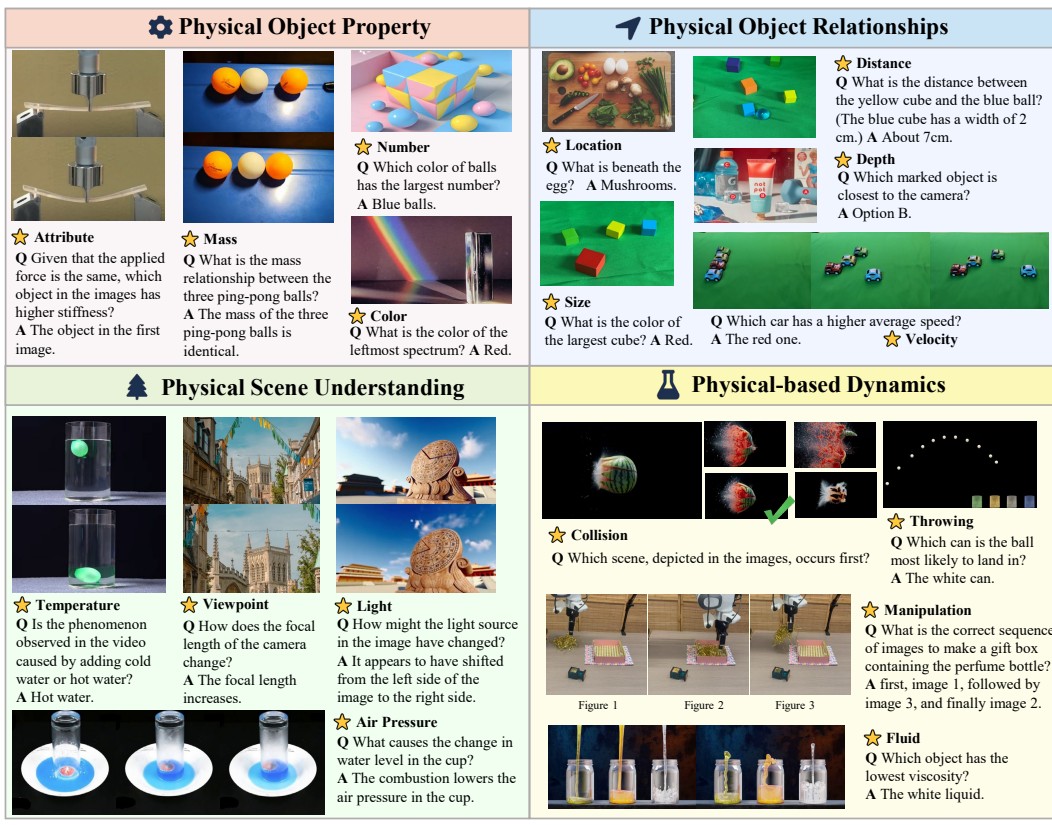

Figure 2: Sampled PhysBench examples from four major dimensions mentioned in Section 3.1. Due to space constraints, we present only the correct answers (as each question in our dataset is a four-option multiple-choice with one correct answer) and defer additional examples to Appendix C.

However, existing datasets often focus solely on image content and commonsense reasoning, neglecting the four fundamental aspects of the physical world mentioned above. To address this gap, we propose PhysBench, which comprehensively evaluates VLMs' perception of the physical world across four major task categories of the physical world: *(1) Physical Object Property*: Assessment of physical attributes of objects such as mass, size, density, tension, friction, bending stiffness, elasticity, and plasticity. *(2) Physical Object Relationships*: Evaluation of spatial relationships involving objects' relative or absolute positions and motions. *(3) Physical Scene Understanding*: Interpretation of environmental factors, including light sources, viewpoints, temperature, *etc.* *(4) Physics-based Dynamics*: Understanding of physical events like collisions, throwing, fluid dynamics, explosions, and other phenomena. Each category corresponds to specific sub-task types and ability types, whose distributions are shown in Figures 3. Detailed examples of specific tasks are illustrated in Figure 2, with additional examples provided in Appendix H. A comprehensive description of sub-task types and ability types is available in Appendix C.

PhysBench is structured as a multiple-choice questionnaire, presenting four options for each question, with only one correct answer. The primary statistics of PhysBench are presented in Table 2 and detailed in Appendix D. Recognizing that different types of tasks possess unique characteristics, we utilize videos and multiple images to effectively convey features that are difficult to capture in a single image—such as elasticity, mass, density, and environmental factors like temperature, humidity, light source, and viewpoint. The dataset also includes objects with similar initial states but differing properties, leading to different future outcomes. This enriches the dataset and allows for a wider range of observable physical behaviors. Consequently, PhysBench draws its data from the internet, real-world captures, and simulations, making it a mixed-format benchmark that integrates text, images, and videos. For convenience, PhysBench-test consists of 10,002 entries, organized into 19 subclasses, as the test set, and 200 entries as the validation set for parameter choosing. We also present 89,998 entries for further research. **The experimental results presented in this paper, unless otherwise specified, are based on the test set**. The performance of VLMs on PhysBench-val can be found in Appendix F.4. Benchmark release details can be found in Appendix B.8.

| Statistic | Number |
|---|---|
| Total questions | 10,002 |
| - only one image | 1,766 (18.6%) |
| - only one video | 2,749 (44.8%) |
| - interleave | 1,902 (20.1%) |
| Unique number of images | 10,058 |
| Unique number of videos | 3,260 |
| 3D Assets | 678 |
| Maximum question length | 48 |
| Maximum choice length | 20 |
| Average question length | 16.5 |
| Average choice length | 4.4 |

Table 2: Key statistics.

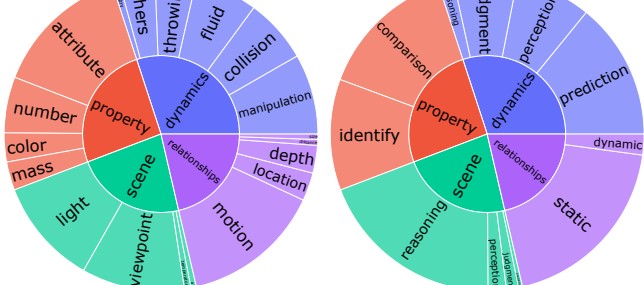

Figure 3: Subtype distribution and ability distribution

## 3.2 DATASET COLLECTION PROCESS

To ensure data quality, all questions were manually annotated by graduate students in STEM fields and further refined through a rigorous review process after collecting and clipping the raw images or videos. To maintain consistency in annotations, we implemented multiple rounds of cleaning and validation throughout the following steps. We have preserved intermediate outputs from the annotation process, such as depth and reflectance maps for simulator-generated data and human-annotated physical principles for many web-sourced videos. The process involves the following sequential steps: *(a) Video Collection.* Videos and images are gathered from web searches, simulations, and real-world captures. The collection process uses predefined simulation rules, LLM-guided queries, and other strategies to find related images or videos (see Appendix A). Human annotators further refine the data by clipping and annotating physical principles in the images or videos. *(b) Video Captioning.* Human-annotated raw videos are processed through automatic filtering, followed by GPT-4o annotations that generate captions with human check. *(c) Questions Design.* For videos annotated with physical principles, we generate physics-related questions using both manual design and GPT-4o, following predefined rules. An automated filter and manual review processes eliminate irrelevant questions. *(d) File Organization.* The remaining valid questions are categorized by task, sub-task, and ability type by human experts. *(e) Quality Check.* The organized dataset undergoes a human review to ensure that the questions are physical world relevant, rely on all input information, are not grounded in common sense, and are accurately categorized with clear questions and corresponding answers. Due to space limitations, the collection guidelines are provided in Appendix B.

## 3.3 CAN VLMS UNDERSTAND THE PHYSICAL WORLD

To assess whether VLMs can understand the physical world, we evaluated 75 representative VLMs on PhysBench and found that a significant performance gap remains between VLMs and human-level understanding. The primary results are presented in Table 3, while detailed analyses of sub-task performance and ability types across the four task categories are provided in Appendix F.3.

**Setup**. Our evaluation was conducted under three configurations: (a) *Image VLMs*, which support only single-image input (e.g., LLaVA-1.5 and BLIP-2); (b) *Video VLMs*, designed for video comprehension (e.g., Chat-UniVi and PLLaVA); and (c) *General VLMs*, which support multiple images and interleaved inputs (e.g., VILA-1.5 and GPT-4o). It is important to note that the data used for evaluating setups (a) and (b) is a subset of PhysBench test subset with interleaved QA pairs removed, whereas setup (c) was evaluated on the full dataset. For most models, we followed the standard protocol outlined in VLMEvalKit Contributors (2023), setting the temperature to 0. For models that do not support multiple images as input, we employed two methods: the *merge* method, where video frames are concatenated into a single image (Fu et al., 2024; Zhang et al., 2024a; Jiang et al., 2024), and the *seq* method, where video frames are input sequentially as individual images. Notably, only models using the *seq* setup can handle interleaved text-image sequences. For details on VLM prompts and hyperparameters, see Appendix E.

**VLMs exhibit a limited understanding of the physical world**. Our evaluation indicates that most models achieve an average accuracy of approximately 40%, which is significantly below human-level performance. Even the best-performing model, GPT-4o, attains only 49.49% accuracy, underscoring the substantial gap between current VLMs and true comprehension of the physical world.

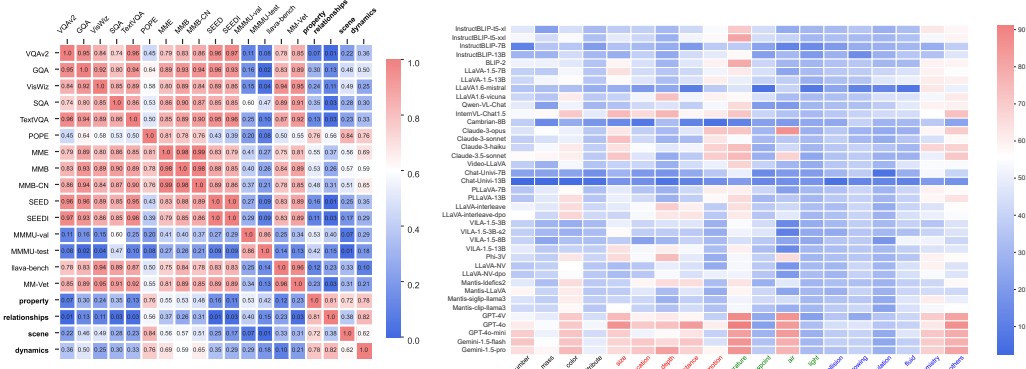

Figure 4: (a) Correlation map between 4 tasks in PhysBench and 15 other vision-language benchmarks. (b) The visualization of model performance across 19 sub-tasks is presented, where different colors represent the respective categories. The four colors, from left to right, represent physical object property, physical object relationships, physical scene, and physical-based dynamics.

| | Size | Format | ⚙Property | ✈Relationships | 🌲Scene | ⚗Dynamics | Avg |
|---|---|---|---|---|---|---|---|
| Random Choice | - | - | 25.00 | 25.00 | 25.00 | 25.00 | 25.00 |
| Human | - | - | 97.10 | 95.67 | 94.91 | 95.68 | 95.87 |
| Image VLM | | | | | | | |
| InstructBLIP-t5-xl (Dai et al., 2024) | 4B | merge | 35.35 | 36.67 | 37.45 | 35.95 | 36.24 |
| InstructBLIP-t5-xxl (Dai et al., 2024) | 12B | merge | 41.11 | 38.47 | 37.89 | 36.42 | 38.51 |
| InstructBLIP-7B (Dai et al., 2024) | 7B | merge | 21.94 | 29.00 | 19.53 | 27.45 | 23.82 |
| InstructBLIP-13B (Dai et al., 2024) | 13B | merge | 31.69 | 33.19 | 23.13 | 30.64 | 29.94 |
| BLIP-2 (Li et al., 2023c) | 12B | merge | 41.70 | 40.83 | 36.25 | 36.93 | 38.61 |
| LLaVA-1.5-7B (Liu et al., 2023a) | 7B | merge | 38.44 | 41.53 | **38.60** | 42.69 | 40.09 |
| LLaVA-1.5-13B (Liu et al., 2023a) | 13B | merge | 41.31 | 42.50 | 34.40 | 44.38 | 40.45 |
| LLaVA1.6-mistral (Liu et al., 2024b) | 7B | merge | 29.77 | 22.22 | 8.54 | 20.58 | 20.30 |
| LLaVA1.6-vicuna (Liu et al., 2024b) | 7B | merge | 40.26 | 59.72 | **38.60** | 42.65 | 42.28🥈 |
| Qwen-VL-Chat (Bai et al., 2023b) | 9B | merge | 35.97 | 43.33 | 26.47 | 41.27 | 35.63 |
| InternVL-Chat1.5 (Chen et al., 2024c) | 26B | merge | **53.08** | **70.14** | 37.01 | **44.78** | **47.51**🥇 |
| Cambrian-8B (Tong et al., 2024) | 8B | merge | 23.27 | 17.92 | 23.02 | 29.29 | 24.61 |
| Claude-3-opus (Anthropic, 2024) | - | merge | 41.97 | 40.97 | 30.63 | 36.50 | 37.00 |
| Claude-3-sonnet (Anthropic, 2024) | - | merge | 37.86 | 40.00 | 32.23 | 36.89 | 36.18 |
| Claude-3-haiku (Anthropic, 2024) | - | merge | 43.28 | 53.33 | 30.06 | 39.93 | 39.44 |
| Claude-3.5-sonnet (Anthropic, 2024) | - | merge | 46.46 | 41.11 | 27.89 | 37.60 | 38.05 |
| Video VLM | | | | | | | |
| Video-LLaVA (Lin et al., 2023a) | 7B | seq | 36.82 | 36.11 | 33.69 | 40.52 | 37.04 |
| Chat-Univi-7B (Jin et al., 2023) | 7B | seq | 19.28 | 20.97 | 18.86 | 28.46 | 22.19 |
| Chat-Univi-13B (Jin et al., 2023) | 13B | seq | 4.30 | 11.53 | 15.67 | 11.47 | 10.36 |
| PLLaVA-7B (Xu et al., 2024) | 7B | seq | **38.02** | 35.83 | **36.34** | 39.89 | **37.94**🥇 |
| PLLaVA-13B (Xu et al., 2024) | 13B | seq | **39.91** | **38.33** | 31.52 | **40.76** | 37.70🥈 |
| General VLM + Interleaved data | | | | | | | |
| LLaVA-interleave (Li et al., 2024d) | 7B | seq | 47.23 | 44.62 | 35.64 | 37.21 | 41.00 |
| LLaVA-interleave-dpo (Li et al., 2024d) | 7B | seq | 47.97 | 42.67 | 33.73 | 38.78 | 40.83 |
| VILA-1.5-3B (Lin et al., 2023b) | 3B | seq | 32.40 | 33.02 | 34.84 | 35.78 | 34.11 |
| VILA-1.5-3B-s2 (Lin et al., 2023b) | 3B | seq | 33.14 | 30.26 | 35.72 | 33.00 | 33.07 |
| VILA-1.5-8B (Lin et al., 2023b) | 8B | seq | 33.41 | 29.88 | 30.85 | 35.91 | 32.85 |
| VILA-1.5-13B (Lin et al., 2023b) | 13B | seq | 40.53 | 40.15 | 31.96 | 36.07 | 37.15 |
| Phi-3V (Abdin et al., 2024) | 4B | seq | 43.67 | 37.92 | 34.93 | 36.92 | 38.42 |
| LLaVA-NV (Zhang et al., 2024b) | 7B | seq | 38.33 | 30.83 | 34.00 | 37.17 | 35.42 |
| LLaVA-NV-dpo (Zhang et al., 2024b) | 7B | seq | 38.83 | 44.31 | 33.86 | 37.21 | 37.43 |
| Mantis-Idefics2 (Jiang et al., 2024) | 8B | seq | 41.97 | 41.44 | 29.53 | 36.56 | 37.39 |
| Mantis-LLaVA (Jiang et al., 2024) | 7B | seq | 44.48 | 30.45 | 36.25 | 34.73 | 36.69 |
| Mantis-siglip-llama3 (Jiang et al., 2024) | 8B | seq | 42.47 | 32.78 | **36.83** | 37.51 | 37.64 |
| Mantis-clip-llama3 (Jiang et al., 2024) | 8B | seq | 40.61 | 35.11 | 32.45 | 38.36 | 36.92 |
| GPT-4V (Achiam et al., 2023) | - | seq | 49.59 | 45.77 | 26.34 | 42.15 | 41.26 |
| GPT-4o (Achiam et al., 2023) | - | seq | 56.91 | **64.80** | 30.15 | **46.99** | **49.49**🥇 |
| GPT-4o-mini (Achiam et al., 2023) | - | seq | 53.54 | 44.24 | 30.59 | 42.90 | 43.15 |
| Gemini-1.5-flash (Team et al., 2023) | - | seq | **57.41** | 52.24 | 34.32 | 40.93 | 46.07 |
| Gemini-1.5-pro (Team et al., 2023) | - | seq | 57.26 | 63.61 | 36.52 | 41.56 | 49.11🥈 |

Table 3: **Evaluation results for 39 VLMs.** The evaluation of General VLMs is based on the data from Video and Image VLM evaluations, with the addition of interleaved data. "Seq" refers to sequential input of frames of videos, while "merge" refers to merging video frames into a single image. **Bold** indicates the best result, and underline indicates the second best in each group.

As shown in Figure 4(b), considerable room for improvement remains, particularly in tasks related to physical scene understanding and physics-based dynamics.

**Closed-source models generally perform better**. As shown in Figure 5(b), the GPT series and Gemini-1.5 models significantly outperform open-source models. Notably, GPT-4 surpasses the best open-source model, LLaVA-interleave, by 20.7%, indicating a substantial gap between open-source and closed-source models. However, we did not observe a clear advantage with Claude, a finding that aligns with results from other benchmarks (Cao et al., 2024; Wu et al., 2024c).

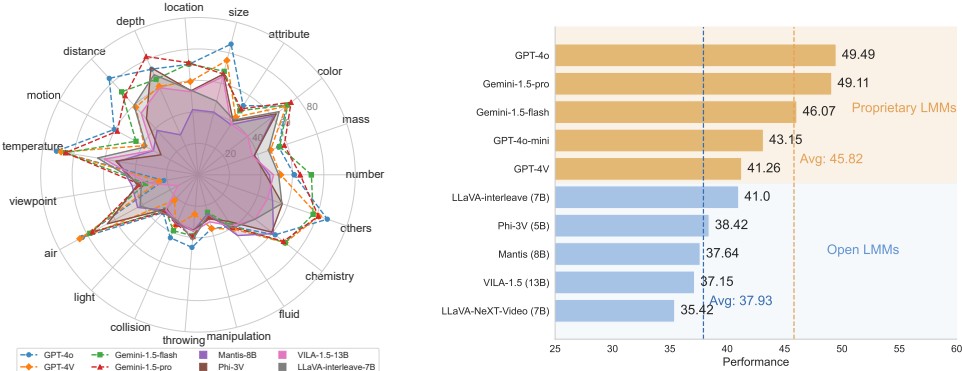

Figure 5: (a) The performance of 8 representative open-source General VLMs across 19 sub-tasks in PhysBench, which support interleaved inputs. The closer it is to the circular boundary, the better. (b) The overall performance of those 8 VLMs. Closed-source models generally perform better.

### 3.4 WHY DO VLMs STRUGGLE WITH PHYSICAL WORLD UNDERSTANDING

To further investigate why VLMs struggle with physical world understanding, we analyzed Phys-Bench and discovered that it differs significantly from common VQA tasks. Additionally, we found that the performance of larger model size or more training data does not result in clear improvements on PhysBench, which may be *due to a lack of physical world knowledge in the training data*. Furthermore, we found that many errors stem from this deficiency; when we augmented the models with physical world knowledge, their performance improved. This further suggests that the gap between VLMs and physical world understanding may be attributed to limitations in the training data.

**Physical world understanding differs significantly from common VQA tasks**. To assess the relationships between our tasks and other VLM benchmarks, we adopted the methodology proposed by (Tong et al., 2024; Fang et al., 2024) to construct a correlation map, as shown in Figure 4(a). Details on the construction of the correlation map are provided in Appendix F.6. Our analysis reveals that PhysBench differs significantly from traditional VLM benchmarks, exhibiting closer alignment with POPE (Li et al., 2023h) in tasks such as hallucination detection, while also showing that performance does not consistently improve with increased data or model scale.

**VLMs's physical world understanding ability does not scale with model size, data, or frames**. *(1) Model Size Scalability*. Figure 6(a) shows that increasing model size using the same dataset significantly enhances performance on common QA tasks. However, this improvement does not extend to PhysBench, where gains are limited or even negative. For instance, while VILA-1.5's performance improves by 7.1% on common QA tasks when scaling from 3B to 7B parameters, it

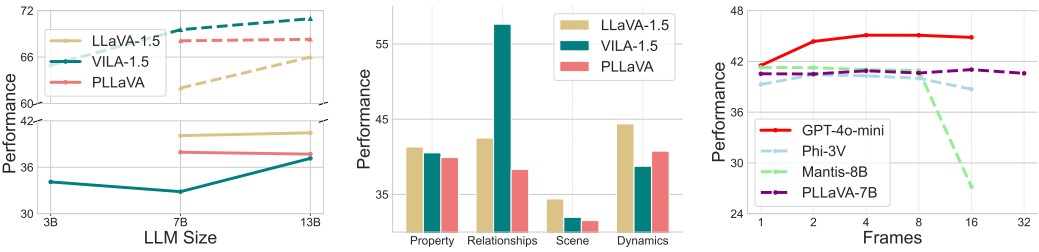

Figure 6: (a) Model size scalability. The solid line shows the average performance across 14 common QA tasks (Table 23), while the dashed line represents PhysBench results. (b) Data scalability. VILA and PLLaVA expand upon LLaVA's architecture by utilizing more data. (c) Frame scalability.

decreases by 3.8% on PhysBench. *(2) Data Scalability*. As shown in Figure 6(b), scaling up the dataset offers limited benefits for physical comprehension. PLLaVA and VILA-1.5, larger-data variants of LLaVA-1.5, exhibit minimal improvement or even a decline in performance on PhysBench compared to LLaVA-1.5. Analysis of the additional data (Appendix D.2) reveals it is predominantly descriptive, focusing on content description rather than enhancing physical understanding. Nevertheless, VILA-1.5's spatial reasoning abilities have significantly improved, aligning with trends observed in other benchmarks (Yu et al., 2023b; Li et al., 2023a). *(3) Frame Scalability*. Figure 6(c) indicates that the three open-source models are insensitive to the number of frames, performing similarly to single-frame inputs, with performance sometimes decreasing as frames increase. This suggests that current models cannot effectively utilize multi-frame information. Notably, increasing the number of frames led Mantis to frequently fail to follow instructions or refuse to answer, and expanding beyond eight frames did not yield further improvements.

**Perceptual and knowledge gaps constitute the majority of errors**. To investigate the poor performance of VLMs on PhysBench, we randomly selected 500 questions and obtained explanations from three models—GPT-4o, Phi-3V, and Gemini-1.5-flash. Expert annotators classified the root causes of the mispredictions into six categories: perception errors, reasoning errors, lack of knowledge, refusal to answer, failure to follow instructions, and annotation errors in the dataset. The distribution of these error types is shown in Figure 7, with selected cases and detailed analyses provided in Appendix I. The error distribution reveals that perceptual errors account for 37%, 40%, and 45% of the mistakes made by GPT-4o, Gemini-1.5-flash, and Phi-3V, respectively, while lack of knowledge constitutes 34%, 35%, and 23% of errors for these models. This analysis suggests that perceptual errors and knowledge gaps are the primary sources of mispredictions, indicating that while the models are adept at extracting information from text and visual inputs, their physical world understanding and complex reasoning abilities remain limited.

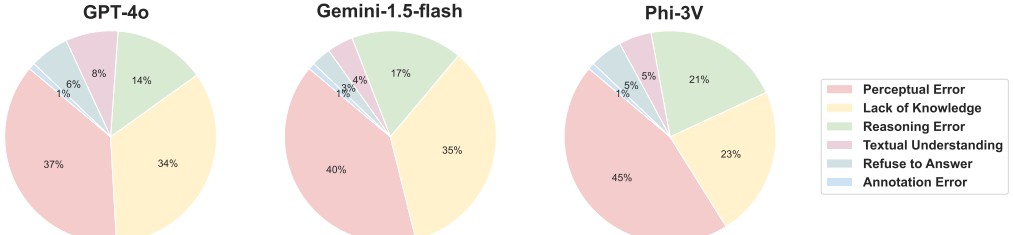

Figure 7: Distribution of error types for GPT-4V, Gemini-1.5-flash, Phi-3V.

**Can VLMs transfer physical world knowledge?** Our error analysis revealed that inadequate physical world knowledge and reasoning capabilities were key contributors to the models' poor performance. To investigate whether introducing additional examples could enhance performance, we conducted tests on 200 entries of PhysBench, pairing each with a similar example. These additional examples were incorporated through fine-tuning or in-context learning. As shown in Figure 9(b), the performance improvements after adding physical world knowledge examples indicate that VLMs can transfer physical knowledge to some extent. This suggests that the original data's lack of physical world knowledge was a significant factor in the models' suboptimal performance.

## 4    PHYSAGENT

Recognizing perceptual inaccuracies and knowledge gaps as key sources of error shown in Section 3.4, we introduce PhysAgent in Section 4.1 to improve VLMs' understanding of the physical world by integrating vision foundation models for enhanced perception and incorporating physical knowledge memory. To verify whether enhancing VLMs' physical understanding facilitates the deployment of embodied agents, we conducted five embodied agent tasks as detailed in Section 4.2.

### 4.1    HOW TO ENHANCE VLMS FOR PHYSICAL WORLD UNDERSTANDING

We propose PhysAgent, a novel framework that integrates knowledge memory and vision foundation models to enhance physical world understanding in VLMs. This framework is inspired by our findings in Section 3.4, where we identified perceptual errors and insufficient knowledge as the

primary causes of mistakes in VLMs. To address these shortcomings, we establish a *knowledge memory* that provides prior physical world knowledge and rules. Additionally, we utilize vision *foundational models* namely Depth Anything (Yang et al., 2024b), SAM (Kirillov et al., 2023), and GroundingDINO (Liu et al., 2023b) to achieve enhanced visual perception. These models enable us to identify object types and spatial locations, and further acquire information about objects' dynamics through VLM reasoning or retrieval from memory. They also help solve problems that VLMs cannot address, such as estimating depth and numerical distances. Unlike prior physical reasoning models that are confined to specific tasks and struggle to adapt to natural language queries, our method aims to fully leverage the reasoning and generalization capabilities of VLMs. Experiments on PhyBench show that PhysAgent improves performance by 18.4% on GPT-4o.

As illustrated in Figure 8, given a question, PhysAgent follows three key steps: (1) *Task-specific Prompt Activation*: PhysAgent first classifies the question (manually or automatically) and activates task-specific prompts, incorporating relevant physical knowledge for different tasks. For instance, for a question about light, it retrieves knowledge on the relationship between light source movement and shadow direction to assist the VLMs. (2) *Foundation Models Integration*: PhysAgent processes the foundation model's outputs, leveraging VLM reasoning capabilities. For example, it identifies objects in the scene using GroundingDINO and retrieves relevant attributes from the knowledge memory. (3) *Chain-of-Thoughts Reasoning*: PhysAgent then engages in chain-of-thought reasoning, conducting a self-verification step to ensure logical consistency before providing the final answer.

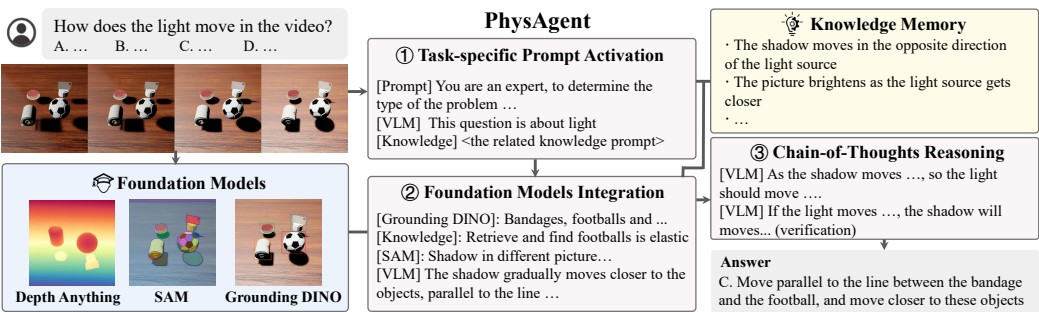

Figure 8: Architecture of PhysAgent. PhysAgent employs a three-step reasoning process to address the problem: activating task-specific prompts, integrating foundation models, and reasoning.

**Baselines**. We utilized three prompt methods: Chain of Thought (CoT) (Kojima et al., 2023), Desp-CoT (Wu et al., 2023d), and Pure Language Reasoning (PLR), in addition to an oracle method, ContPhy (Zheng et al., 2024b), which served as our baseline. Detailed descriptions of the prompt strategies and the implementation of ContPhy are provided in Appendix E.6 and Appendix E.7.

**Results**. The results in Figure 9(a), lead to the following conclusions: *(1) Prompting methods are unstable, and using pure language yields catastrophic results*. As observed, the CoT strategy has minimal impact, while both Desp-CoT and PLR show a decline in performance. This suggests that descriptive prompts are not particularly effective for addressing the questions, implying that our dataset requires a deeper understanding of the videos or images to answer accurately. *(2) ContPhy even worsens performance*. In three out of four tasks, ContPhy underperforms compared to its base model, GPT-4o, due to suboptimal module invocation and limited flexibility in its logical templates, which struggle to adapt to diverse scenarios. Additionally, ContPhy relies on models like RCNN to process visual information instead of directly leveraging GPT-4o, leading to potential information loss and subsequent performance degradation. *(3) PhysAgent consistently improves zero-shot performance*, notably achieving a 49.5% improvement for GPT-4o in Scene. Compared to the CoT, Desp-CoT, and PLR prompting strategies, our method demonstrates consistent improvements.

## 4.2 CAN PHYSICAL WORLD UNDERSTANDING HELP IN EMBODIED APPLICATIONS

Despite gaining significant attention in recent years for their strong generalization capabilities, VLMs as embodied agents (Liu et al., 2024a) still exhibit fundamental operational errors during physical world interactions. In this section, we investigate whether enhancing the physical world perception abilities of VLMs can improve their performance in downstream embodied agent tasks.

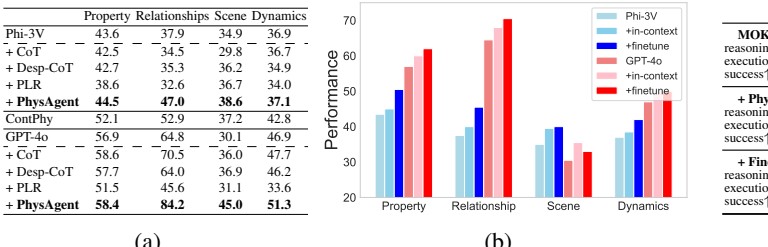

| | Property | Relationships | Scene | Dynamics |
|---|---|---|---|---|
| Phi-3V | 43.6 | 37.9 | 34.9 | 36.9 |
| + CoT | 42.5 | 34.5 | 29.8 | 36.7 |
| + Desp-CoT | 42.7 | 35.3 | 36.2 | 34.9 |
| + PLR | 38.6 | 32.6 | 36.7 | 34.0 |
| + PhysAgent | **44.5** | **47.0** | **38.6** | **37.1** |
| ContPhy | 52.1 | 52.9 | 37.2 | 42.8 |
| GPT-4o | 56.9 | 64.8 | 30.1 | 46.9 |
| + CoT | 58.6 | 70.5 | 36.0 | 47.7 |
| + Desp-CoT | 57.7 | 64.0 | 36.9 | 46.2 |
| + PLR | 51.5 | 45.6 | 31.1 | 33.6 |
| + PhysAgent | **58.4** | **84.2** | **45.0** | **51.3** |

| | Affordance | Force | Color | Location | Tool |
|---|---|---|---|---|---|
| **MOKA** | | | | | |
| reasoning error↓ | 0.3 | 0.5 | 0.1 | 0.1 | 0.3 |
| execution error↓ | 0.1 | 0.3 | 0.2 | 0.2 | 0.3 |
| success↑ | 0.6 | 0.2 | 0.7 | 0.7 | 0.4 |
| **+ PhysAgent** | | | | | |
| reasoning error↓ | 0.0 | 0.2 | 0.0 | 0.1 | 0.2 |
| execution error↓ | 0.2 | 0.3 | 0.2 | 0.2 | 0.3 |
| success↑ | 0.8 | 0.5 | **0.8** | 0.7 | 0.5 |
| **+ Fine-tune** | | | | | |
| reasoning error↓ | 0.0 | 0.1 | 0.1 | 0.0 | 0.0 |
| execution error↓ | 0.1 | 0.3 | 0.1 | 0.2 | 0.3 |
| success↑ | **0.9** | **0.6** | **0.8** | **0.8** | **0.7** |

(a)      (b)      (c)

Figure 9: (a) Performance comparison of various methods. (b) Analysis of physical world knowledge transfer. (c) Performance evaluation across five embodied tasks as described in Figure 10.

To evaluate embodied agents, we designed five fundamental manipulation tasks as shown in Figure 10(a). The specifics of these tasks, along with the corresponding testing methods and language instructions, can be found in Appendix F.5. These tasks require the agents to possess a basic understanding of spatial relations and the physical properties of objects. Specifically, we utilized MuJoCo (Todorov et al., 2012) and the 7-DoF Franka Emika robotic arm from Menagerie (Zakka et al., 2022), building our simulation platform based on MOKA (Liu et al., 2024a) as the embodied agent approach to test these embodied tasks. The VLM we used in these tasks is GPT-4o.

| | |
|---|---|
| **Affordance** | Grasp pot, knife, spoon, monitor, tennis racket. |
| **Force** | Grasp fragile items (egg, ripe persimmon), soft items (jelly, plastic cup), and rigid objects (iron ball). |
| **Color** | Grasp the specific color cube. |
| **Location** | Grab the cube in specific location and move it to the plate. |
| **Tool** | Grasp specific tools, depending on the problem. |

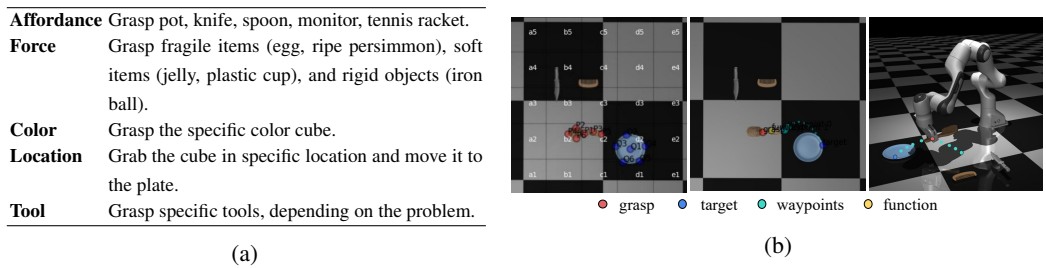

(a)      (b)

Figure 10: (a) Description of each of the testing tasks. (b) Marked observation, predicted affordances, and motion in MOKA. MOKA leverages a VLM to generate key points and waypoints, and then converts these affordance representations into executable motions for the robotic arm.

As illustrated in Figure 10(b), MOKA prompts the VLM to generate key points and additional attributes for affordance representation based on free-form language instructions and visual observations of the environment. Since the five tasks we tested were relatively basic and did not require decomposition into subtasks, we could directly invoke the VLM in a question-answering format to address the operational challenges. This approach ensures seamless compatibility between the pipelines of PhysAgent and MOKA. Once the key points and waypoints were obtained from the VLM, MOKA converted these affordance representations into executable motions for the robotic arm. To evaluate the impact of enhanced physical-world understanding on embodied tasks, we applied two methods to MOKA's VLM: (1) fine-tuning it with PhysBench, and (2) employing PhysAgent to zero-shot assist in reasoning about affordance representations.

As shown in Figure 9(c), we observe consistent improvements after fine-tuning with a subset of PhysBench, indicating that the benchmark's data is of high quality and suitable for use as demonstration data in open-world robotics tasks. Additionally, PhysAgent consistently yields stable zero-shot gains across all five tasks, with significant progress observed in the force task in Figure 10(a).

## 5 CONCLUSION

In conclusion, we introduce PhysBench, a benchmark designed to assess Vision-Language Models' understanding of the physical world. Through experiments on 75 models, we identified significant gaps in physical world understanding, particularly in open-source models, due to inadequate training data. To address this, we developed PhysAgent, a novel framework that improves physical reasoning by 18.4% on GPT-4o. Additionally, we demonstrated the utility of our dataset and approach in robotic tasks, helping to advance the understanding of the physical world in machine intelligence.

**Statement**. We provide a detailed discussion of limitations, broader impacts, ethical considerations, and reproducibility in Appendix J.

ACKNOWLEDGMENTS

Jiageng Mao and Yue Wang acknowledge funding supports from Toyota Research Institute, Dolby, and Google DeepMind. Yue Wang is also supported by a Powell Faculty Research Award.

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

CONTENTS

# A    DETAILED DATASET COLLECTION PROCESS

## A.1    SIMULATION

We use  (Blender, 2018) as our simulation platform. We utilized 679 objects and 470 HDR images to generate simulated videos and images. During each simulation, we concurrently save images or videos of depth, normal, and albedo, as well as the corresponding configuration files, which include the position, angle, movement ,and other properties of the object light source.

**Image generation**. In addition to ambient lighting, we employed two point lights and one sunlight. To ensure data diversity, the positions of the camera and the arrangement of objects (ensuring no overlap and that all objects are captured by the camera) were randomized to some extent. Drawing from the object attribute annotation methods described in Newton (Wang et al., 2023c), we cleaned and re-annotated our data to develop a comprehensive table of objects and their attributes. Utilizing this table, we delineate the relational semantics between different objects and the corresponding queries. Following the approach in BLINK (Fu et al., 2024), each object in the simulated images is demarcated with a bounding box rather than being explicitly mentioned in the text, thereby enhancing the evaluation of the model's image comprehension capabilities.

Despite imposing considerable constraints on our code and meticulously annotating the 3D assets, there remains the possibility of minor object overlaps or incomplete captures of objects by the camera. To ensure that objects in the data are clearly identifiable, we employed GroundingDINO (Liu et al., 2023b) for detection. We only accept images where the labels detected by GroundingDINO match exactly in content and quantity with the generated labels. This process also provides us with the bounding boxes of objects for subsequent annotation. During the later stages of annotation, manual inspection of the images is conducted to ensure accuracy. To improve the detection success rate of GroundingDINO and reduce the probability of false detections, we set the box_threshold to 0.2 and the text_threshold to 0.2. Specifically, these parameter settings were obtained through a grid search, with detailed results presented in Table 14.

**Videos with varying lighting conditions**. We used only one point light source and arranged objects on a plane to render shadows. The variations in lighting include three aspects: the color of the light, the position of the light source, and the intensity of the light. In terms of the light source position, the movement involves translations along the x, y, and z axes. To avoid ambiguity in lateral directions (Du et al., 2024), during the dataset generation, the movement questions are typically framed in terms of moving along the line connecting two objects rather than simply asking for the direction of movement.

**Videos with varying camera conditions**. We used the same lighting and other configurations as in the image generation process. During video recording, we randomly altered the camera's position or shooting angle to capture the videos.

**Fulid**. We used assets from ContPhy (Zheng et al., 2024b) and Unity (Haas, 2014) to generate videos across four types: fluid, rope, cloth, and ball, with 350, 250, 200, and 200 videos respectively. The videos were then manually annotated.

## A.2    WEB

For web data collection, we primarily use predefined topics (e.g., gases) to retrieve relevant videos or images from the internet (such as middle school physics experiments). After filtering and cleaning the data, we proceed with annotation. Additionally, we leverage large language models (LLMs) to generate suitable descriptions of physics-related concepts, which we then use to search for corresponding videos, followed by further cleaning and annotation.

In addition to the network data collection process described in Section 3.2, we employ the following methods to gather data.

**Unsplash.** We use high-quality and high-resolution images from Unsplash Ali et al. (2023). 57,859 images are downloaded, and finally we use only about 6,000 images.

**Manipulation**. We sampled approximately 500 videos from DROID (Khazatsky et al., 2024), Ego4D (Grauman et al., 2022), and MimicPlay (Wang et al., 2023a), providing detailed annotations to generate QA pairs categorized under object-manipulation tasks. The primary focus of these questions is to determine the appropriate sequence of actions based on given instructions, which are derived from the original datasets' descriptions of actions. Figure 42's both first and second examples provide an example of this task. First, we filtered the videos to select those with a strong alignment between the instructions and the visuals, ensuring that the videos were clear, unambiguous, and matched the instructions well. Next, we identified 3-4 keyframes from these videos. The task involved sorting these key frames in the correct order to execute the instructions properly. Additionally, we used FunKPoint (Lai et al., 2021) to annotate the affordance (Gibson, 2014) points in individual images from the original dataset. Specific examples of these annotations can be found in Figure 42's third and fourth examples.

**nuScenes**. We cropped and annotated videos from the nuScenes (Caesar et al., 2019) mini and test datasets, ultimately obtaining 1,356 QA pairs for spatial movement tasks. We categorized the questions into types such as left turn, straight, and right turn, and included arrows on the images to indicate the direction. The questions asked participants to identify which image they might see based on the indicated direction, as illustrated in Figure 28.

**Visual Prompt**. In certain tasks, we utilized Visual Prompts (Fu et al., 2024), and through experimentation, we identified an alternative annotation method, detailed in Appendix F.2. For tasks using Visual Prompts, we set the image size to $1024 \times 1024$ pixels. In this scale, we standardized the Visual Prompt to a red circle with a 30-pixel radius and white text options with a font size of 65 pixels. The positions of the options' centers in the dataset are recorded in the following format:

```
"A": [ 734, 922 ], "B": [ 202, 898 ], "C": [ 343, 115 ], "D": [ 410, 559 ]
```

**Visual Correspondences**. Drawing inspiration from Fu et al. (2024); Sarlin et al. (2020), we also annotated a portion of the corresponding point data using visual prompts. Specific examples can be found in Figure 24.

### A.3 REAL-WORLD

We also collected some real-world videos and images, primarily covering sub-tasks related to light, camera, and physical dynamics such as collisions. An iPhone 13 Pro Max was used as the recording device, and all images are in RGBD format.

## B DATA ANNOTATION PROTOCOL

### B.1 GENERAL GUIDELINES

As previously discussed, there is a significant gap in existing benchmarks, which primarily assess vision-language models (VLMs) based on descriptive tasks without adequately addressing their physical perception and reasoning abilities. To bridge this gap, our benchmark, PhysBench, is designed to provide a comprehensive evaluation framework for physical perception, integrating visual understanding with the assessment of physical properties, spatial relationships, and dynamic phenomena. This approach aims to advance AI systems toward more general-purpose capabilities in real-world physical environments. Our benchmark follows the guidelines outlined below for data collection:

- **General Principles:**
  - Annotations must be accurate, consistent, and adhere to a high standard of academic rigor.
  - It covers multiple tasks and topics to mirror real-world applications.
  - It incorporates diverse visual contexts and physics knowledge to foster a well-rounded evaluation.
  - It offers varying levels of challenge to effectively probe and uncover the potential limitations of current models.

      – It provides robust evaluation settings for deterministic assessments.

- **Specific Instructions:**
  - All questions must contain one or more images.
  - All questions should be written in English.
  - All questions should meet the college-level difficulty.
  - Questions should not be ambiguous and must be answerable with one of the given options.
  - Clearly categorize each question.
  - Annotate all fields, including the question, answer options and other things that follow the format requirement.
- **Review Process:** Ensure that every annotation undergoes a peer review to maintain high standards and minimize errors.

Annotations such as physical properties, spatial relationships, dynamic interactions, and environmental factors are also collected, providing detailed examples that demonstrate the physical perception and reasoning capabilities of the models for further analysis and usage.

## B.2 DATA FORMAT AND STRUCTURE

Detailed examples of annotated question examples as shown in Figure 11 are provided in the guidance to serve as a reference for the annotators.

- **JSON File Format:** The structured JSON format will include fields for number, question type, question text, answer options (for multiple-choice), correct answer, question difficulty, and explanation (if there exists).
- **Naming Conventions:**
  - Each collected sample will be stored in a separate JSON file following a standard naming rule: **subject_{Number}**.json
  - Image Files: **image_{QuesNum}_{ImageNum}**.png

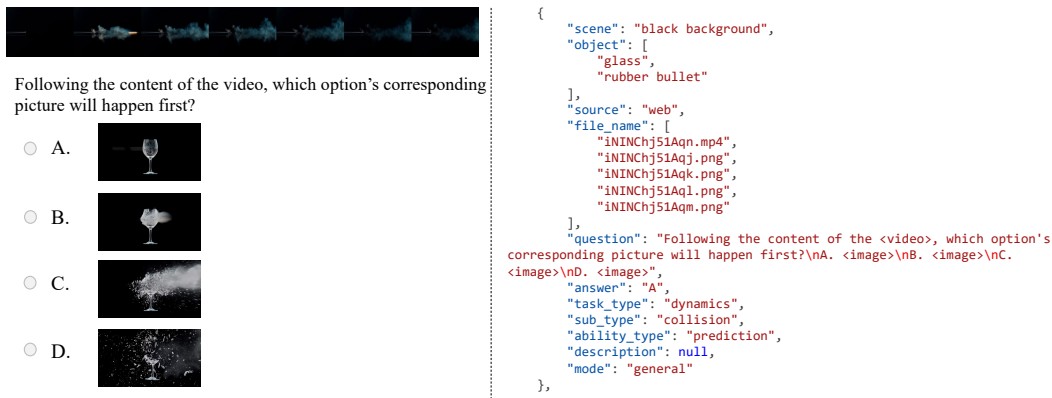

Figure 11: A question-answer pair case in PhysBench and its JSON representation.

## B.3 QUALITY CONTROL AND VALIDATION

- A secondary review team will rigorously vet annotations for quality and adherence to guidelines.
- Regular audits of random samples from the dataset will be conducted to ensure sustained quality and consistency.
- Periodic training sessions will be held to update annotators on best practices and any changes in annotation guidelines.
- Feedback mechanisms will be established to promptly address and rectify any identified errors or inconsistencies in the annotations.

### B.4 HANDLING AMBIGUITIES

Instances of ambiguity or unclear data should be flagged for detailed review. These instances will be collaboratively examined during team meetings to establish a standardized approach for annotation.

### B.5 ETHICAL CONSIDERATIONS

- **Copyright and Licensing:** Adherence to copyright and licensing regulations is strictly enforced. Data from sources that prohibit copying or redistribution will be explicitly avoided.
- **Data Privacy:** Compliance with privacy laws and ethical standards in data handling is paramount. Annotators must avoid collecting questions that contain any private information.
- **Ethical Data Usage:** All data collection and usage must respect ethical guidelines. This includes avoiding biased or harmful content and ensuring that the datasets promote fairness and inclusivity.

### B.6 DATA CONTAMINATION CONSIDERATIONS

The risk of data contamination can be mitigated by assigning annotators to carefully select questions that extend beyond straightforward queries with easily accessible answers. It is essential that tasks rely on provided videos or images for answers rather than the common knowledge of large language models. This approach is beneficial for creating benchmarks that genuinely test the model's ability to comprehend and synthesize information from diverse and challenging sources.

### B.7 ANNOTATION PLATFORM

We developed a GUI-based annotation platform, as shown in Figure 12, designed to assist human experts in the data annotation process. Through this program, experts can easily view various media content, such as videos and images, and perform annotations and edits directly within an intuitive interface. The streamlined layout enhances the user experience, ensuring that experts can complete annotation tasks efficiently and accurately, thereby improving the quality and efficiency of the annotations. The purpose of this tool is to simplify the complex annotation workflow, reduce manual effort, and make the annotation process more efficient.

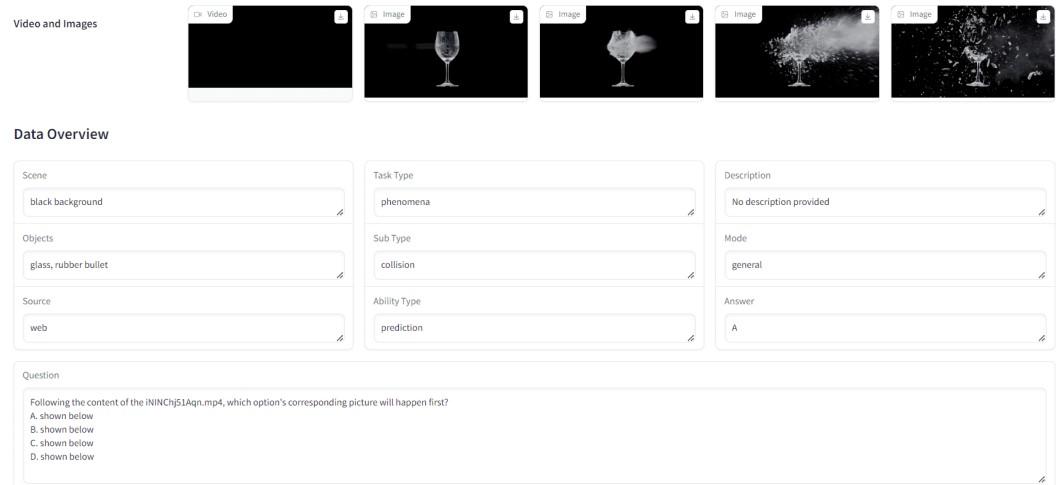

Figure 12: Annotation Platform.

### B.8 BENCHMARK PREPARATION AND RELEASE

For convenience, PhysBench-test consists of 10,002 entries, organized into 19 subclasses, as the test set, and 200 entries as the validation set for parameter choosing (PhysBench-val) since the answers

for the PhysBench will not be publicly available and are hosted similarly to Yue et al. (2024). We also present 89,998 entries for further research. The experimental results presented in this paper, unless otherwise specified, are based on the test set. Importantly, the answer labels for the remaining set will not be publicly released to prevent data leakage, and we will maintain an online evaluation platform. It should be noted that the scores we refer to for PhysBench are based on the entire test dataset, including the validation split.

To ensure that each source dataset is well represented in the validation split and that the distribution of sub-task types and ability types in the validation set is similar to that of the entire dataset, we adopted the following sampling strategy: 1. Randomly sample questions to ensure that the distribution of sub-task and ability types in the validation set matches the full dataset. 2. Randomly sample the remaining questions from each source dataset based on its proportion in the entire dataset.

Additionally, we conducted several quality checks to address any potential errors.

### B.9    MORE DETAILS OF THE ANNOTATION PIPELINE

As described in Section 3.2, all questions were manually annotated by graduate students in STEM fields and subsequently refined through a rigorous review process. The detailed workflow is depicted in Figure 13, where the icon ♣ denotes stages involving human participation. The annotators were divided into three groups, each comprising six individuals. During the annotation process, either a GUI interface similar to Figure 12 or direct editing of JSON files was employed. The first group was responsible for video collection, the final group handled quality checks, and the intermediate group completed the remaining steps. A comprehensive explanation of this process is provided below.

**Video Collection**. Videos and images were sourced through web searches, simulations, and real-world recordings. The collection process utilized predefined simulation rules, LLM-guided queries, and other strategies to identify relevant content. Specifically, for data that could be simulated, we generated it using pre-written simulation scripts, as described in Section A. Additionally, we conducted YouTube searches for videos, leveraging captions generated by GPT to guide annotators in collecting videos. We also sought compilations on topics like "interesting physical phenomena." For real-world recordings, we pre-designed scenarios and objects to capture. Given the complexity of collecting such data, we employed GPT as heuristic tools to expand the search scope and enrich the dataset's diversity. Finally, annotators curated the videos by trimming them to retain only relevant segments and provided detailed physical descriptions, such as the direction of shadow movement or the causes of observed events.

**Question Design**. In the previous step, we utilized GPT to generate annotations for videos. This approach was adopted after observing that directly inputting a video and prompting GPT to generate question-answer pairs in an in-context learning format often led to suboptimal adherence to instructions. Instead of focusing on generating questions, GPT tended to explain the video content. Additionally, inputting videos directly consumed more tokens, increasing computational costs. Furthermore, annotating videos with captions was considered beneficial for subsequent research leveraging our dataset. Given the complexity of collecting such data, we employed GPT as a heuristic strategy to broaden the scope of search and enhance the diversity of our dataset. GPT generated questions using an in-context learning approach, with the templates provided in Appendix E.1. Notably, all final questions were curated and verified by human annotators, with GPT serving only as a reference.

**File Organization**. We presented annotators with examples and detailed classification criteria, as outlined in Appendix C and Appendix H. Annotators were then instructed to categorize tasks accordingly and structure each task in the JSON format shown in Figure 11.

**Quality Check**. After organizing the dataset, we conduct a two-stage review process to ensure its quality. The dataset undergoes human verification to confirm that the questions are relevant to the physical world, rely on all provided input information, are not solely based on common sense, and are accurately categorized with clear questions and corresponding answers. The first stage involves filtering and refining the questions, while the second stage focuses on thorough validation to ensure data accuracy and consistency.

Given the difficulty of acquiring such data, where expressing spe- cific properties often requires multiple images, we undertook a five-step process, spending a total of **4,000 hours** on annotation.

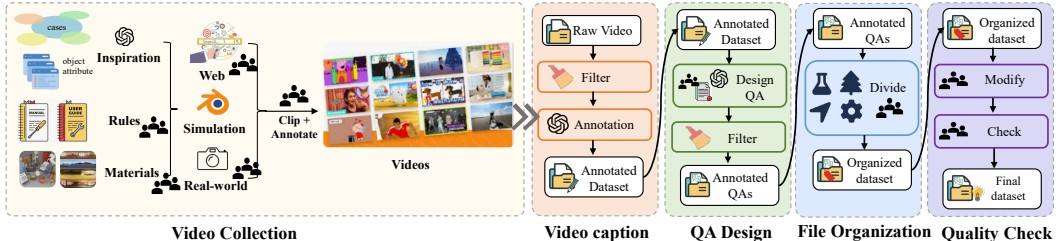

Figure 13: **Annotation Pipeline.** Icon ⚇ indicate stages involving human participation.

## C   DETAILED TASK DESCRIPTION

This section primarily introduces the definitions of the various capabilities and sub-tasks. Specific examples can be found in Appendix H. As illustrated in Figure 1, humans not only accurately comprehend visual content but also draw upon knowledge to explain and reason about the scenes they observe. This is the primary goal of most VQA datasets. Building on this foundation, several studies have focused on the commonsense understanding of the geometric relationships and properties within the 3D world. However, the real physical world encompasses not only 3D geometric relationships but also includes object properties (e.g., elasticity, ductility), physical object relationships (e.g., velocity, acceleration, depth), physical scene and environmental factors understanding (e.g., temperature, camera parameters, lighting conditions), and the principles and mechanisms of physical dynamics (e.g., the sequence of collisions, energy transfer, fluid flow, simple physical and chemical reactions, optical and electromagnetic dynamics). To address this gap, we propose PhysBench, which aims to bring Vision-Language Models closer to achieving spatial intelligence (Gupta et al., 2021; Srivastava et al., 2021).

### C.1   ABILITY DESCRIPTION

Table 4 presents the description of capability types for PhysBench. The tasks are categorized into four types: physical properties, physical object relationships, physical scene, and physics-based dynamics. Each task type corresponds to different capability types such as recognition, comparison, prediction, judgment, reasoning, and perception, with detailed descriptions provided for each. Specifically, the physical properties task type includes the recognition of object properties and comparisons between the physical properties of different objects. The physical object relationships task type distinguishes between static and dynamic relationships of objects. The physical scene understanding task type involves the prediction, judgment, reasoning, and perception of changes in environmental conditions. The physics-based dynamics task type evaluates the ability to predict, judge, reason, and perceive various physics-based dynamics or phenomena.

### C.2   PHYSICAL OBJECT PROPERTY SUB-TASK

Table 5 describes the sub-tasks for object types in PhysBench. The object types are divided into four subcategories: number, mass, color, and attributes. Specifically, the "number" subcategory involves the count of certain objects or changes in their quantity; the "mass" subcategory focuses on approximate changes in object mass, mass estimation, or mass comparison; the "color" subcategory describes the color of objects or color changes; and the "attributes" subcategory covers object characteristics such as rigidity, fluidity, gas, hardness, malleability, elasticity, smoothness, and sharpness.

Notably, our dataset imposes more rigorous evaluations on conventional attributes like mass and color. For instance, in the case of counting tasks, previous works (Kafle & Kanan, 2017; Yi et al., 2019; Chen et al., 2022) typically only require identifying the number of objects in an image. In contrast, our dataset often ties quantities to specific object attributes (e.g., "How many objects of a certain color are outside the plate?" or "How many objects are not blurred by the camera?").

### C.3   PHYSICAL OBJECT RELATIONSHIPS SUB-TASK

Table 6 presents the descriptions of physical object relationships sub-tasks in PhysBench. The relationships types are divided into five subcategories: size, location, depth, distance, and movement.

Table 4: Ability Type Description for PhysBench.

| Task type | Ability type | Description |
|---|---|---|
| Physical Object Property | Identify | Identify a physical property of an object. |
| | Comparison | Comparison of the same physical property between different objects, or changes in a specific property over time. |
| Physical Object Relationships | Static | The static spatial properties of an object. |
| | Dynamic | The dynamic spatial properties of an object, meaning the spatial properties are changing dynamically. |
| Physical Scene Understanding | Prediction | Predict what might happen if a certain environmental condition is changed, or what will happen next. |
| | Judgment | Judge what will be different if a certain environmental condition is modified. |
| | Reasoning | Explain the occurrence reason or condition of the environment. |
| | Perception | Understand what environment condition change has occurred and what its definition is, or determine the environmental attributes that caused the phenomenon to occur. |
| Physics-based Dynamics | Prediction | Predict what might happen if a certain object or object attribute in the video is changed, or what will happen next. |
| | Judgment | Judge what will change if a certain attribute is modified, or, for example, the sequence in which actions occur. |
| | Reasoning | Explaining the occurrence reason or condition of phenomena. |
| | Perception | Understand what phenomenon has occurred and what its definition is, or determine the physical attributes that caused the phenomenon to occur. |

Table 5: Sub-task Description for Physical Object Property Type in PhysBench.

| Sub type | Description |
|---|---|
| Number | The number of certain objects or changes in the number of objects. |
| Mass | Approximate changes in mass, mass estimation, or mass comparisons. |
| Color | Color of an object or changes in color. |
| Attribute | The attributes or types of objects, such as rigid body, fluid, gas, stiffness, malleability, elasticity, smoothness, sharpness, *etc.* |

Specifically, the "size" subcategory relates to the dimensions of an object; the "location" subcategory describes the absolute and relative positions of objects, including tasks related to object localization and spatial information processing; the "depth" subcategory focuses on an object's depth relative to the camera or depth comparisons between different objects; the "distance" subcategory involves the comparison or estimation of distances between objects as well as their absolute size; and the "movement" subcategory addresses the analysis of movement direction, changes in speed, and changes in acceleration.

Table 6: Sub-task Description for Physical Object Relationships Type in PhysBench.

| Sub type | Description |
|---|---|
| Size | The size of the object. |
| Location | Positional relationships (absolute and relative), including directly or indirectly locating objects and other tasks involving spatial information. |
| Depth | Depth of an object relative to the camera or depth comparisons between different objects. |
| Distance | Comparison or estimation of distances between objects or their absolute sizes. |
| Motion | Motion, velocity, acceleration and the direction of movement, changes in speed, or changes in acceleration. |

## C.4 Physical Scene Understanding Sub-task

Table 7 describes the sub-tasks for physical scene understanding types in PhysBench. The physical scene understanding types are divided into four subcategories: temperature, camera, gas, and light. Specifically, the "temperature" subcategory involves temperature and its fluctuations, as well as dynamics caused by these changes; the "camera" subcategory focuses on changes in camera position and the resulting effects; the "gas" subcategory covers conditions of the gas environment, such as high pressure, low pressure, or vacuum states; and the "light" subcategory describes the color tone of the light source (warm or cool), changes in the position of the light source, light intensity, and the nature of the light source (point or surface).

Table 7: Sub-task Description for Physical Scene Understanding Type in PhysBench.

| Sub type | Description |
| --- | --- |
| Temperature | The temperature and its changes, as well as phenomena caused by temperature fluctuations. |
| Viewpoint | The position of the camera and its changes, along with phenomena caused by shifts in camera position. |
| Air | The air environment encompasses conditions such as air pressure, humidity, airflow direction, and intensity. |
| Light | Includes the color tone of the light source (warm or cool), changes in the position of the light source, its intensity, and the nature of the light source (point or surface). |

## C.5 Physics-based Dynamics Sub-task

Table 8 describes the sub-tasks for physics-based dynamics types in PhysBench. The physics-based dynamics types are divided into six subcategories: collision, throwing, manipulation, fluid, chemistry, and other physics-based dynamics. Specifically, the "collision" subcategory includes physics-based dynamics such as friction between objects, collisions, and explosions; the "throwing" subcategory involves physical world dynamics and phenomena such as throwing and falling; the "manipulation" subcategory focuses on the manipulation of deformable objects and affordance-based sequence arrangements; the "fluid" subcategory covers fluid motion, shapes, and other fluid-related dynamics; the "chemistry" subcategory involves basic chemical reactions or other dynamics or phenomena related to chemistry in the physical world; and the "other" subcategory includes various other dynamics related to the physical world.

Table 8: Sub-task Description for Physics-based Dynamics Type in PhysBench

| Sub type | Description |
| --- | --- |
| Collision | Friction between objects, collisions, explosions, and similar dynamics. |
| Throwing | Throwing, falling, and other physical world dynamics. |
| Manipulation | Manipulation of deformable objects and affordance-based sequence arrangements. |
| Fluid | Fluid motion, shapes, and other fluid-related dynamics. |
| Chemistry | Basic chemical reactions or other dynamics involving chemical knowledge in the physical world. |
| Others | Other dynamics related to the physical world. |

All of our questions are designed as multiple-choice, with only one correct answer among the four options. Referring to BLINK (Fu et al., 2024), the visual prompt is set as a red circle with a 10px size, which has been found to be the most suitable.

# D    MORE DATASET ANALYSIS

## D.1    GLOBAL STATICS

**Question Distribution.** Figure 14 further elucidates the distribution of word counts, highlighting the diverse patterns of questions. The average word of the questions within PhysBench is 16.53, while the maximum number of words in a question reaches 48.

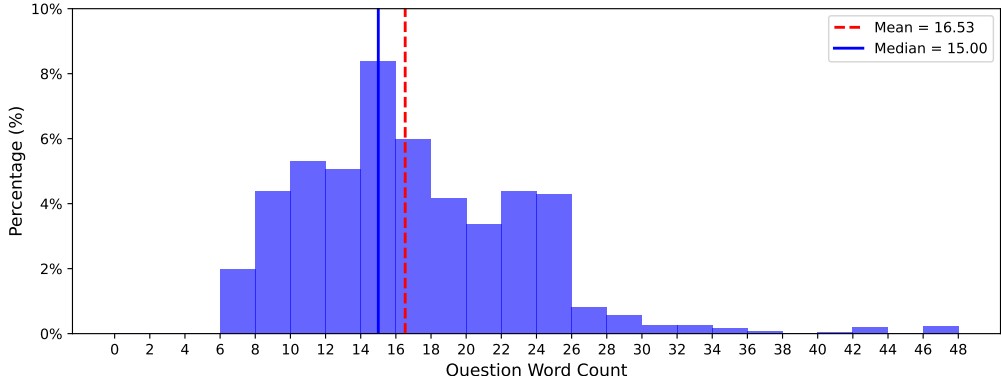

Figure 14: The distribution of the number of words per question in PhysBench. Questions with a length greater than 48 are categorized as 47 for visualization simplicity.

**Option Distribution.** Figure 15 further elucidates the distribution of word counts, highlighting the diverse patterns of options. The average number of words in the options within PhysBench is 4.36, while the maximum number of words in a question reaches 20. It is worth noting that an "option" here refers to the text following a choice, such as "A. Point A," where the character length is 2.

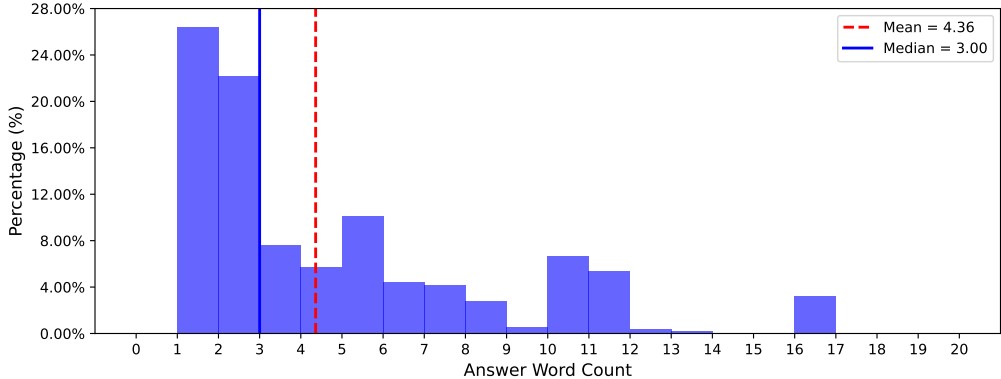

Figure 15: The distribution of the number of words per question in PhysBench. Options with a length greater than 20 are categorized as 20 for visualization simplicity.

**Image Resolution.** The resolution of most images falls within the 1024-2048 range, accounting for approximately 79.7% of the total images. Only four images in the dataset have a resolution below 256. The minimum resolution is 183, while the maximum is 7201.

**Video Resolution.** Similarly, the majority of videos have a resolution between 1024-2048, covering about 98.6% of the total videos. Only 1.3% of videos have a resolution below 1024. The highest video resolution is 1920, and the lowest is 370.

**Video Frames.** Considering that VLMs typically extract keyframes when processing videos—generally selecting 6-8 frames—the total number of frames in the videos doesn't need to be large. However, this doesn't imply that our videos are limited to 6–8 frames. The frame counts are primarily distributed in the ranges of 30–45 frames and 60+ frames, accounting for 59.4% and 35.9% of the total, respectively. The distribution charts for image and video resolution, as well as video frame counts, can be seen in Figure 16.

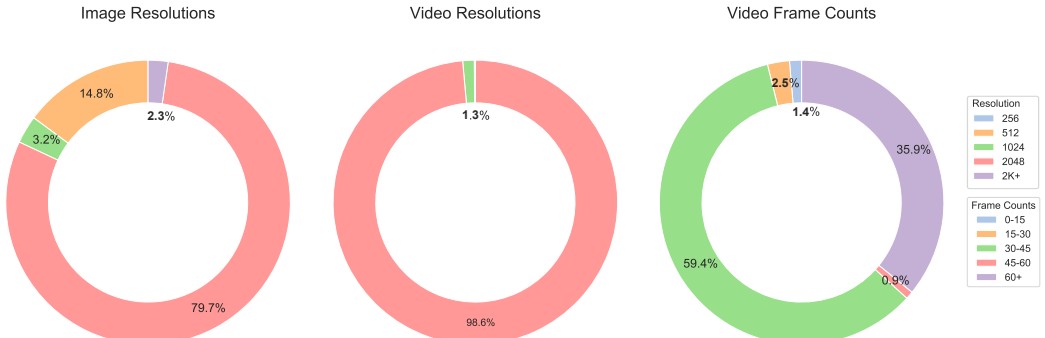

Figure 16: The distribution charts for image and video resolution, as well as video frame counts. From left to right: the distribution of image resolution, the distribution of video resolution, and the distribution of video frame counts.

## D.2 WORD STATICS AND WORD CLOUD

We created a word cloud and word frequency chart for our dataset, as shown in Figure 17. It reveals a significant presence of terms related to physical world perception, such as "direction," "camera," "phenomenon," "effects," "relationship," and "light." Additionally, we generated a word cloud and word frequency chart for the **LLaVA-1.5-13B** training data, which includes 595K pretraining samples and 665K instruction-tuning samples, as illustrated in Figure 18.

**VILA-1.5-13B**. In the pretraining data, CC3M (Sharma et al., 2018) and COYO (Byeon et al., 2022) 25M primarily consist of image captions, while MMC4 (Zhu et al., 2024) 25M includes webpage descriptions, none of which significantly contribute to the model's understanding of the physical world. Therefore, we mainly focus on the instruction-tuning data. The LLaVA-1.5-SFT-665K instruction-tuning data has already been used in LLaVA-1.5-13B, so it is not considered further. Additionally, FLAN (Wei et al., 2021) consists purely of text data and is thus also excluded from consideration.

The data used for VILA-1.5-13B includes the following:

- Captioning: Image Paragraph Captioning (Krause et al., 2017), MSR-VTT (Xu et al., 2016), TextCaps (Sidorov et al., 2020)
- Reasoning: CLEVR (Johnson et al., 2017), NLVR (Suhr et al., 2017), VisualMRC (Tanaka et al., 2021)
- Translation: Multi30k (Elliott et al., 2016)
- VQA: ActivityNet-QA (Yu et al., 2019), DocVQA (Mathew et al., 2021), GQA (Hudson & Manning, 2019), iVQA (Yang et al., 2021), MSRVTT-QA (Xu et al., 2017), MSVD-QA (Xu et al., 2017), OCR-VQA (Mishra et al., 2019), ST-VQA (Biten et al., 2019), Vi-QuAE (Lerner et al., 2022), VQAv2 (Goyal et al., 2017b), Visual Dialog (Das et al., 2017)

According to the guidelines from the VILA repository, the aforementioned instruction-tuning data is all included in M³IT (Li et al., 2023f). Therefore, we use M³IT to analyze the training data for VILA-1.5. The final word frequency statistics and word cloud for the VILA-1.5-13B training data are shown in Figure 19.

**PLLaVA-13B**. PLLaVA-13B is based on LLaVA-Next (Liu et al., 2024b), with LLaVA-Next incorporating additional data from ShareGPT-4V (Chen et al., 2023a) compared to LLaVA-1.5-13B. The main enhancement from LLaVA-Next to PLLaVA-13B is the addition of 783k instructional video-to-text tuning data, enabling LLaVA-Next to handle video input. Specifically, the training data are sourced from the dataset used in VideoChat2 (Li et al., 2023e), which includes data for various video understanding tasks, such as 27k conversation videos from VideoChat (Li et al., 2023d) and Video-ChatGPT (Maaz et al., 2023b), 80k classification task data from Kinetics (Kay et al., 2017) and SthSthV2 (Goyal et al., 2017a), 450k captioned data from Webvid (Bain et al., 2021), YouCook2 (Zhou et al., 2018), TextVR (Wu et al., 2023c), and VideoChat, 117 reasoning data points from NextQA (Xiao et al., 2021b) and CLEVRER (Yi et al., 2019), and 109k annotated question-answering samples from Webvid, TGIF (Li et al., 2016), and Ego4D (Grauman et al., 2022).

We used ShareGPT-4V and downloaded all the training data from the PLLaVA repository, creating a word cloud and word frequency chart of the training data, as shown in Figure 20.

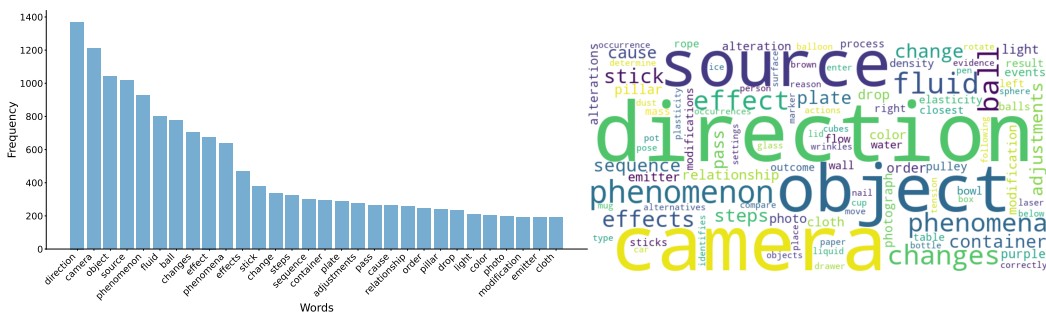

Figure 17: Word Statics and Word Cloud for PhysBench.

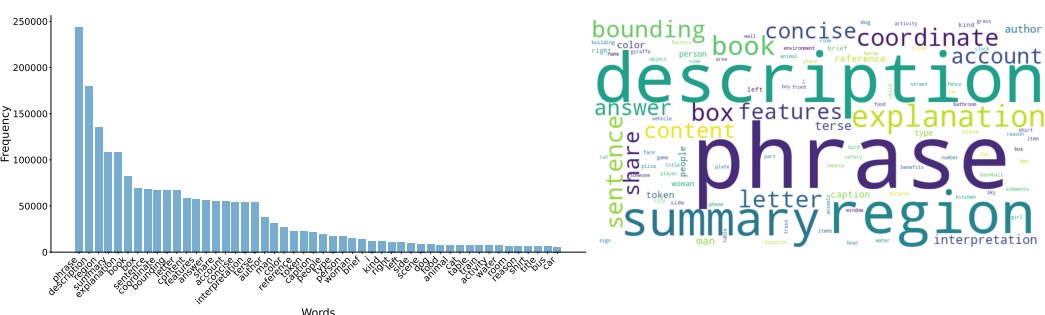

Figure 18: Word Statics and Word Cloud for LLaVA-1.5-13B Training Data.

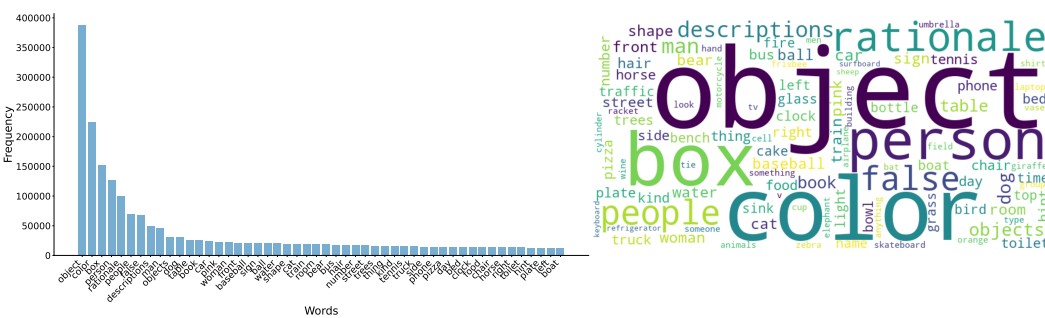

Figure 19: Word Statics and Word Cloud for VILA-1.5-13B Training Data.

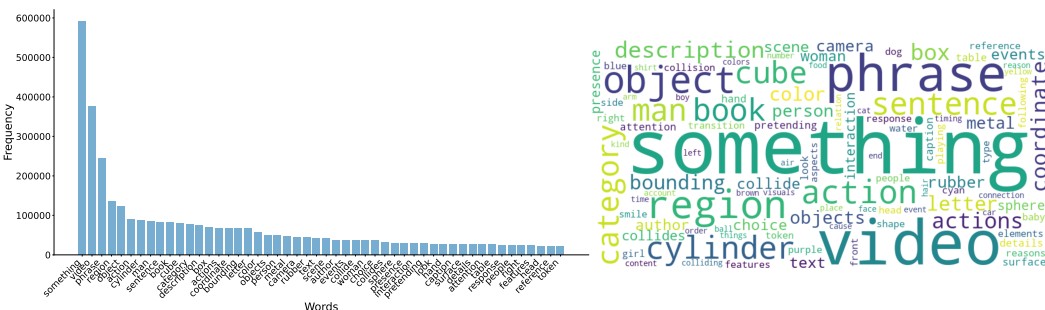

Figure 20: Word Statics and Word Cloud for PLLaVA-13B Training Data.

Observing Figure 18 19 20, we can see that the most frequent words in the training data of LLaVA-1.5-13B, VILA-1.5-13B, and PLLaVA-13B are terms like 'description', 'phrase', 'summary', 'region', and similar words. Keywords such as 'direction' appear much less frequently. We have listed the frequency of several key terms from our dataset in the training data of LLaVA-1.5-13B, VILA-1.5-13B, and PLLaVA-13B, as shown in Figure 21. Although PLLaVA-13B includes words like 'collides' and 'camera', they are primarily used to describe phenomena without addressing the underlying physical mechanisms, which may explain why PLLaVA-13B shows no substantial improvement.

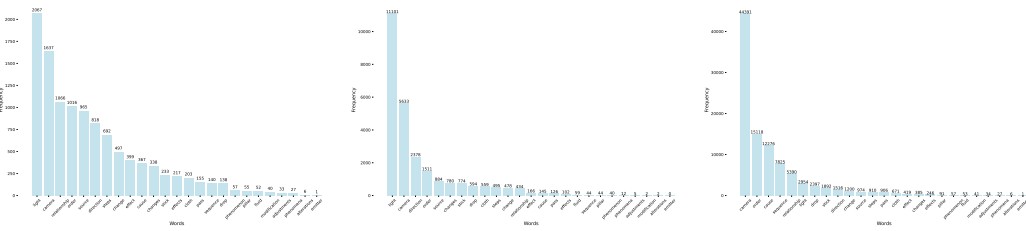

Figure 21: The frequency of common terms in PhysBench within the training data of the LLaVA-1.5-13B, VILA-1.5-13B, and PLLaVA-13B models.

# E MORE DETAILS ON THE SETUP

## E.1 PROMPT FOR LLMS

**Video Cases Prompt**. The general system prompt for LLMs to generate video or image cases is as follows:

> You are an assistant that communicates only in JSON and is an expert in physics. Do not write normal text.
> Your response should be in JSON format with the following string fields:
> 1. 'video description': A description of the video.
> 2. 'question': A multiple-choice question based on the video's content.
> 3. 'answer': A response to the question, with one of the values being 'A', 'B', 'C', or 'D'.
> For the same 'video description', you may provide multiple question-answer pairs, which should be presented in a list format.
> Important notes: 1. The 'question' should not contain details from the 'video description'. 2. The relationship between each question-answer pair is one of juxtaposition, not progression.
> The following is a conversation between the user and the assistant. Ensure that the assistant's answers are as rich and informative as possible.

Then we will the below sentence as the first turn chat and let the LLMs give the more cases in the latter turns. We replace the examples provided in the first turn to enable GPT to generate more diverse responses. Notably, some of the examples generated by GPT do not fully align with the data standards defined in Appendix B. We use these examples solely as references to collect more diverse data. However, as described in Section 3.2, the subsequent data annotation undergoes a rigorous five-step manual process to ensure high data quality.

The process of generating VQA templates is similar: questions are generated based on the captions of the input video or image. However, QA templates are more focused, targeting specific types of relationships, such as those related to spatial positioning.

{"role": "system", "content": (Contents of the color block above)}
{"role": "user", "content": "give me some cases"}
{"role": "assistant", "content":

- **Video Description:** A ball thrown horizontally from a cliff
    - **Questions:**
        1. What causes the ball to fall downward?
           A. Air resistance
           B. Gravitational force
           C. Magnetism
           D. Elastic collision
        2. Which of the following affects how far the ball travels horizontally?
           A. Mass of the ball
           B. Initial velocity of the ball
           C. Color of the ball
           D. Shape of the ball
        3. What happens if the ball is thrown with a higher initial velocity?
           A. It travels farther horizontally
           B. It travels shorter horizontally
           C. It stops mid-air
           D. It moves vertically upward
    - **Answers:** [B, B, A]
- **Video Description:** A feather falling in a vacuum chamber
    - **Questions:**
        1. What happens if the vacuum chamber is filled with air?
           A. The feather falls faster
           B. The feather falls at the same speed
           C. The feather floats
           D. The feather rises upward
        2. What would happen to the action if it happened in a strong magnetic field?
           A. It would fall faster
           B. It would float
           C. It would fall slower
           D. It would remain unaffected
    - **Answers:** [A, D]
},
{"role": "user", "content": "give me some cases"}

**Caption**. video or image caption system prompt is as follows:

You are an outstanding video annotator and physicist.
Your task is to provide a detailed description of a given video based on the user's input video and a brief description of the video.
The description should include, but is not limited to, the following aspects:
1. The content of the video.
2. The principles behind the phenomena observed in the video.
3. The properties of certain objects within the video, such as estimated size, mass, color, and type.
4. Various spatial information, including relative and absolute positions, sizes, distances, and movements.
Ensure that the descriptions are thorough, precise, and reflect your expertise in physics.

**Answer Extraction**. The prompt used to instruct GPT-4o is illustrated as follows:

The following sentences contain answers (one of A, B, C, D) and corresponding analysis.
Your role is to find the answer.
Please return only one of the four letters: A, B, C, or D.
The sentences are:

### E.2  3D ASSETS

The overview of the 3D assets is shown in Figure 22. After defining the object attribute table, we selected 678 objects and annotated their attributes accordingly. For each object, we then adjusted its size and position to ensure that multiple objects (typically 4-5 in our dataset) do not overlap and remain clearly visible. These objects were subsequently used for simulations.

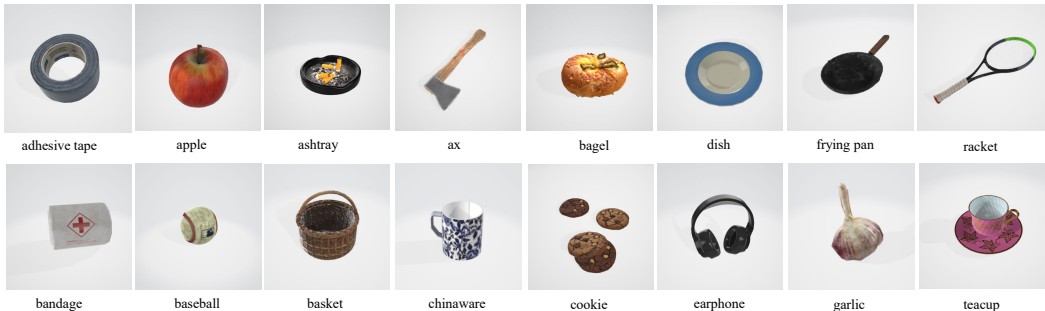

Figure 22: **3D Assets Sample**. We have 678 objects and 470 HDR environment in total for simulation.

### E.3  MODEL HYPERPARAMETERS

In our experiments, we conducted comparisons with some of the most recent and representative MLLMs in the following. In our experiments, we conducted comparisons with some of the most recent and representative MLLMs in the following. We divided the models into three categories: **Image VLM**, **Video VLM**, and **General VLM**, according to whether they support only one image input, support a video input (also supports image), or support interlaced video and multiple images.

Following Zhang et al. (2024a), for Image VLMs, we concatenate multiple frames from a video (defaulting to 8 frames) into a single image, arranged from left to right and bottom to top, before inputting it into the model. Thus, both Image VLMs and Video VLMs were tested on question-answer pairs involving only a single image or video, which represent a subset of PhysBench. In contrast, General VLMs, which support interleaved image-text sequences, were tested on the full PhysBench test split.

For most open-source models, we used the hyperparameter torch_dtype = torch.float16. However, for some models that only support torch_dtype = torch.bfloat16, we used torch_dtype = torch.bfloat16 for those cases. Additionally, for other parameter settings, we generally followed the configurations provided in the original papers, their code repositories, or the examples from Hugging Face.

#### E.3.1  IMAGE VLM

The following is a description of the specific models reviewed, as well as the specific parameter configurations, can be found in Table 9.

- **BLIP-2** (Li et al., 2023c) employs a dual-stage strategy to bridge the modality gap effectively, utilizing a lightweight Q-Former pre-trained on 129 million image-text pairs. In the first stage, the model initiates the learning of vision-language representations by leveraging a frozen image encoder, ViT-g/14 from EVA-CLIP (Fang et al., 2023). In the subsequent stage, a frozen LLM, FlanT5 (Chung et al., 2024), is employed to facilitate vision-to-language generative learning. This innovative approach enables effective zero-shot instructed image-to-text generation. The tested version is BLIP2-t5-xxl (we call it as BLIP-2 in this paper).

- **InstructBLIP** (Dai et al., 2024) is derived from the pre-trained BLIP-2 model (Li et al., 2023c), which integrates a ViT-g/14 image encoder, a Vicuna (Chiang et al., 2023), and a Q-Former that bridges these two components. During the vision-language instruction tuning process, only the Q-Former is fine-tuned, utilizing data from 13 distinct visual question-answering datasets. The tested version is InstructBLIP-t5-xl (we call it as InstructBLIP-

t5-xl in this paper), InstructBLIP-t5-xxl (we call it as InstructBLIP-t5-xxl in this paper), InstructBLIP-vicuna-7b (we call it as InstructBLIP-7B in this paper) and InstructBLIP-vicuna-13b (we call it as InstructBLIP-13B in this paper). It is worth noting that both InstructBLIP-vicuna-7b and InstructBLIP-vicuna-13b provided no responses to a small part of questions. For these instances, we disregarded these questions and did not assign any scores.

- **LLaVA-1.5** Liu et al. (2023a). LLaVA (Liu et al., 2024c) establishes a connection between the visual encoder ViT-L/14 from CLIP (Radford et al., 2021) and the language decoder LLaMA (Touvron et al., 2023) using a lightweight, fully connected (FC) layer. Initially, the system trains this FC layer with 595K image-text pairs, while keeping both the visual encoder and LLM static. Subsequently, LLaVA fine-tunes both the FC layer and the LLM using a dataset of 158K instructional vision-language pairs. LLaVA-1.5 (Liu et al., 2023a) is an enhanced version of LLaVA, trained on additional datasets. The tested version is LLaVa-1.5-7b and LLaVa-1.5-13b (we call it as LLaVA-1.5-7B and LLaVA-1.5-13B in this paper).

- **Qwen-VL-Chat** (Bai et al., 2023b) introduces a series of large-scale vision-language models (LVLMs) designed to perceive and understand both text and images. The Qwen-VL series is built upon the Qwen-LM (Bai et al., 2023a) foundation, augmented with visual capabilities through a meticulously designed visual receptor, input-output interface, and a three-stage training pipeline using a multilingual multimodal cleaned corpus. These models excel in image description, question-answering, grounding, and text-reading by aligning image-caption-box tuples. The series includes Qwen-VL and Qwen-VL-Chat, both of which achieve state-of-the-art performance on a broad range of visual-centric benchmarks and real-world dialog benchmarks, demonstrating their superiority over existing vision-language chatbots. The tested version is Qwen-VL-Chat, as Qwen-VL-Chat have stronger instruction fellow ability than Qwen-VL.

- **LLaVA-NeXT** (Liu et al., 2024b) is the latest iteration in the LLaVA series (Liu et al., 2024c), building upon the foundation of LLaVA-1.5 (Liu et al., 2023a). This new model enhances visual reasoning, OCR, and world knowledge capabilities. It increases input image resolution to four times more pixels, supporting resolutions up to 672x672, 336x1344, and 1344x336. LLaVA-NeXT features an improved visual instruction tuning data mixture, further enhancing its visual reasoning and OCR capabilities. Additionally, it supports better visual conversations across various scenarios and applications, demonstrating improved world knowledge and logical reasoning. Despite these enhancements, LLaVA-NeXT maintains the minimalist design and data efficiency of its predecessor, utilizing less than 1M visual instruction tuning samples. The tested versions are LLaVA-NeXT-mistral-7b (LLaVA1.6-mistral) and LLaVA-NeXT-vicuna-7b (LLaVA1.6-vicuna), with the base models being Mistral-7b (Jiang et al., 2023) and Vicuna-7b (Chiang et al., 2023), respectively (we call it as LLaVA1.6-mistral and LLaVA1.6-vicuna in this paper).

- **InternVL-Chat-V1-5** (Chen et al., 2024c) is an open-source multimodal large language model with enhanced visual understanding through a continuous learning vision encoder, dynamic high-resolution image processing, and a high-quality bilingual dataset. The tested version is InternVL-Chat-V1-5-quantable (we call it as InternVL-Chat1.5), as the GPU memory size restriction of 40G A6000.

- **Cambrian-1** (Tong et al., 2024) is a family of multimodal large language models (MLLMs) designed with a vision-centric approach. This model series addresses the gap between language models and visual representation learning by thoroughly evaluating various visual representations. Cambrian-1 introduces the Spatial Vision Aggregator (SVA), a dynamic and spatially-aware connector that integrates high-resolution vision features with large language models (LLMs) via cross-attention layers (Dai et al., 2024) while reducing the number of tokens. The tested version is Cambrian-1-8b (we call it as Cambrian-8B in this paper).

- **MiniCPM-V** (Yao et al., 2024). MiniCPM-V2 is a robust multimodal large language model designed for efficient end-side deployment. The model is constructed by integrating SigLip-400M (Zhai et al., 2023) and MiniCPM-2.4B (Hu et al., 2024), connected through a perceiver resampler. The latest iteration, MiniCPM-V2.5, further improves upon

its predecessors. Built on SigLip-400M and Llama3-8B-Instruct with a total of 8B parameters, MiniCPM-V2.5 demonstrates significant performance enhancements over MiniCPM-V2. The most recent and advanced model in the MiniCPM-V series, MiniCPM-V2.6, is built on SigLip-400M and Qwen2-7B (Bai et al., 2023a), also with a total of 8B parameters. MiniCPM-V2.6 shows substantial performance improvements over MiniCPM-Llama3-V2.5 and introduces new capabilities for multi-image and video understanding.

- **MolmoE** (Deitke et al., 2024). MolmoE is a family of open vision-language models developed by the Allen Institute for AI. The Molmo models are trained on PixMo, a dataset of 1 million highly curated image-text pairs, and exhibit state-of-the-art performance among multimodal models of similar size, while remaining fully open-source. MolmoE-1B is a multimodal Mixture-of-Experts large language model (LLM) with 1.5B active and 7.2B total parameters, based on OLMoE-1B-7B-0924 (Muennighoff et al., 2024). It closely matches the performance of GPT-4V on both academic benchmarks and human evaluations, achieving state-of-the-art performance among similarly-sized open multimodal models. Molmo 7B-O, based on OLMo-7B-1024 (a preview of the next generation of OLMo models), utilizes OpenAI CLIP as its vision backbone, performing between GPT-4V and GPT-4o on both academic benchmarks and human evaluations. Molmo 7B-D, based on Qwen2-7B and also using OpenAI CLIP as its vision backbone, performs similarly, sitting between GPT-4V and GPT-4o in both academic benchmarks and human evaluations. It powers the Molmo demo available at molmo.allenai.org. Finally, Molmo 72B, based on Qwen2-72B with OpenAI CLIP as the vision backbone, achieves the highest academic benchmark score and ranks second in human evaluations, trailing GPT-4o by only a small margin.

- **Xinyuan-VL** (Group, 2024). Xinyuan-VL-2B is a high-performance multimodal large model developed by the Cylingo Group for end-user applications. It is fine-tuned based on Qwen2-VL-2B (Wang et al., 2024c) and trained on over 5 million multimodal data samples, with additional training on a small amount of plain text data.

- **Aquila-VL** (Gu et al., 2024). The Aquila-VL-2B model is a VLM trained using the LLava framework. The Qwen2.5-1.5B (Wang et al., 2024c) model serves as the LLM, while siglip-so400m-patch14-384 is employed as the vision tower. The model was trained on our custom-built Infinity-MM dataset, which consists of approximately 40 million image-text pairs. This dataset combines open-source data collected from the internet with synthetic instruction data generated using open-source VLM models.

- **DeepSeek-VL** (Lu et al., 2024a). DeepSeek-VL is an open-source Vision-Language (VL) model designed for real-world applications involving vision and language understanding. It demonstrates broad multimodal capabilities, enabling the processing of logical diagrams, web pages, formula recognition, scientific literature, natural images, and embodied intelligence in complex scenarios. The models we tested, with sizes of 1B and 7B parameters, are respectively sourced from deepseek-ai/deepseek-vl-1.3b-chat and deepseek-ai/deepseek-vl-7b-chat.

**PaliGemma 2** (Steiner et al., 2024) represents a significant advancement in vision-language modeling, building upon its predecessor by incorporating the sophisticated Gemma 2 architecture. Following the approach of PaLI-3 (Chen et al., 2023b), the PaliGemma model family utilizes open-source components, specifically combining the SigLIP vision model with Gemma 2 language models (Team et al., 2024). This multimodal system effectively processes both visual and textual inputs to generate multilingual outputs. The architecture has been optimized to achieve superior fine-tuning performance across a comprehensive range of vision-language tasks, including image and short video captioning, visual question answering, optical character recognition, object detection, and instance segmentation. The model variants include paligemma2-10b-ft-docci-448, paligemma2-3b-ft-docci-448.

- **Claude** (Anthropic, 2024) is a large language model developed by Anthropic with a strong focus on safety, interpretability, and alignment. It has been designed to align with human intent and values, using various methodologies like reinforcement learning from human feedback (RLHF) to ensure that the model behaves as intended. Claude integrates mechanisms to reduce harmful or biased outputs and is optimized for interactive dialogue across a range of domains. In this paper, the tested version is Claude-3-opus, Claude-3-sonnet,

Claude-3-haiku, Claude-3.5-sonnet. Except for Claude-3-haiku, where images were used in their original size, the images for the other three models were resized to $128 \times 128$ due to cost considerations. Notably, both Claude-3-sonnet and Claude-3.5-sonnet demonstrated poor adherence to instructions, often failing to provide direct answers as requested in the prompt, instead offering explanations alongside the options. Therefore, we used GPT-4o to extract the answers for scoring, with the prompt provided in Appendix E.1.

### E.3.2 VIDEO VLM

The following is a description of the specific models reviewed, as well as the specific parameter configurations, can be found in Table 10.

- **Chat-Univi** (Jin et al., 2023) is a unified vision-language model capable of comprehending and engaging in conversations involving both images and videos through a unified visual representation. Chat-Univi employs a set of dynamic visual tokens to uniformly represent images and videos, enabling it to efficiently utilize a limited number of visual tokens to capture the spatial details necessary for images and the comprehensive temporal relationships required for videos. This approach is supported by a multiscale representation that allows the model to perceive both high-level semantic concepts and low-level visual details. The tested version is Chat-Univi-7B and Chat-Univi-13B.
- **Video-LLaVA** (Lin et al., 2023a) is a sophisticated multi-modal framework designed to empower large language models (LLMs) with the ability to understand both visual and auditory content in videos. Unlike previous models that handle either visual or audio signals independently, Video-LLaVA integrates both to achieve comprehensive video comprehension. The tested version is Video-LLaVA-7b (we call it as Video-LLaVA in this paper).
- **PLLaVA** (Xu et al., 2024)(Pooling LLaVA) is an advanced video understanding model designed to extend image-based models to video, enabling dense video caption generation. The model employs a simple yet powerful pooling module to bridge image-finetuned Vision-Language Models (Vision-LLM) into the video domain, achieving significant performance improvements in video captioning tasks. The tested versions are PLLaVA-7B and PLLaVA-13B. Notably, since the model uses num_frame as a parameter during loading, our approach for processing images involved duplicating the image for the num_frames input.

We also evaluated VideoChatGPT (Maaz et al., 2023a), Video-LLaMA (Zhang et al., 2023), and VideoChat2 (Li et al., 2024e), and observed that these models exhibit poor instruction-following capabilities. Despite repeatedly prompting them to provide options, the models consistently responded with descriptions of the video or image content rather than addressing the questions posed. As a result, we did not include specific evaluation metrics for these models in our study.

### E.3.3 GENERAL VLM

The following is a description of the specific models reviewed, as well as the specific parameter configurations, can be found in Table 11, 12.

- **LLaVA-NeXT-Interleave** (Li et al., 2024d) is a multimodal large language model that extends LLaVA's capabilities to handle multi-image, video, and 3D data. By using an interleaved data format, it unifies single-image and multi-modal tasks, enabling it to transfer knowledge across different scenarios. The model is trained on the M4-Instruct dataset, which includes over a million samples spanning multi-image, video, and 3D tasks. The tested versions are llava-interleave-qwen-7b-hf (we call it as LLaVA-interleave) and llava-interleave-qwen-7b-dpo-hf (we call it as LLaVA-interleave-dpo in this paper).
- **VILA-1.5** (Lin et al., 2023b) is a multimodal visual language model (VLM) designed to handle both multi-image and video understanding tasks. It is pretrained on large-scale interleaved image-text data, which enhances its ability to perform tasks such as video reasoning and in-context learning. The tested versions are VILA-1.5-3B, VILA-1.5-3B-s2, VILA-1.5-8B and VILA-1.5-13B.
- **NVILA** (Liu et al., 2024f) is a family of open Vision-Language Models (VLMs) optimized for efficient video and multi-image understanding. We enhance VILA's architecture

| Model | Generation Setup |
|---|---|
| BLIP-2 | torch_dtype = torch.float16, max_new_tokens = 200 |
| InstructBLIP-t5-xl | torch_dtype = torch.float16, max_new_tokens = 200 |
| InstructBLIP-t5-xxl | torch_dtype = torch.float16, max_new_tokens = 200 |
| InstructBLIP-7B | torch_dtype = torch.float16, max_new_tokens = 200 |
| InstructBLIP-13B | torch_dtype = torch.float16, max_new_tokens = 200 |
| LLaVA-1.5-7B | torch_dtype = torch.float16, max_new_tokens = 200, do_sample = False |
| LLaVA-1.5-13B | torch_dtype = torch.float16, max_new_tokens = 200, do_sample = False |
| Qwen-VL-Chat | torch_dtype = torch.bfloat16 |
| LLaVA1.6-mistral | torch_dtype = torch.float16, max_new_tokens = 200, do_sample = False |
| LLaVA1.6-vicuna | torch_dtype = torch.float16, max_new_tokens = 200, do_sample = False |
| InternVL-Chat1.5 | torch_dtype = torch.bfloat16, max_new_tokens = 512, num_beams = 1, do_sample = False |
| Cambrian-8B | torch_dtype = torch.float16, max_new_tokens = 512, temperature = 0, num_beams = 1, use_cache = True |
| MiniCPM-V2 | dtype = torch.float32, context = None, sampling = False, temperature = 0.1, max_new_token = 10 |
| MiniCPM-V2.5 | dtype = torch.float32, context = None, sampling = False, temperature = 0.1, max_new_token = 10 |
| MiniCPM-V2.6 | dtype = torch.float32, context = None, sampling = False, temperature = 0.1, max_new_token = 10 |
| MolmoE-1B | dtype = torch.float32, stop_strings = stop_strings="<\|endoftext\|>", dtype = 'auto', max_new_tokens = 200 |
| MolmoE-7B-O | dtype = torch.float32, stop_strings = stop_strings="<\|endoftext\|>", dtype = 'auto', max_new_tokens = 200 |
| MolmoE-7B-D | dtype = torch.float32, stop_strings = stop_strings="<\|endoftext\|>", dtype = 'auto', max_new_tokens = 200 |
| MolmoE-72B | dtype = torch.float32, stop_strings = stop_strings="<\|endoftext\|>", dtype = 'auto', max_new_tokens = 200 |
| Xinyuan-VL | max_new_tokens = 128, resize = (1024, 1024) |
| Aquila-VL | do_sample = False, temperature = 0, max_new_tokens = 4096 |
| DeepSeek-VL-1B dtype = torch.bfloat16, max_new_tokens = 10, do_sample=False, use_cache=True | |
| DeepSeek-VL-7B dtype = torch.bfloat16, max_new_tokens = 10, do_sample=False, use_cache=True | |
| Claude-3-opus | max_new_tokens = 1000, temperature = 0 |
| Claude-3-sonnet | max_new_tokens = 1000, temperature = 0 |
| Claude-3-haiku | max_new_tokens = 1000, temperature = 0 |
| Claude-3.5-sonnet | max_new_tokens = 1000, temperature = 0 |

Table 9: Generating parameters for Image VLM. Parameters not explicitly stated indicate the use of the model's default system settings.

| Model | Generation Setup |
|---|---|
| Video-LLaVA | torch_dtype = torch.float16, max_new_tokens = 1024, temperature = 0.1, use_cache = True, do_sample = True |
| Chat-Univi-7B | torch_dtype = torch.float16, max_new_tokens = 10, temperature = 0, num_beams = 1, use_cache = True, do_sample = False, top_p = None, output_scores = True, return_dict_in_generate = True, length_penalty = 1 |
| Chat-Univi-13B | torch_dtype = torch.float16, max_new_tokens = 10, temperature = 0, num_beams = 1, use_cache = True, do_sample = False, top_p = None, output_scores = True, return_dict_in_generate = True, length_penalty = 1 |
| PLLaVA-7B | conv_mode = conv_eval_videoqabench, max_new_tokens = 256, do_sample = False |
| PLLaVA-13B | conv_mode = conv_eval_videoqabench, max_new_tokens = 256, do_sample = False |

Table 10: Generating parameters for Video VLM. Parameters not explicitly stated indicate the use of the model's default system settings.

through a 'scale-then-compress' approach, increasing spatial and temporal resolutions before compressing visual tokens, enabling efficient processing of high-resolution images and long videos. Through systematic optimization across training, fine-tuning, and deployment phases, NVILA achieves comparable or superior performance to leading VLMs while reducing training costs by 4.5×, fine-tuning memory by 3.4×, and latency by up to 2.8×. The code and models are publicly available for reproducibility.

- **Phi-3V** (Abdin et al., 2024). Phi-3 is a family of open AI models developed by Microsoft, designed to be the most capable and cost-effective small language models (SLMs) available. Phi-3 models outperform other models of the same size and even those in the next size up across various language, reasoning, coding, and math benchmarks. Phi-3V is the VLM based on Phi-3. The tested version is Phi-3V-128k (we call it as Phi-3V in this paper).

- **Phi-3.5V** (AzureML, 2024). Phi-3.5V, released on November 15, 2024, is an advanced version of Phi-3V. This state-of-the-art, lightweight multimodal model is built on datasets comprising synthetic data and curated, publicly available web content, with a focus on high-quality, reasoning-rich data in both text and vision. As part of the Phi-3 model family, the multimodal version supports a context length of 128K tokens. The model has undergone a comprehensive enhancement process, including supervised fine-tuning and direct preference optimization, to ensure accurate adherence to instructions and robust safety measures.

- **mPLUG-Owl3** (Ye et al., 2024). Here's a refined version of your sentence: mPLUG-Owl3 is a state-of-the-art multimodal large language model designed to address the challenges of understanding long image sequences. mPLUG-Owl3 introduces *Hyper Attention*, a method that enhances the speed of visual sequence processing in multimodal large language models by sixfold, enabling the handling of sequences up to eight times longer. Simultaneously, mPLUG-Owl3 maintains exceptional performance across single-image, multi-image, and video tasks. The tested versions are mPLUG-Owl3-1B (mPLUG-Owl3-1B-241014), mPLUG-Owl3-2B (mPLUG-Owl3-2B-241014) and mPLUG-Owl3-7B (mPLUG-Owl3-7B-241101).

- **InternVL2** (Wang et al., 2024e). InternVL 2.0 is the latest iteration in the InternVL series of multimodal large language models. It includes a range of instruction-tuned models, with parameter sizes ranging from 1 billion to 108 billion. In comparison to state-of-the-art open-source multimodal large language models, InternVL 2.0 outperforms most open-source alternatives and demonstrates competitive performance that rivals proprietary commercial models. Its capabilities include document and chart comprehension, infographics question answering, scene text understanding and OCR tasks, scientific and mathematical problem-solving, as well as cultural understanding and integrated multimodal processing. InternVL 2.0 is trained with an 8K context window and incorporates training data consisting of long texts, multiple images, and videos. This training significantly enhances its ability to process and understand these types of inputs, surpassing the capabilities of InternVL

1.5 (Chen et al., 2023c). For larger model variants, we implement a merge-based approach rather than sequential processing for video data to optimize GPU memory consumption, as demonstrated in Table 28.

- **InternVL 2.5** (Gao et al., 2024b) was released on December 9, 2024, representing an advanced iteration in the multimodal large language model (MLLM) series. While maintaining the core architecture of InternVL 2.0, it introduces significant enhancements in training strategies, testing methodologies, and data quality. The model preserves the "ViT-MLP-LLM" paradigm established by its predecessors, InternVL 1.5 and 2.0. This new version integrates a newly incrementally pre-trained InternViT with various pre-trained LLMs, including InternLM 2.5 and Qwen 2.5, utilizing a randomly initialized MLP projector. Consistent with previous implementations, InternVL 2.5 employs a pixel unshuffle operation, reducing visual tokens to one-quarter of their original count, and adopts a dynamic resolution strategy similar to InternVL 1.5, processing images in 448×448 pixel tiles. A significant advancement since InternVL 2.0 is the expansion of capabilities to include multi-image and video data processing. For larger model variants, we implement a merge-based approach rather than sequential processing for video data to optimize GPU memory consumption, as demonstrated in Table 28.

- **LLaVA-NeXT-Video** (Zhang et al., 2024b) is a multimodal large language model designed to excel in video understanding tasks through zero-shot modality transfer. Trained primarily on image data, it demonstrates impressive performance on video tasks by leveraging deep learning models with DPO training and AI feedback. The model supports various deployment scenarios, from cloud environments to edge devices, making it highly versatile. It is part of the broader LLaVA-NeXT suite, focused on advancing visual-language integration. The tested version are LLaVA-NeXT-Video-7B-Qwen and LLaVA-NeXT-Video-7B-Qwen-dpo, we call them as LLaVA-NV and LLaVA-NV-dpo respectively in this paper.

- **Mantis** (Jiang et al., 2024) is a multimodal large language model designed for interleaved multi-image tasks. Built on the LLaMA-3 architecture, it excels in co-reference, reasoning, comparison, and temporal understanding. Mantis uses the Mantis-Instruct dataset, containing 721K examples, to train on various multi-image skills. It achieves state-of-the-art performance on some interleaved benchmarks, while maintaining strong single-image performance on par with CogVLM (Wang et al., 2023b). The tested version are Mantis-Idefics2, Mantis-LLaVA, Mantis-siglip-llama3 and Mantis-clip-llama3.

- **GPT** (Achiam et al., 2023). GPT-4V is an advanced multimodal model that extends GPT-4's capabilities with integrated vision processing, allowing it to understand and generate text based on visual inputs. GPT-4o is an optimized variant designed for better performance in language tasks while maintaining lower computational requirements. GPT-4o-mini is a more lightweight version of GPT-4o, designed for deployment in resource-constrained environments while still providing strong performance in language understanding and generation tasks. The version of GPT-4V used is GPT-4-turbo. Notably, due to token length limitations, all images were resized to 512×512 before being input into GPT. Our testing was conducted around mid-August 2024, using the most advanced model available at that time. The remaining models were also tested during this period.

- **Gemini-1.5** (Team et al., 2023). Gemini-1.5-Flash and Gemini-1.5-Pro are advanced multimodal large language models, each designed with distinct capabilities for different performance needs. Gemini-1.5-Flash emphasizes fast processing and efficient memory usage, making it ideal for tasks requiring speed on devices with limited resources. Our testing of Gemini-1.5-Flash and Gemini-1.5-Pro was conducted around mid-August 2024.

### E.4 PROMPT FOR VLM TEST

Referring to Liu et al. (2024c), during testing, we appended an end prompt to each question-answer pair (i.e., the value corresponding to the "question" key in the Figure 11). The end prompt is as follows:

Answer with the option's letter from the given choices directly. You can only answer one letter from A, B, C, or D.

Consistent with Zhang et al. (2024a), when asking video-related questions to Image VLMs, we prepend the prompt with the following statement:

> This is a series of images sampled at equal intervals from the beginning to the end of a video. Based on the series of images, output the best option for the question.

For Video VLMs, due to their weaker instruction-following capabilities, we provided additional prompts during testing to guide the models in selecting the correct answer rather than simply describing the video content. (Despite this, many models still responded with video descriptions rather than answering the questions, as noted in Appendix E.3.2.) For Video LLMs, if the question involves only a single image input, we prepend the following statement:

> Based on the image, output the best option for the question. You must only output the option.

Add the last line in the prompt:

> The best choice option is:

If the input involves only a single video, we prepend the following statement:

> This is a series of images sampled at equal intervals from the beginning to the end of a video. Based on the series of images, answer the question. Based on the video, output the best option for the question. You must only output the option.

## E.5 REFERENCE DATASETS SUMMARY

Our dataset was entirely annotated by humans and underwent two rounds of rigorous filtering and screening. The data sources we utilized, along with their respective uses, are detailed in Table 13.

## E.6 PROMPT STRATEGIES

In Section 4.1, we explore several prompting strategies:Chain of Thought (CoT) (Kojima et al., 2023) with the prompt "Let's think step-by-step!"; Desp-CoT (Wu et al., 2023d), which begins with an image description prompt; and Pure Language Reasoning (PLR), similar to Mathvista (Lu et al., 2024b). The specific prompt content can be found in Appendix E.6. Following the settings of Kojima et al. (2022); Chen et al. (2024b), we extract the final generated answer through GPT-4o-mini. In this section, we provide the prompts used to implement these three strategies. Note that for Phi-3V:

(1.1) The prompt for CoT (Kojima et al., 2023) is:

> Let's think step by step! Start by selecting the correct option's letter from the given choices, then provide a detailed explanation of your thought process.

(1.2) The prompt for Desp-CoT (Wu et al., 2023d) is:

> Each image or video is followed by a description, which you can refer to. Answer with the option's letter from the given choices directly.

(1.3) The prompt for PLR is:

> Each image or video is replaced by description. Answer with the option's letter from the given choices directly.

For GPT-4o:

(2.1) The prompt for CoT (Kojima et al., 2023) is:

> Let's think step by step! Start by selecting the correct option's letter from the given choices, then provide a detailed explanation of your thought process.

(2.2) The prompt for Desp-CoT (Wu et al., 2023d) is:

> Each image or video is followed by a description, which you can refer to. Answer with the option's letter from the given choices directly.

(2.3) The prompt for PLR is:

> Each image or video is replaced by description. Answer with the option's letter from the given choices directly.

We used the following prompts to generate the corresponding descriptions for each model (for image descriptions, "video" is replaced with "image" where applicable):

> Give me the description of this video.

### E.7 VIPERGPT IMPLEMENTATION

Following the methodology described in ContPhy (Zheng et al., 2024b), since the code for its model, ContPro, has not been released, we implemented an oracle neural-symbolic model using ViperGPT (Surs et al., 2023), which we refer to as ContPhy, and evaluated it on the PhysBench. Similar to ContPhy (Zheng et al., 2024b), we decomposed the question-answering task into four main modules: video perception, physical simulation, program parser, and symbolic execution.

Given a raw video, the video perception module detects objects and their associated static attributes using the MASK-RCNN detector (He et al., 2017). The physical simulator takes point clouds as input and predicts object dynamics across various scenarios using dynamic prediction models (Li et al., 2018; Sulsky et al., 1995). The program parser, powered by a large language model (we use GPT-4o), translates the question query into executable Python programs. Based on the detected object attributes and predicted dynamics, the symbolic executor runs the programs to derive the answer to the question.

Since neuro-symbolic approaches rely on pre-trained domain-specific modules to extract objects' static attributes, physical properties, and dynamic trajectories, this method is not suitable for the complex and diverse tasks in PhysBench. For instance, in ContPhy, simulating four types of tasks—rope, cloth, ball, and fluid—requires pre-training three models: MASK R-CNN (He et al., 2017) based on Detectron2 (Wu et al., 2019), DPI-Net (Li et al., 2018), and Material Point Method (MPM) (Sulsky et al., 1995). Given the variety of tasks in PhysBench, where even individual tasks within certain subcategories can differ significantly, it is impractical to design and call a specific module for every task. Therefore, we designed an Oracle model for the QA pairs generated through specific simulations to ensure that the tasks are similar enough to allow the use of consistent logical processing templates following Zheng et al. (2024b), while GPT-4o directly provided answers for the remaining questions without invoking additional modules or running Python code.

Following the approach in ContPhy, we use ResNet-50 (He et al., 2016) as the backbone for MASK R-CNN to densely detect object locations in each frame and associate static attributes such as color and material. We use the default config from Detectron2, while the number of classes is different across scenarios. Specifically, the batch size is 16 for 8 GPUs thus each mini-batch has 2 images per GPU. We train the model for about $10k$ iterations, with a learning rate of $0.01$. For image size, we keep the original resolution.

We fine-tune the network using the training set data from all simulations. For scenes involving fluid and soft-body dynamics, we adopt MPM, while DPI-Net is used for other tasks. The training data for these two modules is similar to that of MASK R-CNN, and the overall training procedure is comparable to ContPhy. After extracting objects' static attributes, physical properties, and dynamic trajectories, and parsing the natural language query into an executable program, we run the program using the object states as input and produce the predicted answer. Due to the rich diversity and versatile resources of our dataset, it is challenging to ensure that templates perfectly fit all predefined modules. For example, some videos become significantly distorted when rescaled into a square format, and in certain videos, fluids are difficult to extract as points due to their similar color to the environment. For the specific training parameters of MPM and DPI-Net, we have kept them consistent with ContPhy.

### E.8 HUMAN PERFORMANCE

To evaluate human performance on PhysBench, we recruited 12 graduate students in STEM fields and provided monetary compensation for their participation. Each question was assigned to all annotators. To ensure the quality of the results, we followed the methodology of MathVista (Lu et al., 2024b) by implementing qualification questions during participant recruitment. These questions tested basic knowledge of physical world concepts, and only those who answered the qualification questions correctly were deemed eligible for the study. Given the large number of questions in PhysBench, the testing was divided into 10 sessions, delivered as online questionnaires with no time constraints for completion. The average score across all participants was used as the final measure of human performance.

## F MORE EXPERIMENTS RESULTS

### F.1 GROUNDINGDINO CONFIGURATION

GroundingDINO has two key parameters: box_threshold and text_threshold. The box_threshold parameter is used to filter predictions based on the confidence level of the detected bounding boxes, ensuring that only boxes with confidence scores above this threshold are retained. On the other hand, the text_threshold parameter filters predictions based on their relevance to the input text prompt, retaining only those that meet or exceed this threshold.

It is important to note that the filtered phrases may contain a significant number of duplicates. To address this, we consider a prediction successful only if the set of output phrases matches exactly with the set of input phrases, ensuring both completeness and accuracy. The detailed experimental results are presented in Table 14.

### F.2 EFFECT OF VISUAL PROMPTING

To enhance the VLM's responsiveness to visual prompts, we assessed its sensitivity to these prompts. The images used in our tests were all 1024×1024 pixels. We employed two different annotation methods for depth and attribute sub-tasks, as illustrated in Figure 23, drawing on BLINK (Fu et al., 2024) for reference. For depth, the default color for circles is red. Method (a) follows the approach introduced in the BLINK paper, while the second method aligns exactly with the instances in BLINK-eval. Additionally, the prompts for both methods are largely similar to those in the BLINK-eval examples. Similarly, for attributes, we experimented with two annotation methods, denoted as (c) and (d) in Figure 23. However, since the white text doesn't look very clear in the yellow checkbox in the first method, the yellow color was selected.

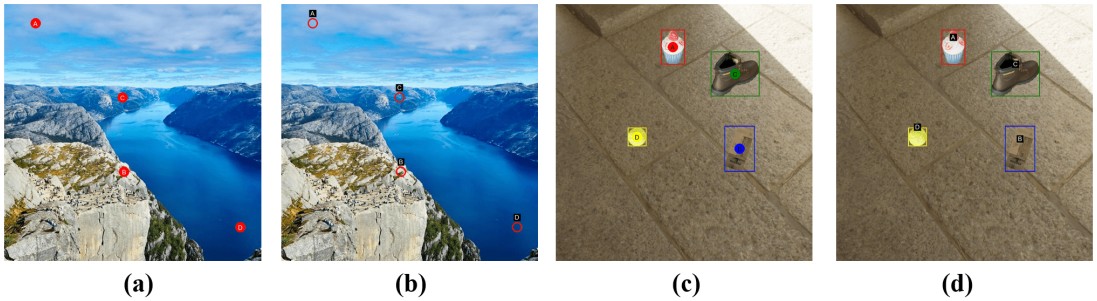

| (a) | (b) | (c) | (d) |

Figure 23: **Several different labeling methods**. (a) (b) are two methods for labeling depth. (c) (d) are two labeling methods for attributes. Attribute also try (a)(b)

The test images comprise 1000 samples, with a uniform circle radius of 20 pixels and a font size of 25 pixels. The detailed experimental results are presented in Table 15. For samples (a) and (b), the text prompts require selecting the location or object indicated by the prompt, while samples (c) and

(d) involve identifying a bounding box or point. For instance, answers for (c) and (d) might be "A. The object enclosed in the red box at point A" using both color and letter identifiers.

Based on the experimental results, we find that the outcomes for (a) and (b) are similar in terms of depth and attribute, though (a) performs slightly better. For attributes, (c) and (d) yield results close to random, likely because the tested models cannot perceive color differences in the bounding boxes, thus failing to answer effectively. However, (a) and (b) significantly improve accuracy. Therefore, we ultimately decided to use the annotation method in (a).

Building on the findings of the previous experiment, we further tested the impact of varying circle sizes using the annotation method, which showed the best model performance. The corresponding results are detailed in Table 16. Additionally, the text size was consistently 5 pixels larger than the center of the circle. Based on the previous experiment, we adopted the annotation method (a) for all subsequent annotations. Therefore, we used method (a) to test depth, with a sample size of 1000 images.

We have tested 6 scales as shown in Figure 24 and the results indicate that for LLaVA-1.5, larger sizes generally yield better results. For Phi-3V and VILA-1.5, the optimal radius is 30 pixels, although performance does not show a significant relationship on either side of 30 pixels. Most models show low sensitivity to size variations, except for LLaVA-1.5-7b. To balance performance and aesthetics, we ultimately selected a radius size of 30 pixels.

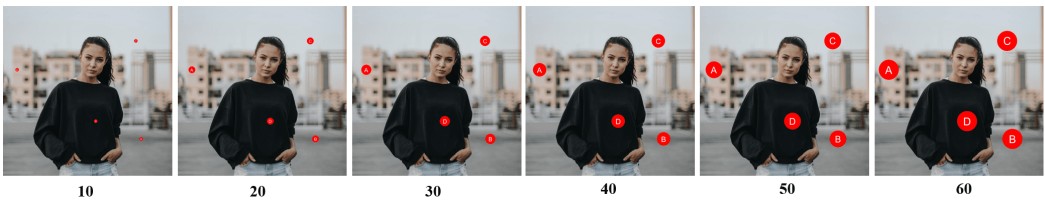

Figure 24: Visual Prompt with Different Size.

### F.3    MORE PHYSBENCH RESULTS

The performance of 39 models across 8 ability categories in PhysBench is presented in Table 17. Note that the ability classifications for the physical scene understanding and physics-based dynamics categories are identical, though their content differs slightly; these are combined here for clarity. Detailed descriptions of the ability classifications can be found in Appendix C.1, while the remaining 36 newly tested models are listed in Appendix K.

The details for Ability Type, Physical Object Property, Physical Object Relationships, Physical Scene Understanding, and Physics-based Dynamics can be found in Table 17,Table 18,Table 19,Table 20,Table 21, respectively.

| Model | Generation Setup |
|---|---|
| LLaVA-interleave | torch_dtype = torch.float16, max_new_tokens = 10, do_sample = False |
| LLaVA-interleave-dpo | torch_dtype = torch.float16, max_new_tokens = 10, do_sample = False |
| VILA-1.5-3B | torch_dtype = torch.float16, max_new_tokens = 4000, temperature = 0.1, num_beams = 1, use_cache = False, do_sample = False, top_p = None |
| VILA-1.5-3B-s2 | torch_dtype = torch.float16, max_new_tokens = 4000, temperature = 0.1, num_beams = 1, use_cache = False, do_sample = False, top_p = None |
| VILA-1.5-8B | torch_dtype = torch.float16, max_new_tokens = 4000, temperature = 0.1, num_beams = 1, use_cache = False, do_sample = False, top_p = None |
| VILA-1.5-13B | torch_dtype = torch.float16, max_new_tokens = 4000, temperature = 0.1, num_beams = 1, use_cache = False, do_sample = False, top_p = None |
| Phi-3V | torch_dtype = torch.float16, max_new_tokens = 500, temperature = 0, do_sample = False, length_penalty = 1, repetition_penalty = 1 (When in error analysis the max_new_tokens is set to 5000) |
| Phi-3.5V | torch_dtype = torch.float16, max_new_tokens = 500, temperature = 0, do_sample = False, length_penalty = 1, repetition_penalty = 1 (When in error analysis the max_new_tokens is set to 5000) |
| LLaVA-NV | torch_dtype = torch.float16, max_new_tokens = 10 |
| LLaVA-NV-dpo | torch_dtype = torch.bfloat16, max_new_tokens = 10 |
| Mantis-Idefics2 | torch_dtype = torch.bfloat16, max_new_tokens = 10, do_sample = False |
| Mantis-LLaVA | torch_dtype = torch.bfloat16, max_new_tokens = 1, num_beams = 1, do_sample = False, length_penalty = 1, repetition_penalty = 1 |
| Mantis-siglip-llama3 | torch_dtype = torch.bfloat16, max_new_tokens = 1, num_beams = 1, do_sample = False |
| Mantis-clip-llama3 | torch_dtype = torch.bfloat16, max_new_tokens = 1, num_beams = 1, do_sample = False |
| mPLUG-Owl3-1B | max_new_tokens = 100, decode_text = True |
| mPLUG-Owl3-2B | max_new_tokens = 100, decode_text = True |
| mPLUG-Owl3-7B | max_new_tokens = 100, decode_text = True |
| InternVL2-1B | dtype = torch.bfloat16, max_new_tokens = 10, do_sample = False, max_num = 12 |
| InternVL2-2B | dtype = torch.bfloat16, max_new_tokens = 10, do_sample = False, max_num = 12 |
| InternVL2-4B | dtype = torch.bfloat16, max_new_tokens = 10, do_sample = False, max_num = 12 |
| InternVL2-8B | dtype = torch.bfloat16, max_new_tokens = 10, do_sample = False, max_num = 12 |
| InternVL2-26B | dtype = torch.bfloat16, max_new_tokens = 10, do_sample = False, max_num = 1 |
| InternVL2-40B | dtype = torch.bfloat16, max_new_tokens = 10, do_sample = False, max_num = 1 |
| InternVL2-76B | dtype = torch.bfloat16, max_new_tokens = 10, do_sample = False, max_num = 1 |
| GPT-4o | max_new_tokens = 300, temperature = 0, seed = 42 (When in error analysis the max_new_tokens is set to 2000) |
| GPT-4o-mini | max_new_tokens = 300, temperature = 0, seed = 42 |
| GPT-4V | max_new_tokens = 300, temperature = 0, seed = 42 |
| Gemini-1.5-flash | stream = True |
| Gemini-1.5-pro | stream = True |

Table 11: Generating parameters for General VLM. Parameters not explicitly stated indicate the use of the model's default system settings.

| Model | Generation Setup |
|-------|------------------|
| InternVL2.5-1B | dtype = torch.bfloat16, max_new_tokens = 10, do_sample = False, max_num = 12 |
| InternVL2.5-2B | dtype = torch.bfloat16, max_new_tokens = 10, do_sample = False, max_num = 12 |
| InternVL2.5-4B | dtype = torch.bfloat16, max_new_tokens = 10, do_sample = False, max_num = 12 |
| InternVL2.5-8B | dtype = torch.bfloat16, max_new_tokens = 10, do_sample = False, max_num = 12 |
| InternVL2.5-26B | dtype = torch.bfloat16, max_new_tokens = 10, do_sample = False, max_num = 1 |
| InternVL2.5-38B | dtype = torch.bfloat16, max_new_tokens = 10, do_sample = False, max_num = 1 |
| InternVL2.5-78B | dtype = torch.bfloat16, max_new_tokens = 10, do_sample = False, max_num = 1 |
| NVILA-8B | default |
| NVILA-15B | default |
| NVILA-Lite-8B | default |
| NVILA-Lite-15B | default |

Table 12: Generation parameters for more models (continued from Table 11). All other parameters not specified here use the default model configurations.

Table 13: Datasets Reference for our PhysBench. 'None' indicates that the dataset does not explicitly state which license is being used.

| Dataset | License | Description of usage |
|---------|---------|----------------------|
| Unsplash Ali et al. (2023) | https://unsplash.com/license | We crawled some images for annotation and labeling. |
| ContPhy Zheng et al. (2024b) | CC-BY 4.0 | We used his code to generate some of the simulation data. |
| ChronoMagic-Bench Yuan et al. (2024) | Apache 2.0 | As source data, we annotated some QA. |
| DROID Khazatsky et al. (2024) | CC-BY 4.0 | As source data, we annotated some QA data belonging to physics-based dynamics–manuplation type. |
| Ego4D Grauman et al. (2022) | CC-BY 4.0 | As source data, we annotated some QA data belonging to physics-based dynamics–manuplation type. |
| MimicPlay Wang et al. (2023a) | CC-BY 4.0 | As source data, we annotated some QA data belonging to physics-based dynamics–manuplation type. |
| nuScenes Caesar et al. (2019) | CC BY-NC-SA 4.0 | As source data, we annotated some QA data belonging to spatial movement type. |
| Netwon Wang et al. (2023c) | CC-BY 4.0 | As reference data for us to create the attribute table for objects. |
| FunKPoint Lai et al. (2021) | None | As source data, we annotated some QA data belonging to physics-based dynamics–manuplation type. |
| PhotoTourism Snavely et al. (2006) | None | As source data, we annotated some QA data belonging to physical scene understanding–camera type. |

Table 14: **GroundingDINO Accuracy with different box threshold and text threshold**. The number of samples tested is 1000.

| box threshold \ text threshold | 0.1 | 0.2 | 0.3 | 0.4 | 0.5 | 0.6 | 0.7 |
|--------------------------------|-----|-----|-----|-----|-----|-----|-----|
| 0.1 | 0.015 | 0 | 0 | 0 | 0 | 0 | 0 |
| 0.2 | 0.015 | 0.075 | 0.005 | 0.005 | 0 | 0 | 0 |
| 0.3 | 0.01 | 0.03 | 0.06 | 0.01 | 0.005 | 0 | 0 |
| 0.4 | 0.005 | 0.015 | 4 | 4 | 0 | 0 | 0 |
| 0.5 | 0.005 | 0.005 | 0 | 0 | 0 | 0 | 0 |
| 0.6 | 0.005 | 0 | 0 | 0 | 0 | 0 | 0 |
| 0.7 | 0.005 | 0 | 0 | 0 | 0 | 0 | 0 |

Table 15: **Comparison of Visual Prompts for Depth and Attribute Annotation**.

| | Depth | | Attribute | | | |
|---|-------|-------|-----------|-----------|-----------|-----------|
| | Prompt (a) | Prompt (b) | Prompt (a) | Prompt (b) | Prompt (c) | Prompt(d) |
| LLaVA-1.5-7b | 27.3 | 27.2 | 31.5 | 29.9 | 25.3 | 25.2 |
| LLaVA-1.5-13b | 21.2 | 20.2 | 41.1 | 38.5 | 26.0 | 25.6 |
| Phi-3V-128k | 78.2 | 75.8 | 41.2 | 40.1 | 24.8 | 25.1 |
| VILA-1.5-8b | 19.9 | 20.3 | 27.9 | 26.1 | 25.9 | 24.7 |

Table 16: **Effects of different visual prompt circle radius on Depth task performance**.

|              | 10   | 20   | 30   | 40   | 50   | 60   |
|--------------|------|------|------|------|------|------|
| LLaVA-1.5-7b | 19.2 | 27.3 | 27.3 | 28.4 | 30.9 | 31.2 |
| LLaVA-1.5-13b| 19.9 | 21.2 | 24.9 | 25.2 | 25.8 | 27.9 |
| Phi-3V-128k  | 74.7 | 78.2 | 78.8 | 76.8 | 73.1 | 73.0 |
| VILA-1.5-8b  | 20.3 | 19.9 | 26.9 | 25.3 | 26.7 | 22.2 |

| Model | Identify | Comparison | Static | Dynamic | Perception | Prediction | Judgment | Reasoning |
|-------|----------|------------|--------|---------|------------|------------|----------|-----------|
| *Image VLM* | | | | | | | | |
| InstructBLIP-t5-xl Dai et al. (2024) | 48.30 | 25.39 | 36.38 | 35.76 | 40.06 | 34.07 | 42.74 | 35.83 |
| InstructBLIP-t5-xxl Dai et al. (2024) | 57.63 | 28.40 | 33.21 | 53.64 | 41.90 | 32.62 | 47.58 | 36.28 |
| InstructBLIP-7B Dai et al. (2024) | 25.02 | 19.31 | 30.98 | 24.50 | 32.21 | 23.74 | 29.91 | 18.28 |
| InstructBLIP-13B Dai et al. (2024) | 39.49 | 25.70 | 32.46 | 35.33 | 35.88 | 27.12 | 33.62 | 20.18 |
| BLIP-2 Li et al. (2023c) | 55.97 | 30.78 | 36.01 | 53.64 | 42.51 | 33.24 | 48.15 | 34.29 |
| LLaVA-1.5-7B Liu et al. (2023a) | 47.78 | 31.26 | 40.30 | 45.70 | 47.40 | 39.90 | 45.01 | 37.38 |
| LLaVA-1.5-13B Liu et al. (2023a) | 52.48 | 32.70 | 42.16 | 42.38 | 51.17 | 40.87 | 43.02 | 32.65 |
| LLaVA1.6-mistral Liu et al. (2024b) | 36.79 | 23.82 | 23.13 | 21.19 | 29.15 | 15.93 | 28.21 | 4.97 |
| LLaVA1.6-vicuna Liu et al. (2024b) | 52.48 | 30.78 | 65.49 | 42.38 | 49.75 | 39.08 | 43.30 | 37.13 |
| Qwen-VL-Chat Bai et al. (2023b) | 51.44 | 23.75 | 44.96 | 39.74 | 45.16 | 37.09 | 50.14 | 24.25 |
| InternVL-Chat1.5 Chen et al. (2024c) | 64.78 | 43.96 | 74.81 | 56.29 | 53.62 | 41.35 | 44.16 | 34.39 |
| Cambrian-8B Tong et al. (2024) | 28.60 | 18.57 | 20.52 | 13.25 | 22.02 | 33.65 | 11.68 | 25.65 |
| Claude-3-opus Anthropic (2024) | 53.71 | 32.70 | 35.82 | 58.28 | 43.93 | 32.90 | 43.30 | 27.83 |
| Claude-3-sonnet Anthropic (2024) | 47.78 | 29.97 | 36.75 | 52.32 | 38.43 | 36.47 | 43.02 | 30.22 |
| Claude-3-haiku Anthropic (2024) | 53.71 | 34.95 | 52.05 | 60.93 | 47.20 | 35.71 | 43.87 | 27.73 |
| Claude-3.5-sonnet Anthropic (2024) | 61.46 | 34.33 | 36.19 | 58.94 | 43.32 | 34.00 | 47.01 | 25.15 |
| *Video VLM* | | | | | | | | |
| Video-LLaVA Lin et al. (2023a) | 47.08 | 29.15 | 35.45 | 35.10 | 42.71 | 40.59 | 43.59 | 31.16 |
| Chat-Univi-7B Jin et al. (2023) | 25.46 | 14.54 | 18.10 | 31.79 | 24.16 | 29.74 | 31.05 | 18.29 |
| Chat-Univi-13B Jin et al. (2023) | 6.10 | 3.07 | 12.50 | 7.95 | 13.86 | 9.00 | 9.69 | 17.10 |
| PLLaVA-7B Xu et al. (2024) | 47.95 | 30.31 | 31.72 | 49.67 | 42.71 | 39.42 | 40.46 | 34.74 |
| PLLaVA-13B Xu et al. (2024) | 50.39 | 31.60 | 36.94 | 43.71 | 44.44 | 39.77 | 41.60 | 29.17 |
| *General VLM + Interleaved Data* | | | | | | | | |
| LLaVA-interleave Li et al. (2024d) | 59.37 | 37.68 | 44.21 | 49.02 | 47.40 | 39.07 | 28.52 | 32.74 |
| LLaVA-interleave-dpo Li et al. (2024d) | 61.73 | 37.41 | 42.90 | 36.60 | 47.81 | 40.84 | 28.77 | 31.40 |
| VILA-1.5-3B Lin et al. (2023b) | 38.01 | 28.46 | 32.15 | 40.52 | 39.65 | 36.15 | 32.99 | 33.65 |
| VILA-1.5-3B-s2 Lin et al. (2023b) | 38.71 | 29.22 | 29.12 | 39.87 | 39.04 | 30.97 | 31.71 | 34.99 |
| VILA-1.5-8B Lin et al. (2023b) | 39.06 | 29.15 | 28.97 | 39.87 | 37.41 | 42.48 | 21.23 | 30.54 |
| VILA-1.5-13B Lin et al. (2023b) | 50.74 | 32.90 | 39.25 | 47.71 | 44.24 | 35.94 | 31.97 | 29.44 |
| Phi-3V Abdin et al. (2024) | 55.01 | 34.68 | 36.64 | 52.94 | 47.81 | 35.06 | 32.35 | 32.74 |
| LLaVA-NV Zhang et al. (2024b) | 47.78 | 31.26 | 29.85 | 38.56 | 42.41 | 38.76 | 31.63 | 32.22 |
| LLaVA-NV-dpo Zhang et al. (2024b) | 47.52 | 32.22 | 45.28 | 40.52 | 43.53 | 38.35 | 32.01 | 31.74 |
| Mantis-Idefics2 Jiang et al. (2024) | 55.36 | 31.88 | 41.18 | 41.83 | 42.92 | 36.76 | 33.25 | 27.00 |
| Mantis-LLaVA Jiang et al. (2024) | 59.81 | 32.76 | 29.28 | 41.18 | 41.18 | 36.28 | 30.43 | 33.75 |
| Mantis-siglip-llama3 Jiang et al. (2024) | 55.01 | 32.63 | 32.72 | 30.72 | 42.41 | 40.84 | 31.46 | 34.47 |
| Mantis-clip-llama3 Jiang et al. (2024) | 52.40 | 31.60 | 34.71 | 36.60 | 40.16 | 42.75 | 31.33 | 30.68 |
| GPT-4V Achiam et al. (2023) | 65.13 | 37.41 | 44.05 | 64.71 | 55.15 | 34.99 | 41.43 | 24.41 |
| GPT-4o Achiam et al. (2023) | 71.23 | 45.73 | 64.20 | 71.90 | 56.37 | 42.75 | 43.99 | 28.63 |
| GPT-4o-mini Achiam et al. (2023) | 74.46 | 36.86 | 42.90 | 60.13 | 55.45 | 41.25 | 35.04 | 27.96 |
| Gemini-1.5-flash Team et al. (2023) | 74.89 | 43.55 | 51.51 | 60.78 | 54.23 | 39.05 | 33.76 | 31.69 |
| Gemini-1.5-pro Team et al. (2023) | 72.28 | 45.46 | 63.57 | 65.36 | 53.41 | 37.92 | 37.08 | 34.85 |

Table 17: **Evaluation Results for 39 VLMs Categorized by Ability Dimensions**. PhysBench includes two evaluation dimensions: ability and task. Table 3 presents results categorized by the task-type dimension.

| Model | Number | Mass | Color | Attribute |
|---|---|---|---|---|
| Image VLM | | | | |
| InstructBLIP-t5-xl Dai et al. (2024) | 47.14 | 22.18 | 44.33 | 31.37 |
| InstructBLIP-t5-xxl Dai et al. (2024) | 52.86 | 43.00 | 54.00 | 33.22 |
| InstructBLIP-7B Dai et al. (2024) | 11.23 | 34.93 | 35.57 | 20.66 |
| InstructBLIP-13B Dai et al. (2024) | 29.12 | 28.67 | 43.33 | 30.77 |
| BLIP-2 Li et al. (2023c) | 51.30 | 50.85 | 47.33 | 34.63 |
| LLaVA-1.5-7B Liu et al. (2023a) | 38.99 | 36.86 | 52.67 | 35.42 |
| LLaVA-1.5-13B Liu et al. (2023a) | 41.59 | 52.22 | 56.67 | 35.63 |
| LLaVA1.6-mistral Liu et al. (2024b) | 34.49 | 32.42 | 32.67 | 26.76 |
| LLaVA1.6-vicuna Liu et al. (2024b) | 41.07 | 35.15 | 61.33 | 36.41 |
| Qwen-VL-Chat Bai et al. (2023b) | 48.35 | 21.50 | 51.33 | 30.45 |
| InternVL-Chat1.5 Chen et al. (2024c) | 53.73 | 57.68 | 71.00 | 48.12 |
| Cambrian-8B Tong et al. (2024) | 38.13 | 14.68 | 25.67 | 18.52 |
| Claude-3-opus Anthropic (2024) | 45.75 | 54.95 | 50.33 | 35.77 |
| Claude-3-sonnet Anthropic (2024) | 38.30 | 44.71 | 50.00 | 33.64 |
| Claude-3-haiku Anthropic (2024) | 40.73 | 53.24 | 62.00 | 38.18 |
| Claude-3.5-sonnet Anthropic (2024) | 56.67 | 58.02 | 60.33 | 36.69 |
| Video VLM | | | | |
| Video-LLaVA Lin et al. (2023a) | 38.13 | 28.33 | 53.00 | 34.49 |
| Chat-Univi-7B Jin et al. (2023) | 22.70 | 22.87 | 23.00 | 16.32 |
| Chat-Univi-13B Jin et al. (2023) | 2.43 | 3.07 | 4.00 | 5.32 |
| PLLaVA-7B Xu et al. (2024) | 35.36 | 48.46 | 56.33 | 33.07 |
| PLLaVA-13B Xu et al. (2024) | 34.84 | 47.44 | 65.00 | 35.06 |
| General VLM + Interleaved Data | | | | |
| LLaVA-interleave Li et al. (2024d) | 51.30 | 47.44 | 65.33 | 41.66 |
| LLaVA-interleave-dpo Li et al. (2024d) | 55.29 | 41.30 | 67.33 | 42.23 |
| VILA-1.5-3B Lin et al. (2023b) | 39.17 | 29.01 | 27.00 | 31.37 |
| VILA-1.5-3B-s2 Lin et al. (2023b) | 38.82 | 30.72 | 28.00 | 32.29 |
| VILA-1.5-8B Lin et al. (2023b) | 39.86 | 24.23 | 33.33 | 32.72 |
| VILA-1.5-13B Lin et al. (2023b) | 48.01 | 36.52 | 40.00 | 38.47 |
| Phi-3V Abdin et al. (2024) | 43.15 | 37.88 | 63.00 | 40.81 |
| LLaVA-NV Zhang et al. (2024b) | 35.70 | 38.57 | 55.67 | 35.70 |
| LLaVA-NV-dpo Zhang et al. (2024b) | 36.40 | 39.59 | 54.67 | 36.34 |
| Mantis-Idefics2 Jiang et al. (2024) | 47.49 | 32.76 | 65.33 | 36.69 |
| Mantis-LLaVA Jiang et al. (2024) | 58.41 | 40.27 | 61.33 | 35.98 |
| Mantis-siglip-llama3 Jiang et al. (2024) | 45.41 | 37.88 | 61.00 | 38.11 |
| Mantis-clip-llama3 Jiang et al. (2024) | 42.29 | 40.96 | 57.67 | 36.05 |
| GPT-4V Achiam et al. (2023) | 52.51 | 48.81 | 71.00 | 43.79 |
| GPT-4o Achiam et al. (2023) | 61.18 | 54.27 | 71.00 | 52.52 |
| GPT-4o-mini Achiam et al. (2023) | 69.67 | 50.85 | 72.33 | 43.29 |
| Gemini-1.5-flash Team et al. (2023) | 72.10 | 54.95 | 72.33 | 48.65 |
| Gemini-1.5-pro Team et al. (2023) | 64.99 | 58.02 | 75.00 | 50.04 |

Table 18: Evaluation Results for 39 VLMs in PhysBench Physical Object ✿Property Sub-task (the fifth last column of Table 3).

| Model | Size | Location | Depth | Distance | Motion |
|---|---|---|---|---|---|
| Image VLM | | | | | |
| InstructBLIP-t5-xl Dai et al. (2024) | 37.50 | 38.72 | 30.53 | 41.67 | 53.93 |
| InstructBLIP-t5-xxl Dai et al. (2024) | 48.21 | 41.35 | 27.72 | 51.67 | 57.30 |
| InstructBLIP-7B Dai et al. (2024) | 33.93 | 30.92 | 26.35 | 20.00 | 37.08 |
| InstructBLIP-13B Dai et al. (2024) | 26.79 | 39.10 | 25.26 | 43.33 | 48.31 |
| BLIP-2 Li et al. (2023c) | 58.93 | 44.74 | 29.47 | 46.67 | 59.55 |
| LLaVA-1.5-7B Liu et al. (2023a) | 64.29 | 43.23 | 35.09 | 41.67 | 51.69 |
| LLaVA-1.5-13B Liu et al. (2023a) | 57.14 | 46.24 | 32.63 | 45.00 | 60.67 |
| LLaVA1.6-mistral Liu et al. (2024b) | 26.79 | 19.55 | 24.56 | 10.00 | 37.08 |
| LLaVA1.6-vicuna Liu et al. (2024b) | 39.29 | 51.88 | 72.98 | 51.67 | 59.55 |
| Qwen-VL-Chat Bai et al. (2023b) | 33.93 | 42.11 | 45.96 | 38.33 | 52.81 |
| InternVL-Chat1.5 Chen et al. (2024c) | 75.00 | 66.17 | 76.14 | 58.33 | 61.80 |
| Cambrian-8B Tong et al. (2024) | 14.29 | 7.89 | 31.93 | 11.67 | 5.62 |
| Claude-3-opus Anthropic (2024) | 62.50 | 38.72 | 31.23 | 46.67 | 71.91 |
| Claude-3-sonnet Anthropic (2024) | 73.21 | 42.11 | 28.77 | 31.67 | 62.92 |
| Claude-3-haiku Anthropic (2024) | 64.29 | 54.14 | 47.37 | 51.67 | 68.54 |
| Claude-3.5-sonnet Anthropic (2024) | 71.43 | 46.24 | 25.96 | 36.67 | 69.66 |
| Video VLM | | | | | |
| Video-LLaVA Lin et al. (2023a) | 39.29 | 40.60 | 27.72 | 43.33 | 50.56 |
| Chat-Univi-7B Jin et al. (2023) | 28.57 | 19.55 | 17.89 | 25.00 | 35.96 |
| Chat-Univi-13B Jin et al. (2023) | 17.86 | 16.17 | 6.32 | 11.67 | 15.73 |
| PLLaVA-7B Xu et al. (2024) | 48.21 | 34.59 | 28.42 | 41.67 | 60.67 |
| PLLaVA-13B Xu et al. (2024) | 50.00 | 42.86 | 28.42 | 38.33 | 57.30 |
| General VLM + Interleaved Data | | | | | |
| LLaVA-interleave Li et al. (2024d) | 48.21 | 53.95 | 69.47 | 60.00 | 37.48 |
| LLaVA-interleave-dpo Li et al. (2024d) | 41.07 | 53.26 | 62.46 | 63.33 | 36.17 |
| VILA-1.5-3B Lin et al. (2023b) | 50.00 | 40.89 | 45.61 | 46.67 | 28.42 |
| VILA-1.5-3B-s2 Lin et al. (2023b) | 57.14 | 37.80 | 37.19 | 36.67 | 26.76 |
| VILA-1.5-8B Lin et al. (2023b) | 37.50 | 22.34 | 29.47 | 40.00 | 31.47 |
| VILA-1.5-13B Lin et al. (2023b) | 64.29 | 52.92 | 60.35 | 53.33 | 32.57 |
| Phi-3V Abdin et al. (2024) | 66.07 | 53.61 | 74.39 | 48.33 | 26.69 |
| LLaVA-NV Zhang et al. (2024b) | 32.14 | 35.40 | 49.47 | 45.00 | 26.00 |
| LLaVA-NV-dpo Zhang et al. (2024b) | 32.14 | 35.74 | 53.68 | 45.00 | 50.55 |
| Mantis-Idefics2 Jiang et al. (2024) | 39.29 | 58.42 | 65.61 | 65.00 | 32.50 |
| Mantis-LLaVA Jiang et al. (2024) | 37.50 | 47.08 | 31.93 | 46.67 | 26.56 |
| Mantis-siglip-llama3 Jiang et al. (2024) | 41.07 | 41.58 | 27.72 | 38.33 | 32.23 |
| Mantis-clip-llama3 Jiang et al. (2024) | 55.36 | 39.86 | 33.68 | 43.33 | 33.89 |
| GPT-4V Achiam et al. (2023) | 75.00 | 59.45 | 61.40 | 58.33 | 38.87 |
| GPT-4o Achiam et al. (2023) | 85.71 | 70.45 | 73.33 | 83.33 | 60.58 |
| GPT-4o-mini Achiam et al. (2023) | 69.64 | 51.55 | 57.19 | 63.33 | 39.21 |
| Gemini-1.5-flash Team et al. (2023) | 67.86 | 70.79 | 66.32 | 71.67 | 44.81 |
| Gemini-1.5-pro Team et al. (2023) | 66.07 | 71.48 | 82.11 | 68.33 | 58.30 |

Table 19: Evaluation Results for 39 VLMs in PhysBench Object ⬈Relationships Sub-task (the forth last column of Table 3).

| Model | Temperature | Viewpoint | Air | Light |
|---|---|---|---|---|
| Image VLM | | | | |
| InstructBLIP-t5-xl Dai et al. (2024) | 70.59 | 43.26 | 26.00 | 30.03 |
| InstructBLIP-t5-xxl Dai et al. (2024) | 80.88 | 43.36 | 40.00 | 30.30 |
| InstructBLIP-7B Dai et al. (2024) | 36.76 | 22.94 | 12.00 | 14.74 |
| InstructBLIP-13B Dai et al. (2024) | 57.35 | 21.61 | 30.61 | 19.86 |
| BLIP-2 Li et al. (2023c) | 79.41 | 40.90 | 30.00 | 29.48 |
| LLaVA-1.5-7B Liu et al. (2023a) | 61.76 | 44.58 | 24.00 | 31.76 |
| LLaVA-1.5-13B Liu et al. (2023a) | 50.00 | 40.06 | 38.00 | 28.03 |
| LLaVA1.6-mistral Liu et al. (2024b) | 45.59 | 5.66 | 40.00 | 8.01 |
| LLaVA1.6-vicuna Liu et al. (2024b) | 57.35 | 44.67 | 32.00 | 31.76 |
| Qwen-VL-Chat Bai et al. (2023b) | 66.18 | 20.55 | 20.00 | 29.66 |
| InternVL-Chat1.5 Chen et al. (2024c) | 72.06 | 40.43 | 34.00 | 31.48 |
| Cambrian-8B Tong et al. (2024) | 11.76 | 22.05 | 20.00 | 24.66 |
| Claude-3-opus Anthropic (2024) | 72.06 | 29.59 | 86.00 | 27.93 |
| Claude-3-sonnet Anthropic (2024) | 54.41 | 31.39 | 36.00 | 31.30 |
| Claude-3-haiku Anthropic (2024) | 73.53 | 26.86 | 42.00 | 29.94 |
| Claude-3.5-sonnet Anthropic (2024) | 75.00 | 27.80 | 68.00 | 24.48 |
| Video VLM | | | | |
| Video-LLaVA Lin et al. (2023a) | 67.65 | 39.40 | 30.00 | 26.02 |
| Chat-Univi-7B Jin et al. (2023) | 50.00 | 14.80 | 20.00 | 20.75 |
| Chat-Univi-13B Jin et al. (2023) | 20.59 | 13.29 | 16.00 | 17.93 |
| PLLaVA-7B Xu et al. (2024) | 60.29 | 40.81 | 22.00 | 31.12 |
| PLLaVA-13B Xu et al. (2024) | 66.18 | 30.63 | 36.00 | 30.39 |
| General VLM + Interleaved Data | | | | |
| LLaVA-interleave Li et al. (2024d) | 64.71 | 36.48 | 41.82 | 32.85 |
| LLaVA-interleave-dpo Li et al. (2024d) | 61.76 | 35.34 | 47.27 | 30.21 |
| VILA-1.5-3B Lin et al. (2023b) | 54.41 | 43.26 | 14.55 | 26.11 |
| VILA-1.5-3B-s2 Lin et al. (2023b) | 51.47 | 46.18 | 10.91 | 25.48 |
| VILA-1.5-8B Lin et al. (2023b) | 44.12 | 36.29 | 23.64 | 25.02 |
| VILA-1.5-13B Lin et al. (2023b) | 60.29 | 32.52 | 14.55 | 30.21 |
| Phi-3V Abdin et al. (2024) | 52.94 | 38.93 | 65.45 | 29.48 |
| LLaVA-NV Zhang et al. (2024b) | 52.94 | 36.29 | 18.18 | 31.12 |
| LLaVA-NV-dpo Zhang et al. (2024b) | 51.47 | 35.25 | 20.00 | 31.94 |
| Mantis-Idefics2 Jiang et al. (2024) | 63.24 | 27.43 | 47.27 | 29.39 |
| Mantis-LLaVA Jiang et al. (2024) | 58.82 | 41.28 | 23.64 | 30.57 |
| Mantis-siglip-llama3 Jiang et al. (2024) | 51.47 | 42.41 | 43.64 | 30.57 |
| Mantis-clip-llama3 Jiang et al. (2024) | 52.94 | 33.93 | 27.27 | 30.21 |
| GPT-4V Achiam et al. (2023) | 88.24 | 25.07 | 85.45 | 22.02 |
| GPT-4o Achiam et al. (2023) | 91.18 | 22.05 | 83.64 | 32.67 |
| GPT-4o-mini Achiam et al. (2023) | 82.35 | 28.56 | 83.64 | 27.75 |
| Gemini-1.5-flash Team et al. (2023) | 85.29 | 33.74 | 78.18 | 30.48 |
| Gemini-1.5-pro Team et al. (2023) | 85.29 | 37.32 | 76.36 | 31.85 |

Table 20: Evaluation Results for 39 VLMs in PhysBench Physical 🌲Scene Understanding Sub-task (the third last column of Table 3).

| Model | Collision | Throwing | Manipulation | Fluid | Chemistry | Others |
|---|---|---|---|---|---|---|
| Image VLM | | | | | | |
| InstructBLIP-t5-xl Dai et al. (2024) | 31.49 | 34.68 | 27.89 | 32.41 | 62.16 | 58.16 |
| InstructBLIP-t5-xxl Dai et al. (2024) | 32.72 | 31.35 | 29.90 | 33.33 | 52.70 | 58.84 |
| InstructBLIP-7B Dai et al. (2024) | 19.97 | 26.60 | 27.96 | 29.94 | 27.03 | 38.10 |
| InstructBLIP-13B Dai et al. (2024) | 28.26 | 15.44 | 27.89 | 35.65 | 33.78 | 46.60 |
| BLIP-2 Li et al. (2023c) | 33.64 | 35.39 | 30.40 | 31.17 | 59.46 | 58.84 |
| LLaVA-1.5-7B Liu et al. (2023a) | 36.25 | 37.77 | 41.21 | 45.37 | 59.46 | 55.10 |
| LLaVA-1.5-13B Liu et al. (2023a) | 33.18 | 34.92 | 46.73 | 50.15 | 60.81 | 60.20 |
| LLaVA1.6-mistral Liu et al. (2024b) | 24.12 | 11.64 | 12.81 | 7.56 | 62.16 | 50.34 |
| LLaVA1.6-vicuna Liu et al. (2024b) | 33.03 | 35.39 | 44.97 | 45.22 | 52.70 | 61.90 |
| Qwen-VL-Chat Bai et al. (2023b) | 32.57 | 33.25 | 37.19 | 47.84 | 59.46 | 59.86 |
| InternVL-Chat1.5 Chen et al. (2024c) | 35.94 | 39.90 | 42.71 | 43.67 | 55.41 | 73.47 |
| Cambrian-8B Tong et al. (2024) | 24.88 | 39.43 | 28.89 | 30.56 | 36.49 | 23.47 |
| Claude-3-opus Anthropic (2024) | 35.79 | 28.98 | 25.63 | 27.78 | 58.11 | 69.73 |
| Claude-3-sonnet Anthropic (2024) | 38.25 | 30.64 | 26.13 | 35.19 | 44.59 | 57.48 |
| Claude-3-haiku Anthropic (2024) | 35.94 | 39.90 | 34.42 | 30.40 | 67.57 | 68.37 |
| Claude-3.5-sonnet Anthropic (2024) | 34.25 | 37.05 | 27.64 | 28.09 | 50.00 | 71.09 |
| Video VLM | | | | | | |
| Video-LLaVA Lin et al. (2023a) | 35.48 | 38.48 | 29.15 | 46.30 | 54.05 | 53.74 |
| Chat-Univi-7B Jin et al. (2023) | 29.80 | 37.77 | 5.78 | 26.39 | 52.70 | 40.48 |
| Chat-Univi-13B Jin et al. (2023) | 5.38 | 12.59 | 12.81 | 11.27 | 20.27 | 17.01 |
| PLLaVA-7B Xu et al. (2024) | 33.79 | 40.86 | 28.89 | 41.20 | 59.46 | 57.14 |
| PLLaVA-13B Xu et al. (2024) | 36.25 | 39.43 | 29.90 | 39.20 | 58.11 | 63.61 |
| General VLM + Interleaved Data | | | | | | |
| LLaVA-interleave Li et al. (2024d) | 31.98 | 41.57 | 26.30 | 40.43 | 45.95 | 56.40 |
| LLaVA-interleave-dpo Li et al. (2024d) | 33.63 | 40.38 | 28.59 | 44.60 | 50.00 | 54.22 |
| VILA-1.5-3B Lin et al. (2023b) | 31.07 | 38.00 | 30.88 | 39.35 | 35.14 | 45.78 |
| VILA-1.5-3B-s2 Lin et al. (2023b) | 25.64 | 35.63 | 30.76 | 33.02 | 40.54 | 45.78 |
| VILA-1.5-8B Lin et al. (2023b) | 35.29 | 37.05 | 18.58 | 52.16 | 51.35 | 41.42 |
| VILA-1.5-13B Lin et al. (2023b) | 32.43 | 35.87 | 30.88 | 38.73 | 45.95 | 46.87 |
| Phi-3V Abdin et al. (2024) | 33.18 | 35.39 | 28.95 | 35.34 | 59.46 | 56.68 |
| LLaVA-NV Zhang et al. (2024b) | 29.71 | 41.57 | 29.35 | 44.44 | 59.46 | 45.90 |
| LLaVA-NV-dpo Zhang et al. (2024b) | 29.41 | 39.43 | 30.92 | 44.44 | 54.05 | 46.45 |
| Mantis-Idefics2 Jiang et al. (2024) | 32.88 | 33.25 | 28.59 | 38.43 | 41.89 | 57.49 |
| Mantis-LLaVA Jiang et al. (2024) | 27.75 | 40.14 | 28.11 | 40.74 | 43.24 | 41.96 |
| Mantis-siglip-llama3 Jiang et al. (2024) | 34.09 | 35.87 | 25.33 | 46.14 | 60.81 | 50.14 |
| Mantis-clip-llama3 Jiang et al. (2024) | 36.95 | 41.33 | 22.92 | 45.06 | 52.70 | 56.13 |
| GPT-4V Achiam et al. (2023) | 35.29 | 25.18 | 35.22 | 41.20 | 70.27 | 81.20 |
| GPT-4o Achiam et al. (2023) | 43.74 | 46.32 | 35.22 | 39.20 | 62.16 | 86.92 |
| GPT-4o-mini Achiam et al. (2023) | 37.41 | 39.90 | 25.33 | 46.76 | 71.62 | 78.75 |
| Gemini-1.5-flash Team et al. (2023) | 38.86 | 38.95 | 24.61 | 38.43 | 70.27 | 77.38 |
| Gemini-1.5-pro Team et al. (2023) | 34.24 | 39.43 | 27.62 | 39.81 | 68.92 | 80.93 |

Table 21: Evaluation Results for 39 VLMs in PhysBench Physics-based ⚖Dynamics Sub-task (the second last column of Table 3).

## F.4 PHYSBENCH-VAL RESULTS

| | Size | Format | ✿Property | ➤Relationships | ♠Scene | ⚖Dynamics | Avg |
|---|---|---|---|---|---|---|---|
| Random Choice | - | - | 25.00 | 25.00 | 25.00 | 25.00 | 25.00 |
| *Image VLM* | | | | | | | |
| InstructBLIP-t5-xl Dai et al. (2024) | 4B | merge | 40.54 | 47.62 | 45.95 | 44.44 | 44.69 |
| InstructBLIP-t5-xxl Dai et al. (2024) | 12B | merge | 48.65 | 52.38 | 43.24 | 49.21 | 48.60 |
| InstructBLIP-7B Dai et al. (2024) | 7B | merge | 28.57 | 30.95 | 36.36 | 35.48 | 33.14 |
| InstructBLIP-13B Dai et al. (2024) | 13B | merge | 30.56 | 32.50 | 60.00 | 36.51 | 37.80 |
| BLIP-2 Li et al. (2023c) | 12B | merge | 51.35 | 52.38 | 37.84 | 46.03 | 46.93 |
| LLaVA-1.5-7B Liu et al. (2023a) | 7B | merge | 37.84 | 54.76 | 43.24 | 52.38 | 48.04 |
| LLaVA-1.5-13B Liu et al. (2023a) | 13B | merge | 56.76 | 52.38 | 29.73 | 42.86 | 45.25 |
| LLaVA1.6-mistral Liu et al. (2024b) | 7B | merge | 43.24 | 30.95 | 35.14 | 39.68 | 37.43 |
| LLaVA1.6-vicuna Liu et al. (2024b) | 7B | merge | 59.46 | 42.86 | 37.84 | 49.21 | 47.49 |
| Qwen-VL-Chat Bai et al. (2023b) | 9B | merge | 40.54 | 45.24 | 40.54 | 44.44 | 43.02 |
| InternVL-Chat1.5 Chen et al. (2024c) | 26B | merge | 62.16 | 61.90 | 54.05 | 57.14 | **58.66**🥇 |
| Cambrian-8B Tong et al. (2024) | 8B | merge | 8.11 | 21.43 | 21.62 | 28.57 | 21.23 |
| Claude-3-opus Anthropic (2024) | - | merge | 45.95 | 61.90 | 45.95 | 57.14 | 53.63🥈 |
| Claude-3-sonnet Anthropic (2024) | - | merge | 40.54 | 66.67 | 43.24 | 38.10 | 46.37 |
| Claude-3-haiku Anthropic (2024) | - | merge | 56.76 | 59.52 | 35.14 | 49.21 | 50.28 |
| Claude-3.5-sonnet Anthropic (2024) | - | merge | 54.05 | 69.05 | 40.54 | 50.79 | 53.63🥈 |
| *Video VLM* | | | | | | | |
| Video-LLaVA Lin et al. (2023a) | 7B | seq | 43.24 | 33.33 | 48.65 | 52.38 | 45.25 |
| Chat-Univi-7B Jin et al. (2023) | 7B | seq | 24.32 | 28.57 | 29.73 | 33.33 | 29.61 |
| Chat-Univi-13B Jin et al. (2023) | 13B | seq | 8.11 | 21.43 | 18.92 | 17.46 | 16.76 |
| PLLaVA-7B Xu et al. (2024) | 7B | seq | 48.65 | 47.62 | 32.43 | 53.97 | 46.93🥈 |
| PLLaVA-13B Xu et al. (2024) | 13B | seq | 51.35 | 50.00 | 40.54 | 58.73 | **51.40**🥇 |
| *General VLM + Interleaved data* | | | | | | | |
| LLaVA-interleave Li et al. (2024d) | 7B | seq | 48.65 | 59.57 | 43.24 | 51.90 | 51.50 |
| LLaVA-interleave-dpo Li et al. (2024d) | 7B | seq | 40.54 | 48.94 | 48.65 | 50.63 | 48.00 |
| VILA-1.5-3B Lin et al. (2023b) | 3B | seq | 37.84 | 51.06 | 43.24 | 36.71 | 41.50 |
| VILA-1.5-3B-s2 Lin et al. (2023b) | 3B | seq | 35.14 | 51.06 | 43.24 | 36.71 | 41.00 |
| VILA-1.5-8B Lin et al. (2023b) | 8B | seq | 24.32 | 29.79 | 43.24 | 36.71 | 34.00 |
| VILA-1.5-13B Lin et al. (2023b) | 13B | seq | 32.43 | 51.06 | 37.84 | 45.57 | 43.00 |
| Phi-3V Abdin et al. (2024) | 4B | seq | 54.05 | 61.70 | 51.35 | 49.37 | 53.50 |
| LLaVA-NV Zhang et al. (2024b) | 7B | seq | 45.95 | 40.43 | 32.43 | 46.15 | 42.21 |
| LLaVA-NV-dpo Zhang et al. (2024b) | 7B | seq | 48.65 | 41.86 | 29.73 | 46.15 | 42.56 |
| Mantis-Idefics2 Jiang et al. (2024) | 8B | seq | 40.54 | 46.81 | 43.24 | 45.57 | 44.50 |
| Mantis-LLaVA Jiang et al. (2024) | 7B | seq | 43.24 | 29.79 | 45.95 | 35.44 | 37.50 |
| Mantis-siglip-llama3 Jiang et al. (2024) | 8B | seq | 67.57 | 34.04 | 40.54 | 40.51 | 44.00 |
| Mantis-clip-llama3 Jiang et al. (2024) | 8B | seq | 54.05 | 42.55 | 35.14 | 44.30 | 44.00 |
| GPT-4V Achiam et al. (2023) | - | seq | 62.16 | 68.09 | 43.24 | 67.09 | 62.00 |
| GPT-4o Achiam et al. (2023) | - | seq | 72.97 | 74.47 | 54.05 | 62.03 | **65.50**🥇 |
| GPT-4o-mini Achiam et al. (2023) | - | seq | 64.86 | 57.45 | 45.95 | 63.29 | 59.00 |
| Gemini-1.5-flash Team et al. (2023) | - | seq | 64.86 | 74.47 | 54.05 | 63.29 | 64.50 |
| Gemini-1.5-pro Team et al. (2023) | - | seq | 59.46 | 70.21 | 56.76 | 68.35 | 65.00🥈 |

Table 22: **Evaluation results for 39 vision-language models in PhysBench-val.** Note that the evaluation of General VLMs is based on the data from Video and Image VLM evaluations, with the addition of interleaved data, meaning that the full test dataset of PhysBench is being assessed. In this context, "seq" refers to the sequential input of images after frame selection from videos, while "merge" refers to merging video frames into a single image.

### F.5 EMBODIED TASKS DETAILED DESCRIPTION

To further validate the effectiveness of our data and method, we built a simulation platform using MuJoCo (Todorov et al., 2012) and the Franka Emika Panda from Menagerie (Zakka et al., 2022), and conducted tests on five embodied tasks. Using the proposed mark-based visual prompting technique with GroundedSAM (Ren et al., 2024) and farthest point sampling (Qi et al., 2017), MOKA (Liu et al., 2024a) converts affordance reasoning into a series of visual question-answering problems that pre-trained VLMs can solve. The setup matches MOKA, and the general visual setup can be seen in Figure 25(a)(b). Our tabletop environment only has one top-down camera, which is the primary camera used in MOKA to capture RGBD images. For each task, we report the number of successes out of 10 trials following the setting of Liu et al. (2024a).

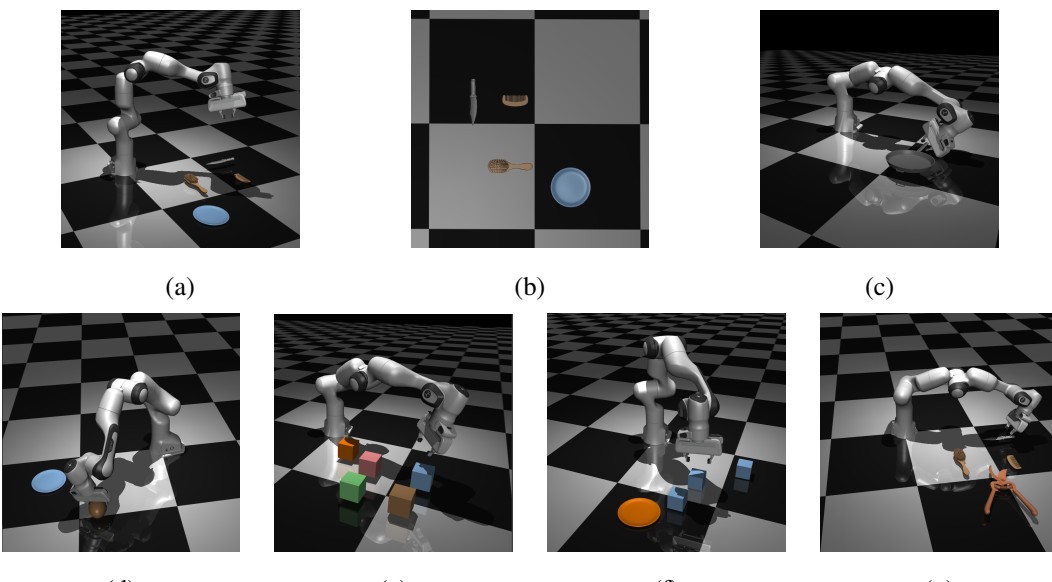

(a)  (b)  (c)

(d)  (e)  (f)  (g)

Figure 25: (a) Overview of the simulation platform. (b) Top-down view of the area. (c) Affordance test: Testing whether the robotic arm correctly grasps the object. (d) Force test: Testing whether the robotic arm can properly grasp deformable, fragile, and rigid objects. (e) Color test: Testing whether the robotic arm can pick up the correct colored object among identical ones. (f) Number test: Testing whether the robotic arm can grasp a specific number of objects. (g) Tool test: Testing whether the robotic arm can select the correct tool given a specific scenario.

We present the specific language instructions given to the VLM in Figure 26. It is important to note that, for each task, we only provide a single example. For instance, while we tested five objects in the Affordance task—pot, knife, spoon, monitor, and tennis racket—we only used the tennis racket as an example here. For each task, we report the success rate over 10 trials, following the MOKA protocol. "Specifically, the test content and evaluation methods for each task are as follows: (1) Affordance test: This test evaluates whether the robotic arm can correctly grasp various objects. Figure 25(c) illustrates the robotic arm successfully grasping a pot. In total, we tested the grasping ability on 10 items, including a pot, knife, spoon, spatula, monitor, tennis racket, phone, and others. Specifically, we tested 5 objects—pot, knife, spoon, monitor, and tennis racket—attempting to grasp each object twice. (2) Force Test: This test evaluates the robotic arm's capacity to properly grasp deformable, fragile, and rigid objects. Due to simulation constraints, the evaluation was based on the robotic arm's output metrics. We tested fragile items (e.g., egg, ripe persimmon), soft items (e.g., jelly, plastic cup), and rigid objects (e.g., iron ball), with two attempts per object. It should be noted that the simulation system models all objects as rigid bodies, meaning that breakage during grasping is not depicted. Furthermore, while using the MOKA system to control the Panda robotic arm, we were unable to directly manipulate the gripper's size. Instead, we provided the VLM with approximate object dimensions, allowing the VLM to determine the necessary gripping force to evaluate success or failure." (3) Color test: This test evaluates whether the robotic arm can accurately pick up the correctly colored object from a set of identical items. As shown in Figure 25(e), the blocks are identical except for their color, and the task requires selecting the object of the designated color. We tested five colors—blue, pink, brown, green, and orange—conducting two trials for each color.

(4) Location test: This test evaluates whether the robotic arm can correctly grasp objects at specific locations. The goal is to ensure that the robotic arm accurately grasps the required objects based on their positions. Specifically, we tested with three blocks, requiring the arm to grasp the middle block (4 attempts), the block farthest from the plate (3 attempts), and the block closest to the plate (3 attempts). (5) Tool test: This test assesses whether the robotic arm can select the appropriate tool for a given task. For instance, as shown in Figure 25(g), the task is: "If you need to cut a watermelon, which tool should you grasp?" In this scenario, the robotic arm is expected to grasp the fruit knife. In total, we posed 5 questions, with 2 attempts per question, requiring the robotic arm to select and grasp different target tools for each task.

On the other hand, these five tasks require minimal consideration of height (or depth) information, making the evaluation more fundamental. For MOKA's waypoints selected from free space, their height must be explicitly specified for accurate deprojection into 3D space, as they are not anchored to any objects. For this reason, in typical tabletop manipulation scenarios, MOKA primarily focuses on cases where the waypoints are at the same height as the target point.

| | |
|---|---|
| **Affordance** | Grasp the tennis racket. |
| **Force** | Grasp the egg. |
| **Color** | Grasp the blue cube. |
| **Location** | Grab the block farthest from the plate and move it to the plate. |
| **Tool** | Grasp the tools for cutting a watermelon. |

Figure 26: The language description of the testing tasks.

We present an overview of our implementation of MOKA in Algorithm 1. Our experiments aim to enhance the VLM's ability to understand the physical world and validate its impact on downstream embodied agent tasks. Specifically, we employ two methods to improve the VLM: first, fine-tuning the VLM using PhysBench, and second, incorporating the PhysAgent to assist during VLM inference. It is worth noting that the five tasks we address are relatively fundamental, unlike the complex multi-action tasks described in the MOKA paper, which require hierarchical decomposition from high- to low-level actions. In our case, the tasks can be executed directly without such decomposition."

---

**Algorithm 1** MOKA Pipeline

---

1: **Input:** Vision-language Model $\mathcal{M}$, Task instruction $l$, text prompt for low-level reasoning $p_{low}$ and initial observation $s$
2: Get observation $s$ from the top-down camera
3: Propose keypoint and waypoint candidates and get annotated image $f(s_k)$
4: Query $\mathcal{M}$ for low-level motion reasoning, obtain $y_{low} = \mathcal{M}([p_{low}, l, f(s)])$
5: Execute $y_{low}$ on the robot in simulation

---

## F.6    CORRELATION MAP

Following the approach of Tong et al. (2024); Fang et al. (2024; 2025), we used the Pearson correlation coefficient to construct a relationship matrix. The data used to build this matrix can be found in Table 23.

| | VQAv2 | GQA | VisWiz | SQA | TextVQA | POPE | MME | MMB | MMBCN | SEED | SEEDI | MMMUval | MMMUtest | LLaVA-bench |
|---|---|---|---|---|---|---|---|---|---|---|---|---|---|---|
| LLaVA-1.5-7B | 78.5 | 62.0 | 50.0 | 66.8 | 58.2 | 85.9 | 1510.7 | 64.3 | 58.3 | 61.5 | 67.0 | 33.2 | 31.1 | 63.4 |
| LLaVA-1.5-13B | 80.0 | 63.3 | 53.6 | 71.6 | 61.3 | 85.9 | 1531.3 | 67.7 | 63.6 | 62.4 | 68.2 | 36.4 | 33.6 | 70.7 |
| InstructBLIP-7B | 61.1 | 49.2 | 34.5 | 60.5 | 50.1 | 78.8 | 1210.1 | 36.0 | 23.7 | 53.4 | 58.8 | 32.9 | 30.6 | 60.9 |
| InstructBLIP-13B | 62.3 | 49.5 | 33.4 | 63.1 | 50.7 | 78.9 | 1212.8 | 42.0 | 25.0 | 55.2 | 61.7 | 35.7 | 33.8 | 58.2 |
| Qwen-VL-Chat | 78.2 | 57.5 | 38.9 | 68.2 | 61.5 | 85.6 | 1487.5 | 60.6 | 56.7 | 58.2 | 65.4 | 35.9 | 32.9 | 64.1 |
| VILA-1.5-3B | 80.4 | 61.5 | 53.5 | 69.0 | 60.4 | 85.9 | 1442.4 | 63.4 | 52.7 | 60.9 | 67.9 | 33.3 | 30.8 | 75.9 |
| VILA-1.5-3B-s2 | 79.8 | 61.4 | 61.3 | 69.6 | 63.4 | 85.3 | 1431.7 | 62.8 | 52.2 | 60.0 | 66.4 | 32.8 | 31.3 | 76.7 |
| VILA-1.5-8B | 80.9 | 61.9 | 58.7 | 79.9 | 66.3 | 84.4 | 1577.0 | 72.3 | 66.2 | 64.2 | 71.4 | 36.9 | 36.0 | 80.0 |
| VILA-1.5-13B | 82.8 | 64.3 | 62.6 | 80.1 | 65.0 | 86.3 | 1569.6 | 74.9 | 66.3 | 65.1 | 72.6 | 37.9 | 33.6 | 80.8 |
| BLIP-2 | 41.0 | 44.6 | 29.4 | 61.0 | 42.1 | 85.3 | 1293.8 | 44.0 | 27.0 | 46.4 | 49.7 | 35.4 | 34.0 | 56.2 |

Table 23: The performance of the 10 models used to construct the correlation map across 15 other VLM benchmarks.

The correlation presented in Figure 4(a) illustrates the relationships between the four major categories in PhysBench and other tasks. Additionally, we provide a detailed correlation map between PhysBench and 15 other vision-language benchmarks in Figure 27 below.

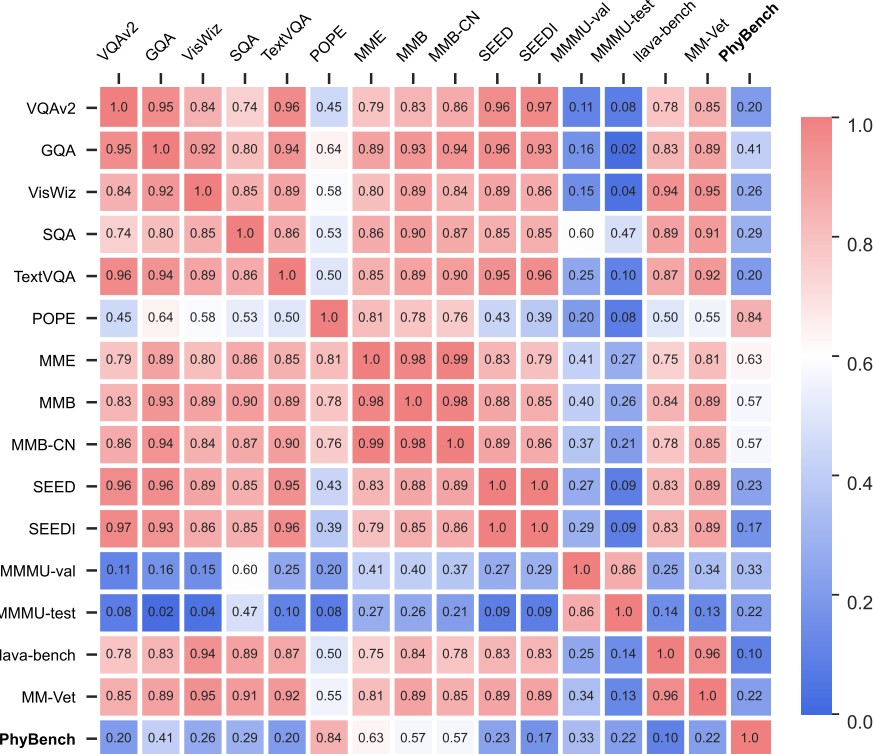

Figure 27: Correlation map between PhysBench and 15 other vision-language benchmarks.

### F.7 PERFORMANCE ON RELATED BENCHMARKS

To further evaluate the contribution of our data and model to understanding the physical world, we conducted tests on three existing benchmarks related to physical-world perception. Notably, these benchmarks focus on specific aspects of physical-world perception, whereas our PhysBench provides a more comprehensive and holistic evaluation, as summarized in Table 1. Furthermore, since these datasets were not originally designed for VLMs, we applied necessary preprocessing to adapt them for our evaluations.

**Setup.** *EmbSpatial* (Du et al., 2024) is a benchmark designed to evaluate spatial understanding within embodied environments with source image come from MP3D (Chang et al., 2017), ScanNet (Dai et al., 2017) and AI2-THOR (Kolve et al., 2017). We utilized its benchmark dataset for testing purposes. *ContPhy* (Zheng et al., 2024b) is a benchmark aimed at assessing visual models' capabilities in perceiving continuous physical phenomena and properties. It comprises four simulation systems based on Unity3D: Fluid Hourglass, Rope-Pulley System, Cloth Magic Trick, and Ball Playground. Since ContPhy primarily targets visual and physical models, we selected 200 items for each of the four categories (split evenly between property-based and dynamics-based tasks) and translate them into multiple-choice format suitable for VLMs. The question prompts were modified to better align with the answering capabilities of VLMs. *Physion++* (Tung et al., 2023) evaluates the impact of physical properties such as mass, friction, elasticity, and deformability on physical phenomena. It leverages the ThreeDWorld simulation platform (Gan et al., 2020) to generate a series of videos, each paired with a corresponding question. The videos consist of an inference phase, where artificial systems can identify objects' mechanical properties, followed by a prediction phase, where the model must predict whether two specified objects will collide after the video ends. Physion++ is primarily designed for vision models such as ResNet (He et al., 2016) and VGG (Simonyan & Zisserman, 2014), with answers in the form of binary classification (yes/no), derived by converting model outputs into probabilities. To adapt Physion++ for VLMs, we processed the videos and

reformulated the questions into natural language prompts, adding necessary contextual hints. For example, if a video includes a transition phase, we explicitly include a statement like, "The black screen in the video marks a transition in objects or scenes" in the prompt. Due to the small size of the test set, we combined the train and test sets, resulting in 250 VQA pairs for evaluation. During fine-tuning with PhysBench, we modified the options to an open-ended format. The results, presented in Table 24, are reported in terms of accuracy. All parameter settings and configurations are kept consistent with those outlined in the main text.

Table 24: Performance on Three Related Benchmarks.

|  | ContPhy-Property | ContPhy-Dynamics | Physion++ | EmbSpatial |
|---|---|---|---|---|
| Phi-3V | 49.25 | 43.25 | 68.80 | 55.91 |
| Phi-3V + finetune | 64.50 | 62.75 | 82.20 | 66.95 |
| Phi-3V + PhysAgent | 52.00 | 44.75 | 78.40 | 62.28 |

**Results**. As presented in Table 24, leveraging PhysBench data for fine-tuning or in a zero-shot setting with PhysAgent leads to performance improvements across the benchmarks, particularly in Physion++, where improvements of 19.50% and 9.6% are observed, with fine-tuning achieving the most significant gains. These results highlight the effectiveness of our data and methods in enhancing the capability of Vision-Language Models to comprehend the physical world.

# G   MORE RELATED WORKS

**Vision-Language Models**. Vision-Language Models (VLMs) are large language models that integrate visual modalities, such as images and videos, with language knowledge (Wu et al., 2023b; Huang et al., 2024; Bai et al., 2022; Zhan et al., 2024; Dai et al., 2024; Bai et al., 2024b). Notable models like BLIP-2 (Li et al., 2023c) and LLaVA (Liu et al., 2024c) have advanced image-captioning datasets and visual instruction tuning, with LLaVA-Next further improving single-image performance at higher computational costs (Liu et al., 2024b). Subsequent models, such as QwenVL (Bai et al., 2023b), CogVLM (Wang et al., 2023b), and Yi-VL (AI et al., 2024), have followed a similar architecture to LLaVA. As single-image and text interaction technologies continue to mature, many recent VLMs (Alayrac et al., 2022; Peng et al., 2023; Pan et al., 2024; Lin et al., 2023b) are now capable of handling complex visual tasks with interleaved images or videos, enabling VLMs to tackle more sophisticated tasks (Lu et al., 2024b; Yu et al., 2024b) and paving the way for interactions with the real physical world.

**Vision-Language Benchmarks** VLMs (Liu et al., 2024c; Achiam et al., 2023; Pan et al., 2024; Yu et al., 2023a; Chow et al., 2024; **?**; Li et al., 2024b;c) have inherited and advanced many intriguing features from text-only LMs. Benchmarks for VLMs have rapidly emerged to evaluate performance in areas such as image question answering (Ying et al., 2024), in-context response (Yu et al., 2023b), chart understanding (Li et al., 2024f), and web comprehension (Liu et al., 2024d; Zhou et al., 2023). Some benchmarks cover spatial relations understanding (Li et al., 2023a), but often overlook the ability to devise complex spatial action plans based on physical world comprehension. Recently, new benchmarks have also emerged that focus on the ability to understand multiple images in long contexts (Zhang et al., 2024a; Kil et al., 2024; Jiang et al., 2024; Wu et al., 2023a; Li et al., 2024e; Ge et al., 2024) and complex realistic environments (Fu et al., 2024; Bai et al., 2024a). However, these benchmarks—whether based on answering questions from images, videos, or tables, or using visual prompts (Fu et al., 2024; Yu et al., 2024a)—ultimately rely on responding to the content of the given images rather than the true perception of the physical world, thus falling short of achieving spatial intelligence (Gupta et al., 2021; Yang et al., 2024a).

**Video Benchmarks**. With the growing interest in video understanding, the development of benchmarks for VLMs has become increasingly emphasized. In video comprehension, the research community has made significant strides, particularly for short videos. There are specialized benchmarks for temporal perception (Yu et al., 2019; Wu et al., 2024a), action understanding (Liu et al., 2024e; Mangalam et al., 2024), video classification (Kay et al., 2017), video reasoning (Xiao et al., 2021a; Xie et al., 2023), video captioning (Miech et al., 2019; Xu et al., 2016), video question-answering (Zhou et al., 2024; Li et al., 2023g; 2024e), long video comprehension (Wu et al., 2024b; Chandrasegaran et al., 2024), video generation (Bansal et al., 2024), and interleaved video-text

Table 25: **Comparison between PhysBench and other vision-language benchmarks**. In the format, I, T, V present text, image, and video. Annotated means annotate the existing dataset, like MSCOCO Karpathy & Fei-Fei (2015). LLaVA[Wd]: LLaVA-Bench(In-the-Wild)-Detail Liu et al. (2024c). Reasoning means that it requires the VLMs' reasoning ability to answer the question.

| Dataset | Size (k) | Format | Interleaving | Source | Domain | Reasoning |
|---|---|---|---|---|---|---|
| VQA-v2 Goyal et al. (2017b) | 1,105,904 | I+T | ✗ | Annotated | Image content | ✗ |
| GQA Hudson & Manning (2019) | 22,669,678 | I+T | ✗ | Annotated | Image content | ✗ |
| VizWiz Gurari et al. (2018) | 32,000 | I+T | ✗ | Annotated | Image content | ✗ |
| TextVQA Singh et al. (2019) | 45,000 | I+T | ✗ | Annotated | Image content | ✗ |
| OKVQA Marino et al. (2019) | 14,000 | I+T | ✗ | Annotated | Image content | ✗ |
| SEED Li et al. (2023a) | 19,000 | V+I+T | ✗ | Annotated | Image and video content | ✗ |
| MMBench Liu et al. (2023c) | 3,000 | I+T | ✗ | Annotated | Image content | ✗ |
| MME Yin et al. (2023) | 1,297 | I+T | ✗ | Annotated | Image content | ✗ |
| POPE Li et al. (2023h) | 18,000 | I+T | ✗ | Annotated | Image hallucination detection | ✗ |
| MM-Vet Yu et al. (2023b) | 200 | I+T | ✗ | Annotated | Image chat | ✗ |
| LLaVA[Wd] Liu et al. (2024c) | 60 | I+T | ✗ | Annotated | Image chat | ✗ |
| SQA[I] Lu et al. (2022) | 6,000 | I+T | ✗ | Annotated | Image content | ✗ |
| NLVR2 Suhr et al. (2018) | 6,967 | I+T | ✗ | Annotated | Image content | ✗ |
| MathVista Lu et al. (2024b) | 6,141 | I+T | ✗ | Annotated | Math | ✓ |
| BLINK Fu et al. (2024) | 1,901 | I+T | ✓ | Annotated, Chart | Visual prompt | ✗ |
| Mantis-eval Jiang et al. (2024) | 217 | I+T | ✓ | Annotated | Image chat | ✗ |
| Q-Bench Wu et al. (2023a) | 2,990 | I+T | ✓ | Annotated | Image content | ✗ |
| MMMU Yue et al. (2024) | 11,500 | I+T | ✓ | Annotated, Web, Textbook | Image content | ✗ |
| PhysBench | 10,002 | V+I+T | ✓ | Annotated, Web, Simulation, Real-world | Physical property and dynamics | ✓ |

question-answering (Wang et al., 2024a). However, these works primarily focus on evaluating video content and do not explore the underlying mechanisms of video representation or address true physical world perception. Furthermore, during our experiments, we observed significant challenges with current video VLMs in following instructions and answering questions, as the models frequently output descriptions of the video rather than directly addressing the posed questions.

**Interleaved Vision-Language Benchmarks**. VLMs are increasingly processing longer and more complex inputs. Along with this development, several benchmarks with interleaved inputs have emerged (Li et al., 2023b; Wang et al., 2024b; Meng et al., 2024). For example, SEED-Bench (Li et al., 2023a; 2024a) focuses on video understanding, BLINK (Fu et al., 2024) introduces visual prompts, and NLVR2 (Suhr et al., 2018) differentiates between two images. However, all of these benchmarks still primarily assess content description based on images, without evaluating physical understanding or perception abilities. Furthermore, current interleaved benchmarks only involve images and text, while PhysBench is a dataset that interweaves video, image, and text inputs. A detailed comparison with the previously mentioned benchmarks and other vision-language benchmarks can be found in Table 25.

**Science-related Benchmarks**. In addition to the benchmarks related to physical world comprehension mentioned in Section 2, there are also benchmarks that assess models' understanding through middle or university-level physics exam questions. SciQ (Welbl et al., 2017), ScienceQA (Lu et al., 2022), E-EVAL (Hou et al., 2024), MMLU-STEM (Hendrycks et al., 2020), and C-Eval-STEM (Huang et al., 2023b) include some physics-related questions, but these datasets often allow questions to be answered simply by analyzing the provided images, lacking the complexity of reasoning and computational tasks. JEEBench (Arora et al., 2023) requires multistep reasoning with physics knowledge but is limited in scope and purely text-based. SciBench (Wang et al., 2024f), OlympiadBench (He et al., 2024), MathVista (Lu et al., 2024b), and OCWCourses (Lewkowycz et al., 2022) provide college-level physics questions. However, these benchmarks mainly consist of homework and exam-style questions, focusing more on mathematical reasoning (Zheng et al., 2024a) and general knowledge rather than true physical world comprehension. In contrast, our PhysBench is the first systematic and comprehensive question-answering benchmark specifically designed for understanding the real physical world.

Table 26: A comparison between PhysBench and other physical understanding benchmarks not in question-answering format.

| | Property | Attribute | Location | Velocity | Temperature | Camera | Light | Collision | Manipulation | Fluid | Interleaved | Size |
|---|---|---|---|---|---|---|---|---|---|---|---|---|
| Physics 101 Wu et al. (2016) | ✗ | ✗ | ✗ | ✗ | ✗ | ✗ | ✗ | ✓ | ✗ | ✗ | ✗ | 17,408 |
| IntPhys Riochet et al. (2018) | ✓ | ✗ | ✗ | ✗ | ✗ | ✗ | ✗ | ✓ | ✗ | ✗ | ✗ | 15,000 |
| ESPRIT Rajani et al. (2020) | ✗ | ✗ | ✗ | ✗ | ✗ | ✗ | ✗ | ✓ | ✗ | ✗ | ✗ | 2,441 |
| CRAFT Ates et al. (2020) | ✓ | ✗ | ✗ | ✗ | ✗ | ✗ | ✗ | ✓ | ✗ | ✗ | ✗ | 57,000 |
| CoPhy Baradel et al. (2019) | ✓ | ✗ | ✓ | ✗ | ✗ | ✗ | ✗ | ✓ | ✗ | ✗ | ✗ | 216,000 |
| PhysBench | ✓ | ✓ | ✓ | ✓ | ✓ | ✓ | ✓ | ✓ | ✓ | ✓ | ✓ | 10,002 |

**3D Scence VQA**. Recently, multi-modal 3D perception has garnered increasing attention due to its connection to the physical world, driving rapid advancements in the field. SQA3D (Ma et al., 2023) highlights the importance of "situations" within contextual understanding. EmbodiedScan (Wang et al., 2024d) and SceneVerse (Jia et al., 2025) expand the scope by collecting more scenes or annotating additional objects, scaling annotations to the millions. OpenEQA (Majumdar et al., 2024), SpatialRGPT-Bench (Cheng et al., 2024), MMScan (Lyu et al., 2024), and MSNN (Linghu et al., 2024) integrate comprehensive annotations, adapting the task into a VQA format. However, these studies primarily focus on geometric relationships, which represent only a subset of the broader understanding of the physical world, as discussed in Section 2. Our work prioritizes a more holistic evaluation of the physical world perception capabilities of VLMs across four major task categories: Physical Object Properties, Physical Object Relationships, Physical Scene Understanding, and Physics-based Dynamics. Additionally, the spatial components in our dataset differ significantly from those in existing Spatial VQA datasets. While such benchmarks typically rely on 3D point cloud scenes or interleaved 2D images from multiple viewpoints, our dataset uses interleaved images to capture physical world dynamics, such as viewpoint rotations and the progression of physical phenomena.

**VLMs for Robotic Manipulation**. Recently, two main approaches have been proposed for applying Vision-Language Models (VLMs) (Liu et al., 2024c; Achiam et al., 2023; Team et al., 2023; Mao et al., 2023a;b) to robotic manipulation: *(a) directly generating actions* (Kim et al., 2024; Octo Model Team et al., 2024; Zawalski et al., 2024a; Niu et al., 2024; Zawalski et al., 2024b) and *(b) employing VLMs as agents* (Liu et al., 2024a; Nasiriany et al., 2024; Huang et al., 2023a). Approach (a) involves directly outputting actions, which requires extensive training—for instance, OpenVLA was trained using 64×A100 GPUs—and produces embodied-specific actions that necessitate targeted fine-tuning. In contrast, approach (b) generates affordance representations through VLMs, which are subsequently converted into actions. This method requires less training and exhibits stronger generalization capabilities. However, it suffers from weaker perception capabilities in the physical world, leading to performance limitations (Liu et al., 2024a; Mao et al., 2024).

## H MORE EXAMPLES

We use red color as right answer. We use red to indicate the correct answer (correct answer). It is important to note that, to reduce difficulty and facilitate evaluation, we employed a multiple-choice format rather than open-ended responses. For space-saving purposes, the example figures in the main text do not display the available options.

### H.1 PHYSICAL OBJECT PROPERTY SUB-TASK

**Question**

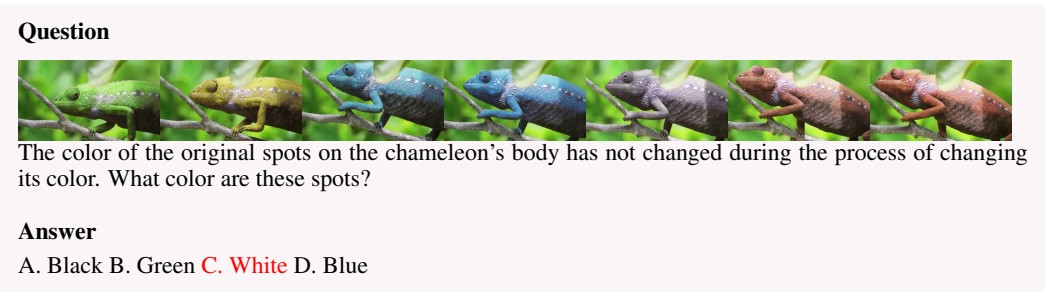

The color of the original spots on the chameleon's body has not changed during the process of changing its color. What color are these spots?

**Answer**

A. Black B. Green C. White D. Blue

Figure 28: Example for property color. Ability Type is identify.

### H.2 PHYSICAL OBJECT RELATIONSHIPS SUB-TASK

### H.3 PHYSICAL SCENE UNDERSTANDING SUB-TASK

### H.4 PHYSICS-BASED DYNAMICS SUB-TASK

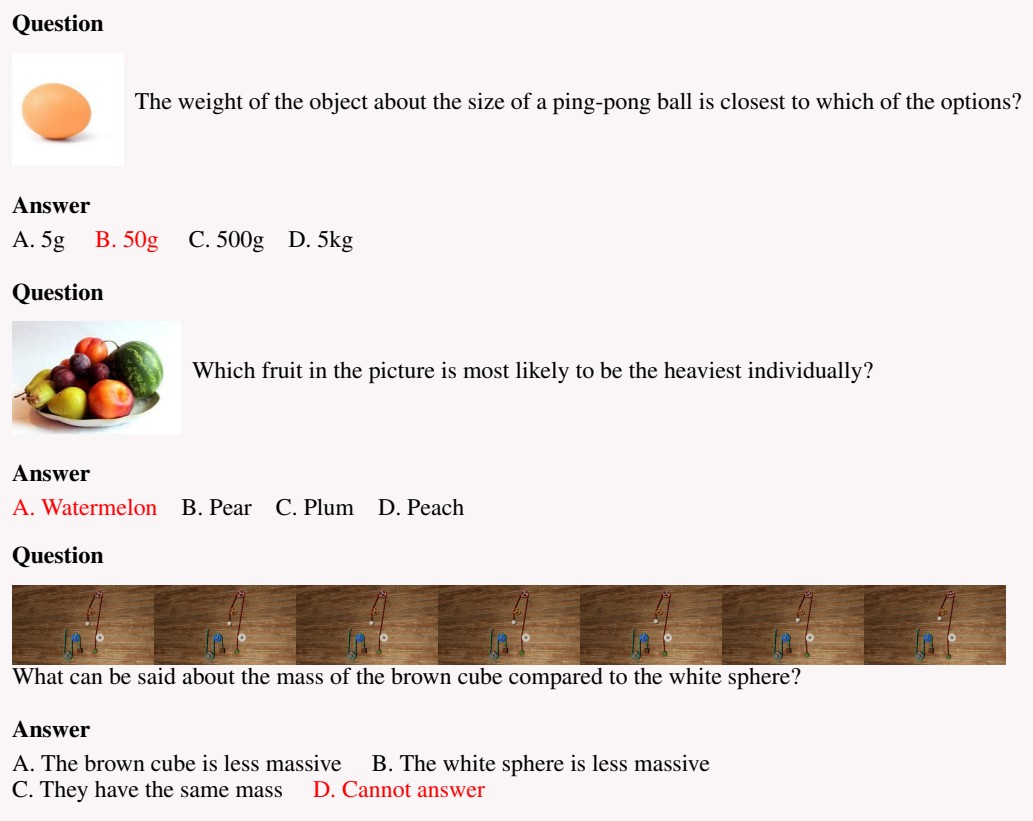

**Question**

The weight of the object about the size of a ping-pong ball is closest to which of the options?

**Answer**

A. 5g    B. 50g    C. 500g    D. 5kg

**Question**

Which fruit in the picture is most likely to be the heaviest individually?

**Answer**

A. Watermelon    B. Pear    C. Plum    D. Peach

**Question**

What can be said about the mass of the brown cube compared to the white sphere?

**Answer**

A. The brown cube is less massive    B. The white sphere is less massive
C. They have the same mass    D. Cannot answer

Figure 29: Three examples related to property mass, categorized by the following ability types: identification, comparison, and comparison.

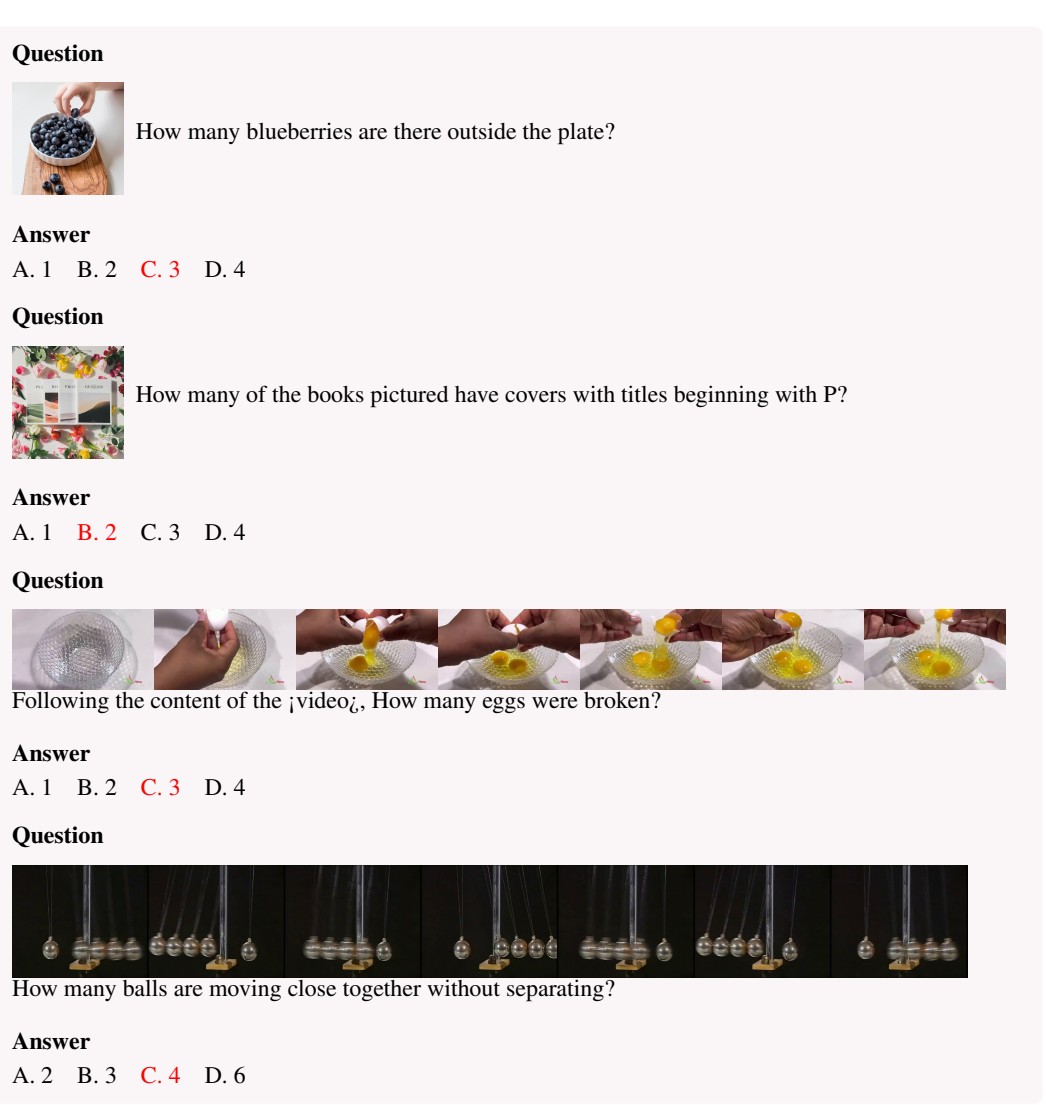

Figure 30: Four examples related to property number, ability types are all identification.

**Question**

 In the photo, which point with option signifies the object with the most sharp?

**Answer**

A. Point A    B. Point B    C. Point C    D. Point D

**Question**

 Can you tell me which point with option in the image points to the object that has the least brittle?

**Answer**

A. Point A    B. Point B    C. Point C    D. Point D

**Question**

 Which point with option in the photograph pinpoints the object with the most stiff?

**Answer**

A. Point A    B. Point B    C. Point C    D. Point D

**Question**

 Which point with option in the image marks the object that exhibits the most elastic?

**Answer**

A. Point A    B. Point B    C. Point C    D. Point D

**Question**

 Which point with option in the photograph captures the object that has the most malleable?

**Answer**

A. Point A    B. Point B    C. Point C    D. Point D

**Question**

 Which point with option in the image isolates the object with the most soft?

**Answer**

A. Point A    B. Point B    C. Point C    D. Point D

Figure 31: Six examples of property attributes include sharpness, brittleness, stiffness, elasticity, malleability, and softness.

**Question**

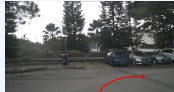 Moving according to the arrow in the picture, which of the following options are you most likely to encounter?

**Answer**

A. 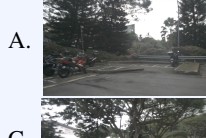 B.

C. 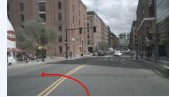 D.

**Question**

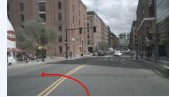 Following the direction indicated by the arrow in the picture, which of the following options are you most likely to encounter?

**Answer**

A. B.

C. D.

**Question**

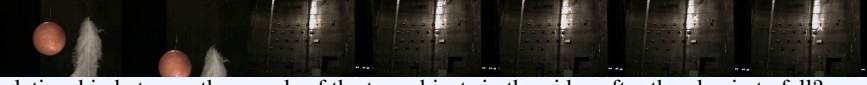 What is the relationship between the speeds of the two objects in the video after they begin to fall?

**Answer**

A. Both objects have the same speed.
B. The egg consistently falls faster than the feather.
C. The feather consistently falls faster than the egg.
D. Initially, the egg falls faster, but the feather eventually surpasses it.

Figure 32: Three examples for relationships motion, categorized by the following ability types: static, static and dynamic.

**Question**

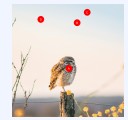 Determine which point is nearest to the camera:

**Answer**

A. Point A is nearest   B. Point B is nearest   C. Point C is nearest   D. Point D is nearest

Figure 33: An example for relationships depth. Ability Type is static.

**Question**

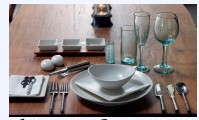 Find the point that is closest to the river:

**Answer**

A. Point A is the closest    B. Point B is the closest    C. Point C is the closest    D. Point D is the closest

**Question**

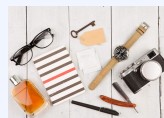 What value is the distance between each of the three small bowls placed side by side closest to?

**Answer**

A.3mm    B.3cm    C.10cm    D.20cm

**Question**

What is farthest from the key in the picture?

**Answer**

A. Pen    B. Camera    C. Watch    D. Glasses

Figure 34: Three examples for relationships distance. Ability Type is static, static and dynamic.

**Question**

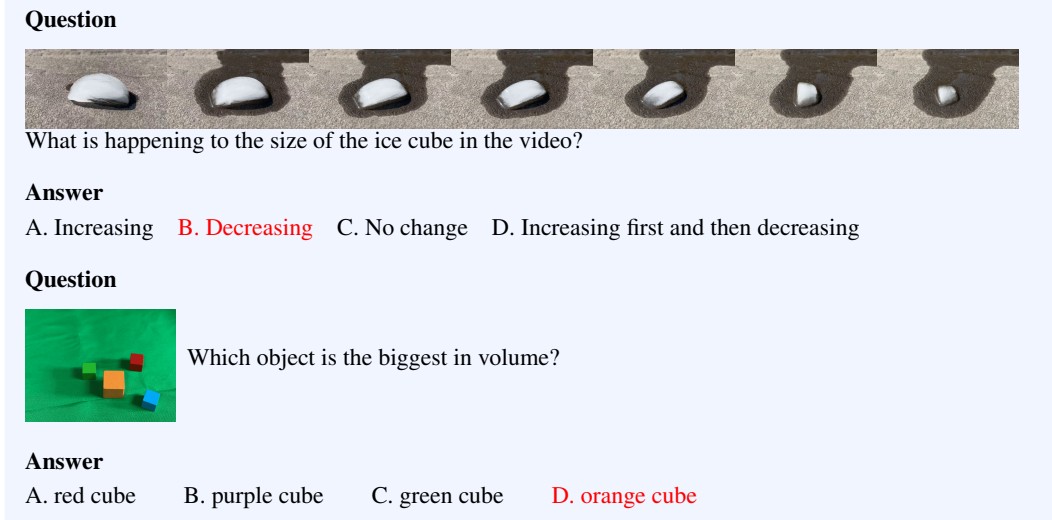

What is happening to the size of the ice cube in the video?

**Answer**

A. Increasing    B. Decreasing    C. No change    D. Increasing first and then decreasing

**Question**

Which object is the biggest in volume?

**Answer**

A. red cube    B. purple cube    C. green cube    D. orange cube

Figure 35: Two examples for relationships size. Ability Type is dynamic and static.

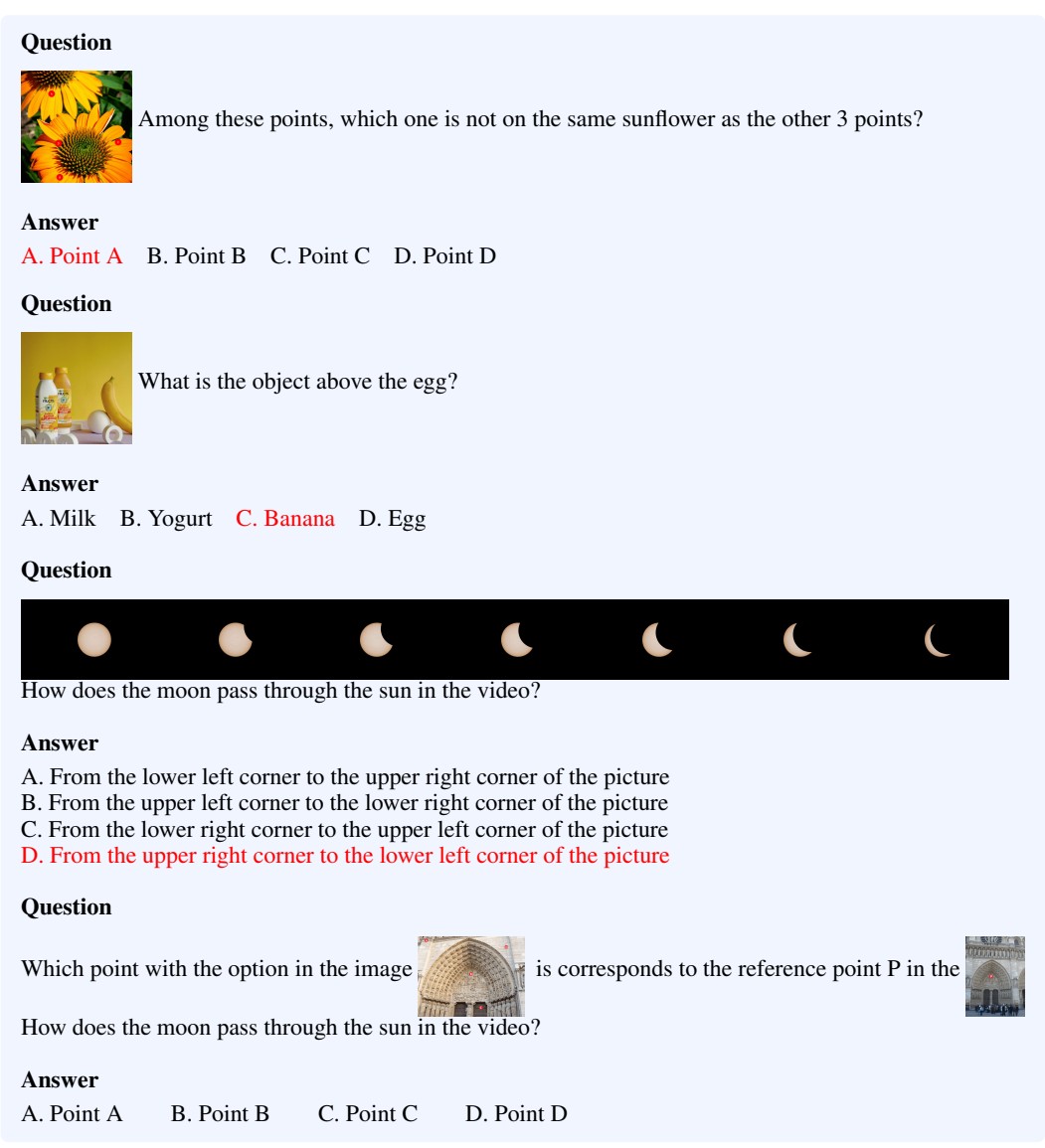

Figure 36: Four examples for relationships location. Ability Type is static, static, dynamic and static.

**Question**

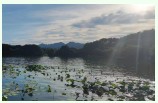 Where is the light source in the picture?

**Answer**

A. In the clouds on the upper right of the screen
B. In the lower left corner of the screen
C. A little above the center of the screen
D. In the exact center of the screen

**Question**

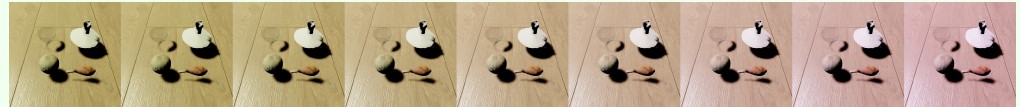

Reflecting on the events in the video, which of the following alterations to the light source is most likely to result in the phenomenon observed?

**Answer**

A. The color of the light changes from yellow to pink
B. It's just that the light source is weaker and the light source position remains the same
C. Move parallel to the line between the drumstick and the ballet skirt
D. It's just that the light source is stronger and the light source position remains the same

**Question**

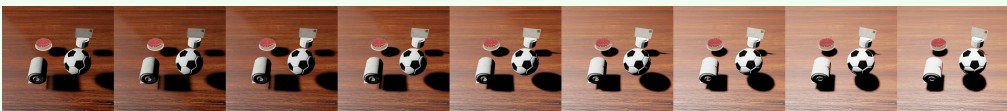

From the events in the video, which of the listed changes to the light source is most likely to have resulted in the observed phenomenon?

**Answer**

A. Move parallel to the line between the postcard and the cake
B. The color of the light changes from orange to blue
C. The color of the light changes from lime yellow to green
D. The light source moves downward

**Question**

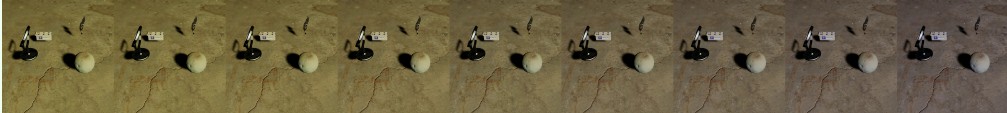

Taking into account the phenomena observed in the video, which of the following changes to the light source is most likely to have led to this result?

**Answer**

A. It's just that the light source is stronger and the light source position remains the same
B. It's just that the light source is weaker and the light source position remains the same
C. The color of the light changes from red to purple
D. The color of the light changes from yellow to blue

Figure 37: Four examples illustrating scene environmental lighting conditions. The corresponding ability types are perception, reasoning, reasoning, and reasoning.

**Question**

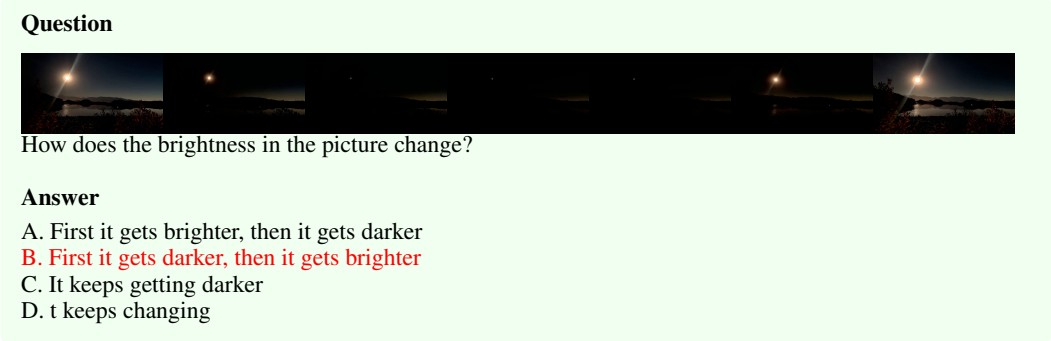

How does the brightness in the picture change?

**Answer**

A. First it gets brighter, then it gets darker
B. First it gets darker, then it gets brighter
C. It keeps getting darker
D. t keeps changing

Figure 38: Examples for scene environmental lighting conditions (Continued). Ability Type is judgement.

**Question**

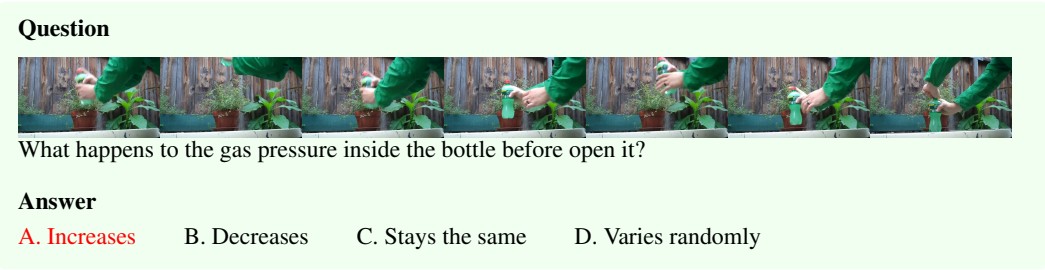

What happens to the gas pressure inside the bottle before open it?

**Answer**

A. Increases          B. Decreases          C. Stays the same          D. Varies randomly

Figure 39: Example for scene environmental air conditions and the ability type of it is perception.

**Question**

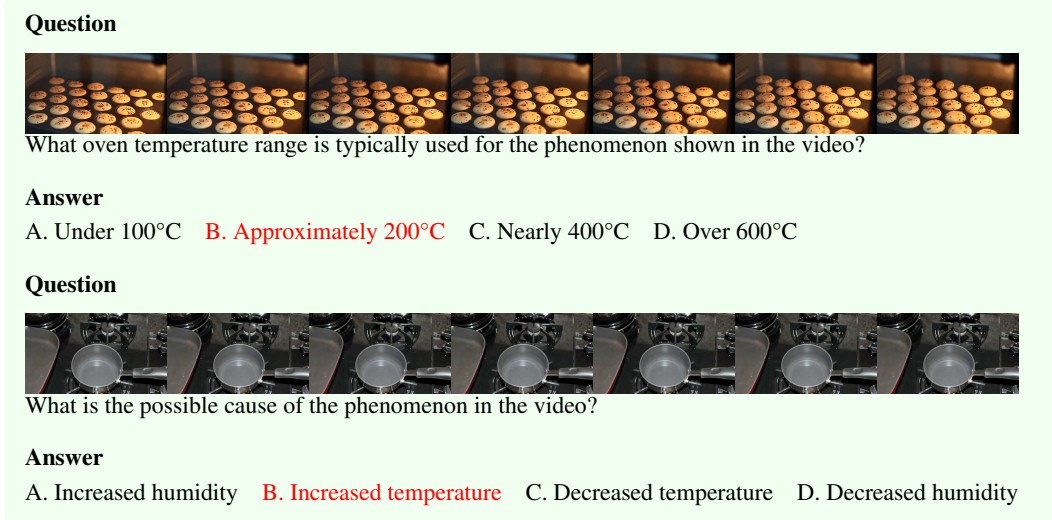

What oven temperature range is typically used for the phenomenon shown in the video?

**Answer**

A. Under 100°C    B. Approximately 200°C    C. Nearly 400°C    D. Over 600°C

**Question**

What is the possible cause of the phenomenon in the video?

**Answer**

A. Increased humidity    B. Increased temperature    C. Decreased temperature    D. Decreased humidity

Figure 40: Two examples for scene temperature conditions and the ability types of them are all perception.

**Question**

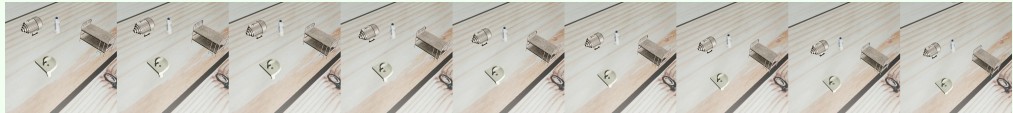

At the beginning of the video, how does the camera's focal length change?

**Answer**

A. Focus length remains unchanged
B. Focus length increases
C. Focus length decreases
D. Unknown

**Question**

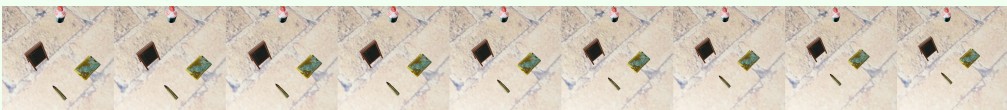

Based on the phenomenon in the video, which of the following camera changes can produce the effect in the video?

**Answer**

A. Move parallel to the line between the motor and the wash basin
B. Move parallel to the line between the bunk bed and the wash basin
C. The camera moves upward or downward
D. The camera rotates along the horizontal axis (left or right)

**Question**

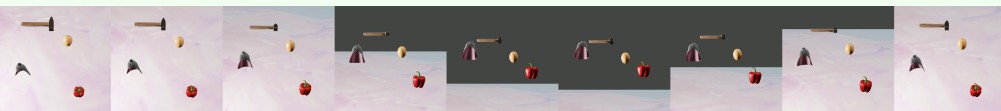

From the video, which of these camera changes could be responsible for the depicted phenomenon?

**Answer**

A. The camera is farther away from the objects
B. Move parallel to the line between the cupcake and the sponge
C. The camera is closer to the objects
D. Move parallel to the line between the cupcake and the blackboard

**Question**

Based on the phenomenon in the video, which of the following camera changes can produce the effect in the video?

**Answer**

A. The camera rotates along the vertical axis (upside or downside).
B. The camera is closer to the objects
C. The camera rotates along the horizontal axis (left or right).
D. The camera moves upward or downward

Figure 41: Four examples illustrating scene viewpoint conditions. The corresponding ability types are perception, reasoning, reasoning and reasoning.

**Question**

Select the option that shows the correct procedure to put on a pair of gloves.

image #1: 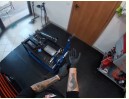    image #2: 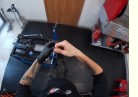    image #3: 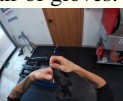

**Answer**

A. 1 - 3 - 2     B. 3 - 2 - 1
C. 2 - 1 - 3     D. 1 - 2 - 3

**Question**

Which of the following options lists the steps in the correct sequence to put the carrot in the microwave?

image #1: 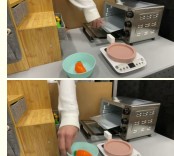    image #2: 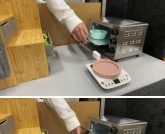

image #3: 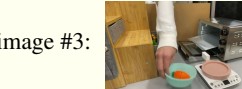    image #4:

**Answer**

A. 2 - 1 - 4 - 3     B. 1 - 3 - 2 - 4
C. 2 - 3 - 1 - 4     D. 2 - 4 - 3 - 1

**Question**

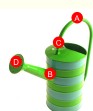 To poke the watering can, which point is most suitable?

**Answer**

A.      B.      C.      D.

**Question**

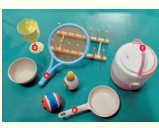 In order to pick up the cup, which of the following color points has reasonable affordance?

**Answer**

A.      B.      C.      D.

**Question**

What operation is used to transform the object from Image 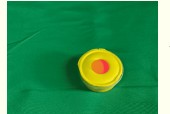 to Image 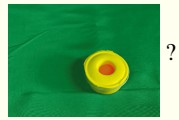 ?

**Answer**

A. Remove the small ball from the clay block.
B. Add another ball to the clay block.
C. Air-dry the clay block.
D. Press the small ball into the clay block.

Figure 42: Five examples illustrating dynamics manipulation. The corresponding ability types are judgment, judgment, perception, perception and reasoning.

**Question**

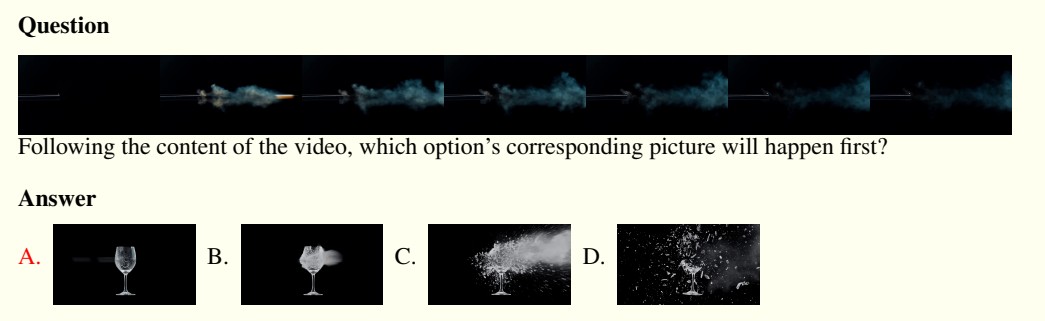

Following the content of the video, which option's corresponding picture will happen first?

**Answer**

A.     B.     C.     D.

Figure 43: Example for dynamics collision and the ability type of it is prediction.

**Question**

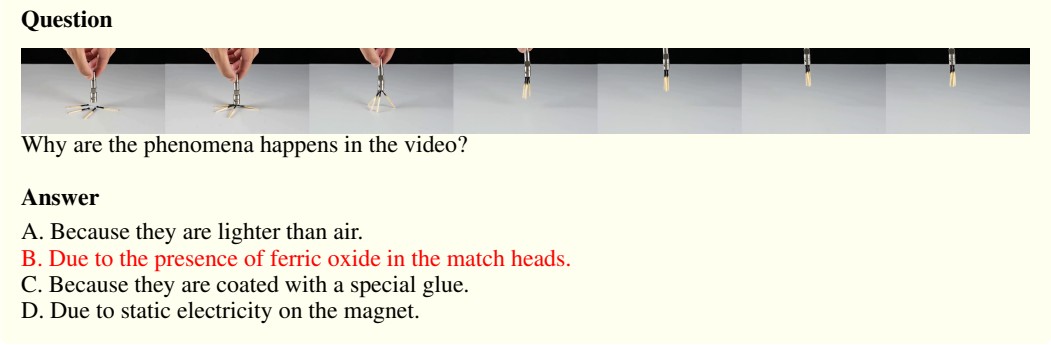

Why are the phenomena happens in the video?

**Answer**

A. Because they are lighter than air.
B. Due to the presence of ferric oxide in the match heads.
C. Because they are coated with a special glue.
D. Due to static electricity on the magnet.

Figure 44: Example for dynamics chemistry and the ability type of it is perception.

**Question**

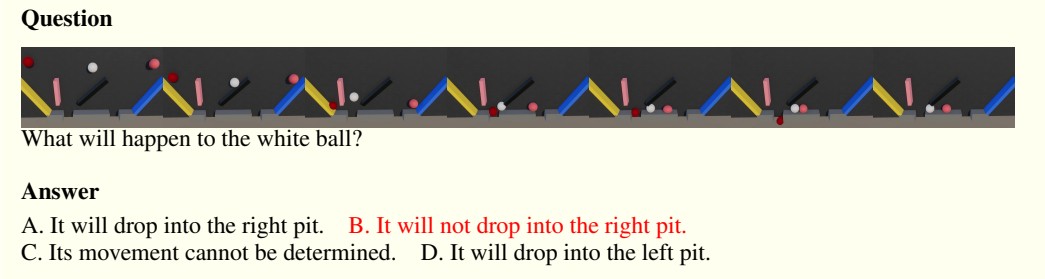

What will happen to the white ball?

**Answer**

A. It will drop into the right pit.     B. It will not drop into the right pit.
C. Its movement cannot be determined.     D. It will drop into the left pit.

Figure 45: Example for dynamics throwing and the ability type of it is prediction.

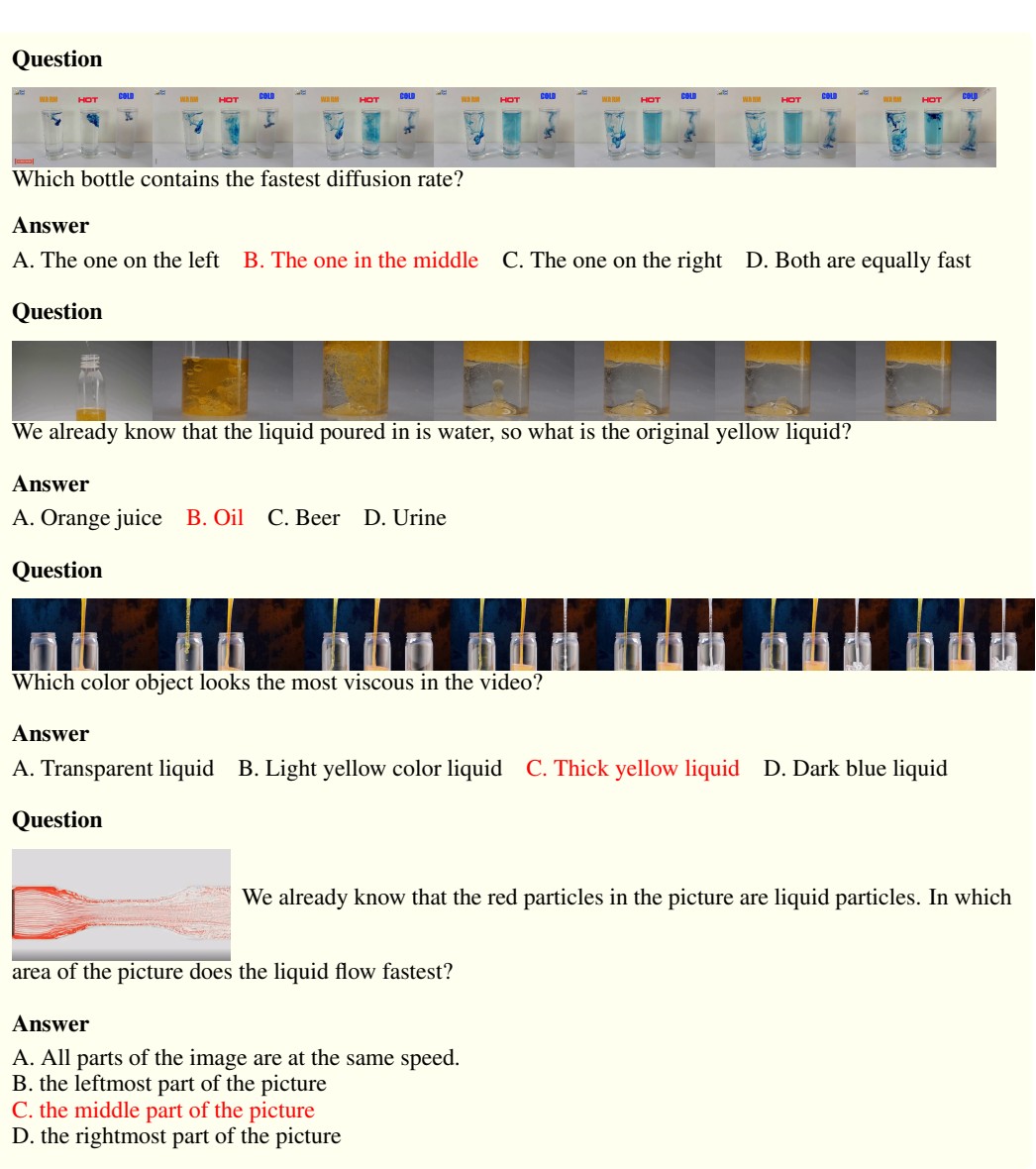

**Question**

Which bottle contains the fastest diffusion rate?

**Answer**

A. The one on the left    B. The one in the middle    C. The one on the right    D. Both are equally fast

**Question**

We already know that the liquid poured in is water, so what is the original yellow liquid?

**Answer**

A. Orange juice    B. Oil    C. Beer    D. Urine

**Question**

Which color object looks the most viscous in the video?

**Answer**

A. Transparent liquid    B. Light yellow color liquid    C. Thick yellow liquid    D. Dark blue liquid

**Question**

We already know that the red particles in the picture are liquid particles. In which area of the picture does the liquid flow fastest?

**Answer**

A. All parts of the image are at the same speed.
B. the leftmost part of the picture
C. the middle part of the picture
D. the rightmost part of the picture

Figure 46: Four examples illustrating dynamics fluid. The corresponding ability types are perception, reasoning, perception and perception.

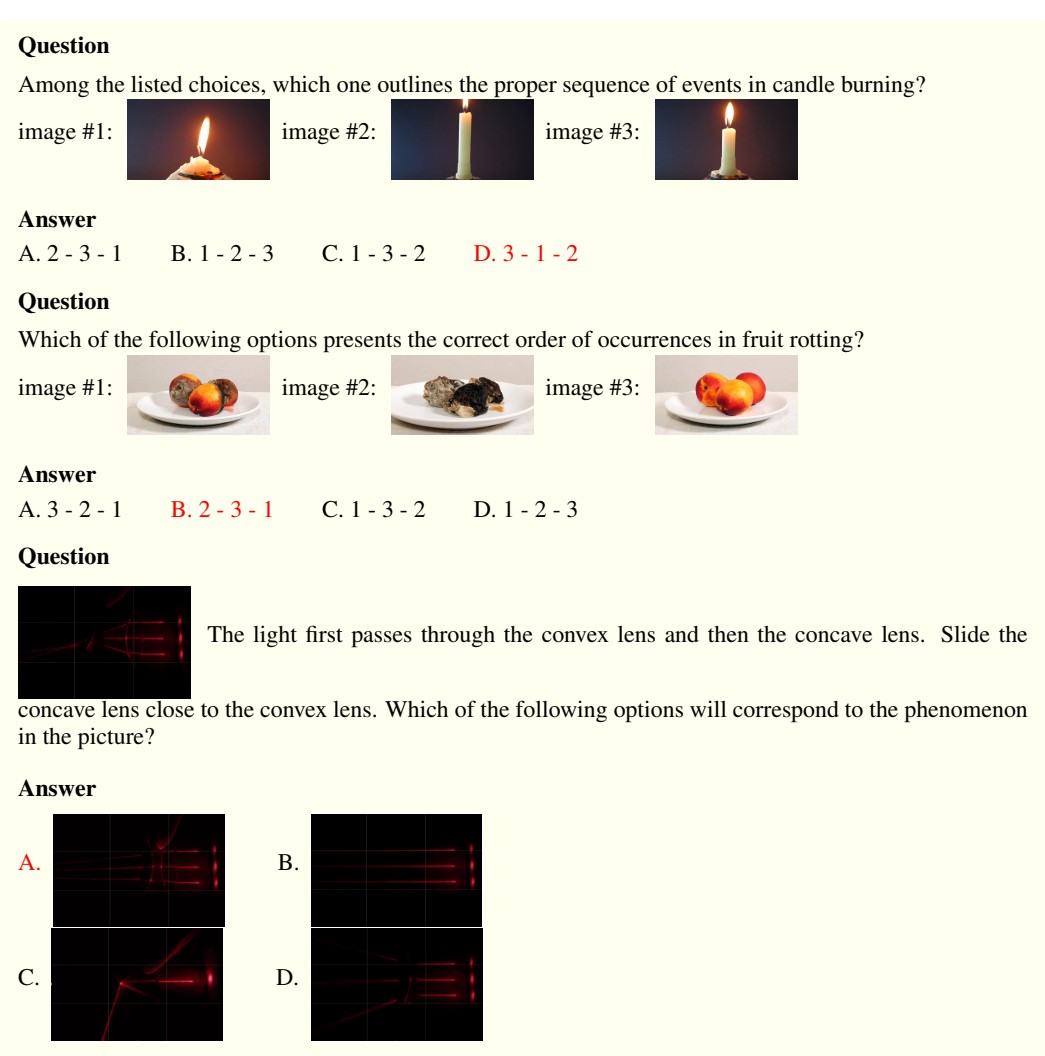

Figure 47: Three examples illustrating dynamics others. The corresponding ability types are reasoning, reasoning and prediction.

# I ERROR STUDY

## I.1 DETAILED STATICS

In this section, we present a case study analysis of the error types made by GPT-4V, Gemini-1.5-flash, and Phi-3V across various tasks. The errors are classified into the following five categories. Other less frequent error types are not included in this analysis. For the analysis, as mentioned in Section 3.4, we selected 500 samples for each model, but due to space limitations, we present only 60 of them here, as shown in Table 27.

**Perception Error** : VLMs fail to recognize, classify, or detect the objects or content in images. VLMs are constrained by the representation power of visual encoders, and these errors account for the majority of them. See examples in Figure 48, Figure 50, *etc.*

**Reasoning Error** : VLMs can recognize the text and visual content exactly but make errors in reasoning, leading to incorrect results. See examples in Figure 65, Figure 81, *etc.*

**Lack of Knowledge** : VLMs do not have specific knowledge, so they finally get a wrong answer. See examples in Figure 57, Figure 71.

**Refuse to Answer** : VLMs refuse to answer questions and stop answering immediately. See examples in Figure 54.

**Fail to Follow Instruction** : VLMs fail to correctly understand instructions and provide erroneous answers. For example, VLMs may not understand the specified conditions in the instruction (see Figure 74).

## I.2 MAIN REASON ANALYSIS

**Perceptual Errors**: Perceptual errors can be classified into two categories: basic perceptual errors and domain-specific perceptual errors. Basic perceptual errors occur when the model successfully understands the given information but fails to interpret fundamental visual objects correctly. In contrast, domain-specific perceptual errors arise when the model misinterprets visual inputs due to a lack of understanding of specialized conditions. Moreover, GPT-4V often exhibits a bias towards textual information, prioritizing text over visual inputs—a trend highlighted in recent studies Cui et al. (2023). A notable example is shown in Figure 68, where the model mistakenly identified the plate as part of the background, leading to incorrect depth estimation. This underscores the importance of seeking a balanced approach to enhance the model's interpretative capabilities.

**Reasoning Errors**: Sometimes, even when the model correctly interprets the text, images, and question, it fails to establish a rigorous logical chain, leading to incorrect conclusions. These types of mistakes, known as reasoning errors, are illustrated in Figures 65 and 78. In the first example, the model mistakenly assumes that friction can only exist if the machine rotates faster than the object, overlooking the presence of another object, which results in an incorrect answer. In the second example, while the model correctly identifies that the temperature of the wrapped paper is significantly lower than that of the other object, it incorrectly attributes this phenomenon to the flame's temperature, leading to an erroneous conclusion.

**Lack of Knowledge**: Another major cause of errors is the model's lack of relevant knowledge. A notable example is shown in Figure 71, where the model incorrectly assumes that light bending in water is due to refraction. The model clearly does not understand the concept of total internal reflection in water, leading to an incorrect conclusion.

**Other Errors**: The remaining errors, such as those related to textual understanding, refusal to answer, annotation mistakes, and answer extraction issues, account for only a small proportion. However, they remain significant and should not be overlooked.

## I.3 CASE STUDY

Table 27: Table index of case study figures by meta-task with associated error categories.

| Case Figure | Meta-task | Subtask | GPT-4o | Gemini-1.5-flash | Phi-3V |
|---|---|---|---|---|---|
| Figure 48 | Scene | Light | Perception Error | Perception Error | Perception Error |
| Figure 49 | Dynamics | Collision | Reasoning Error | Perception Error | Reasoning Error |
| Figure 50 | Dynamics | Throwing | Perception Error | Perception Error | Perception Error |
| Figure 51 | Scene | Light | Perception Error | Perception Error | Perception Error |
| Figure 52 | Scene | Viewpoint | Perception Error | Reasoning Error | Reasoning Error |
| Figure 53 | Scene | Viewpoint | Perception Error | Reasoning Error | Success |
| Figure 54 | Dynamics | Chemistry | Reasoning Error | Refuse to Answer | Success |
| Figure 55 | Dynamics | Manipulation | Perception Error | Perception Error | Perception Error |
| Figure 56 | Dynamics | Manipulation | Perception Error | Perception Error | Perception Error |
| Figure 57 | Property | Attribute | Perception Error | Perception Error | Reasoning Error |
| Figure 58 | Property | Color | Perception Error | Reasoning Error | Perception Error |
| Figure 59 | Property | Mass | Reasoning Error | Success | Perception Error |
| Figure 60 | Property | Mass | Success | Success | Success |
| Figure 61 | Property | Number | Perception Error | Perception Error | Refuse to Answer |
| Figure 62 | Property | Number | Perception Error | Perception Error | Success |
| Figure 63 | Property | Attribute | Lack of Knowledge | Lack of Knowledge | Perception Error |
| Figure 64 | Relationships | Motion | Perception Error | Perception Error | Reasoning Error |
| Figure 65 | Relationships | Motion | Reasoning Error | Reasoning Error | Reasoning Error |
| Figure 66 | Relationships | Depth | Reasoning Error | Reasoning Error | Success |
| Figure 67 | Relationships | Depth | Success | Success | Success |
| Figure 68 | Relationships | Distance | Perception Error | Reasoning Error | Success |
| Figure 69 | Relationships | Distance | Reasoning Error | Reasoning Error | Lack of Knowledge |
| Figure 70 | Relationships | Size | Perception Error | Reasoning Error | Reasoning Error |
| Figure 71 | Dynamics | Others | Lack of Knowledge | Lack of Knowledge | Lack of Knowledge |
| Figure 72 | Relationships | Size | Success | Success | Success |
| Figure 73 | Scene | Viewpoint | Perception Error | Perception Error | Success |
| Figure 74 | Relationships | Location | Perception Error | Perception Error | Fail to follow instruction |
| Figure 75 | Relationships | Location | Perception Error | Perception Error | Reasoning Error |
| Figure 76 | Scene | Viewpoint | Perception Error | Perception Error | Success |
| Figure 77 | Scene | Temperature | Perception Error | Perception Error | Perception Error |
| Figure 78 | Scene | Temperature | Reasoning Error | Reasoning Error | Reasoning Error |
| Figure 79 | Dynamics | Air | Perception Error | Perception Error | Perception Error |
| Figure 80 | Dynamics | Air | Reasoning Error | Reasoning Error | Reasoning Error |
| Figure 81 | Dynamics | Manipulation | Reasoning Error | Reasoning Error | Reasoning Error |
| Figure 82 | Relationships | Depth | Reasoning Error | Perception Error | Perception Error |
| Figure 83 | Dynamics | Fluid | Perception Error | Perception Error | Perception Error |
| Figure 84 | Dynamics | Others | Perception Error | Perception Error | Perception Error |
| Figure 85 | Dynamics | Others | Perception Error | Perception Error | Perception Error |
| Figure 86 | Dynamics | Collision | Perception Error | Perception Error | Perception Error |
| Figure 87 | Scene | Light | Reasoning Error | Perception Error | Success |
| Figure 88 | Scene | Viewpoint | Perception Error | Perception Error | Perception Error |
| Figure 89 | Relationships | Location | Success | Perception Error | Perception Error |
| Figure 90 | Property | Attribute | Perception Error | Reasoning Error | Perception Error |
| Figure 91 | Dynamics | Manipulation | Perception Error | Success | Perception Error |
| Figure 92 | Scene | Viewpoint | Perception Error | Perception Error | Perception Error |
| Figure 93 | Scene | Light | Perception Error | Perception Error | Perception Error |
| Figure 94 | Relationships | Location | Perception Error | Perception Error | Perception Error |
| Figure 95 | Relationships | Location | Perception Error | Perception Error | Perception Error |
| Figure 96 | Dynamics | Others | Reasoning Error | Reasoning Error | Reasoning Error |
| Figure 97 | Scene | Viewpoint | Perception Error | Reasoning Error | Success |
| Figure 98 | Scene | Light | Reasoning Error | Reasoning Error | Reasoning Error |
| Figure 99 | Dynamics | Manipulation | Perception Error | Success | Reasoning Error |
| Figure 100 | Dynamics | Collision | Perception Error | Perception Error | Success |
| Figure 101 | Property | Attribute | Perception Error | Perception Error | Reasoning Error |
| Figure 102 | Scene | Light | Success | Reasoning Error | Success |
| Figure 103 | Scene | Light | Perception Error | Perception Error | Perception Error |
| Figure 104 | Scene | Light | Perception Error | Perception Error | Reasoning Error |
| Figure 105 | Scene | Light | Success | Perception Error | Reasoning Error |
| Figure 106 | Scene | Light | Reasoning Error | Perception Error | Reasoning Error |
| Figure 107 | Scene | Light | Perception Error | Perception Error | Perception Error |

## Scene: Light

**Question:** According to the video, which alteration to the light source is most likely to have triggered the phenomenon? <video 1>

**Option:**
(A) The color of the light changes from yellow to cyan
(B) The color of the light changes from green to red
(C) It's just that the light source is weaker and the light source position remains the same
(D) It's just that the light source is stronger and the light source position remains the same

<video 1>

The answer is C. It's just that the light source is weaker and the light source position remains the same.

Here's the reasoning:

1. Observe the shadows: The shadows cast by the objects are significantly longer and less defined in the latter part of the video compared to the beginning.
2. Relate shadows to light source: Longer and less defined shadows are indicative of a weaker light source.
3. Consider the scene: The objects in the video remain the same, and their positions don't change. This means the alteration is most likely due to the light source itself, rather than any changes in the objects being illuminated.
4. Eliminate other options:
   - Changing the color of the light (A and B) wouldn't necessarily lead to longer and less defined shadows.
   - A stronger light source (D) would result in shorter and sharper shadows.

Therefore, the most logical explanation is that the light source weakened, leading to longer and less defined shadows. ❌

Choice: C. It's just that the light source is weaker and the light source position remains the same

Reasoning Chain:

1. Observation of Images:
   - The series of images show a consistent scene with a ball, a backpack, and some other objects on a wooden surface.
   - The shadows and lighting on the objects remain consistent in terms of direction and shape throughout the series of images.

2. Analysis of Light Intensity:
   - The primary noticeable change in the images is the overall brightness.
   - The images progressively become darker, indicating a reduction in the intensity of the light source.

3. Elimination of Color Change Options:
   - Option A (Yellow to Cyan): A change in light color from yellow to cyan would result in a noticeable shift in the color tones of the objects and shadows. This is not observed in the images.
   - Option B (Green to Red): Similarly, a change from green to red would also alter the color tones significantly, which is not evident in the images.

4. Elimination of Light Source Strengthening:
   - Option D (Stronger Light Source): If the light source were getting stronger, the images would progressively become brighter. This is contrary to what is observed.

5. Conclusion:
   - The consistent direction and shape of the shadows indicate that the position of the light source has not changed.
   - The progressive darkening of the images suggests that the light source is becoming weaker. ❌

Therefore, the most likely alteration to the light source that triggered the phenomenon is that the light source is weaker while its position remains the same.

The phenomenon in the image is the casting of shadows on the wooden floor, which suggests that there is a light source above the objects. The shadows are consistent in shape and size, indicating that the light source is not moving. The most likely alteration to the light source that would trigger this phenomenon is a change in the light source's intensity. Therefore, the correct answer is C. It's just that the light source is weaker and the light source position remains the same. ❌

Figure 48: A sample case of physical scene understanding–light. Back to List of Figures.

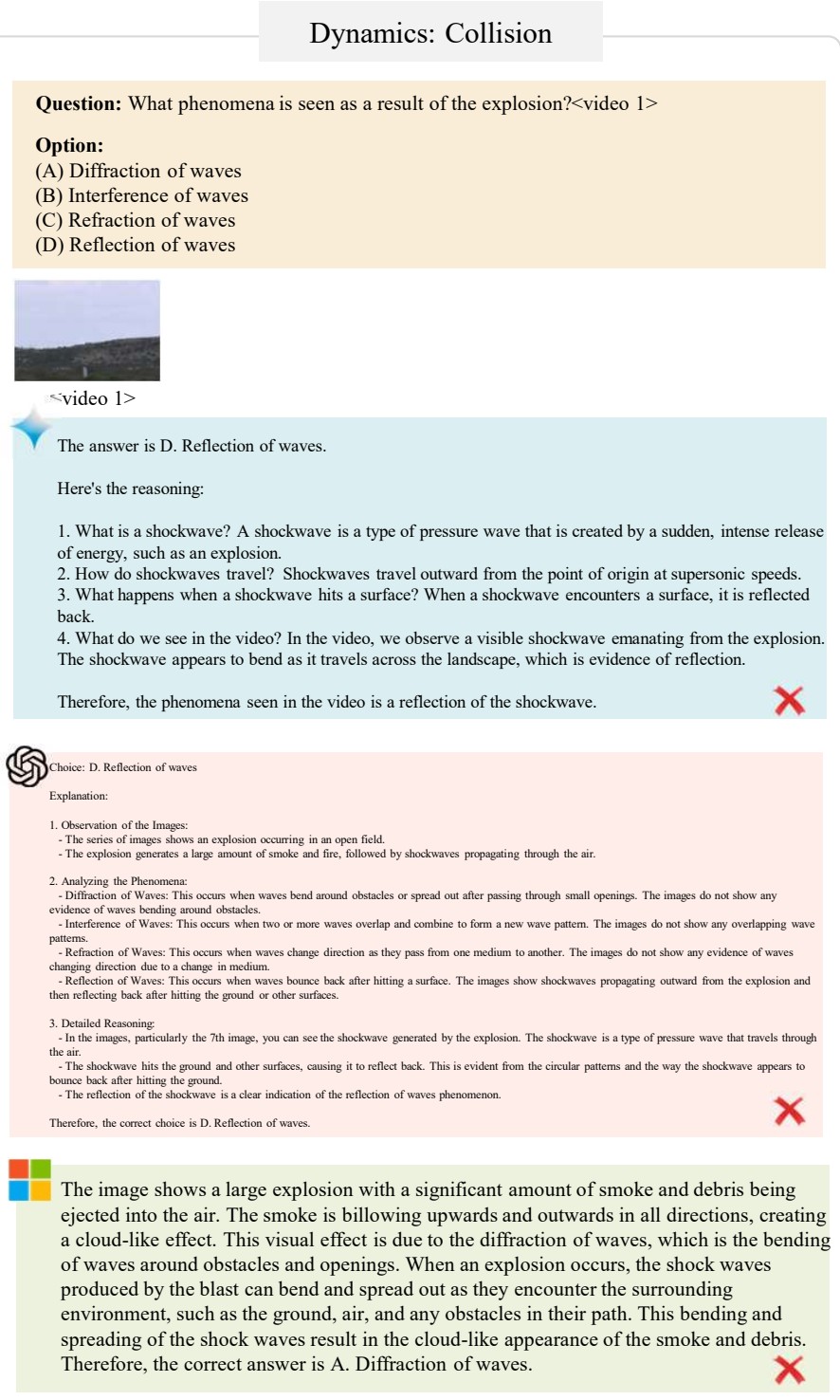

Figure 49: A sample case of physics-based dynamics——collision. Back to List of Figures.

## Dynamics: Throwing

**Question:** If we removed the green floating wall and other balls, which pit would the black ball drop into?<video 1>

**Option:**
(A) The left pit
(B) The right pit
(C) None of the above
(D) Both pits

<video 1>

The answer is A. The left pit.

Here is the reasoning:

1. The black ball is sitting on the second platform from the left.
2. There is a white ball to the right of the black ball.
3. The green floating wall is the only thing preventing the black ball from falling into the left pit.
4. Removing the green wall would remove the obstacle between the black ball and the left pit.
5. Therefore, the black ball would fall into the left pit. ✗

Choice: B. The right pit

Reasoning:

1. Initial Position and Movement:
   - The black ball starts near the top of the screen, positioned above the right pit.
   - The green floating wall is currently preventing the black ball from falling directly into the right pit.

2. Removing the Green Floating Wall:
   - If the green floating wall is removed, the black ball will no longer have an obstacle in its path.
   - The black ball will fall straight down due to gravity.

3. Path of the Black Ball:
   - Without the green floating wall, the black ball will fall directly downward.
   - The black ball is positioned directly above the right pit.

4. Conclusion:
   - Since the black ball is directly above the right pit and there are no other obstacles in its path, it will fall into the right pit.

Therefore, the black ball will drop into the right pit if the green floating wall and other balls are removed. ✗

To determine where the black ball would drop into, we need to analyze the image and the position of the pits. The black ball is located above the left pit and below the right pit. Since the pits are not visible from the front, we cannot see if there is a barrier or if the pits are empty. However, based on the position of the black ball, it is more likely to drop into the pit on the left side of the image. Therefore, the answer is A. The black ball would drop into the left pit. ✗

Figure 50: A sample case of physics-based dynamics——throwing. Back to List of Figures.

## Scene: Light

**Question:** Observing the phenomena in the video, which of the following alterations to the light source is most likely to produce such an effect?<video 1>

**Option:**
(A) It's just that the light source is weaker and the light source position remains the same
(B) It's just that the light source is stronger and the light source position remains the same
(C) The color of the light changes from limeyellow to blue
(D) The color of the light changes from red to pink

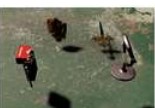

<video 1>

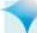 The answer is A. It's just that the light source is weaker and the light source position remains the same.

Here's the reasoning:

1. Observation: The objects in the video cast elongated shadows, indicating that the light source is positioned at a low angle. The shadows are also relatively faint and indistinct, suggesting the light source is not very strong.

2. Elimination:
    * B. Stronger light source: A stronger light source would produce sharper, more defined shadows. The shadows in the video are too blurry for a strong light source.
    * C & D. Color changes: While color changes can affect the perceived appearance of objects, they wouldn't directly cause the elongated and faint shadows observed in the video.

3. Conclusion: The most likely explanation is that the light source is weaker. A weaker light source produces less intense light, resulting in softer, more blurred shadows. The low angle of the light source would still contribute to the elongated shadow shapes.

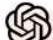 Choice: B. It's just that the light source is stronger and the light source position remains the same

Detailed Explanation:

1. Observation of Shadows:
    - In the series of images, the shadows of the objects on the ground are clearly visible and consistent in shape and position.
    - The shadows do not change in size, shape, or direction, indicating that the position of the light source remains constant.

2. Intensity of Shadows:
    - The shadows appear to be well-defined and dark, suggesting that the light source is strong and casting distinct shadows.
    - If the light source were weaker, the shadows would be less defined and lighter in appearance.

3. Color of Light:
    - The images do not show any noticeable change in the color of the light. The ground and objects maintain a consistent color throughout the series of images.
    - If the color of the light had changed (options C or D), there would be a visible shift in the color tones of the objects and the ground.

4. Conclusion:
    - Since the shadows are well-defined and there is no change in the color of the light, the most likely alteration is that the light source has become stronger while its position remains the same.
    - Therefore, the correct choice is B. It's just that the light source is stronger and the light source position remains the same.

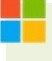 The most likely explanation for the observed effect is that the light source is weaker and the light source position remains the same. This is because a weaker light source would result in less intense shadows, making them appear more diffuse and less defined. Additionally, if the position of the light source remains the same, the shadows would still be cast in the same direction, but they would be less pronounced due to the reduced intensity of the light. This would result in the shadows appearing more blurred and less distinct, as seen in the video. Therefore, the correct answer is A. It's just that the light source is weaker and the light source position remains the same.

Figure 51: A sample case of physical scene understanding–light. Back to List of Figures.

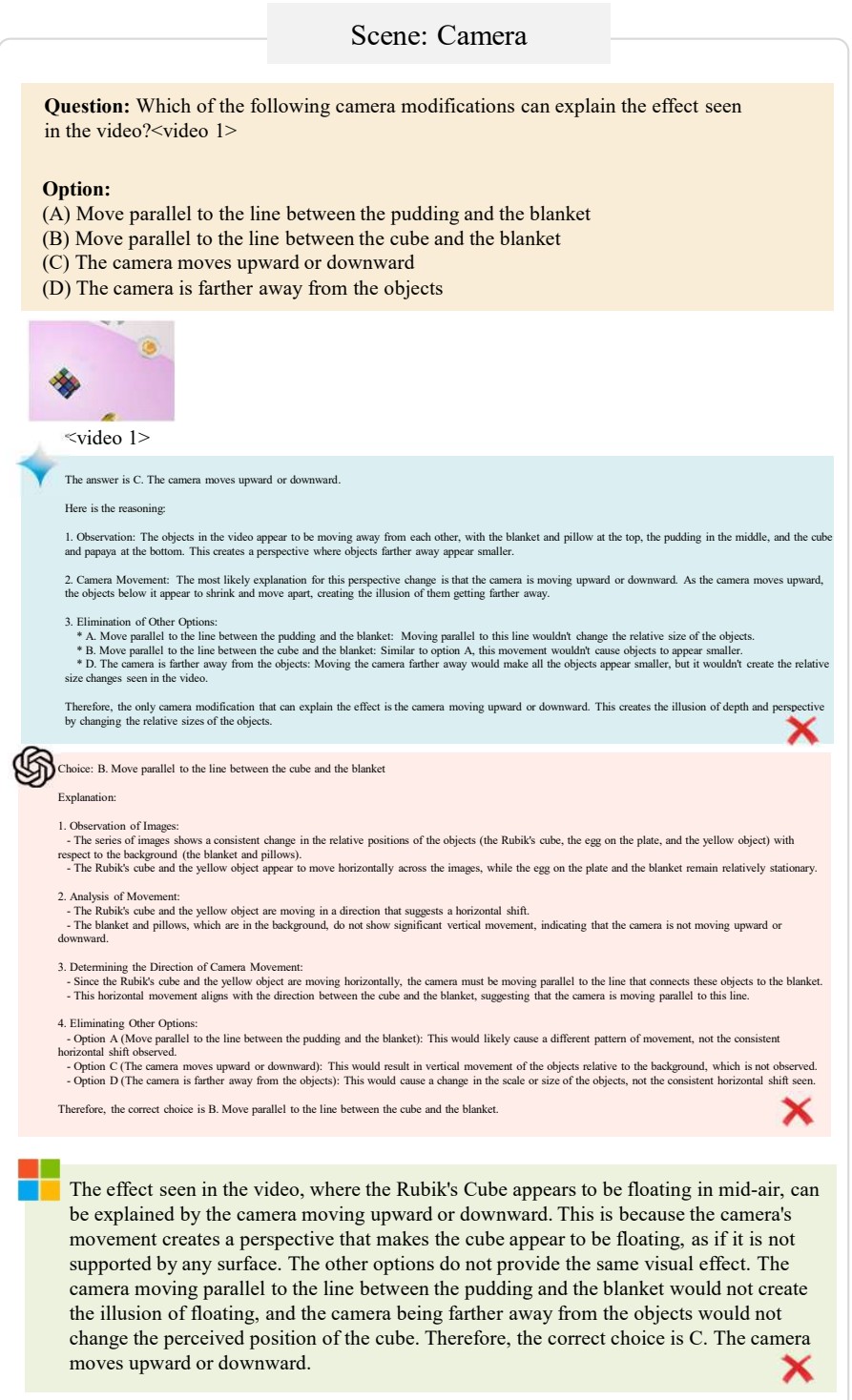

Figure 52: A sample case of physical scene understanding–viewpoint. Back to List of Figures.

## Scene: Camera

**Question:** In light of the effects observed in the video, which of the following camera adjustments might achieve the depicted outcome?<video 1>

**Option:**
(A) The camera is closer to the objects
(B) Move parallel to the line between the cereal and the skiboot
(C) Move parallel to the line between the cereal and the cymbal
(D) The camera is farther away from the objects

<video 1>

The answer is D. The camera is farther away from the objects.

Here's the step-by-step reasoning:

1. Observe the effect: The objects in the video appear smaller and further apart than they likely are in reality. This suggests a shift in perspective.

2. Consider camera positioning: A change in perspective is primarily achieved by moving the camera closer or further away from the scene.

3. Eliminate options:
   - A. The camera is closer to the objects: This would make the objects appear larger and closer together, the opposite of what we see in the video.
   - B. Move parallel to the line between the cereal and the skiboot: This movement would change the angle of view, but not necessarily the overall size or distance between objects.
   - C. Move parallel to the line between the cereal and the cymbal: Similar to option B, this would adjust the angle but not the scale of the objects.

4. Conclusion: The only remaining option, D, is consistent with the observed effect. Moving the camera further away from the objects would make them appear smaller and further apart, matching the video. ❌

C. Move parallel to the line between the cereal and the cymbal

Detailed Explanation:

1. Observation of Movement:
   - In the series of images, the objects appear to shift positions relative to each other. Specifically, the cereal box and the cymbal seem to change their relative positions more significantly compared to the other objects.

2. Relative Position Analysis:
   - The cereal box and the cymbal are positioned on opposite sides of the frame. As the images progress, the cereal box moves from the left side of the frame towards the right, while the cymbal moves from the right side of the frame towards the left.

3. Determining Camera Movement:
   - The observed effect suggests that the camera is moving in a direction that changes the relative positions of the cereal box and the cymbal. This indicates a movement parallel to the line connecting these two objects.

4. Eliminating Other Options:
   - Option A (The camera is closer to the objects): This would result in the objects appearing larger but would not necessarily change their relative positions in the way observed.
   - Option B (Move parallel to the line between the cereal and the skiboot): This would primarily affect the relative positions of the cereal and the skiboot, but the significant changes are observed between the cereal and the cymbal.
   - Option D (The camera is farther away from the objects): This would result in the objects appearing smaller but would not necessarily change their relative positions in the way observed.

5. Conclusion:
   - The most plausible explanation for the observed changes in the relative positions of the objects is that the camera is moving parallel to the line between the cereal and the cymbal. This movement would cause the cereal box and the cymbal to appear to shift positions relative to each other, as seen in the images. ❌

To achieve the depicted outcome, where the objects appear larger and the background appears smaller, the camera needs to be closer to the objects. This is because the perspective of the image is such that the objects are in the foreground and the background is in the distance, creating a sense of depth. By moving the camera closer, the objects will appear larger due to the decrease in distance, and the background will appear smaller due to the increase in distance. Therefore, the correct choice is:

A. The camera is closer to the objects.

This choice is based on the principles of perspective and depth of field in photography, where objects closer to the camera appear larger, and objects further away appear smaller. The other options (B, C, and D) do not align with the desired outcome of making the objects appear larger and the background smaller. ✅

Figure 53: A sample case of physical scene understanding–viewpoint. Back to List of Figures.

## Dynamics: Chemistry

**Question:** What action is displayed in the video?<video 1>

**Option:**
(A) Candle being blown out deliberately
(B) Candle causing wax to overflow
(C) Candle burning with a double wick
(D) Candle being used in a romantic setting

<video 1>

The answer is C. Candle burning with a double wick.

Here's the reasoning:

1. Visual Cues: The video shows a tall, cylindrical candle with a honeycomb pattern. The flame at the top suggests the candle is burning.

2. Flame Observation: The flame appears to be unusually tall and wide, indicating a larger-than-normal amount of fuel being consumed.

3. Double Wick Inference: This suggests that the candle ✗

B. Candle causing wax to overflow

Detailed Explanation:
1. Observation of Candle Burning: The series of images shows a candle burning over time. The flame is consistent and does not show signs of being blown out deliberately (which would typically show a sudden change in the flame's behavior).

2. Wax Overflow: In the last two images, there is a noticeable overflow of wax on the left side of the candle. This indicates that the candle has been burning for a while, causing the wax to melt and overflow.

3. No Double Wick: The images do not show any indication of a double wick. The flame is singular and centered, which is typical for a single-wick candle.

4. No Romantic Setting: The background is a solid red color with no additional elements that would suggest a romantic setting (such as dim lighting, additional candles, or romantic decor).

Based on these observations, the correct choice is B. Candle causing wax to overflow. ✗

The image shows a single candle with a flame at the top, set against a red background with repeated logos of 'THE 100% NATURAL ROLLING CANDLES' and 'THE NATURAL CANDLE FACTORY EST. 1969'. The repeated logos suggest that the candle is a product being advertised, and the presence of the flame indicates that the candle is currently burning. The action displayed in the video is not explicitly shown, but based on the context of the image, it is reasonable to infer that the candle is being used, likely for its intended purpose, which could be for lighting, ambiance, or decoration. The most likely action is that the candle is being used in a romantic setting, as candles are often associated with romance and intimate settings. Therefore, the choice is:

D. Candle being used in a romantic setting ✓

Figure 54: A sample case of physics-based dynamics——chemistry. Back to List of Figures.

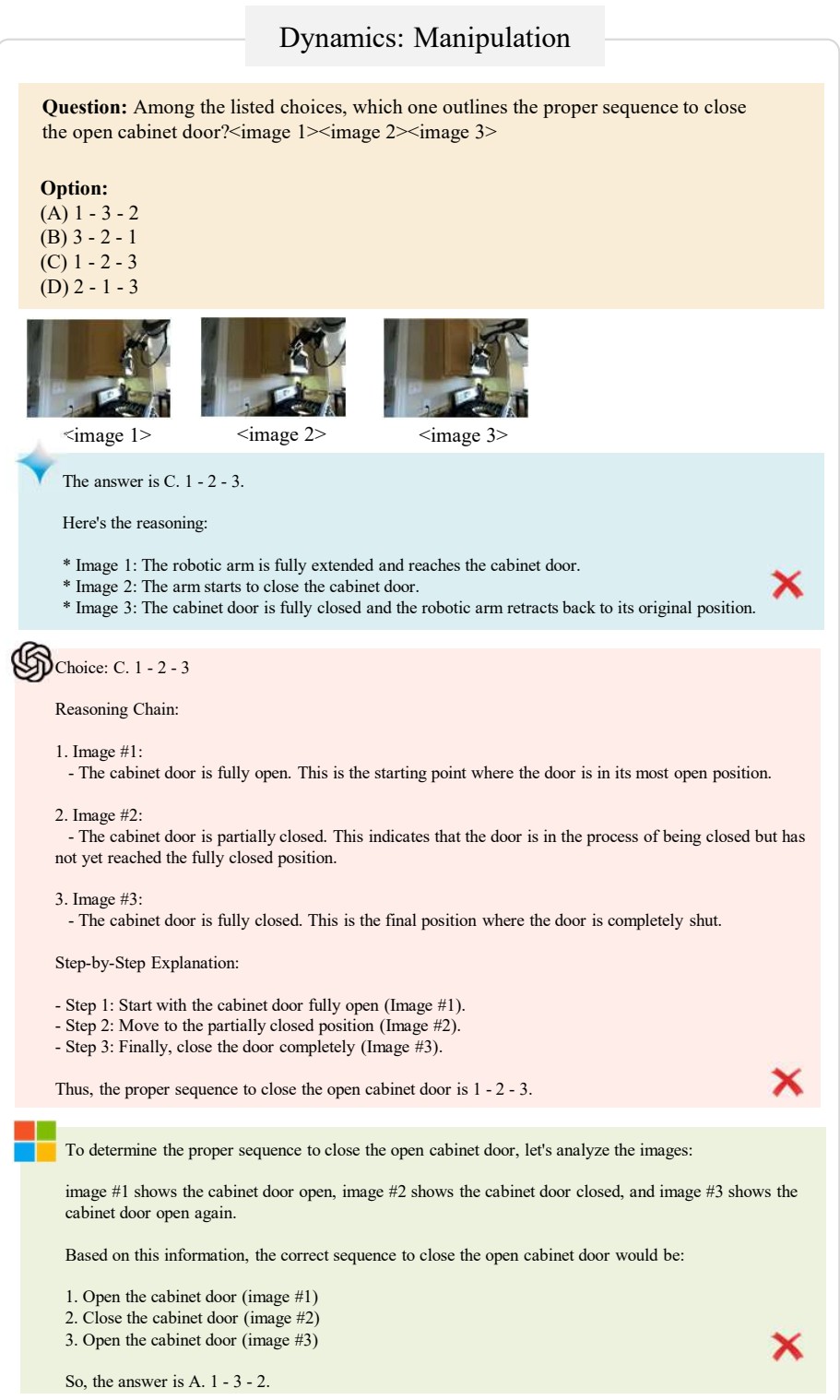

Figure 55: A sample case of physics-based dynamics——manipulation. Back to List of Figures.

## Dynamics: Manipulation

**Question:** Which of the provided options demonstrates the accurate steps to close the microwave?<video 1>

**Option:**
(A) Remove the blue floating wall and other balls
(B) Remove the green floating wall and other balls
(C) None of the above
(D) Remove both the blue and green floating walls

<image 1>          <image 2>          <image 3>

The answer is B. 1 - 2 - 3.

The robot arm first moves towards the microwave door (image #1), then presses the door down to close it (image #2), and finally, it moves back to its original position (image #3). ✗

Choice: A. 1 - 3 - 2

Reasoning Chain:

1. Image #1:
   - The robot arm is approaching the microwave door, which is open. This is the initial step where the robot is preparing to close the door.

2. Image #3:
   - The robot arm is now in contact with the microwave door, starting to push it closed. This is the intermediate step where the door is being moved towards the closed position.

3. Image #2:
   - The microwave door is fully closed, and the robot arm is no longer in contact with it. This is the final step where the door has been successfully closed.

Therefore, the correct sequence of steps to close the microwave is 1 - 3 - 2. ✗

The image shows a kitchen with a microwave oven and a refrigerator. The correct sequence to close the microwave would be to first ensure that the door is open, then to close the door, and finally to secure it with the latch. Therefore, the correct sequence is:

1. Open the microwave door.
2. Close the microwave door.
3. Close the latch.

The correct option is B. 1 - 2 - 3. ✗

Figure 56: A sample case of physics-based dynamics——throwing. Back to List of Figures.

## Property: Attribute

**Question:** How does the flexibility of the left cloth compare to the other cloth?<video 1>

**Option:**
(A) It is more flexible
(B) It is equally flexible
(C) It is harder to bend
(D) It is softer

<video 1>

A. It is more flexible

Here's the reasoning:

1. Observation: We see two cloths on the table, one brown and one plaid.
2. Cloth Movement: Both cloths are draped over the objects on the table. The brown cloth drapes in a more fluid and uneven manner.
3. Flexibility Inference: The more fluid and uneven drape of the brown cloth indicates that it can bend and conform to the shape of the objects more easily. This suggests it is more flexible.
4. Comparison: The plaid cloth, on the other hand, drapes in a more rigid and uniform manner, implying less flexibility.

Therefore, based on the way the cloths drape, we can infer that the brown cloth is more flexible than the plaid cloth.

A. It is more flexible

Step-by-step explanation:

1. Initial Observation: In the first two images, we see two cloths placed on a table with some objects underneath them. The left cloth is orange, and the right cloth is checkered.

2. Cloth Placement: In the third image, both cloths are being placed on the objects. The left cloth (orange) and the right cloth (checkered) are both pinned at the corners.

3. Cloth Behavior: As the cloths are draped over the objects, we can observe their behavior:
   - In the fourth image, the left cloth (orange) starts to show more pronounced draping and bending around the object compared to the right cloth (checkered).
   - In the fifth image, the left cloth (orange) continues to show more flexibility, conforming more closely to the shape of the object underneath it.
   - In the sixth and seventh images, the left cloth (orange) appears to be more fluid and flexible, showing more folds and curves compared to the right cloth (checkered), which maintains a more structured and less flexible appearance.

4. Final Observation: By the last image, the left cloth (orange) has clearly shown more flexibility in its ability to drape and conform to the shape of the object underneath it, while the right cloth (checkered) appears stiffer and less flexible.

Based on these observations, the left cloth (orange) is more flexible compared to the right cloth (checkered).

To determine the flexibility of the left cloth compared to the other cloth, we need to analyze the image and look for any differences in the way the cloths are positioned or how they appear to move. The left cloth is positioned on the left side of the table, while the other cloth is on the right side. We can observe that the left cloth is not as stretched out as the other cloth, which is more extended towards the right side of the table. This suggests that the left cloth is less flexible than the other cloth. Therefore, the choice is A. It is less flexible.

Figure 57: A sample case of physical object property–attribute. Back to List of Figures.

## Property: Color

**Question:** What color does the chameleon turn from blue in the video?<video 1>

**Option:**
(A) Red
(B) Grey
(C) Purple
(D) Orange

<video 1>

The answer is A. Red.

Here's the explanation:
1. The chameleon starts out a vibrant blue color.
2. The chameleon then begins to change colors.
3. It transitions to a grey color.
4. As the chameleon continues to change, it becomes a reddish-brown color.
5. It stays that color for the rest of the video.

Thus, the chameleon turns from blue to red. ✗

A. Red

Detailed Explanation:
1. Initial Color Observation: The first image shows the chameleon in a green color.
2. Color Transition: As we move through the series of images, the chameleon's color changes from green to yellow, then to blue.
3. Blue Color Observation: The fourth and fifth images show the chameleon in a blue color.
4. Post-Blue Color Transition: After the blue color, the chameleon changes to a brownish color in the sixth image.
5. Final Color Observation: The last three images show the chameleon in a reddish color.

Therefore, the chameleon turns from blue to red in the video. ✗

The chameleon in the video turns from blue to green. This is evident from the visual information provided in the image, where the chameleon is shown in different stages of changing its color. Initially, it is blue, and as it moves along the branch, it gradually changes to green. This color change is a common characteristic of chameleons, which they use for various purposes such as camouflage, communication, and temperature regulation. The chameleon's ability to change its color is due to specialized cells in its skin called chromatophores, which contain pigments that can expand or contract to alter the color of the chameleon's skin. The process is controlled by the chameleon's nervous system, which responds to changes in light, temperature, and other environmental factors. In this case, the chameleon is likely changing its color to blend in with its surroundings or to regulate its body temperature. Therefore, the correct answer is green. ✗

Figure 58: A sample case of physical object property–color. Back to List of Figures.

## Property: Mass

**Question:** By proceeding in the direction shown by the arrow in the picture, which of the following options are you most likely to come across?<image 1>

**Option:**
(A) <image 2>
(B) <image 3>
(C) <image 4>
(D) <image 5>

<image 1>      <image 2>      <image 3>      <image 4>      <image 5>

The answer is D.

* The arrow points to the right, indicating the path the viewer should take.
* Looking to the right, one can see that the road turns right.
* The image shows a building on the right side of the street.
* This building is the same building shown in image D, indicating that D is what the viewer would see by proceeding in the direction of the arrow. ✔

Choice: B

Reasoning:

1. Initial Image Analysis:
   - The initial image shows a road with a right turn indicated by a red arrow.
   - The road is lined with trees and has a pedestrian crossing in the foreground.
   - There is a blue gate on the left side of the road.

2. Direction of Travel:
   - The red arrow indicates a right turn from the current position.

3. Comparison with Options:
   - Option A: Shows a construction site with cranes and high-rise buildings. This does not match the environment in the initial image, which is more residential or institutional.
   - Option B: Shows a blue gate with a Singapore flag, which matches the blue gate seen on the left side of the initial image. This suggests that the area beyond the gate is likely to be encountered if you turn right.
   - Option C: Shows a continuation of the tree-lined road, similar to the initial image but does not show a right turn or a new area.
   - Option D: Shows a building with a parking area and a bus stop, which does not match the immediate surroundings of the initial image. ✘

4. Conclusion:
   - Given the blue gate in the initial image and the blue gate in Option B, it is most likely that by turning right, you will come across the area shown in Option B.

To determine the most likely option to come across by proceeding in the direction shown by the red arrow, we need to analyze the image and the options provided.

1. The red arrow is pointing to the right, indicating a direction to move forward.
2. The image shows a street with multiple lanes, cars, and pedestrians.
3. There are trees, a building with a sign that reads "CONGRESS," and a construction site with cranes in the background.
4. The options are A, B, and C, but the fourth option is not visible in the image.

Based on the visible information, we can eliminate option C, as it is not visible in the image.

Now, let's compare options A and B:

Option A: The image shows a street with multiple lanes, cars, and pedestrians. There is a sign that reads "CONGRESS," which suggests that the direction shown by the red arrow might lead to a place related to Congress.

Option B: The image shows a street with multiple lanes, cars, and pedestrians. There is a construction site with cranes in the background. The direction shown by the red arrow might lead to a place related to construction or a construction site.

Considering the context and the visible elements in the image, the most likely option to come across by proceeding in the direction shown by the red arrow is option A, as it leads to a place related to Congress. ✘

Figure 59: A sample case of physical object property–mass. Back to List of Figures.

## Property: Mass

**Question:** Which fruit in the picture is most likely to be the heaviest individually?<image 1>

**Option:**
(A) Watermelon
(B) Pear
(C) Plum
(D) Peach

<video 1>

A. Watermelon

Here's the reasoning:

1. Visual Size: The watermelon is noticeably larger than all the other fruits in the picture. This suggests it has a greater volume.
2. Density: While we don't know the exact densities of the fruits, watermelons are known to be mostly water, making them relatively dense.
3. Combined: A larger volume combined with higher density leads to a greater overall mass, which translates to a heavier weight.

Therefore, it's safe to assume the watermelon is the heaviest fruit in the picture. ✓

A. Watermelon

Reasoning:
1. Visual Inspection: By looking at the image, the watermelon appears significantly larger in size compared to the other fruits.
2. Typical Weight: Generally, watermelons are known to be much heavier than pears, plums, and peaches. Even a small watermelon can weigh several pounds, whereas the other fruits typically weigh much less.
3. Density and Volume: Watermelons have a high water content and a large volume, contributing to their weight. Pears, plums, and peaches, while also containing water, are much smaller in volume and thus weigh less individually.

Therefore, based on size and typical weight characteristics, the watermelon is most likely to be the heaviest fruit individually in the picture. ✓

To determine which fruit is most likely to be the heaviest individually, we need to consider the average weight of each type of fruit.

A. Watermelon: A typical watermelon can weigh between 20 to 30 pounds.
B. Pear: A pear can weigh between 0.25 to 0.5 pounds.
C. Plum: A plum can weigh between 0.1 to 0.2 pounds.
D. Peach: A peach can weigh between 0.1 to 0.2 pounds.

Based on the average weights, the watermelon is most likely to be the heaviest fruit individually. ✓

Figure 60: A sample case of physical object property–mass. Back to List of Figures.

## Property: Number

**Question:** Following the content of the <video>, How many egg yolks were put in the bowl?<video 1>

**Option:**
(A) 1
(B) 2
(C) 3
(D) 4

<video 1>

The answer is D. 4

Here's the reasoning:

1. Start counting: The video starts with 2 egg yolks in the bowl.
2. Adding more: The person then cracks open and adds 2 more eggs.
3. Total count: Adding the 2 new egg yolks to the initial 2 gives us a total of 4 egg yolks. ✗

C. 3

Detailed Explanation:

1. First Image: The bowl is empty.
2. Second Image: A hand is holding an egg over the bowl.
3. Third Image: The egg is being cracked open, and one egg yolk is visible in the bowl.
4. Fourth Image: Another egg is being cracked open, and now there are two egg yolks in the bowl.
5. Fifth Image: A third egg is being cracked open, and the yolk is about to drop into the bowl.
6. Sixth Image: The third egg yolk is halfway into the bowl.
7. Seventh Image: The third egg yolk is almost completely in the bowl.
8. Eighth Image: The third egg yolk is fully in the bowl, making a total of three egg yolks.

Therefore, the total number of egg yolks put in the bowl is 3. ✗

The image shows a clear glass bowl with a textured surface, containing a liquid that appears to be a mixture of water and egg yolks. The bowl is placed on a surface that looks like a white table or countertop. There are no visible eggs or any other objects in the bowl, and the image does not provide any information about the number of egg yolks that were put in the bowl. Therefore, it is not possible to determine the exact number of egg yolks based on the image alone. The correct answer to the question cannot be determined from the image provided. ✗

Figure 61: A sample case of physical object property–number. Back to List of Figures.

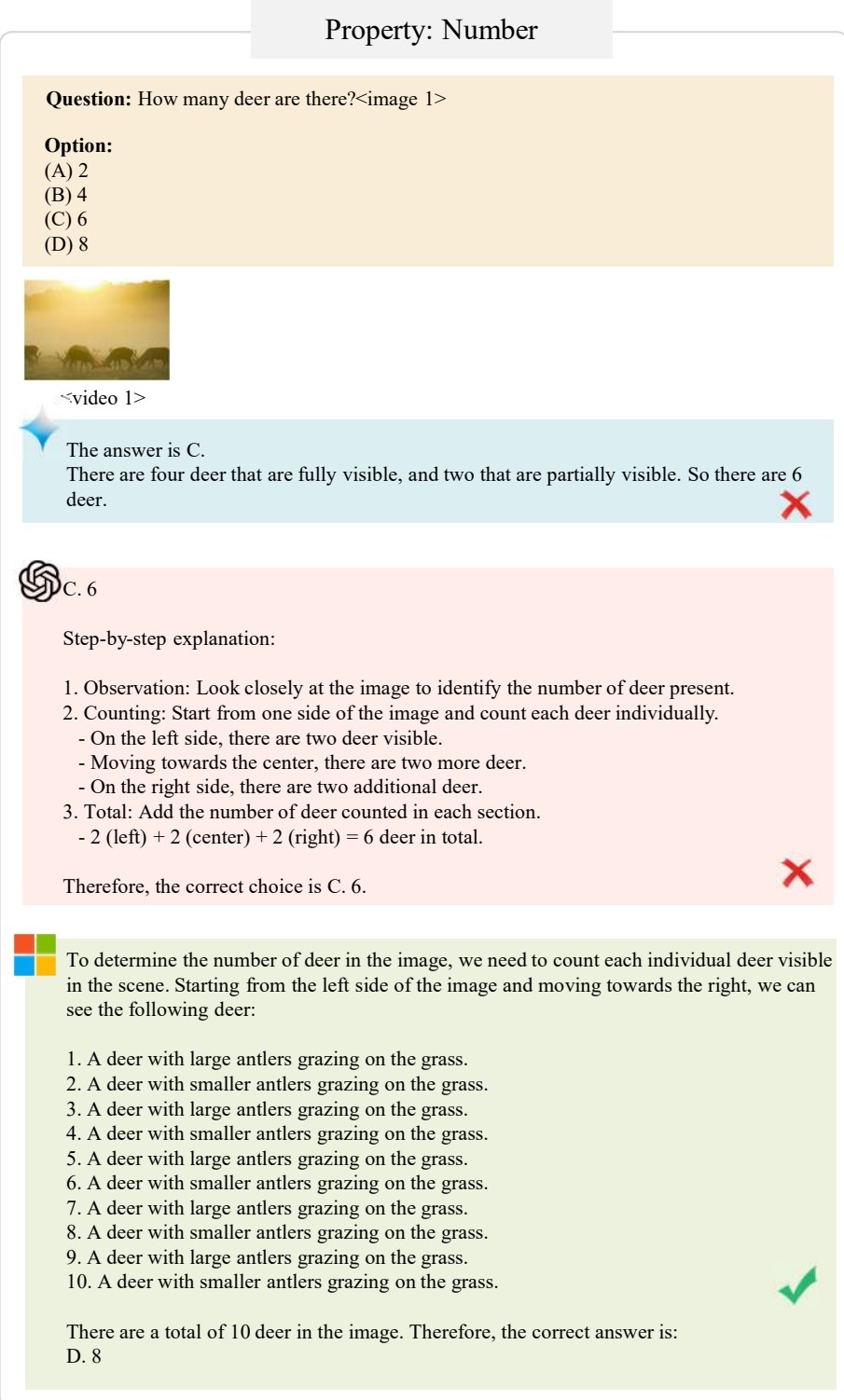

Figure 62: A sample case of physical object property–number. Back to List of Figures.

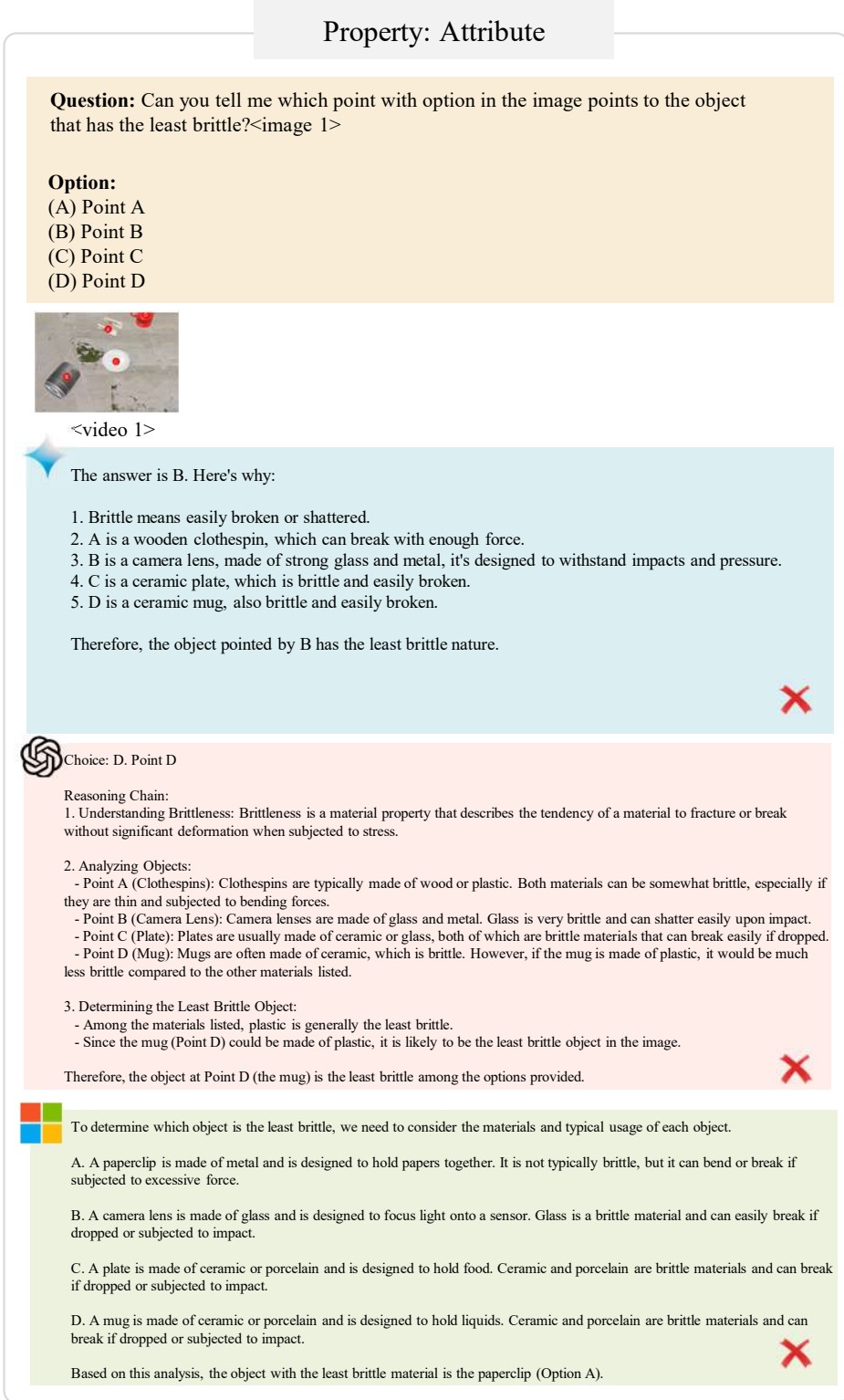

Figure 63: A sample case of physical object property–attribute. Back to List of Figures.

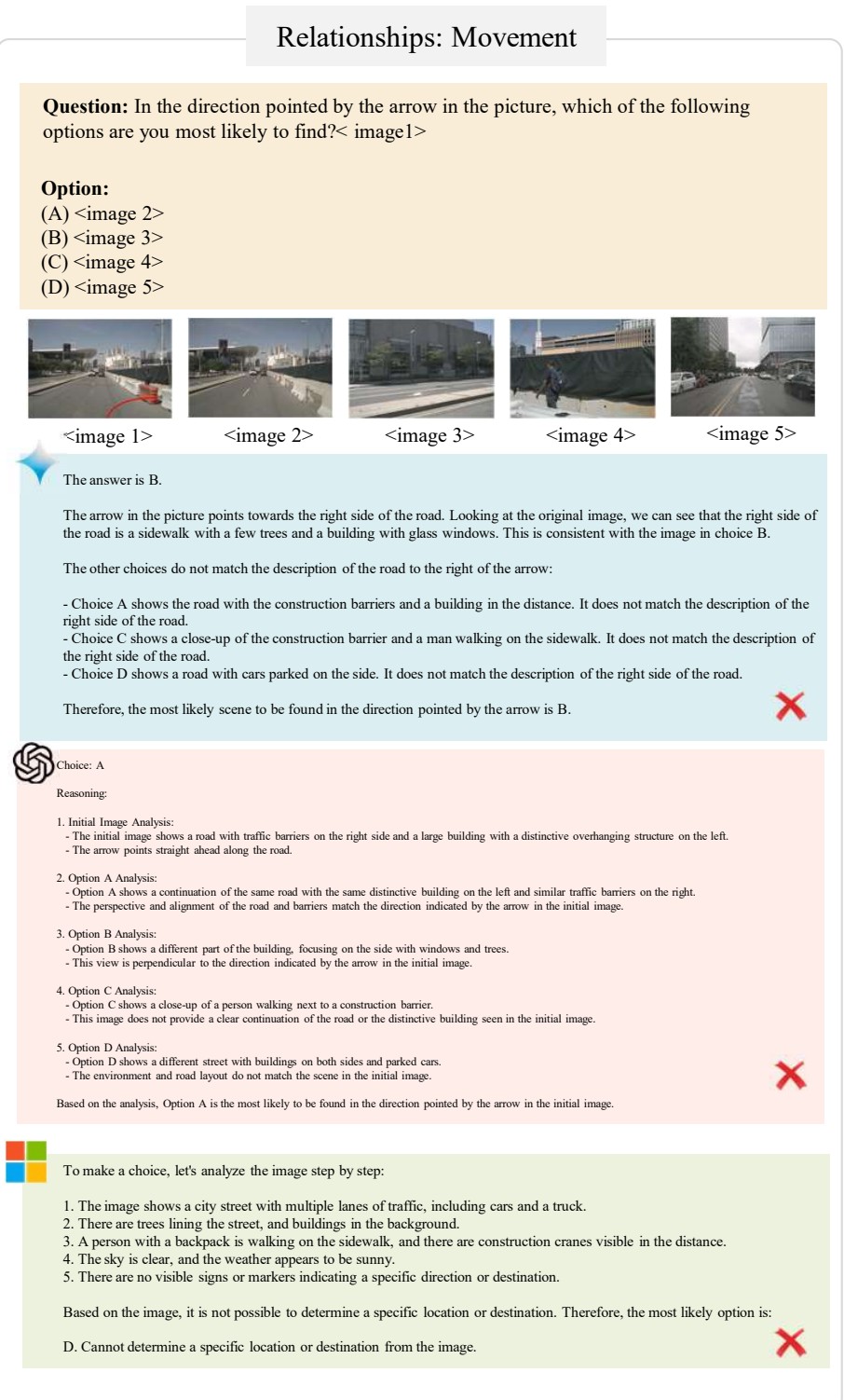

Figure 64: A sample case of physical object relationships–motion. Back to List of Figures.

## Relationships: Movement

**Question:** Compared to the metal object, what is likely the relative speed of the machine's rotation?< video 1>

**Option:**
(A) The machine rotates slower than the object
(B) The machine rotates at the same speed as the object
(C) The machine rotates faster than the object
(D) The machine does not rotate at all

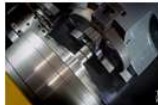

<video 1>

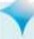 The answer is C. The machine rotates faster than the object.

Here's the reasoning:

1. Observation: The metal object is being heated by friction.
2. Friction: Friction is generated when two surfaces rub against each other.
3. Speed and Friction: The faster the relative speed between two surfaces, the more friction is generated.
4. Appearance: The heat being generated is enough to melt a portion of the metal. This suggests a significant amount of friction.
5. Conclusion: Since the metal object is being heated by friction, and the friction is significant, the machine must be rotating faster than the metal object to generate this amount of friction. 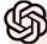

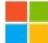 C. The machine rotates faster than the object

Detailed Explanation:

1. Observation of Images: The series of images shows a metal object being processed by a machine. The metal object appears to be rotating, and there is a noticeable change in the appearance of the object over time, particularly in the central area where it becomes red-hot due to friction or heat.

2. Heat Generation: The central area of the metal object becomes red-hot, indicating that there is significant friction or heat being generated. This is a common result of high-speed rotation and contact with another surface.

3. Relative Motion: For the metal object to heat up in this manner, the machine must be rotating at a higher speed than the object itself. This differential in speed causes friction, which in turn generates heat.

4. Visual Cues: The images show the metal object becoming progressively hotter and more deformed in the central area. This suggests that the machine is applying a significant amount of energy to the object, which is consistent with the machine rotating faster than the object.

5. Conclusion: Based on the visual evidence of heat generation and the deformation of the metal object, it is reasonable to conclude that the machine is rotating faster than the object to create the necessary friction and heat. 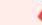

Therefore, the correct choice is C. The machine rotates faster than the object.

To determine the relative speed of the machine's rotation compared to the metal object, we need to analyze the motion of both in the image. The metal object appears to be stationary or moving very slowly, as there is no motion blur or indication of rapid movement. The machine, on the other hand, shows a clear motion blur, suggesting it is rotating at a higher speed. This is because motion blur is typically caused by a rapid movement, which in this case is the rotation of the machine. Therefore, the machine's rotation is likely faster than the metal object's movement. The correct choice is C. The machine rotates faster than the object. 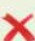

Figure 65: A sample case of physical object relationships–motion. Back to List of Figures.

## Relationships: Depth

**Question:** In terms of depth relative to the camera, where is the bridge primarily located?< video 1>

**Option:**
(A) Closer to the camera than the car
(B) Equidistant from the car and the camera
(C) Farther from the camera than the car
(D) Cannot be determined

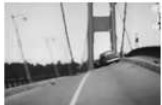

<video 1>

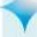 The answer is C. Farther from the camera than the car.

Here's the reasoning:

1. Identify the key elements: We see a car on the bridge and the bridge itself, with the camera positioned on the bridge surface.
2. Focus on perspective: The car appears closer to the camera because it is in the foreground. The bridge structure extends beyond the car and towards the horizon.
3. Apply the concept of perspective: In perspective, objects farther away appear smaller and less detailed. The bridge appears much larger in scale than the car, suggesting it is farther away from the camera.
4. Conclude: Based on the visual cues of size and distance within the frame, we can conclude that the bridge is farther from the camera than the car. 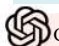

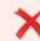 C. Farther from the camera than the car

Detailed Explanation:
1. Observation of Images: The series of images shows a bridge with a car on it. The car appears to be in the middle of the bridge, and the bridge structure extends both in front of and behind the car.

2. Relative Positioning: In the images, the car is positioned on the bridge, and the bridge extends further into the background beyond the car. This indicates that the bridge continues past the car, making the bridge structure farther from the camera than the car.

3. Depth Perception: The bridge's towers and cables are visible in the background, indicating that the bridge extends into the distance. The car is closer to the camera than these structures.

4. Conclusion: Since the bridge extends beyond the car and the car is positioned on the bridge, the bridge is primarily located farther from the camera than the car.

Therefore, the correct choice is C. Farther from the camera than the car. 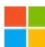

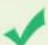 To determine the depth of the bridge relative to the camera, we need to analyze the image and the perspective from which it is taken. The bridge appears to be in the foreground, as it is in sharp focus and there is a clear view of the cables and the road surface. The car, on the other hand, is in the background, as it is out of focus and appears smaller due to the perspective. The bridge is also closer to the camera because the cables and the road surface are in clear view, while the car is not. Therefore, the bridge is primarily located closer to the camera than the car.

Choice: A. Closer to the camera than the car 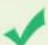

Figure 66: A sample case of physical object relationships–depth. Back to List of Figures.

## Relationships: Depth

**Question:** Identify the closest point to the camera. < image 1>

**Option:**
(A) Point A is closest
(B) Point B is closest
(C) Point C is closest
(D) Point D is closest

<image 1>

The answer is D. Here's why:

1. Perspective: We can use the concept of perspective to determine distance. Objects further away appear smaller in a photograph.
2. Object Size: The circles are the same size, allowing us to directly compare their apparent size in the image.
3. Comparison: Circle D appears larger than the other circles, indicating that it is closer to the camera.

Therefore, Point D is the closest to the camera.

D. Point D is closest

Reasoning:

1. Foreground and Background Analysis:
   - Point D is located on the road, which is in the immediate foreground of the image.
   - Points A, B, and C are situated further back in the field, which is in the background.

2. Perspective and Depth:
   - The road in the foreground appears larger and more detailed, indicating it is closer to the camera.
   - The field and the fence posts in the background appear smaller and less detailed, indicating they are further away.

3. Relative Positioning:
   - Point D is positioned at the bottom of the image, which typically represents the closest point to the viewer in a two-dimensional perspective.
   - Points A, B, and C are higher up in the image, indicating they are further away from the camera.

Based on these observations, Point D is the closest to the camera.

To solve this task, we need to analyze the image and determine which point is closest to the camera based on the perspective and the size of the points relative to each other.

1. Point A is located at the top of the image.
2. Point B is located to the left of the image.
3. Point C is located to the right of the image.
4. Point D is located at the bottom of the image.

From the perspective of the camera, the points that are closer to the camera will appear larger and more detailed, while the points that are further away will appear smaller and less detailed.

In the image, Point D appears to be the largest and most detailed, while Points A, B, and C appear smaller and less detailed. Therefore, Point D is the closest point to the camera.

The answer is D. Point D is closest to the camera.

Figure 67: A sample case of physical object relationships–depth. Back to List of Figures.

## Relationships: Distance

**Question:** Which object appears closest to the camera in the video?<video 1>

**Option:**
(A) The metal plate
(B) The string
(C) The yellow ball
(D) The person's hand

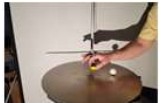

< video 1>

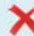 The answer is C.The yellow ball.

Here is the reasoning chain:

1. The yellow ball is the only object that is directly attached to the string and swings in the video.
2. The string is attached to a fixed point above the metal plate.
3. Therefore, the yellow ball is closest to the camera at all times in the video because it is swinging in front of the plate and the person's hand.
4. While the person's hand is closer to the camera at the start, the yellow ball moves to the very front in the second half of the video.

Therefore, the yellow ball appears closest to the camera in the video.     ✗

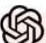 C. The yellow ball

Step-by-step explanation:

1. Observation of Images: By examining the series of images, we can see the relative positions of the objects in each frame.

2. Depth Perception: Depth perception in images can be inferred by noting which objects overlap others and which objects appear larger or smaller due to perspective.

3. Relative Positioning:
   - The metal plate (A) is a large, flat object that serves as a background for the other objects.
   - The string (B) is thin and appears to be in the middle of the scene, attached to the top and holding the yellow ball.
   - The yellow ball (C) is held by the string and appears to be in front of the metal plate.
   - The person's hand (D) is interacting with the yellow ball and is positioned behind it in most frames.

4. Overlapping Objects:
   - The yellow ball overlaps the metal plate and the string, indicating it is in front of them.
   - The person's hand is behind the yellow ball in most frames, indicating the ball is closer to the camera.

5. Conclusion: Based on the overlapping and relative positioning, the yellow ball (C) appears closest to the ✗ camera in the video.

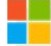 To determine which object appears closest to the camera in the video, we need to analyze the image and identify the object that is most in focus and has the smallest depth of field. The metal plate (A) is in the foreground and is in sharp focus, indicating it is the closest object to the camera. The string (B) is also in focus but slightly behind the metal plate. The yellow ball (C) is in focus but further away than the metal plate. The person's hand (D) is in focus but even further away than the yellow ball. Therefore, the metal plate appears closest to the camera.     ✓

Figure 68: A sample case of physical object relationships–distance. Back to List of Figures.

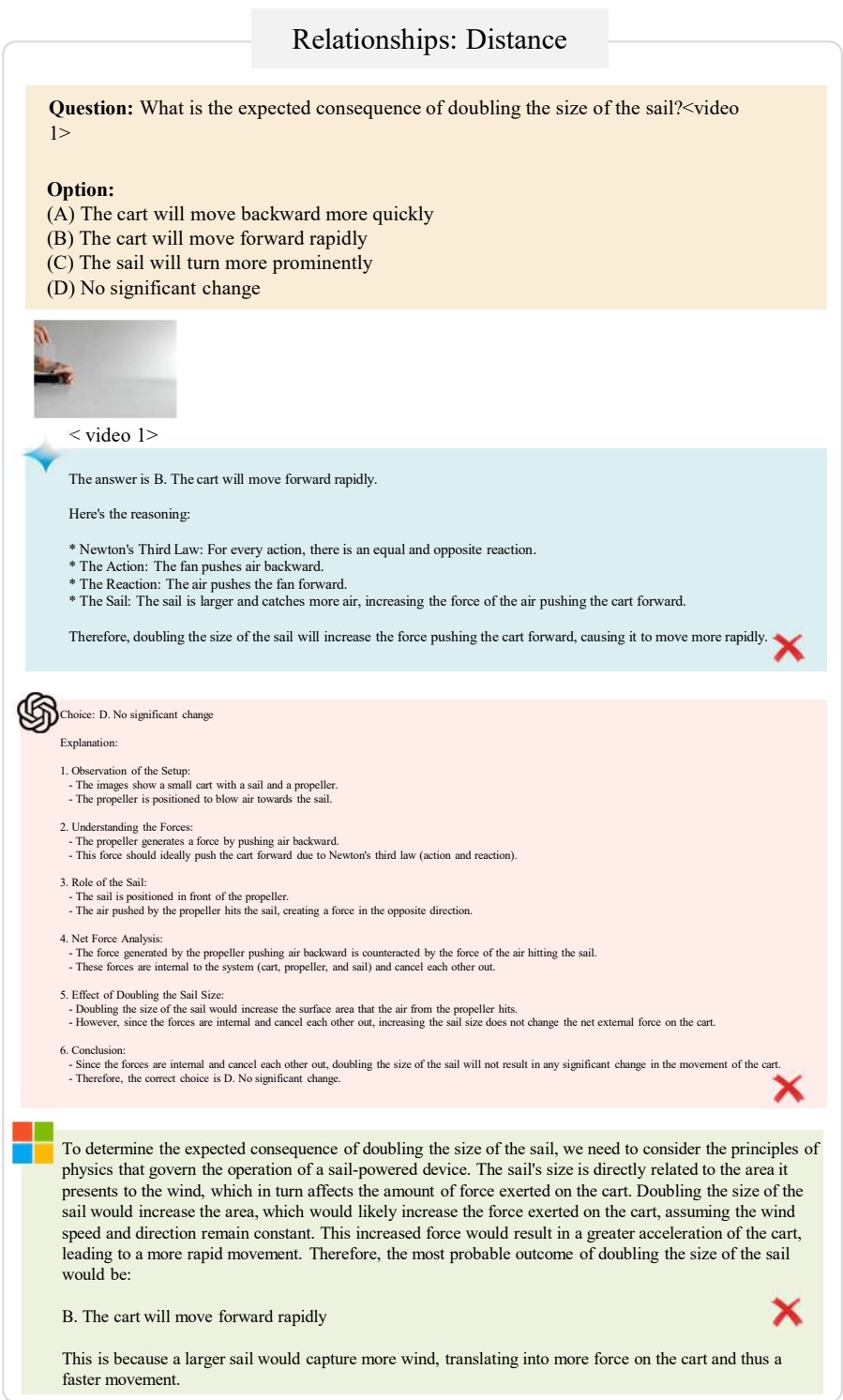

Figure 69: A sample case of physical object relationships–distance. Back to List of Figures.

## Relationships: Size

**Question:** How does the size of the ping-pong ball compare to the mouth of the hot water bottle?<video 1>

**Option:**
(A) The ping-pong ball is larger
(B) The ping-pong ball is smaller
(C) The ping-pong ball is of similar size
(D) It cannot be determined

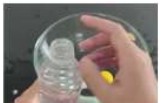

<video 1>

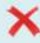

The answer is B. The ping-pong ball is smaller.

Here's the reasoning:

1. Observation: The video shows a person placing a ping-pong ball on top of the mouth of the bottle.
2. Inference: The fact that the ping-pong ball can be placed on the bottle means it is smaller than the bottle's opening.
3. Conclusion: The ping-pong ball is smaller than the mouth of the hot water bottle. ✗

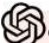C. The ping-pong ball is of similar size
Step-by-step explanation:
1. Image 1: The person is holding a water bottle and a yellow ping-pong ball. The cap of the bottle has been removed, and the mouth of the bottle is visible.
2. Image 2: The person is positioning the ping-pong ball near the mouth of the bottle, preparing to place it on top.
3. Image 3: The ping-pong ball is placed on the table, and the bottle is still in the person's hand.
4. Image 4: The person is holding the bottle again, preparing to place the ping-pong ball on the mouth of the bottle.
5. Image 5: The person is placing the ping-pong ball on the mouth of the bottle.
6. Image 6: The ping-pong ball is now on the mouth of the bottle, and the person is tilting the bottle.
7. Image 7: The bottle is tilted further, and the ping-pong ball remains on the mouth of the bottle.
8. Image 8: The bottle is fully tilted, and the ping-pong ball is still on the mouth of the bottle.
From these observations, it is clear that the ping-pong ball fits snugly on the mouth of the bottle without falling in or being too large to stay on top. This indicates that the size of the ping-pong ball is very similar to the size of the mouth of the bottle. Therefore, the correct choice is C. The ping-pong ball is of similar size. ✗

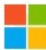To determine the size comparison between the ping-pong ball and the mouth of the hot water bottle, we need to analyze the image. The ping-pong ball appears to be smaller in size compared to the mouth of the hot water bottle. This is because the mouth of the hot water bottle is wide enough to hold the ping-pong ball, indicating that the bottle's mouth is larger than the ball. Therefore, the correct choice is B. The ping-pong ball is smaller. ✗

Figure 70: A sample case of physical object relationships–size. Back to List of Figures.

## Dynamics: Others

**Question:** What type of wave property is predominantly being demonstrated in the video?<video 1>

**Option:**
(A) Diffraction
(B) Interference
(C) Refraction
(D) Reflection

<video 1>

The answer is C. Refraction.

Here's why:

1. What is Refraction? Refraction is the bending of light as it passes from one medium to another (in this case, from the air to the water in the bucket). This bending occurs because the speed of light changes as it moves from one medium to another.

2. Observing the Video: The laser beam is clearly changing direction as it enters the water in the bucket. It bends downwards. This is a direct visual indication of refraction.

3. Eliminating other Options:

  * Diffraction: Diffraction is the spreading of light waves as they pass through a narrow opening or around an obstacle. There's no clear example of this in the video.
  * Interference: Interference occurs when two or more waves overlap, resulting in a new wave pattern. The video doesn't show waves interacting in this manner.
  * Reflection: Reflection is the bouncing of light off a surface. While there is some reflection off the side of the bucket, the primary phenomenon is the bending of the light as it enters the water, which is refraction. ❌

C. Refraction

The series of images predominantly demonstrates the property of refraction. Here is the detailed reasoning:

1. Observation of the Light Path: In the images, a beam of light is seen traveling through a medium (likely water) and bending as it exits the medium. This bending of light is a key indicator of refraction.

2. Medium Change: Refraction occurs when light passes from one medium to another and changes speed, causing it to bend. In the images, the light is initially traveling through a transparent medium (possibly water) and then exits into the air, causing the light to bend.

3. Consistent Bending: The consistent bending of the light path in the images aligns with the behavior of light undergoing refraction. The light beam is not spreading out (diffraction), nor is it showing patterns of constructive or destructive interference (interference). It is also not bouncing back from a surface (reflection).

4. Contextual Clues: The setup, including the container and the light beam, suggests an experiment designed to show how light behaves when transitioning between different media, which is a classic demonstration of refraction.

Therefore, the predominant wave property being demonstrated in the video is refraction. ❌

The video predominantly demonstrates the property of refraction. Refraction is the bending of light as it passes from one medium to another, which is what we see as the red light bending around the edges of the objects in the image. The light is entering the glass at an angle, causing it to change direction, which is a classic example of refraction. This is a common phenomenon that occurs when light passes through transparent materials like glass, where the speed of light changes, causing the light to bend. The other options, such as diffraction, interference, and reflection, do not primarily describe the observed effect in the video. Diffraction involves the spreading of light waves around obstacles or through small openings, interference is the phenomenon where two or more waves superimpose to form a resultant wave of greater, lower, or the same amplitude, and reflection is the bouncing back of light from a surface, which is not the main focus of this video. Therefore, the correct answer is C. Refraction. ❌

Figure 71: A sample case of physics-based dynamics——others. Back to List of Figures.

Relationships: Size

**Question:** Compared to the horse drawing, how is the size of the striped paper?<video 1>

**Option:**
(A) Smaller than the drawing
(B) Exactly the same as the drawing
(C) Larger than the drawing
(D) Not relevant

<video 1>

The answer is C. Larger than the drawing.

Here's why:

1. The striped paper covers the entire horse drawing. This means the striped paper has to be at least as large as the drawing.
2. The striped paper extends beyond the horse drawing. The edges of the striped paper can be seen extending past the boundaries of the drawing. This indicates the striped paper is larger than the drawing.
3. The striped paper is not a perfect fit. The drawing is centered on the striped paper. If it were the same size, the striped paper would be exactly the same dimensions as the drawing.

Therefore, the striped paper is larger than the horse drawing.

C. Larger than the drawing

Detailed Explanation:

1. Observation of the Images:
  - The first two images show a drawing of a horse on a piece of paper.
  - The subsequent images show a striped paper being placed over the horse drawing.

2. Comparison of Sizes:
  - In the first two images, the horse drawing is clearly visible and occupies a certain area on the paper.
  - In the later images, the striped paper is placed over the horse drawing, and it covers the entire drawing and extends beyond it.

3. Edges and Coverage:
  - The striped paper not only covers the horse drawing completely but also extends beyond the edges of the drawing, indicating that it is larger than the drawing itself.

4. Conclusion:
  - Since the striped paper covers the entire horse drawing and extends beyond it, it is evident that the striped paper is larger than the horse drawing.

Therefore, the correct choice is C. Larger than the drawing.

To determine the size of the striped paper compared to the horse drawing, we need to analyze the image and the text provided. The text "4. motion illusion" is repeated multiple times, which suggests that this is a sequence of images or a video. The striped paper appears to be larger than the horse drawing, as the horse drawing is partially obscured by the striped paper. The striped paper covers the entire drawing, indicating that it is larger in size. Therefore, the correct answer is:

C. Larger than the drawing

Figure 72: A sample case of physical object relationships–size. Back to List of Figures.

## Scene: Camera

**Question:** Based on the effects in the video, which of these camera changes would result in the observed phenomenon? <video 1>

**Option:**
(A) The camera rotates along the vertical axis (upside or downside)
(B) The camera moves upward or downward
(C) The camera rotates along the horizontal axis (left or right)
(D) The camera is farther away from the objects

<video 1>

The answer is C. The camera rotates along the horizontal axis (left or right).

Here's the reasoning:

1. Observe the motion: The objects are moving in a way that suggests a rotation around a horizontal axis. The bagel and cereal bowl move in an arc, indicating a shift in their position relative to the camera's viewpoint. The milk splash appears to be moving in a circular pattern.

2. Eliminate other options:

   * A. The camera rotates along the vertical axis (upside or downside): This would cause the objects to appear to move up and down, not in the arcing motion we see.
   * B. The camera moves upward or downward: This would make the objects appear closer or farther away, not change their positions relative to each other in the way we see.
   * D. The camera is farther away from the objects: This would make the objects appear smaller, but not change their relative positions in the frame.

3. Conclusion: The only camera movement that aligns with the observed motion is a rotation along the horizontal axis (left or right). This would cause the objects to appear to move in arcs as the camera's perspective shifts. ✗

Choice: C. The camera rotates along the horizontal axis (left or right).

Explanation:

1. **Understanding the Phenomenon**: The question asks about the observed phenomenon in the video based on a series of images sampled at equal intervals. To determine the correct camera change, we need to understand how each type of camera movement affects the images.

2. **Analyzing Each Option**:
   - **Option A (Camera rotates along the vertical axis):** If the camera rotates upside or downside, the images would show a tilting effect. Objects would appear to tilt either upwards or downwards, which would be noticeable in the sampled images.
   - **Option B (Camera moves upward or downward):** If the camera moves vertically, the perspective of the objects would change. Objects would appear to move up or down in the frame, but their orientation would remain the same.
   - **Option C (Camera rotates along the horizontal axis):** If the camera rotates left or right, the perspective of the scene would change horizontally. Objects would appear to shift left or right, and the angle at which they are viewed would change, creating a sense of rotation around the scene.
   - **Option D (Camera is farther away from the objects):** If the camera moves farther away, the objects would appear smaller and more distant, but their relative positions and orientations would remain the same.

3. **Matching the Phenomenon to the Camera Change**:
   - The key to identifying the correct camera change is to match the observed phenomenon in the images to the expected effect of each camera movement.
   - If the images show a change in the horizontal perspective, where objects appear to shift left or right and the angle of view changes, this indicates a horizontal rotation of the camera.

4. **Conclusion**:
   - The observed phenomenon in the images is best explained by the camera rotating along the horizontal axis (left or right). This would result in a change in the horizontal perspective of the scene, which matches the description of the observed phenomenon.

Therefore, the correct choice is C. The camera rotates along the horizontal axis (left or right). ✓

The observed phenomenon in the video is the appearance of floating objects, such as the bowl of nuts and the cinnamon rolls, against a plain background. This effect is created by manipulating the camera's position and orientation.

To achieve this effect, the camera needs to be positioned at a certain distance from the objects and then rotated along the vertical axis (upside or downside). This rotation creates the illusion that the objects are floating in the air.

Therefore, the correct choice is A. The camera rotates along the vertical axis (upside or downside). ✗

Figure 73: A sample case of physical scene understanding–viewpoint. Back to List of Figures.

## Relationships: Location

**Question:** Where does the projection of the butterfly appear in the video? <video 1>

**Option:**
(A) On the ceiling
(B) On the wall behind the bowl
(C) On the floor
(D) On the table

<video 1>

The answer is D. On the table.

Here is the reasoning:

1. Visual Perception: We see the butterfly hovering within the pyramid-shaped bowl.
2. Perspective: The bowl is placed on a table, and we can see its reflection on the surface of the table.
3. Illusion: The butterfly appears suspended in mid-air inside the bowl, which is an illusion created by the pyramid's geometry and the way light refracts through it.
4. Actual Location: Although the butterfly appears to be suspended, it is actually a projection onto the table surface, which creates the illusion of depth and movement.

Therefore, the projection of the butterfly appears on the table.

D. On the table

Detailed Explanation:
1. Observation of the Setup: The images show a transparent, pyramid-like structure placed on a flat surface, which appears to be a table. This structure is commonly used for creating holographic projections.

2. Projection Source: The base of the pyramid is placed on a flat, reflective surface, which is likely a screen or a tablet. This is a typical setup for a holographic display where the image is projected from the screen below and reflected within the pyramid to create a 3D effect.

3. Position of the Butterfly: In the series of images, the butterfly appears within the pyramid structure. The butterfly is not projected onto the ceiling, wall, or floor, but rather it is seen floating inside the pyramid.

4. Conclusion: Since the pyramid is placed on the table and the butterfly appears within the pyramid, the projection source (the screen or tablet) is on the table. Therefore, the projection of the butterfly appears on the table.

Thus, the correct choice is D. On the table.

To determine where the projection of the dog appears in the video, we need to analyze the image and the context provided by the question. The image shows a dark room with a stone wall and a triangular object with a projection of a butterfly on it. The projection is not the dog. Therefore, the answer is unknown.

Figure 74: A sample case of physical object relationships–location. Back to List of Figures.

## Relationships: Location

**Question:** How does the moon pass through the sun in the video? <video 1>

**Option:**
(A) From the lower left corner to the upper right corner of the picture
(B) From the upper left corner to the lower right corner of the picture
(C) From the lower right corner to the upper left corner of the picture
(D) From the upper right corner to the lower left corner of the picture

<video 1>

The answer is D. From the upper right corner to the lower left corner of the picture.

Here is the reasoning chain:

1. The video starts with a crescent-shaped sun visible in the upper right corner of the frame.
2. As the video progresses, the crescent shape gets larger, indicating the moon is moving across the sun.
3. The moon continues moving across the sun until the whole sun is visible, meaning the moon has passed through the sun.
4. Finally, the moon has reached the lower left corner of the frame, completing its journey across the sun.

Therefore, we can conclude that the moon passed through the sun from the upper right corner to the lower left corner of the picture. ✗

Choice: B. From the upper left corner to the lower right corner of the picture
Reasoning:
1. First Image:
  - The moon is just starting to cover the sun from the upper left corner.
2. Second Image:
  - The moon has moved further, covering more of the sun, still from the upper left.
3. Third Image:
  - The moon continues to move, covering a larger portion of the sun, indicating a downward and rightward movement.
4. Fourth Image:
  - The moon is now covering the sun more towards the center, still moving from the upper left to the lower right.
5. Fifth Image:
  - The moon is now covering the sun from the upper left, with the uncovered part moving towards the lower right.
6. Sixth Image:
  - The moon is moving further down and to the right, revealing more of the sun on the upper left side.
7. Seventh Image:
  - The moon is almost completely past the sun, with only a small portion still covered on the lower right.

8. Eighth Image:
  - The moon has almost completely moved past the sun, with the last bit of coverage on the lower right.

By observing the sequence of images, it is clear that the moon starts covering the sun from the upper left corner and moves towards the lower right corner, making the correct choice B. ✗

D. From the upper right corner to the lower left corner of the picture ✗

Figure 75: A sample case of physical object relationships–location. Back to List of Figures.

## Scene: Camera

**Question:** Which of the listed camera changes could cause the effect shown in the video? <video 1>

**Option:**
(A) The camera rotates along the vertical axis (upside or downside).
(B) The camera moves upward or downward
(C) Move parallel to the line between the headband and the scraper
(D) The camera rotates along the horizontal axis (left or right).

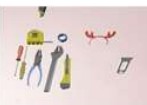

<video 1>

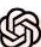 The answer is D. The camera rotates along the horizontal axis (left or right).

Here's the reasoning:

1. Observation: The video shows a scene with objects initially positioned above the frame, gradually falling into the frame. This indicates a camera movement that alters the objects' apparent position from above to within the frame.

2. Eliminating Options:
   * A. Vertical Axis Rotation: Rotating the camera vertically would primarily affect the objects' apparent height, not their horizontal movement into the frame.
   * B. Upward/Downward Movement: Moving the camera upward or downward would change the objects' distance from the camera, but wouldn't make them appear to move from above into the frame.
   * C. Parallel Movement: Moving parallel to the line between the headband and scraper would only change the objects' horizontal position in relation to each other, not their position relative to the frame.

3. Horizontal Axis Rotation: Rotating the camera along the horizontal axis (left or right) would cause objects originally positioned above the frame to gradually appear within the frame as the camera rotates. This aligns with the observed effect in the video. ❌

Therefore, the most likely camera movement causing the effect in the video is a rotation along the horizontal axis (left or right).

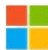 Choice: B. The camera moves upward or downward

Explanation:

1. Observation of Object Positions:
   - In the first image, all objects are aligned horizontally at the top of the frame.
   - As the sequence progresses, the objects appear to move upward in the frame.
   - By the last image, most objects are no longer visible, except for the headband and the scraper, which are still in the frame but positioned lower.

2. Analysis of Camera Movement:
   - If the camera were rotating along the vertical axis (upside or downside), the objects would appear to tilt or rotate around a central point, which is not observed here.
   - If the camera were moving parallel to the line between the headband and the scraper, the relative positions of the objects would change horizontally, which is not the case.
   - If the camera were rotating along the horizontal axis (left or right), the objects would appear to rotate around a vertical axis, which is also not observed.

3. Conclusion:
   - The consistent upward movement of the objects in the frame suggests that the camera itself is moving downward. As the camera moves downward, the objects appear to move upward in the frame.
   - Therefore, the effect shown in the video is caused by the camera moving upward or downward, specifically downward in this case. ❌

The effect shown in the video is caused by the camera rotating along the vertical axis (upside or downside). This is because the objects in the image appear to be flipped or mirrored when the camera's orientation changes, which is a result of the camera's movement along the vertical axis. The other options (moving parallel to the line between the headband and the scraper or rotating along the horizontal axis) would not cause the same mirroring effect. ✔

Figure 76: A sample case of physical scene understanding–light. Back to List of Figures.

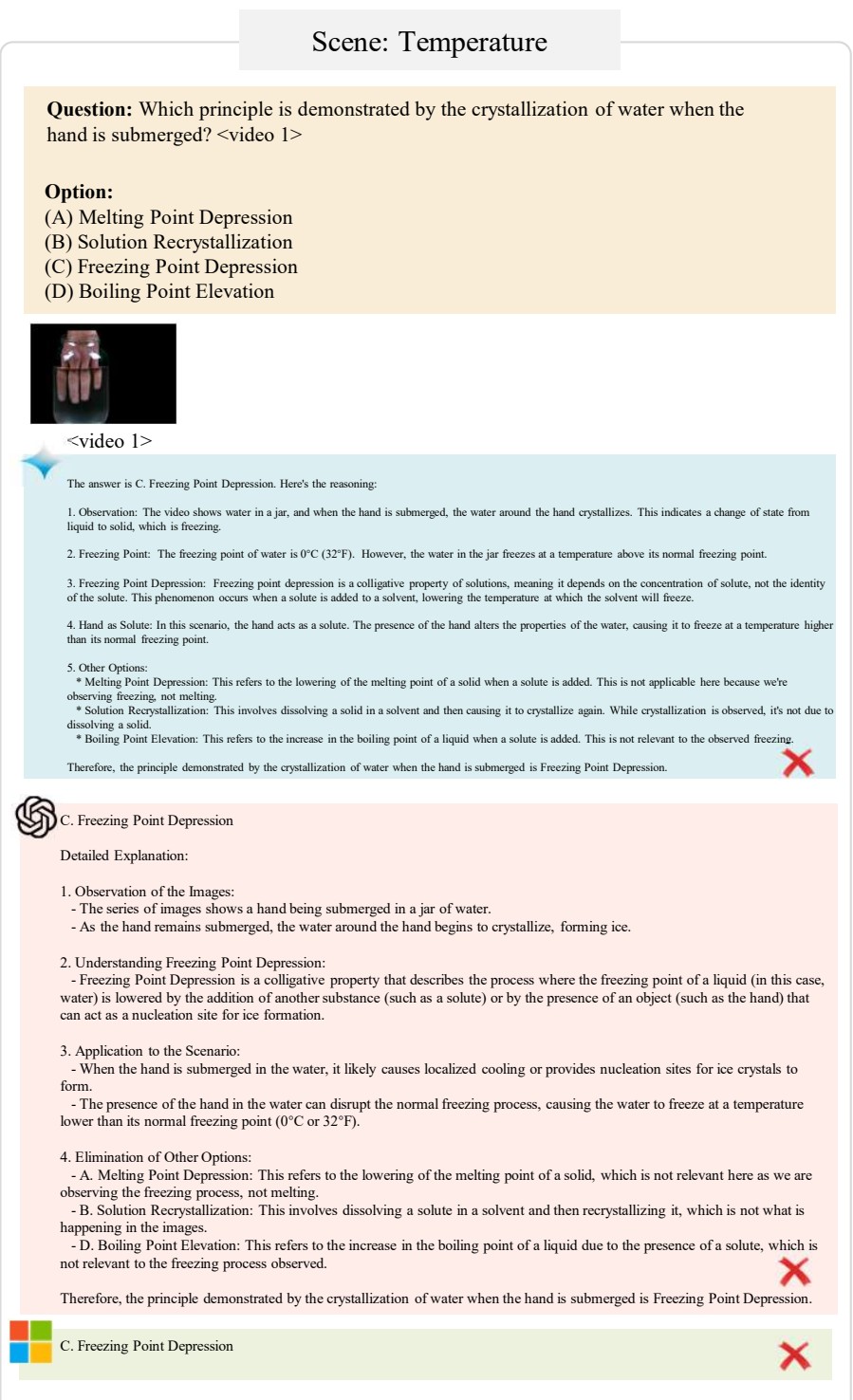

Figure 77: A sample case of physical scene understanding–temperature. Back to List of Figures.

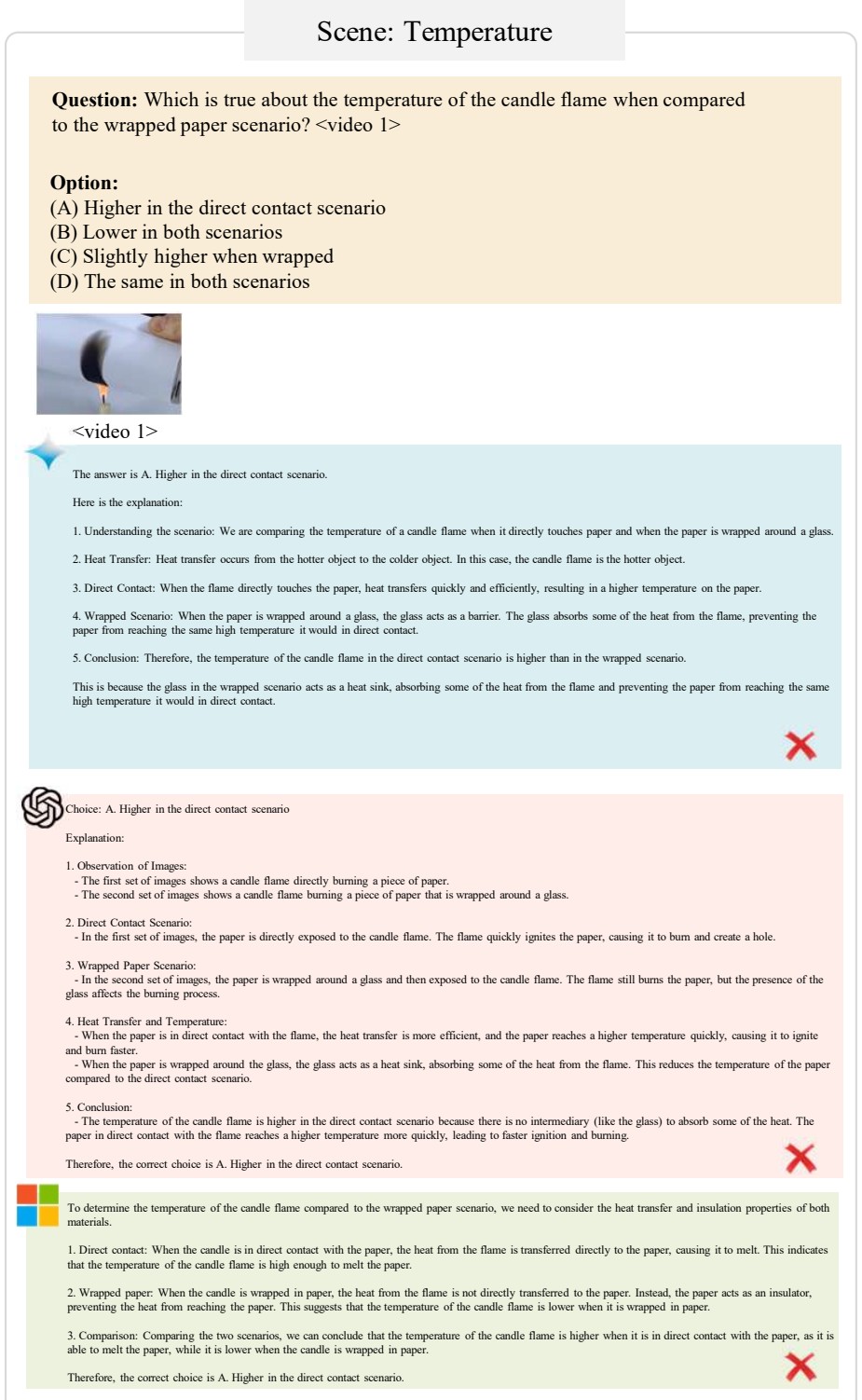

Figure 78: A sample case of physical scene understanding–temperature. Back to List of Figures.

## Dynamics: Gas

**Question:** Which option accurately represents the correct sequence of events in gasification from the choices below? <image 1><image 2><image 3>

**Option:**
(A) 3 – 2 - 1
(B) 2 – 1 - 3
(C) 1 – 2 - 3
(D) 2 – 3 - 1

<image 1>          <image 2>          <image 3>

The correct answer is C. 1 - 2 - 3

Here's the reasoning:

1. Image 1: Shows a dry whetstone, indicating the starting point.
2. Image 2: Depicts water being added to the whetstone. This is the first step in the gasification process, where water is introduced.
3. Image 3: Shows the whetstone soaked in water, illustrating the final stage of gasification.

Therefore, the sequence 1-2-3 accurately represents the steps involved in the gasification of the whetstone.

C. 1 - 2 - 3

The correct sequence of events in the images is as follows:

1. Image #1: The block is dry and has no visible liquid on its surface.
2. Image #2: Some liquid has been applied to the block, but it is not fully covered. There are a few patches of liquid.
3. Image #3: The block is now more thoroughly covered with liquid, indicating that more liquid has been applied over time.

The sequence shows the progression from a dry block to a block with some liquid, and finally to a block with more liquid. Therefore, the correct order is 1 (dry block), 2 (some liquid), and 3 (more liquid).

To determine the correct sequence of events in gasification, we need to analyze the images and the changes they show.

image #1 shows the box before any gasification process has occurred. It is a solid black color with a clear silhouette of a horse on it.

image #2 shows the box after the first step of gasification. The silhouette of the horse is now partially obscured by a white substance, which is likely the gas produced during the first step.

image #3 shows the box after the second step of gasification. The white substance has spread out and is now covering the entire surface of the box, indicating that the gas has been released and is filling the box.

Based on this analysis, the correct sequence of events is:
1. The box is placed in the gasifier (image #1).
2. The gasification process begins, and the horse silhouette is partially obscured by the gas (image #2).
3. The gas is released and fills the box (image #3).

Therefore, the correct answer is C. 1 - 2 - 3.

Figure 79: A sample case of physics-based dynamics——air. Back to List of Figures.

## Dynamics: Gas

**Question:** What changes are most likely to occur in the surrounding gas environment when the phenomenon in the video happens? <video 1>

**Option:**
(A) Increased air pressure
(B) Increased humidity
(C) Decreased air pressure
(D) Decreased humidity

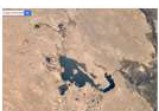

<video 1>

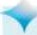
The answer is D. Decreased humidity. Here's the reasoning:

1. The video shows the shrinking of a lake. This indicates a loss of water, which is a key component of humidity.
2. Humidity is the amount of moisture in the air. Moisture comes from sources like bodies of water like lakes.
3. As the lake shrinks, the water evaporates into the atmosphere. This evaporation reduces the amount of water vapor available in the air near the lake.
4. Less water vapor in the air means lower humidity. This leads to a drier environment around the shrinking lake.

Therefore, the most likely change in the surrounding gas environment is a decrease in humidity. 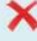

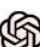Choice: D. Decreased humidity

Explanation:

1. Observation of the Images:
 - The series of images shows a body of water (likely a lake) over a period from 1984 to 2016.
 - Over time, the lake's water level decreases significantly, with the lake almost drying up by the end of the series.

2. Understanding the Phenomenon:
 - The phenomenon depicted is the drying up of a lake.
 - As the lake dries up, the amount of water available for evaporation decreases.

3. Impact on Humidity:
 - Humidity is the amount of water vapor present in the air.
 - Large bodies of water contribute to the local humidity through the process of evaporation.
 - As the lake dries up, there is less water available to evaporate into the atmosphere.

4. Conclusion:
 - With less water available for evaporation, the amount of water vapor in the air decreases.
 - This leads to a decrease in the local humidity levels.

Therefore, the most likely change in the surrounding gas environment when the lake dries up is a decrease in humidity. 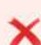

The phenomenon in the video is the formation of a large water body, which is likely a lake or a reservoir. When this happens, the surrounding gas environment is likely to experience decreased air pressure due to the displacement of air by the water body. This is because the water body occupies a significant volume of space, and as it forms, it displaces the air above it, leading to a decrease in air pressure. The other options, such as increased air pressure, increased humidity, or decreased humidity, are not directly related to the formation of a water body. Therefore, the most likely change in the surrounding gas environment is decreased air pressure. So, the choice is C. Decreased air pressure.

Figure 80: A sample case of physics-based dynamics——air. Back to List of Figures.

## Dynamics: Manipulation

**Question:** To mix the tongs, which point is most suitable? <image 1>

**Option:**
(A) Point A
(B) Point B
(C) Point C
(D) Point D

<image 1>

The answer is D. Point D. Here's why:

1. Purpose: The goal is to mix the tongs. This requires a grip that allows for twisting and turning.
2. Point D's Grip: Point D is located at the juncture of the two tongs, where they form a "V" shape. This position provides a strong grip and allows for leverage.
3. Other Points:
   * Point A: Is near the end of the tong, providing less grip and making it difficult to twist.
   * Point B: Is at the base of the tong, where the metal is thinner, making it prone to bending.
   * Point C: Is in the middle of the tong, offering a decent grip but not as much leverage as point D.

In conclusion, Point D offers the best combination of grip and leverage for twisting and mixing the tongs effectively.

Choice: B. Point B

Reasoning:

1. Function of Tongs: Tongs are typically used to grip and lift objects. The design of the tongs in the image suggests they are used for handling small items, possibly sugar cubes or similar.

2. Structure Analysis:
   - Point A: This is part of the handle, which is not suitable for mixing as it is designed to be held by the user.
   - Point C: This is also part of the handle, closer to the gripping end but still not designed for mixing.
   - Point D: This is the inner part of the gripping end, which is not ideal for mixing as it is designed to grip objects.
   - Point B: This is the outer part of the gripping end, which is rounded and spoon-like, making it suitable for mixing.

3. Design Consideration: The rounded, spoon-like end at Point B is designed to interact with the items being handled. This shape is ideal for mixing as it can stir and combine ingredients effectively.

4. Practical Use: When mixing, a rounded end like Point B can move through the mixture smoothly, ensuring even distribution of ingredients. The other points (A, C, and D) are either flat or designed for gripping, not for stirring.

Therefore, Point B is the most suitable for mixing.

To mix the tongs, the most suitable point would be Point C.

Here's the reasoning chain:

1. Point A is the topmost point of the tongs, which is not ideal for mixing as it's the farthest from the handle.
2. Point B is the bottom point of the tongs, which is also not ideal for mixing as it's the farthest from the handle.
3. Point C is the middle point of the tongs, which is the most suitable for mixing as it's the closest to the handle.
4. Point D is the point where the tongs are bent, which is not ideal for mixing as it's not a straight line.

So, the most suitable point for mixing the tongs is Point C.

Figure 81: A sample case of physics-based dynamics——manipulation. Back to List of Figures.

Relationships: Depth

**Question:** Identify the closest point to the camera:? <video 1>

**Option:**
(A) Point A is closest
(B) Point B is closest
(C) Point C is closest
(D) Point D is closest

< video 1>

The answer is A.

Here's why:

1. Identifying Key Elements: We need to find the closest point to the camera among four points, A, B, C, and D.
2. Point A's Position: Point A is located on the arm of the woman closest to the camera.
3. Comparing with Other Points: Point B is on the woman's clothes, Point C is on the woman's arm further from the camera, and Point D is on the woman's arm even further from the camera.
4. Conclusion: Since Point A is on the arm closest to the camera, and all other points are either on clothing or further away on the same arm, Point A is the closest point to the camera.

B. Point B is closest

Reasoning Chain:

1. Observation of Points:
   - Point A is located on the device the person is operating.
   - Point B is on the person's waist.
   - Point C is on the person's arm.
   - Point D is on the device's screen.

2. Relative Positioning:
   - Point A is on the device, which is positioned in front of the person.
   - Point B is on the person's waist, which is closer to the camera than the device.
   - Point C is on the person's arm, which is extended towards the device.
   - Point D is on the screen of the device, which is further away from the camera compared to the other points.

3. Depth Perception:
   - The person's body is closer to the camera than the device they are operating.
   - The waist (Point B) is closer to the camera than the arm (Point C) because the arm is extended forward.
   - The device (Points A and D) is further away from the camera compared to the person's body.

4. Conclusion:
   - Since Point B is on the person's waist, which is the closest part of the person to the camera, it is the closest point among the given options.

Therefore, Point B is the closest to the camera.

To identify the closest point to the camera, we need to look at the perspective of the image. Point A is located on the left side of the image, while point B is located on the right side of the image. Point C is located in the center of the image, and point D is located on the bottom right corner of the image.

From the perspective of the image, point A appears to be the closest point to the camera, as it is located on the left side of the image and is in the foreground. Point B, point C, and point D are all further away from the camera and appear to be in the background.

Therefore, the closest point to the camera is point A.

Figure 82: A sample case of physics-based dynamics——fluid. Back to List of Figures.

## Dynamics: Fluid

**Question:** Based on the visual phenomena in the video, which alteration to the light source is most likely to be the cause? <video 1>

**Option:**
(A) It's just that the light source is weaker and the light source position remains the same
(B) The color of the light changes from purple to yellow
(C) It's just that the light source is stronger and the light source position remains the same
(D) The color of the light changes from green to limeyellow

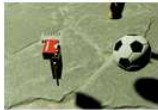

< video 1>

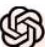

The answer is D. Blue container.

Here's the reasoning:

1. The cyan fluid is initially blocked by the purple stick.
2. If the purple stick were removed, the cyan fluid would flow downwards freely.
3. There's no other obstacle in the path of the cyan fluid, so it would continue downwards until it reaches the blue container.
4. Therefore, the cyan fluid would flow into the blue container. 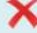

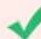Choice: B. Purple container

Explanation:

1. Initial Setup:
   - The images show a setup where colored fluids flow from the top to the bottom through a series of sticks that direct the flow into different containers.

2. Observation of Fluid Paths:
   - The cyan fluid starts from the top left and flows downwards.
   - The purple stick is positioned such that it directs the cyan fluid towards the green container.

3. Effect of Removing the Purple Stick:
   - If the purple stick is removed, the cyan fluid will no longer be directed towards the green container.
   - Instead, the cyan fluid will follow the path of least resistance, which is directly downwards.

4. New Path of Cyan Fluid:
   - Without the purple stick, the cyan fluid will flow straight down.
   - The next obstacle in its path is the orange stick, which will direct the cyan fluid towards the right.

5. Final Destination:
   - After being directed to the right by the orange stick, the cyan fluid will flow into the purple container.

Therefore, if the purple stick were removed, the cyan fluid would flow into the purple container. 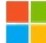

If the purple stick were removed, the cyan fluid would flow into the purple container. This is because the purple stick is acting as a barrier between the cyan and purple containers, preventing the cyan fluid from flowing into the purple container. With the purple stick removed, there would be no obstruction, allowing the cyan fluid to flow into the purple container.

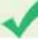

Figure 83: A sample case of physics-based dynamics——fluid. Back to List of Figures.

## Dynamics: Others

**Question:** What happens to the green laser beam as it passes through the prism?
<video 1>

**Option:**
(A) It bends and splits into a spectrum of colors
(B) It reflects off the surface
(C) It remains straight and unchanged
(D) It scatters in different directions

< video 1>

The answer is A. It bends and splits into a spectrum of colors.

Here is the reasoning chain:

1. The video shows a green laser beam passing through a prism.
2. A prism is a triangular piece of glass that bends light.
3. When light passes through a prism, it bends because different wavelengths of light travel at different speeds through the glass.
4. This bending of light causes the different wavelengths to separate, creating a spectrum of colors.

Therefore, the green laser beam bends and splits into a spectrum of colors as it passes through the prism. ✗

Choice: A. It bends and splits into a spectrum of colors

Explanation:

1. Observation of the Images:
  - The images show a green laser beam passing through a prism.
  - The laser beam is incident at an angle to the surface of the prism.
  - The beam bends as it enters the prism and bends again as it exits the prism.

2. Refraction:
  - The bending of the laser beam as it enters and exits the prism is due to refraction. Refraction occurs when light passes from one medium to another with a different refractive index, causing the light to change speed and direction.

3. Snell's Law:
  - The images show measurements of the angles of incidence and refraction, which can be explained by Snell's Law: $n_1 \sin(\theta_1) = n_2 \sin(\theta_2)$, where $n_1$ and $n_2$ are the refractive indices of the two media, and $\theta_1$ and $\theta_2$ are the angles of incidence and refraction, respectively.

4. Dispersion:
  - Although the images primarily show the bending of the green laser beam, a prism typically causes dispersion, which is the splitting of light into its constituent colors. This is because different wavelengths of light refract by different amounts.

5. Conclusion:
  - The green laser beam bends as it passes through the prism due to refraction. While the images do not explicitly show the splitting into a spectrum of colors, the bending and the context of using a prism suggest that dispersion is also occurring. Therefore, the correct choice is that the laser beam bends and splits into a spectrum of colors.

Thus, the correct answer is A. It bends and splits into a spectrum of colors. ✗

When a green laser beam passes through a prism, it undergoes refraction, which is the bending of light as it passes from one medium to another. The prism is made of a material with a different refractive index than the air, causing the light to change direction. As the green laser beam enters the prism, it slows down and bends towards the normal line (an imaginary line perpendicular to the surface at the point of incidence). This bending causes the light to spread out into its component colors, creating a spectrum. This phenomenon is known as dispersion. Therefore, the correct answer is:

A. It bends and splits into a spectrum of colors. ✗

Figure 84: A sample case of physics-based dynamics——others. Back to List of Figures.

## Dynamics: Others

**Question:** What is shown happening in the video? <video 1>

**Option:**
(A) Can't tell
(B) Flowers bloom
(C) Flowers grow
(D) Flowers wither

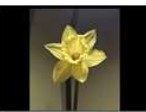

< video 1>

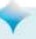 The answer is A. Can't tell.

Here's the reasoning:

1. The video is static: The video only shows a single image of a flower.
2. No movement or change: There is no visual indication of growth, blooming, or withering.
3. Lack of context: The video provides no information about the flower's life cycle or the time period captured. ✗

Therefore, based on the provided information, we cannot determine what is happening in the video.

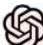 B. Flowers bloom

Step-by-step explanation:

1. Observation of Initial Images: The first few images show a flower that is partially open. The petals are not fully spread out, and the central part of the flower is not fully visible.

2. Progression Over Time: As we move through the series of images, the petals of the flower gradually spread out more, and the central part of the flower becomes more prominent and visible.

3. Final Images: In the last few images, the flower appears to be fully open, with the petals spread wide and the central part fully visible.

4. Conclusion: The sequence of images shows the flower transitioning from a partially open state to a fully open state, indicating that the flower is blooming. ✗

Therefore, the correct choice is B. Flowers bloom.

The image shows a single yellow daffodil in focus with a blurred background. The daffodil appears to be in full bloom with a bright yellow center and petals that are slightly curled at the edges. There is no clear indication of movement or change over time, so it is not possible to determine what is happening in the video based on this still image. Therefore, the answer is A. Can't tell. 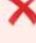

Figure 85: A sample case of physics-based dynamics——others. Back to List of Figures.

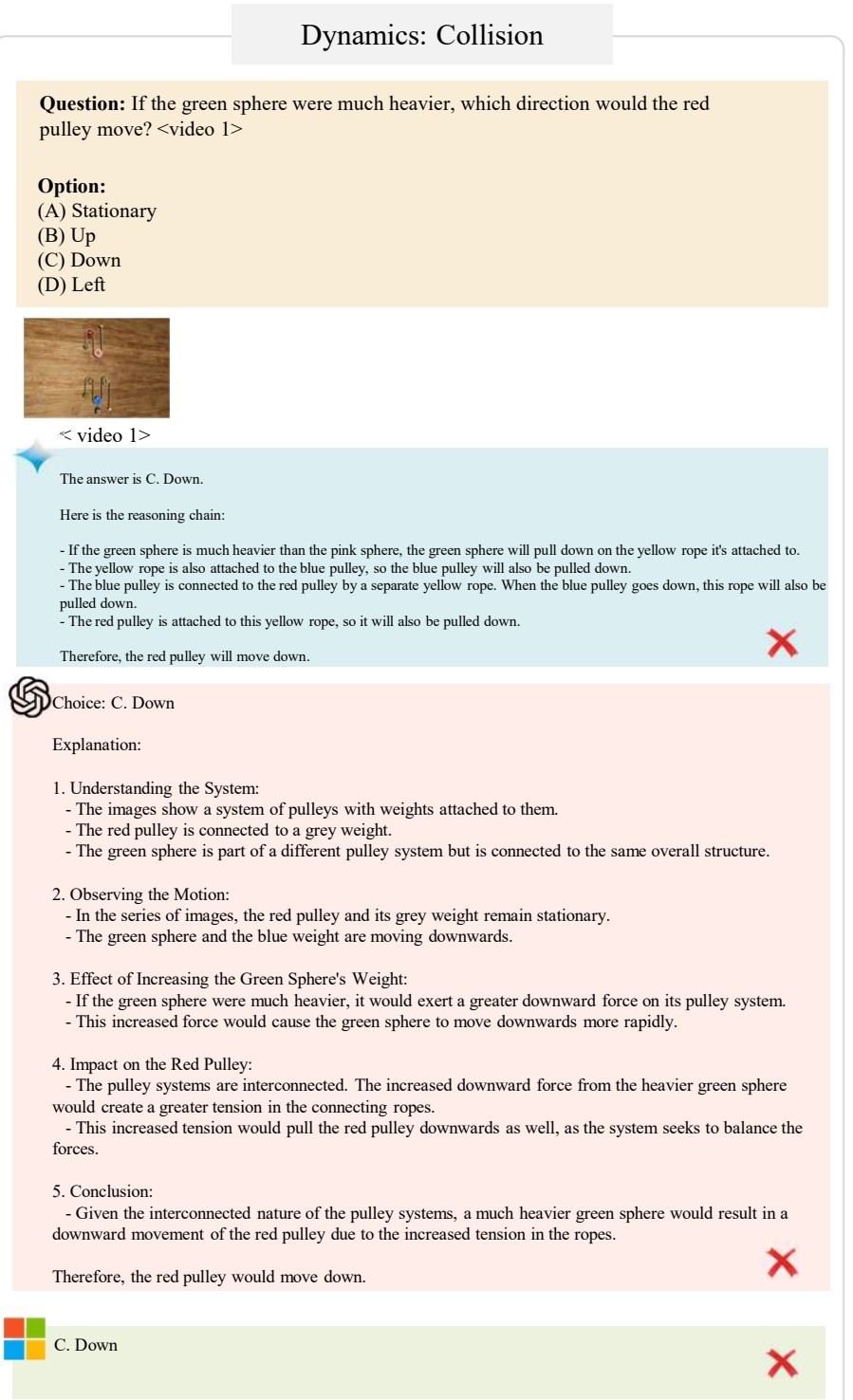

Figure 86: A sample case of physics-based dynamics——collision. Back to List of Figures.

## Scene: Light

**Question:** Considering the occurrences in the video, which light source modification listed below is most likely responsible for the phenomenon? <video 1>

**Option:**
(A) The color of the light changes from orange to purple
(B) The color of the light changes from blue to limeyellow
(C) The light source moves downward
(D) Move parallel to the line between the cigarette and the peach

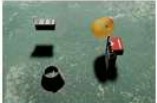

< video 1>

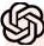 The answer is C. The light source moves downward.
Here's the reasoning:
1. Observe the shadows: Notice how the shadows of all the objects are elongated and pointing in the same direction. This indicates that the light source is positioned at an angle, casting long shadows.
2. Compare the shadows: The shadow of the cigarette box is longer than the shadow of the peach. This means the cigarette box is closer to the light source.
3. Analyze the change: The video shows the shadow of the cigarette box becoming shorter. This means the light source is moving closer to the peach.
4. Deduce the movement: Since the shadow is shrinking, the light source must be moving downward, towards the objects on the surface.
Therefore, the most likely light source modification is the light source moving downward. ❌

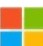 Choice: C. The light source moves downward
Explanation:
1. Observation of Shadows:
  - In the series of images, the shadows of the objects change position and shape.
  - Initially, the shadows are longer and more stretched out.
  - As the sequence progresses, the shadows become shorter and more directly beneath the objects.
2. Analysis of Shadow Behavior:
  - Shadows become shorter and more directly beneath objects when the light source moves closer to the objects (i.e., downward).
  - If the light source were moving parallel to the line between the cigarette and the peach, the shadows would shift horizontally rather than becoming shorter.
  - Changes in the color of the light (options A and B) would not affect the length and position of the shadows in the manner observed.
3. Conclusion:
  - The observed phenomenon of shadows becoming shorter and more directly beneath the objects is consistent with the light source moving downward.
  - Therefore, the most likely modification responsible for the observed changes in the video is that the light source moves downward. ❌

To determine the most likely cause of the phenomenon, let's analyze each option step by step:

1. The color of the light changes from orange to purple: This option suggests that the light source itself is changing color, which is unlikely as the color of the light is not mentioned in the video.

2. The color of the light changes from blue to limeyellow: This option suggests that the light source is changing color, which is also unlikely as the color of the light is not mentioned in the video.

3. The light source moves downward: This option suggests that the light source is moving downward, which could cause the shadow to change shape. However, the shadow of the apple is not affected by the movement of the light source, so this option is not the most likely cause.

4. Move parallel to the line between the cigarette and the peach: This option suggests that the light source is moving parallel to the line between the cigarette and the peach. This could cause the shadow of the apple to change shape as the light source moves, making it the most likely cause of the phenomenon.

Based on the analysis, option 4 is the most likely cause of the phenomenon. The shadow of the apple changes shape as the light source moves parallel to the line between the cigarette and the peach. ✔

Figure 87: A sample case of physical scene understanding–light. Back to List of Figures.

## Scene: Camera

**Question:** Which of the following camera modifications could produce the visual effects seen in the video? <video 1>

**Option:**
(A) The camera is closer to the objects
(B) Move parallel to the line between the jewel and the balletskirt
(C) The camera moves upward or downward
(D) Move parallel to the line between the jewel and the poncho

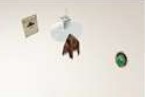

< video 1>

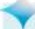 The answer is C. The camera moves upward or downward.
Here's the reasoning:
1. Observe the visual effect: The objects appear to be getting smaller and further away from the viewer. This is a classic sign of a camera moving away from the scene.
2. Eliminate options:
   * A. The camera is closer to the objects: This would make the objects appear larger, not smaller.
   * B. Move parallel to the line between the jewel and the balletskirt: This movement would keep the objects at a similar size and perspective, not create the shrinking effect.
   * D. Move parallel to the line between the jewel and the poncho: Similar to option B, this would not cause the objects to shrink.
3. Conclusion: The only remaining possibility is that the camera is moving upward or downward, creating the illusion of objects shrinking and moving further away as the camera gains distance. ✗

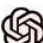 B. Move parallel to the line between the jewel and the ballet skirt
Detailed Explanation:
1. Observation of Images:
   - The series of images shows a consistent set of objects: a "WANTED" poster, a ballet skirt, a poncho, and a jewel.
   - The relative positions of these objects change slightly from one image to the next.
2. Analysis of Object Movement:
   - The "WANTED" poster and the ballet skirt appear to move slightly to the left in each subsequent image.
   - The jewel appears to move slightly to the right in each subsequent image.
3. Determining Camera Movement:
   - If the camera were moving closer to the objects (Choice A), we would expect the objects to appear larger over time, which is not observed.
   - If the camera were moving upward or downward (Choice C), we would expect a vertical shift in the objects' positions, which is not observed.
   - If the camera were moving parallel to the line between the jewel and the poncho (Choice D), we would expect the relative positions of the jewel and the poncho to change more significantly, which is not observed.
4. Conclusion:
   - The observed effect is consistent with the camera moving parallel to the line between the jewel and the ballet skirt (Choice B). This movement would cause the objects on the left (the "WANTED" poster and the ballet skirt) to appear to move leftward and the object on the right (the jewel) to appear to move rightward, which matches the observed changes in the images. ✗

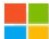 The visual effects seen in the video, where the objects appear to be floating or hovering in the air, could be produced by the camera moving upward or downward. This is because when the camera moves in a vertical direction, it creates a parallax effect, making the objects appear to be at different distances from the camera, giving the illusion of them floating. The other options, such as moving parallel to the line between the jewel and the poncho or being closer to the objects, would not produce the same effect. Therefore, the correct answer is C. The camera moves upward or downward. ✗

Figure 88: A sample case of physical scene understanding–viewpoint. Back to List of Figures.

## Relationships: Location

**Question:** How does the moon pass through the sun in the video? <video 1>

**Option:**
(A) From the lower left corner to the upper right corner of the picture
(B) From the upper left corner to the lower right corner of the picture
(C) From the lower right corner to the upper left corner of the picture
(D) From the upper right corner to the lower left corner of the picture

< video 1>

The answer is D. From the upper right corner to the lower left corner of the picture.
Here is a breakdown of the reasoning:
1. The video shows a partial solar eclipse. This means the moon is passing in front of the sun, but not completely covering it.
2. The initial frame of the video shows the moon partially covering the sun, with a crescent-shaped portion of the sun visible.
3. The shape of the crescent indicates that the moon is entering from the upper right of the screen and moving towards the lower left.
4. As the video progresses, the crescent of sunlight becomes smaller, indicating that the moon is moving across the sun from right to left.
Therefore, the correct answer is D: From the upper right corner to the lower left corner of the picture.   ✗

Choice: A. From the lower left corner to the upper right corner of the picture

Reasoning:

1. Initial Observation:
  - The first image shows a total solar eclipse, where the moon completely covers the sun, leaving only the corona visible.

2. Subsequent Images:
  - In the second image, a small crescent of the sun starts to appear on the lower right side of the moon.
  - As we progress through the images, the crescent of the sun becomes larger and moves towards the upper left side of the moon.

3. Direction of Movement:
  - The sequence of images shows the illuminated part of the sun moving from the lower right to the upper left.
  - This indicates that the moon is moving in the opposite direction, from the lower left to the upper right.

4. Conclusion:
  - Since the illuminated part of the sun appears on the lower right and moves towards the upper left, the moon must be moving from the lower left to the upper right.

Therefore, the correct choice is A. From the lower left corner to the upper right corner of the picture.   ✓

D. From the upper right corner to the lower left corner of the picture   ✗

Figure 89: A sample case of physical object relationships–location. Back to List of Figures.

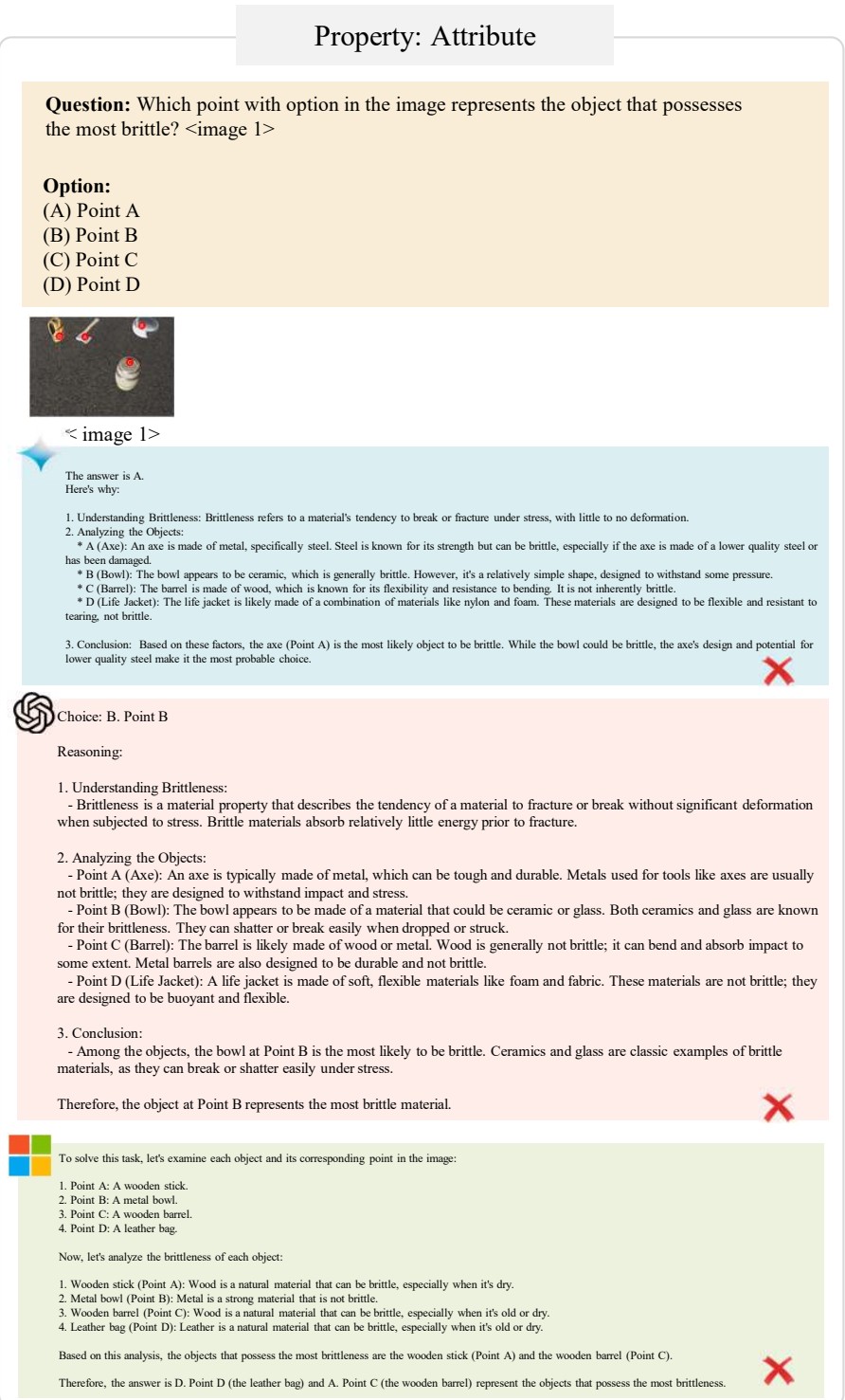

**Property: Attribute**

**Question:** Which point with option in the image represents the object that possesses the most brittle? <image 1>

**Option:**
(A) Point A
(B) Point B
(C) Point C
(D) Point D

< image 1>

The answer is A.
Here's why:

1. Understanding Brittleness: Brittleness refers to a material's tendency to break or fracture under stress, with little to no deformation.
2. Analyzing the Objects:
  * A (Axe): An axe is made of metal, specifically steel. Steel is known for its strength but can be brittle, especially if the axe is made of a lower quality steel or has been damaged.
  * B (Bowl): The bowl appears to be ceramic, which is generally brittle. However, it's a relatively simple shape, designed to withstand some pressure.
  * C (Barrel): The barrel is made of wood, which is known for its flexibility and resistance to bending. It is not inherently brittle.
  * D (Life Jacket): The life jacket is likely made of a combination of materials like nylon and foam. These materials are designed to be flexible and resistant to tearing, not brittle.

3. Conclusion: Based on these factors, the axe (Point A) is the most likely object to be brittle. While the bowl could be brittle, the axe's design and potential for lower quality steel make it the most probable choice. ✗

Choice: B. Point B

Reasoning:

1. Understanding Brittleness:
  - Brittleness is a material property that describes the tendency of a material to fracture or break without significant deformation when subjected to stress. Brittle materials absorb relatively little energy prior to fracture.

2. Analyzing the Objects:
  - Point A (Axe): An axe is typically made of metal, which can be tough and durable. Metals used for tools like axes are usually not brittle; they are designed to withstand impact and stress.
  - Point B (Bowl): The bowl appears to be made of a material that could be ceramic or glass. Both ceramics and glass are known for their brittleness. They can shatter or break easily when dropped or struck.
  - Point C (Barrel): The barrel is likely made of wood or metal. Wood is generally not brittle; it can bend and absorb impact to some extent. Metal barrels are also designed to be durable and not brittle.
  - Point D (Life Jacket): A life jacket is made of soft, flexible materials like foam and fabric. These materials are not brittle; they are designed to be buoyant and flexible.

3. Conclusion:
  - Among the objects, the bowl at Point B is the most likely to be brittle. Ceramics and glass are classic examples of brittle materials, as they can break or shatter easily under stress.

Therefore, the object at Point B represents the most brittle material. ✗

To solve this task, let's examine each object and its corresponding point in the image:

1. Point A: A wooden stick.
2. Point B: A metal bowl.
3. Point C: A wooden barrel.
4. Point D: A leather bag.

Now, let's analyze the brittleness of each object:

1. Wooden stick (Point A): Wood is a natural material that can be brittle, especially when it's dry.
2. Metal bowl (Point B): Metal is a strong material that is not brittle.
3. Wooden barrel (Point C): Wood is a natural material that can be brittle, especially when it's old or dry.
4. Leather bag (Point D): Leather is a natural material that can be brittle, especially when it's old or dry.

Based on this analysis, the objects that possess the most brittleness are the wooden stick (Point A) and the wooden barrel (Point C).

Therefore, the answer is D. Point D (the leather bag) and A. Point C (the wooden barrel) represent the objects that possess the most brittleness. ✗

Figure 90: A sample case of physical object property–attribute. Back to List of Figures.

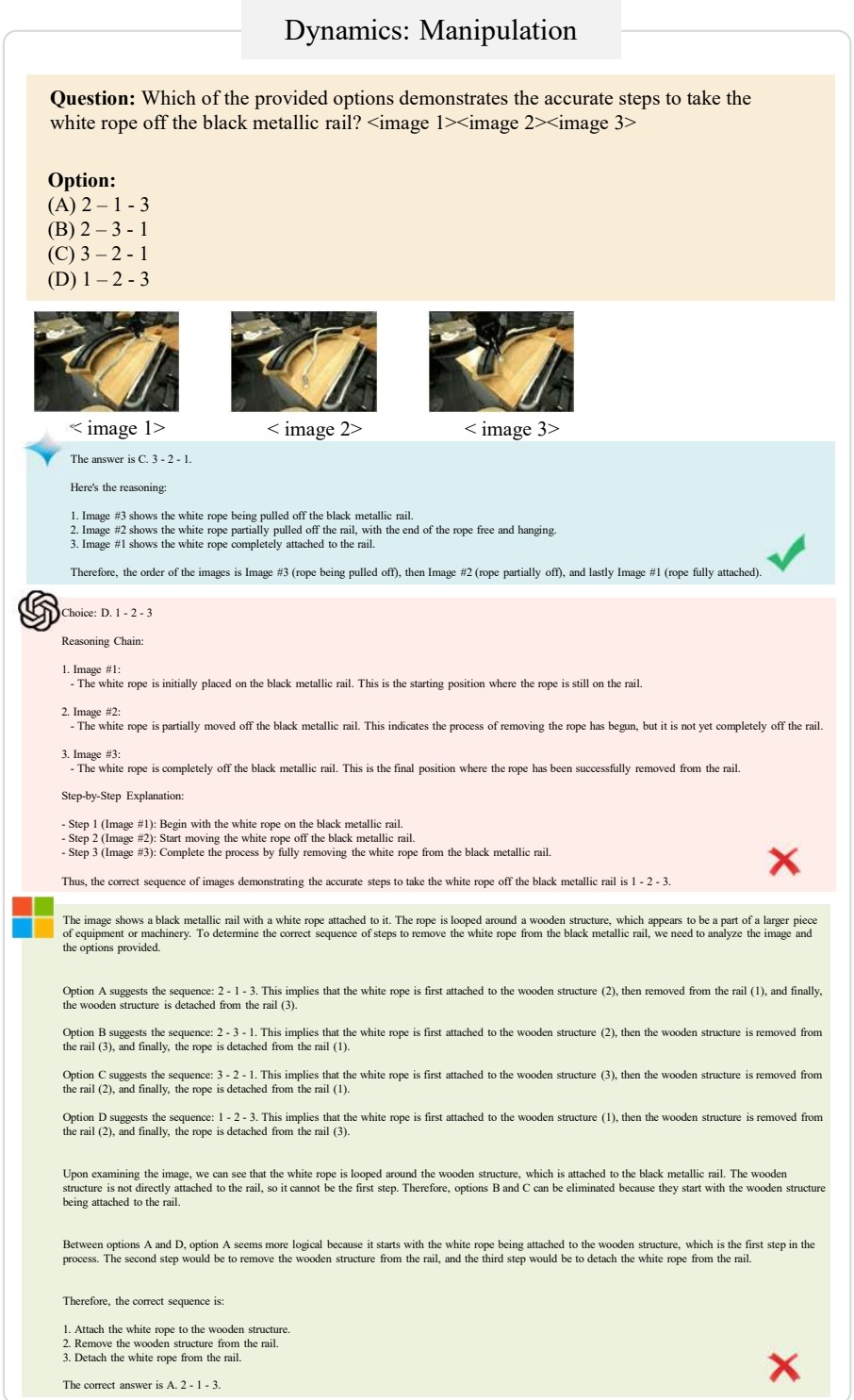

Figure 91: A sample case of physics-based dynamics——manipulation. Back to List of Figures.

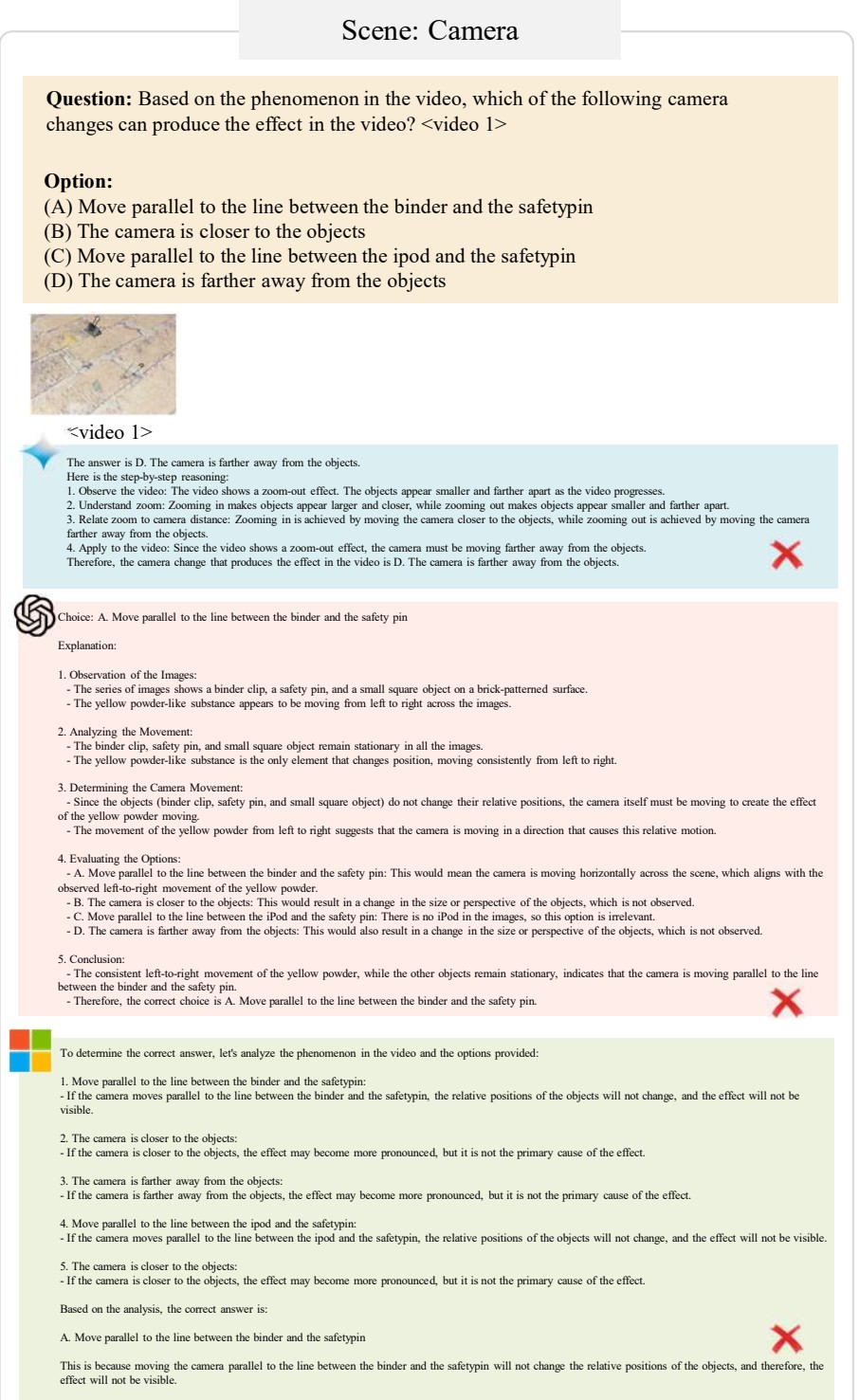

Figure 92: A sample case of physical scene understanding–viewpoint. Back to List of Figures.

## Scene: Light

**Question:** Referring to the phenomenon shown in the video, which light source change from the options below would most likely be the cause? <video 1>

**Option:**
(A) The color of the light changes from limeyellow to pink
(B) It's just that the light source is weaker and the light source position remains the same
(C) It's just that the light source is stronger and the light source position remains the same
(D) Move parallel to the line between the backpack and the cake

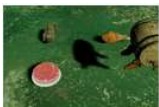

<video 1>

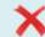 The answer is D. Move parallel to the line between the backpack and the cake.
Here's the reasoning:
1. Observe the shadows: The shadows in the video are elongated and clearly defined, indicating a light source that is positioned relatively close to the objects.
2. Analyze the shadow changes: The shadows of the objects are moving over time. This indicates that the light source itself is also moving, not just changing in intensity or color.
3. Identify the shadow movement pattern: The shadows appear to be moving in a direction that is parallel to the line connecting the backpack and the cake.
4. Deduce the light source movement: For the shadows to move in this way, the light source must also be moving in a direction parallel to the line connecting the backpack and the cake.
5. Eliminate other options:
   * A. Color change wouldn't affect shadow movement.
   * B. A weaker light source would make the shadows less defined, not move them.
   * C. A stronger light source wouldn't change the direction of the shadows.
Therefore, the most likely cause of the shadow changes in the video is a light source moving parallel to the line between the backpack and the cake.

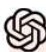 Choice: D. Move parallel to the line between the backpack and the cake
Reasoning Chain:
1. Observation of Shadows:
   - The series of images shows a consistent shadow cast on the ground.
   - The shadow's shape and position change slightly across the images.
2. Analysis of Shadow Movement:
   - The shadow appears to move horizontally across the images.
   - The objects (backpack, cake, ham, and barrel) remain stationary, indicating that the change is due to the light source movement.
3. Light Source Position:
   - The shadow's movement suggests that the light source is moving.
   - The direction of the shadow's movement indicates the direction of the light source's movement.
4. Direction of Light Source Movement:
   - The shadow moves in a manner that suggests the light source is moving parallel to the line between the backpack and the cake.
   - This movement would cause the shadow to shift horizontally as observed in the images. 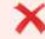
5. Elimination of Other Options:
   - Option A (Color Change): The images do not show any change in the color of the light.
   - Option B (Weaker Light Source): A weaker light source would result in a dimmer scene, not a change in shadow position.
   - Option C (Stronger Light Source): A stronger light source would result in a brighter scene, not a change in shadow position.
Therefore, the most likely cause of the phenomenon shown in the video is that the light source is moving parallel to the line between the backpack and the cake.

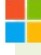 The most likely cause of the change in light source is that the light source is weaker and the light source position remains the same. This is because the shadow of the backpack and the cake appears to be getting larger, indicating that the light source is moving further away from the objects. The other options do not explain the change in shadow size.

Figure 93: A sample case of physics-based dynamics——collision. Back to List of Figures.

## Relationships: Location

**Question:** Relative to the movement of the rope, what is the position of the hand initiating the wave? <video 1>

**Option:**
(A) At the rope's middle point.
(B) At the free end of the rope.
(C) Near the rope's starting end.
(D) At both ends simultaneously.

<video 1>

The answer is C. Near the rope's starting end.

Here's the breakdown:

1. Observe the motion: The rope is shown oscillating in a standing wave pattern. Standing waves occur when waves traveling in opposite directions interfere with each other.
2. Identify the fixed point: The rope is fixed at the top, indicating that the end attached to the ceiling is a node (a point of no displacement).
3. Recognize the pattern: The hand holding the rope is moving up and down, generating the wave. The rope oscillates, creating alternating nodes (N) and antinodes (A) along its length.
4. Analyze the hand's position: The hand is always near the rope's starting end, not at the middle or the free end, and it's not at both ends simultaneously.
5. Confirm the choice: The hand's movement is responsible for initiating the wave and creating the standing wave pattern, which aligns with the choice of being near the rope's starting end.

Therefore, the position of the hand initiating the wave is near the rope's starting end. ✗

C. Near the rope's starting end
Explanation:
1. Observation of the Images:
  - The images show a rope exhibiting different harmonic modes (1st, 2nd, 3rd, and 5th harmonics).
  - The hand initiating the wave is visible in each image, and it is consistently positioned at one end of the rope.

2. Understanding Harmonics:
  - Harmonics are standing wave patterns that occur at specific frequencies.
  - The 1st harmonic (fundamental frequency) has one antinode (A) in the middle and nodes (N) at the ends.
  - The 2nd harmonic has two antinodes and three nodes.
  - The 3rd harmonic has three antinodes and four nodes.
  - The 5th harmonic has five antinodes and six nodes.

3. Position of the Hand:
  - In each image, the hand is seen moving the rope to create these standing wave patterns.
  - The hand is always at the same end of the rope, which is the starting end where the wave is initiated.
4. Conclusion:
  - Since the hand is consistently at the same end of the rope and not in the middle or at both ends, the correct answer is that the hand is near the rope's starting end. ✗

Therefore, the correct choice is
C. Near the rope's starting end.

To determine the position of the hand initiating the wave, we need to analyze the image and the movement of the rope. The rope is shown in motion, and we can see the hand at the end of the rope. The hand appears to be at the end of the rope, as it is the only hand visible and the rope is extending from it. The rope is moving from the top left to the bottom right of the image, indicating that the hand is pulling the rope from the top left to the bottom right. Therefore, the hand initiating the wave is at the end of the rope. ✗

The correct answer is:
C. At both ends simultaneously

Figure 94: A sample case of physics-based dynamics——throwing. Back to List of Figures.

## Relationships: Location

**Question:** What is the position of the electromagnet relative to the can? <video 1>

**Option:**
(A) Below the can
(B) On top of the can
(C) Next to the can
(D) None of the above

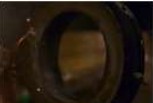

<video 1>

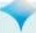 The answer is A. Below the can.

Here's the reasoning chain:

1. Visual Inspection: In the beginning, we see the can spinning and then we see sparks coming out of the machine that is used to launch the can.
2. Inference: We can infer that the sparks are related to the launching of the can.
3. Electromagnet: The sparks are coming out of the electromagnet.
4. Location: The electromagnet is located below the can because the sparks are coming out from below the can.

Therefore, the electromagnet is located below the can. ❌

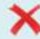 C. Next to the can

Detailed Explanation:
1. Initial Observation: In the first image, we see a can positioned inside a circular device, which appears to be an electromagnet. The can is held horizontally.

2. Sequence of Events: As we move through the images, the can is shown being ejected from the circular device. This suggests that the electromagnet is responsible for propelling the can.

3. Final Position: In the last image, the can is seen lying on a table, and the electromagnet is visible next to it. This confirms that the electromagnet was positioned horizontally relative to the can.

4. Conclusion: Since the electromagnet is seen next to the can in the final image and the can was ejected horizontally, it is clear that the electromagnet was positioned next to the can during the process. ❌

 The image shows a close-up of a cylindrical object with a metallic surface, which appears to be a can due to its shape and the visible text and logo. The electromagnet is positioned around the can, suggesting it is being used to hold or manipulate the can. The exact position of the electromagnet relative to the can cannot be determined with precision without additional context, but it is in close proximity to the can, likely encircling it. Therefore, the most accurate choice would be:

C.Next to the can

The electromagnet is not below, on top, or in a completely different position from the can; it is in close proximity to the can, which is indicated by the visible text and logo on the can and the electromagnet's position around it. ❌

Figure 95: A sample case of physical object relationships–location. Back to List of Figures.

## Dynamics: Others

**Question:** What is the primary reason for the insect's behavior as shown in the video?
<video 1>

**Option:**
(A) To hunt for food
(B) To minimize light refraction interference
(C) To hide from predators
(D) To communicate with other insects

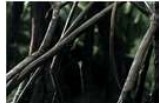

<video 1>

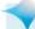 The answer is C. To hide from predators.

Here's why:

1. The insect's movement is erratic and seemingly random. This is a classic predator avoidance strategy.
2. The insect is camouflaged with its surroundings. Its green color blends in with the vegetation, making it harder for predators to spot.
3. The insect is in a complex environment. The branches provide numerous hiding places and escape routes.
4. The insect's behavior is typical of prey animals. Many insects use a combination of camouflage and rapid, unpredictable movements to avoid being caught by predators. ✗

Therefore, the insect's behavior in the video is most likely driven by a desire to avoid becoming prey.

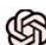 C. To hide from predators

Detailed Explanation:

1. Observation of the Insect's Behavior:
   - The series of images shows an insect initially stationary on a branch.
   - Suddenly, the insect makes a rapid movement, causing water to splash.

2. Analysis of the Environment:
   - The environment appears to be a natural habitat with branches and water, suggesting a place where predators might be present.
   - The rapid movement and splash indicate a sudden reaction, which is often a defensive mechanism.

3. Purpose of the Movement:
   - The rapid movement and splash are not consistent with hunting behavior, as hunting typically involves stealth and precision rather than sudden, conspicuous movements.
   - The behavior does not seem to be related to minimizing light refraction interference, as this would not typically involve such a dramatic action.
   - There is no indication of communication with other insects, which would likely involve more specific and repeated signals rather than a single rapid movement.

4. Conclusion:
   - The most plausible reason for the insect's sudden movement and splash is to evade a threat, suggesting it is trying to hide from predators. This type of behavior is common in many insects as a means of escaping or confusing predators.

Therefore, the primary reason for the insect's behavior as shown in the video is to hide from predators. ✗

 The insect's behavior of bending its legs and positioning itself in a way that creates a shadow on the glass surface is likely a form of camouflage. This behavior helps the insect to blend in with the surroundings, making it less visible to predators. The shadow creates a visual effect that mimics the texture and shape of the glass, providing a form of protection. This is a common survival strategy in the animal kingdom, where many species have evolved to use their environment to their advantage. The insect is not actively hunting for food, nor is it trying to communicate with other insects, and there is no indication that it is trying to minimize light refraction interference. Therefore, the most plausible reason for this behavior is to hide from predators. ✗

Figure 96: A sample case of physics-based dynamics——fluid. 

## Scene: Camera

**Question:** Based on the visual effects in the video, which of the following changes to the camera might create this effect? <video 1>

**Option:**
(A) The camera rotates along the horizontal axis (left or right).
(B) The camera rotates along the vertical axis (upside or downside).
(C) The camera moves upward or downward
(D) The camera is closer to the objects

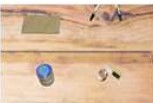

<video 1>

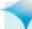 The answer is C. The camera moves upward or downward.

Here is the reasoning:

1. Visual Analysis: The video shows objects appearing to float and move upward. This indicates a change in the camera's position relative to the objects.

2. Eliminating Other Options:
 * A. Rotating along the horizontal axis: This would cause objects to appear to move left or right, not upward.
 * B. Rotating along the vertical axis: This would cause objects to appear to move forward or backward, not upward.
 * D. The camera is closer to the objects:  This would make objects appear larger, not float upward. 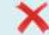

3. Conclusion: The only option that logically explains the upward movement of objects is the camera moving upward. As the camera rises, the objects appear to move upwards from the camera's perspective.

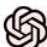 Choice: C. The camera moves upward or downward

Explanation:

1. Initial Observation:
 - In the first few images, the objects (peanut butter jar, syringe, tripod, etc.) are placed on a wooden surface.
 - As the sequence progresses, the objects appear to move upward relative to the wooden surface.

2. Analysis of Object Movement:
 - The peanut butter jar, syringe, and other items gradually rise above the wooden surface.
 - By the fourth image, the objects are completely off the wooden surface and appear to be floating in the air.
 - This upward movement continues until the last image, where the objects are still above the wooden surface.

3. Camera Movement Consideration:
 - Horizontal Rotation (Left or Right): If the camera were rotating horizontally, the objects would appear to move sideways rather than upward.
 - Vertical Rotation (Upside or Downside): If the camera were rotating vertically, the perspective of the objects would change, but they would not uniformly rise above the surface.
 - Closer to Objects: If the camera were moving closer, the objects would appear larger but would not rise above the surface.
 - Upward or Downward Movement: If the camera moves upward, the objects would appear to rise above the surface as the camera's perspective changes.

4. Conclusion:
 - The consistent upward movement of the objects relative to the wooden surface indicates that the camera is moving upward. This creates the effect of the objects rising above the surface as the camera's perspective changes.

Therefore, the correct choice is C. The camera moves upward or downward. 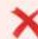

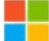 The visual effects in the video are created by the camera rotating along the vertical axis (upside or downside). This is evident from the way the camera appears to be moving up and down, causing the objects to appear to rotate around it. The rotation creates the illusion of the objects moving in a circle, which is a common technique used in 360-degree videos to give the viewer a sense of immersion and movement. The camera's rotation around the vertical axis allows the viewer to see different angles of the objects, making them appear to be in motion. This effect is achieved by capturing multiple frames of the scene from different angles and stitching them together to create a seamless, 360-degree view. The camera's movement along the vertical axis is what gives the viewer the impression of the objects rotating around the camera.

Figure 97: A sample case of physical scene understanding–viewpoint. Back to List of Figures.

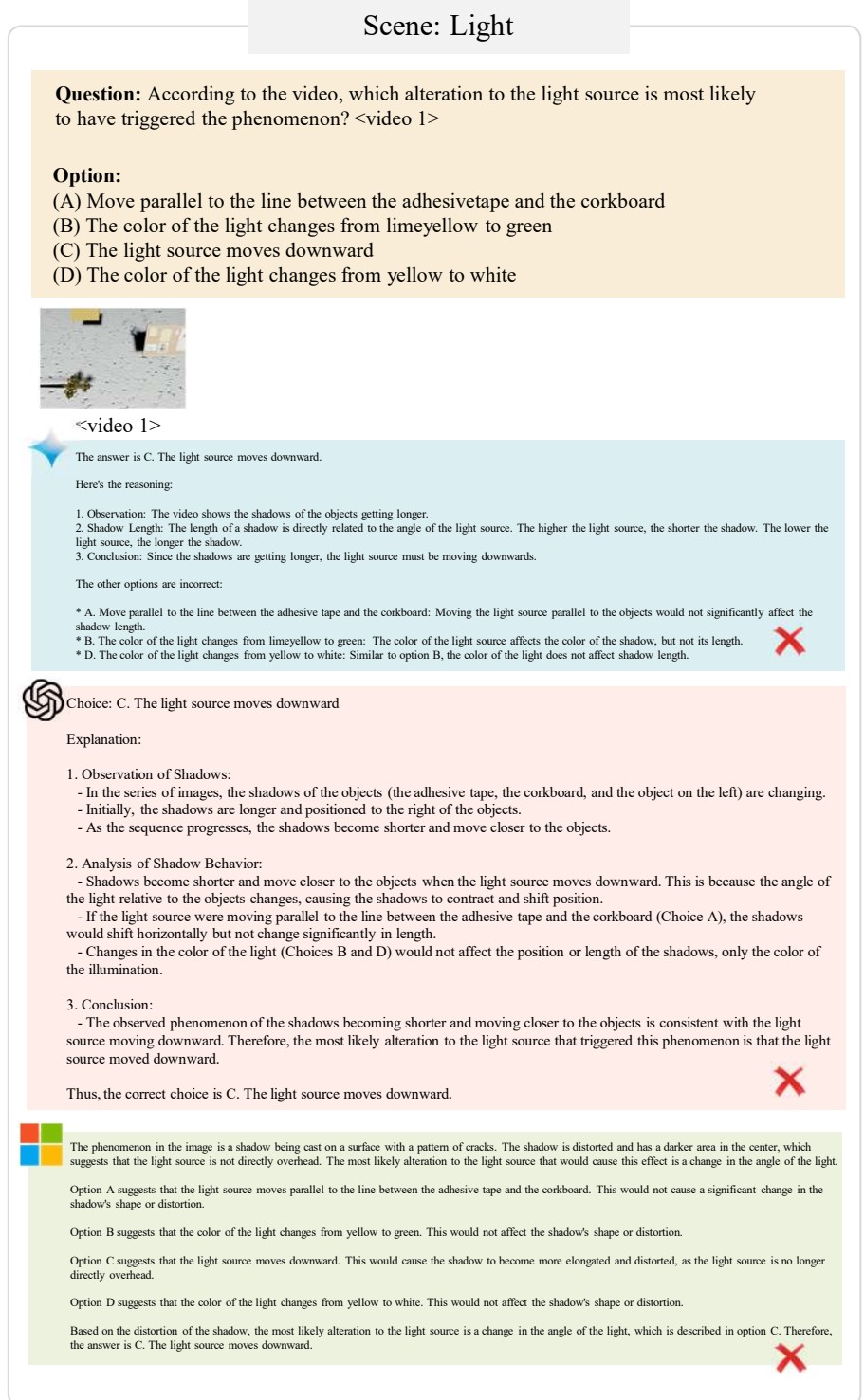

Figure 98: A sample case of physical scene understanding–light. Back to List of Figures.

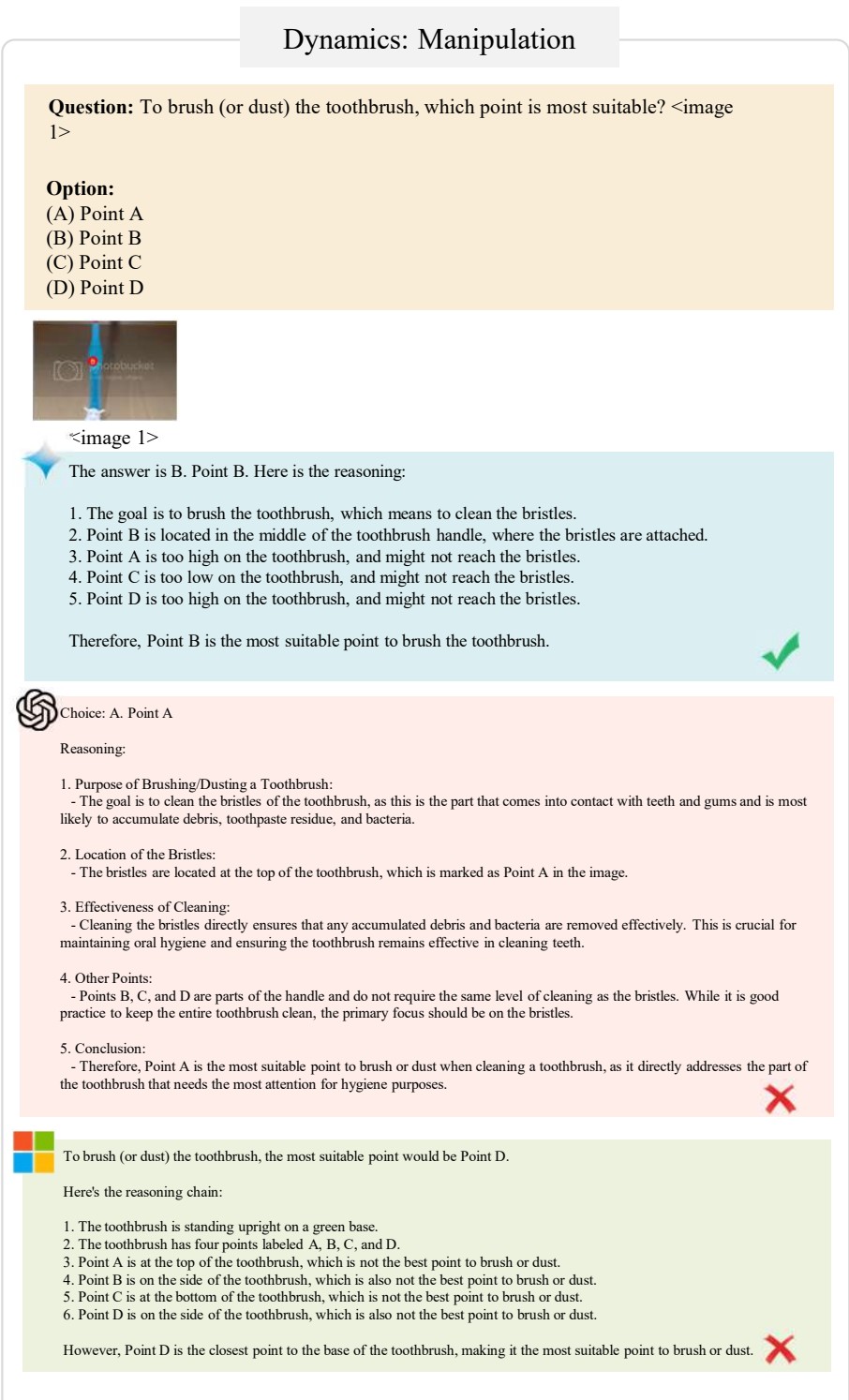

Figure 99: A sample case of physics-based dynamics——manipulation. Back to List of Figures.

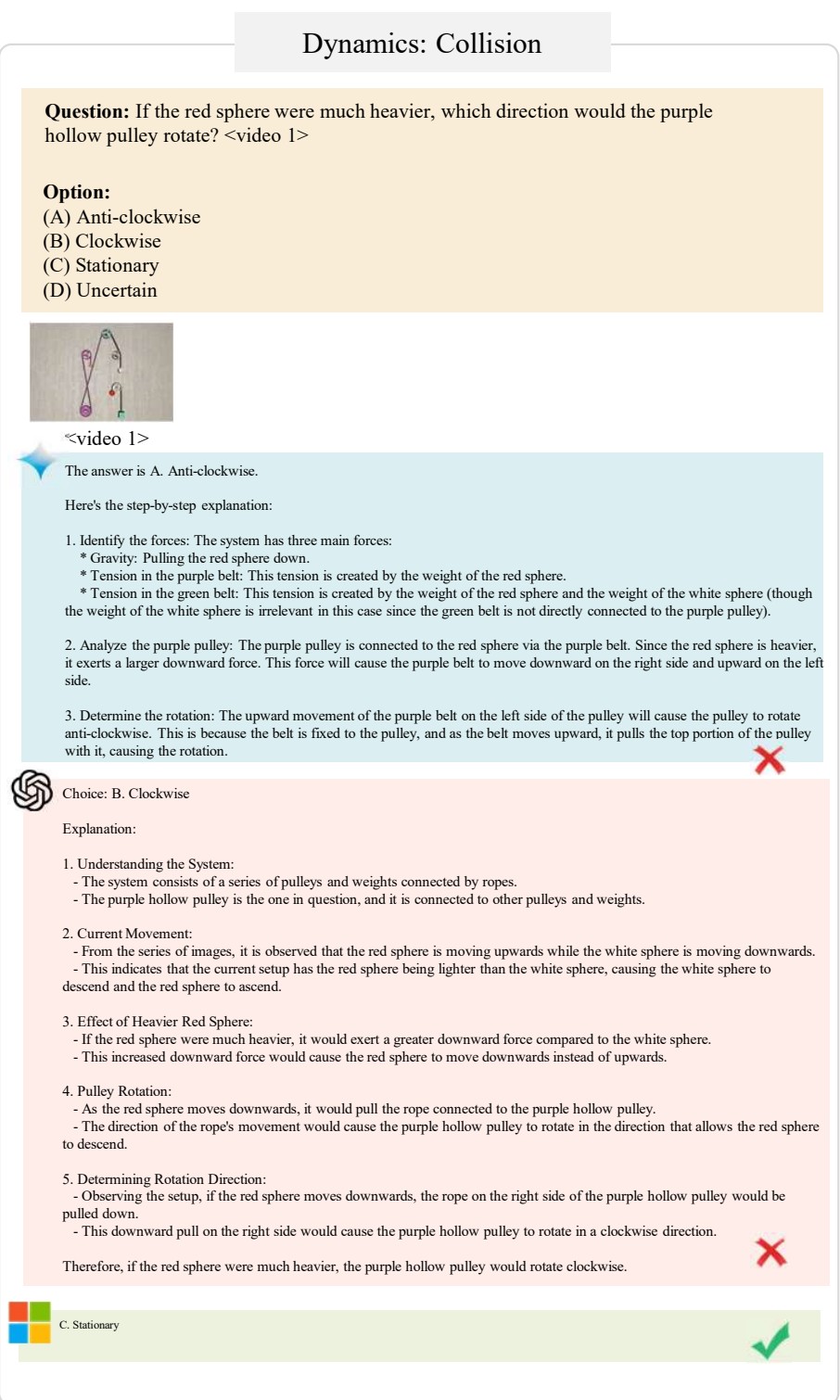

Figure 100: A sample case of physics-based dynamics——collision. Back to List of Figures.

## Property: Attribute

**Question:** Reflecting on the phenomena in the video, which of these camera adjustments could cause the observed effect? <video 1>

**Option:**
(A) Move parallel to the line between the screwdriver and the lemonade
(B) The camera is farther away from the objects
(C) The camera is closer to the objects
(D) Move parallel to the line between the screwdriver and the tool

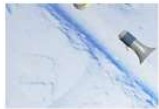

<video 1>

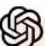 The answer is A. Move parallel to the line between the screwdriver and the lemonade.
Here is the step-by-step reasoning:
1. Observe the image: The image shows a set of tools and a glass of lemonade on a seemingly flat surface. The perspective is distorted, creating a sense of depth where the objects appear to be on a curved or slanted surface.
2. Identify the distortion: The distortion is most noticeable in the lines running parallel to the edges of the image. These lines appear curved, indicating a change in perspective.
3. Consider the camera movements: The question asks about camera adjustments that could cause this effect.
  * A. Move parallel to the line between the screwdriver and the lemonade: Moving the camera parallel to the line would cause the objects further away from the camera to appear smaller, creating the curved lines effect.
  * B. The camera is farther away from the objects: This would make the objects appear smaller overall, but not necessarily cause the curved lines effect.
  * C. The camera is closer to the objects: This would make the objects appear larger overall, not smaller, and wouldn't create the curved lines effect.
  * D. Move parallel to the line between the screwdriver and the tool: This movement is irrelevant to the observed distortion, as it would not affect the relative positions of the objects.
4. Conclude: Therefore, the most likely explanation for the observed distortion is A. Move parallel to the line between the screwdriver and the lemonade, as this movement would cause the perspective change and the resulting curved lines. ❌

Choice: A. Move parallel to the line between the screwdriver and the lemonade
Explanation:
1. Observation of the Images:
  - The series of images shows a set of tools and objects, including a screwdriver and a lemonade, with the background remaining constant.
  - The objects appear to be moving relative to the camera, but their relative positions to each other remain unchanged.
2. Analysis of Movement:
  - The objects are not changing in size, which indicates that the camera is not moving closer or farther away from the objects (eliminating choices B and C).
  - The objects are not moving in a way that suggests a change in perspective along the line between the screwdriver and the tool (eliminating choice D).
3. Direction of Movement:
  - The objects appear to be moving in a direction that is consistent with the camera moving parallel to the line between the screwdriver and the lemonade.
  - This parallel movement would cause the objects to shift position in the frame while maintaining their relative distances and sizes.
4. Conclusion:
  - The observed effect is consistent with the camera moving parallel to the line between the screwdriver and the lemonade, as this would cause the objects to shift in the manner observed without changing their relative positions or sizes.
Therefore, the correct choice is A. Move parallel to the line between the screwdriver and the lemonade. ❌

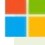 The observed effect in the video is caused by the camera adjustment where the camera is moved parallel to the line between the screwdriver and the tool. This is because when the camera is moved parallel to the line between the screwdriver and the tool, the objects appear to be in different positions, creating the illusion of depth and distance. This is known as the "Parallax Error" or "Parallax Shift" in photography, where the apparent position of an object changes when viewed from different angles or distances. The other options, such as moving the camera farther away or closer to the objects, or moving the camera parallel to the line between the screwdriver and the lemonade, would not create the same effect. Therefore, the correct answer is A. Move parallel to the line between the screwdriver and the tool. ❌

Figure 101: A sample case of physical object property–attribute. Back to List of Figures.

## Scene: Light

**Question:** Referring to the phenomenon shown in the video, which light source change from the options below would most likely be the cause? <video 1>

**Option:**
(A) It's just that the light source is farther away from the objects
(B) It's just that the light source is closer to the objects
(C) It's just that the light source is stronger
(D) It's just that the light source is weaker

<video 1>

The answer is D. It's just that the light source is weaker.

Here is the step-by-step reasoning:

1. Observe the shadows: The shadows of the objects are significantly larger and more elongated than their actual size.

2. Understand shadow formation: Shadows are formed when an object blocks light. The size and shape of the shadow depend on the position of the light source and the object.

3. Apply logic: If the light source is farther away, the shadows would be smaller and less defined. If the light source is closer, the shadows would be larger and more defined. If the light source is stronger, the shadows would be darker.

4. Conclude: Since the shadows are large and elongated, it suggests that the light source is weak, causing the light to spread out more and create larger shadows.

5. Eliminate other options: Option A and B are incorrect because the shadows are getting larger, not smaller. Option C is incorrect because a stronger light source would create darker, more defined shadows. ✗

Choice: A. It's just that the light source is farther away from the objects

Explanation:

1. Observation of Shadows:
   - In the series of images, the shadows of the objects (apple, flower, and artichoke) become progressively longer and more defined.
   - The direction of the shadows also changes, indicating a change in the position of the light source.

2. Light Source Distance:
   - When a light source moves farther away from objects, the shadows cast by the objects become longer. This is because the angle at which the light hits the objects becomes more acute, causing the shadows to stretch out.
   - The images show a clear progression of shadows becoming longer, which is consistent with the light source moving farther away.

3. Light Intensity:
   - The intensity of the light does not appear to change significantly in the images. The objects and their surroundings remain similarly illuminated throughout the sequence.
   - If the light source were simply getting stronger or weaker, we would expect to see changes in the brightness of the objects and their shadows, but the length of the shadows would not necessarily change in the same way.

4. Conclusion:
   - The most plausible explanation for the observed phenomenon is that the light source is moving farther away from the objects, causing the shadows to lengthen. This matches the pattern seen in the images, where the shadows progressively become longer and more defined.

Therefore, the correct choice is A. It's just that the light source is farther away from the objects. ✗

The phenomenon shown in the video is most likely caused by a change in the light source. When the light source is farther away from the objects, the shadows become longer and more diffused, as seen in the image. This is because the light has to travel a greater distance, causing it to spread out and become less concentrated. The shadows are also less defined and more elongated, which is a result of the light being less direct. Therefore, the most likely cause of this phenomenon is that the light source is farther away from the objects. So, the choice is A. It's just that the light source is farther away from the objects. ✗

Figure 102: A sample case of physical scene understanding–light. Back to List of Figures.

## Scene: Light

**Question:** Taking into account the phenomena observed in the video, which of the following changes to the light source is most likely to have led to this result? <video 1>

**Option:**
(A) It's just that the light source is stronger and the light source position remains the same
(B) The color of the light changes from red to cyan
(C) It's just that the light source is weaker and the light source position remains the same
(D) The color of the light changes from white to orange

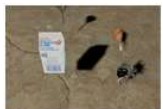

<video 1>

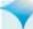 The answer is C. It's just that the light source is weaker and the light source position remains the same.

Here's the reasoning:

1. Observe the shadows: The shadows in the video are elongated and faint. This indicates that the light source is low and weak.

2. Consider the objects: The objects in the video are not brightly illuminated, and their details are not sharp. This reinforces the idea of a weak light source.

3. Eliminate other options:
   * A. Stronger light source: A stronger light source would result in sharper shadows and brighter objects, contradicting the observation.
   * B. Color change: While color change would affect the appearance of the objects, it wouldn't explain the faint and elongated shadows.
   * D. Color change to orange: Similar to option B, a color change to orange wouldn't explain the weakened illumination observed in the shadows. ✗

Therefore, the most likely explanation is that the light source has become weaker, leading to the fainter shadows and dimmer objects observed in the video.

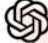 Choice: A. It's just that the light source is stronger and the light source position remains the same

Explanation:

1. Observation of Shadows:
   - In the series of images, the shadows of the objects (milk carton, chicken leg, and other items) become progressively darker and more defined.
   - The position and shape of the shadows remain consistent throughout the images.

2. Light Source Intensity:
   - The increasing darkness and definition of the shadows suggest that the light source is becoming stronger. When a light source becomes stronger, it casts more pronounced and darker shadows because the contrast between the illuminated areas and the shadowed areas increases.

3. Light Source Position:
   - The consistency in the position and shape of the shadows indicates that the position of the light source has not changed. If the light source had moved, the direction and length of the shadows would have changed as well.

4. Color of Light:
   - There is no noticeable change in the color of the light in the images. The objects and their shadows do not exhibit any color shifts that would suggest a change in the light color from red to cyan or from white to orange. ✗

Based on these observations, the most likely explanation for the phenomena observed in the video is that the light source has become stronger while its position has remained the same. Therefore, the correct choice is A. It's just that the light source is stronger and the light source position remains the same.

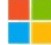 C. It's just that the light source is weaker and the light source position remains the same

The shadows in the image are consistent in shape and size, indicating that the light source is not moving. The only change observed is the shadow of the milk carton becoming less distinct, which suggests that the light source has become weaker. This could be due to the light source being moved further away or the light source itself becoming dimmer. The position of the shadows relative to the objects and the light source remains the same, so the light source position does not change. Therefore, the most likely explanation is that the light source has become weaker, leading to less distinct shadows. ✗

Figure 103: A sample case of physical object property–attribute. Back to List of Figures.

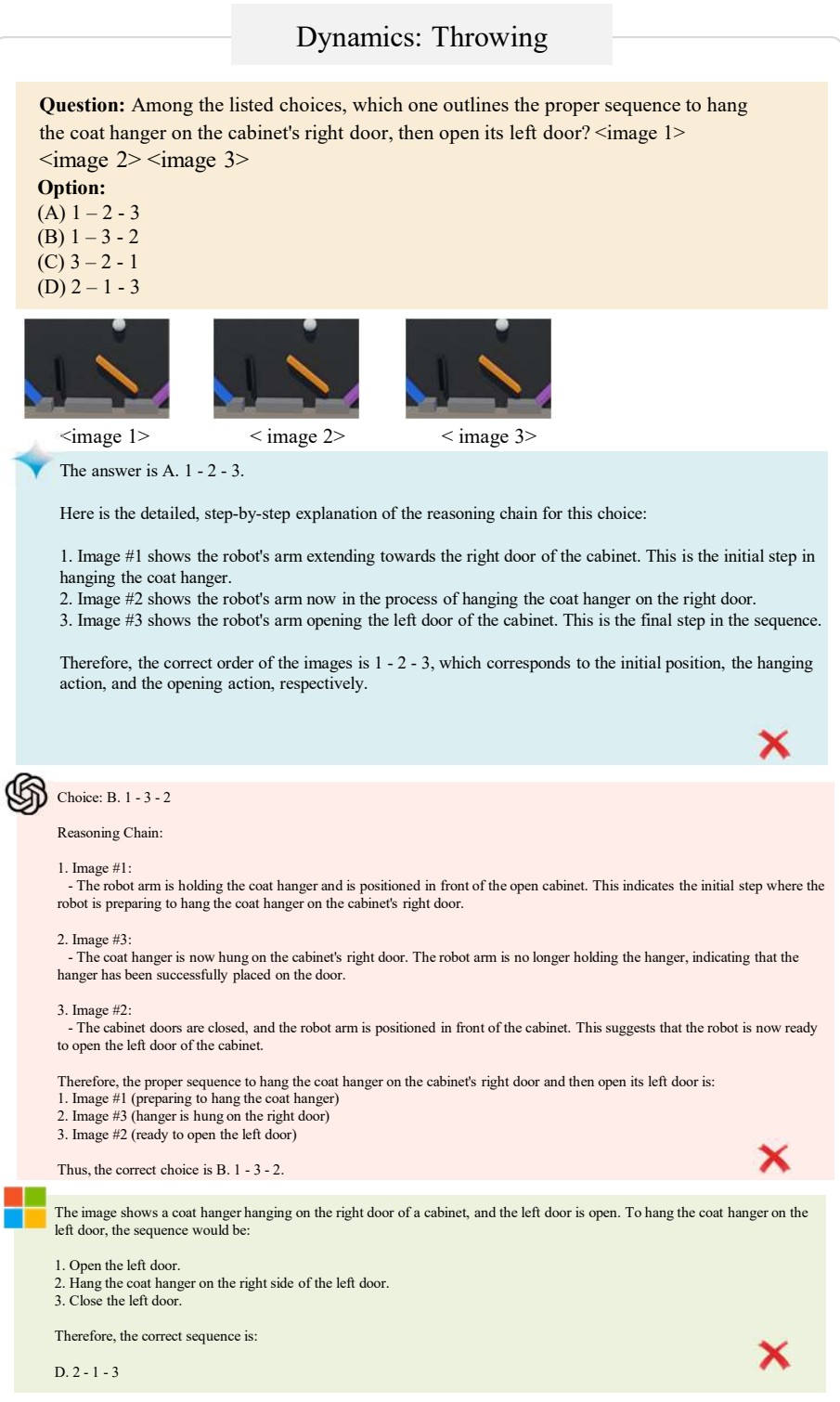

Figure 104: A sample case of physics-based dynamics——throwing. Back to List of Figures.

## Relationships: Size

**Question:** Which object is the biggest in volume? <image 1>

**Option:**
(A) red cube
(B) red cube
(C) green ball
(D) purple cube

<image 1>

The answer is C.
Here's why:
- Although we don't have exact measurements, we can visually assess the relative sizes.
- The green ball is significantly larger than the cubes, even considering the blue cube's shadow.
- This makes the green ball the biggest in volume among the given objects. ✗

Choice: B. purple cube

Explanation:
1. Visual Inspection: By looking at the image, we can see several objects: a green ball, a red cube, a yellow cube, a green cube, and a purple cube.
2. Relative Size Comparison:
   - The green ball appears to be smaller in volume compared to the cubes.
   - Among the cubes, the purple cube is noticeably larger than the red, yellow, and green cubes.
3. Conclusion: Based on the visual comparison, the purple cube is the largest in volume among the objects listed.

Therefore, the biggest object in volume is the purple cube (Choice B). ✗

To determine which object has the biggest volume, we need to compare the volumes of the different shapes.

A cube has equal sides, so its volume is calculated by multiplying the length of one side by itself twice. The purple cube has sides of length 1, so its volume is 1 * 1 * 1 = 1 cubic unit. The yellow cube has sides of length 1, so its volume is also 1 * 1 * 1 = 1 cubic unit. The red cube has sides of length 1, so its volume is 1 * 1 * 1 = 1 cubic unit.

A sphere has a radius, which is the distance from the center to any point on its surface. The green sphere has a radius of 1, so its volume is calculated by multiplying the cube of the radius by 4/3 * pi. The volume of the green sphere is 4/3 * pi * 1^3 = 4/3 * pi cubic units.

Comparing the volumes, we can see that the green sphere has the biggest volume, followed by the purple, yellow, and red cubes, which all have the same volume.

So, the answer is C. green ball. ✗

Figure 105: A sample case of physical object relationships–size. Back to List of Figures.

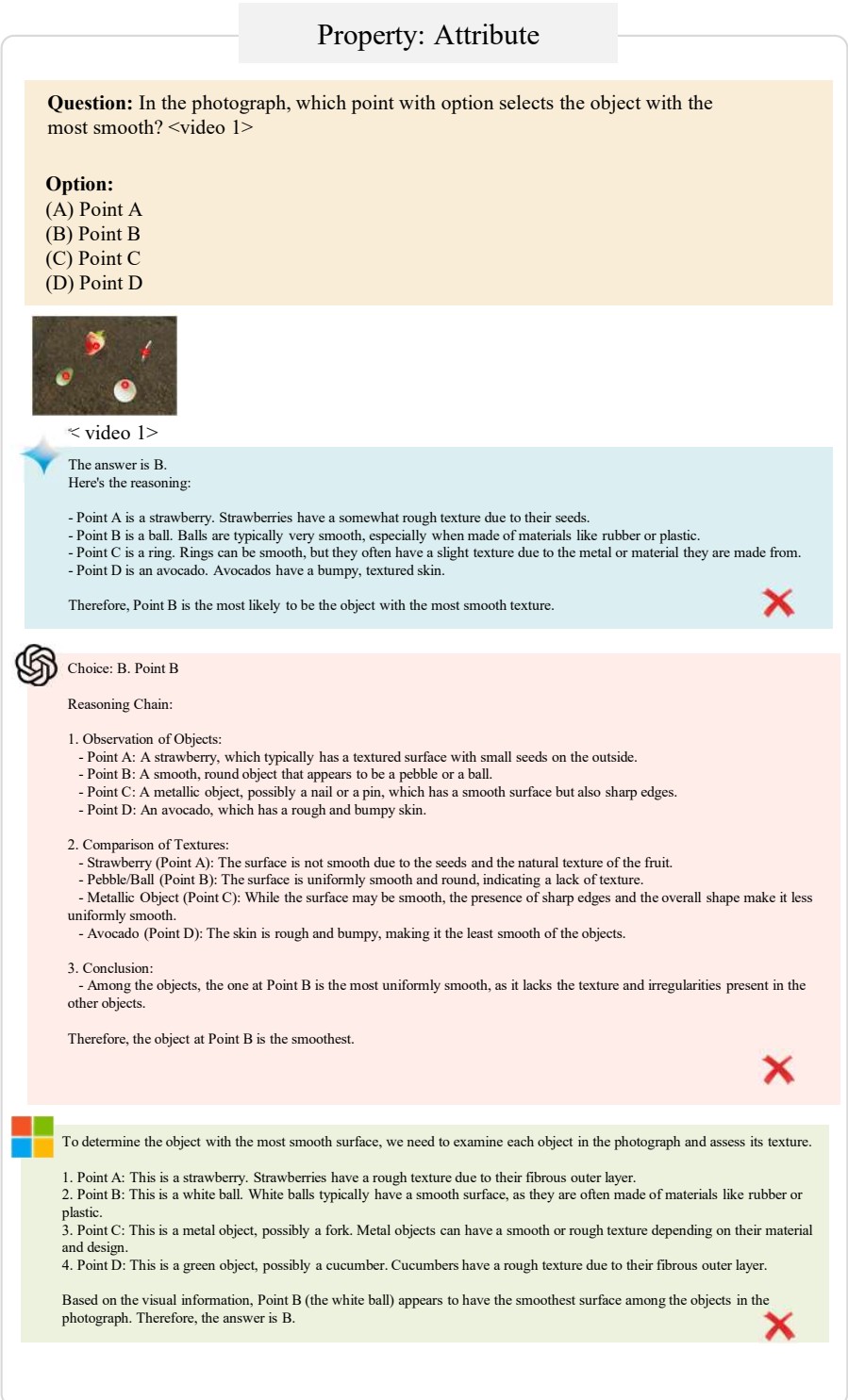

Figure 106: A sample case of physics-based dynamics——fluid. Back to List of Figures.

## Relationships: Depth

**Question:** Which point lies closest to the camera? <video 1>

**Option:**
(A) Point A lies closest
(B) Point B lies closest
(C) Point C lies closest
(D) Point D lies closest

< video 1>

The answer is C.
Here's how we got there:
1. The vehicle is closest to the camera.
2. Point C is on the road in front of the vehicle and is the closest point to it, making it the closest point to the camera.
3. Point B, D, and A are all in the distance behind the car, so they are further away from the camera.
Therefore, Point C is the closest to the camera. ✗

C. Point C lies closest

Reasoning:
1. Point A is in the sky, far in the background, and is part of the distant horizon and clouds.
2. Point B is on the van, which is further down the road and not the closest object to the camera.
3. Point C is on the dirt road, very close to the bottom edge of the image, indicating it is near the camera.
4. Point D is also on the dirt road but is further away from the camera compared to Point C.

Therefore, Point C is the closest to the camera. ✗

To solve this task, let's analyze the image step by step.

1. The image shows a car driving on a dirt road in a desert-like environment.
2. There are four points labeled A, B, C, and D in the image.
3. Point A is located in the sky, point B is on the left side of the car, point C is on the right side of the car, and point D is on the ground, slightly behind the car.
4. The car appears to be closest to point B, as it is the only point that is directly in line with the car. ✗

Based on this analysis, the answer is B. Point B lies closest to the camera.

Figure 107: A sample case of physical object relationships–depth. Back to List of Figures.

## J DISCUSSION AND STATEMENT

### J.1 LIMITATION

Portions of our data are constructed based on pre-existing datasets, as detailed in Table 13. We have made every effort to ensure that the images included in this paper comply with applicable copyright laws and are appropriately credited. Should you be the copyright holder of any image used in our work and believe that its usage conflicts with your licensing agreements, don't hesitate to get in touch with us directly. We are committed to promptly addressing any legitimate concerns.

Although PhysBench is categorized into 4 major categories and 19 subcategories, making it the first benchmark aimed at evaluating a vision-language model's understanding of the physical world, it still does not encompass all aspects of the real physical environment. We invested thousands of hours in data collection, organization, and annotation, following multiple rigorous processes and repeated reviews to ensure data quality. Nevertheless, minor issues may persist in a small portion of the dataset.

In the experiments described in Section 3.4 and 4.1, we utilized GPT-4o-mini to extract answers, which, while currently being one of the most reliable and reproducible methods, may still unavoidably introduce hallucinations. Considering the challenges of using LLMs for evaluation, such as hallucinations and knowledge limitations, assessing open-ended formats is particularly difficult (Yu et al., 2023b; Ge et al., 2024; Li et al., 2023b). Moreover, automatically evaluating the quality of reasoning processes presents additional challenges (Lu et al., 2024b). To address these difficulties and simplify testing, we adopted a multiple-choice format. Nevertheless, PhysBench remains highly challenging, with even the most advanced GPT-4o model achieving less than 50% accuracy. We believe that exploring more complex evaluation formats is a crucial direction for future research.

These challenges are not unique to our dataset but are common across various datasets in VLMs. Nevertheless, we believe that the potential benefits outweigh the associated risks, promoting continued advancement and societal progress. To the best of our knowledge, PhysBench and PhysAgent represent the first effort aimed at benchmarking and enhancing VLMs for physical world understanding, marking a significant step forward in the field. We are committed to the ongoing refinement of our dataset to further improve machine intelligence's comprehension of the physical world, advancing embodied AI toward human-level capabilities.

### J.2 BOARDER IMPACT

The broader impact of PhysBench and PhysAgent carries both potential benefits and risks upon deployment and release. While some considerations pertain specifically to the nature of the dataset, others reflect broader challenges inherent to instruction-following vision-language models (VLMs). Below, we outline key risks and corresponding mitigation strategies.

**Biases.** PhysAgent may inherit biases from its foundational models, both in vision and language foundation models. These biases can manifest in skewed outcomes or unfair representations, necessitating careful evaluation and mitigation efforts.

**Anticipated Societal Implications.** A major societal concern is the potential misuse of the dataset and system, including the generation of fabricated content, which may contribute to misinformation, privacy infringements, and other detrimental outcomes. To mitigate these risks, strict adherence to ethical guidelines and ongoing oversight are essential.

**Environmental Considerations.** In alignment with environmental sustainability goals, we commit to publicly releasing the dataset and scripts to reduce unnecessary carbon emissions by regenerating similar datasets. Throughout our experiments, we ensure compliance with model and data licensing requirements.

## J.3 ETHICS STATEMENT

This study does not raise any ethical concerns, as it exclusively utilizes publicly available pre-existing models, with no involvement of subjective evaluations. All research presented in this paper strictly adheres to the ethical guidelines set forth by the ICLR Code of Ethics.

## J.4 REPRODUCIBILITY STATEMENT

We have adhered to the standard baseline settings employed by existing evaluation benchmarks or the original testing benchmarks of specific models. All necessary implementation details of our method are provided in Appendix E and F. Furthermore, we are committed to releasing the data and code under an open-access license, accompanied by comprehensive instructions to ensure the accurate reproduction of the primary experimental results presented in this paper. All research conducted complies fully with the ICLR Reproducibility Requirements.

# K LATEST RESULTS

The models listed in Table 3 are current as of August 2024. Given the rapid evolution of VLMs, we evaluated an additional 36 models in December 2024, as shown in Table 28. A description of these newly added models and their hyperparameters, along with those of the original 39 models, can be found in Appendix E.3. Due to space limitations, we have not combined the two tables, but a consolidated version is available on our project page at **Our Project Page**. We will continue to update the results to reflect the latest advancements in VLMs.

In Figure 6(a), the largest model size tested was 13B. In this experiment, we have updated to larger models to evaluate whether increasing the model size significantly impacts performance on Phys-Bench. Based on the new models added in Table 3, we present Figure 108 below.

The results in Figure 108 reveal that models ranging from 1B to larger sizes generally show improvements, while the performance between 3B and 13B does not exhibit significant gains, and in some cases, such as with VILA and PLLaVA, a decrease in performance is observed. This observation constitutes the primary area of interest in Figure 6(a). Notably, at larger scales, particularly at 26B and 40B, we observe substantial improvements compared to previous sizes. However, the scalability patterns in PhysBench differ from traditional VQA benchmarks, which typically demonstrate a strong positive correlation between model size and performance. We selected widely-used benchmarks including TextVQA (Singh et al., 2019), MathVista (Lu et al., 2024b), and MMMU (Yue et al., 2024) for comparison. The relationship between model size and performance on these benchmarks is illustrated in Figures 111, 112, and 113, which demonstrate this consistent scaling pattern.

In contrast, PhysBench exhibits less pronounced scalability compared to these benchmarks, with performance not always correlating positively with model size, as shown in Figures 108, 109, and 110. This phenomenon is particularly evident in the Scene subcategory of PhysBench, where performance remains relatively stagnant in the 5-20B parameter range, only showing notable improvements with models exceeding 25B parameters, as demonstrated in Figure 109.

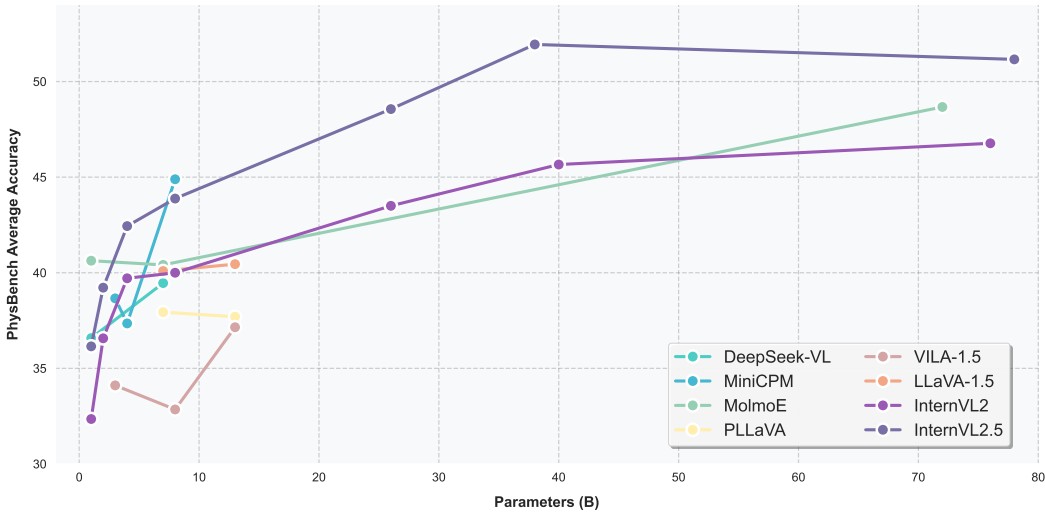

Figure 108: The performance of models of different sizes on PhysBench.

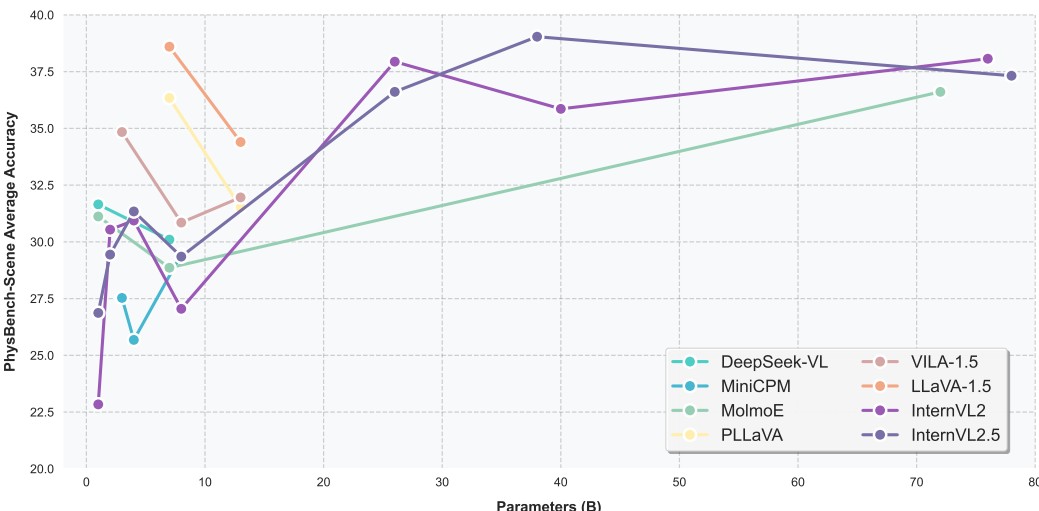

Figure 109: The performance of models of different sizes on PhysBench-Scene.

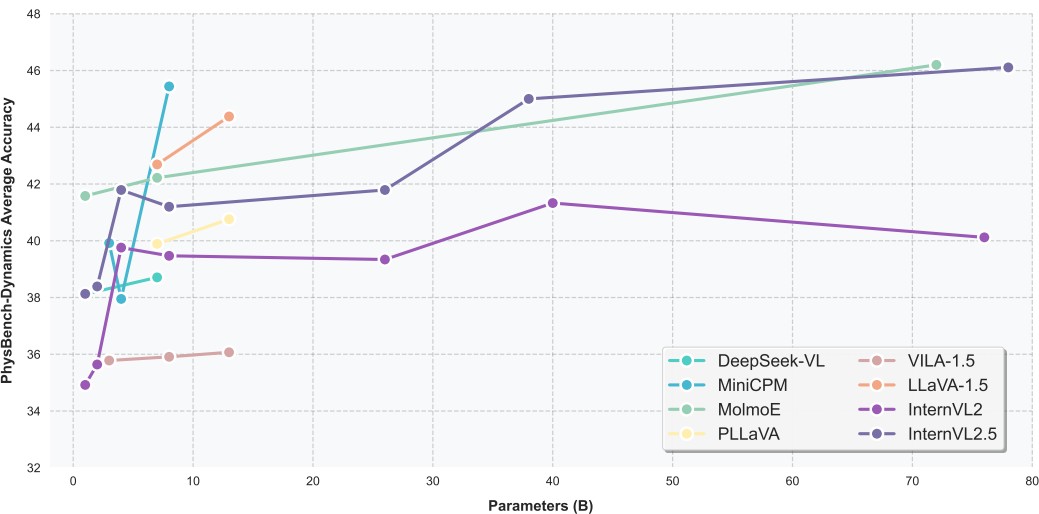

Figure 110: The performance of models of different sizes on PhysBench-Dynamics.

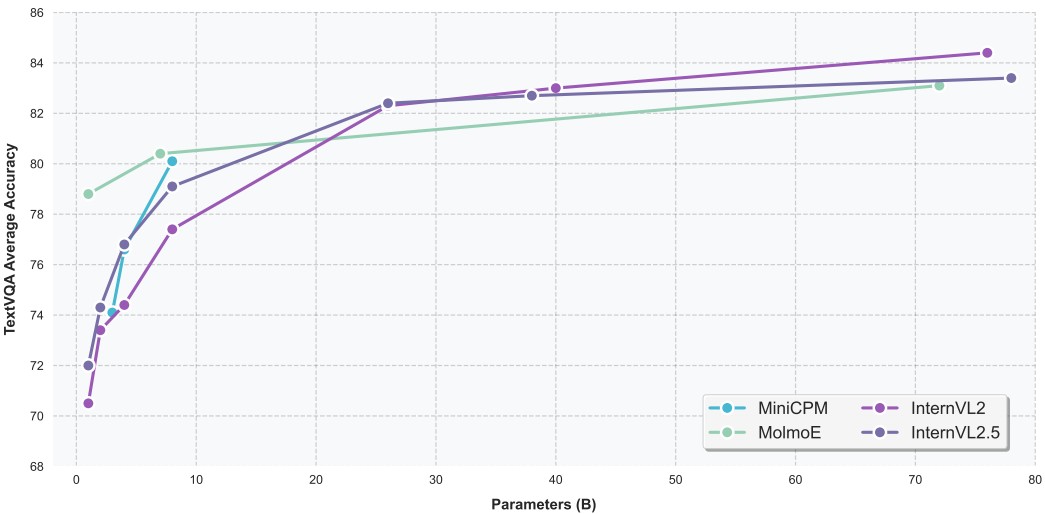

Figure 111: The performance of models of different sizes on TextVQA (Singh et al., 2019).

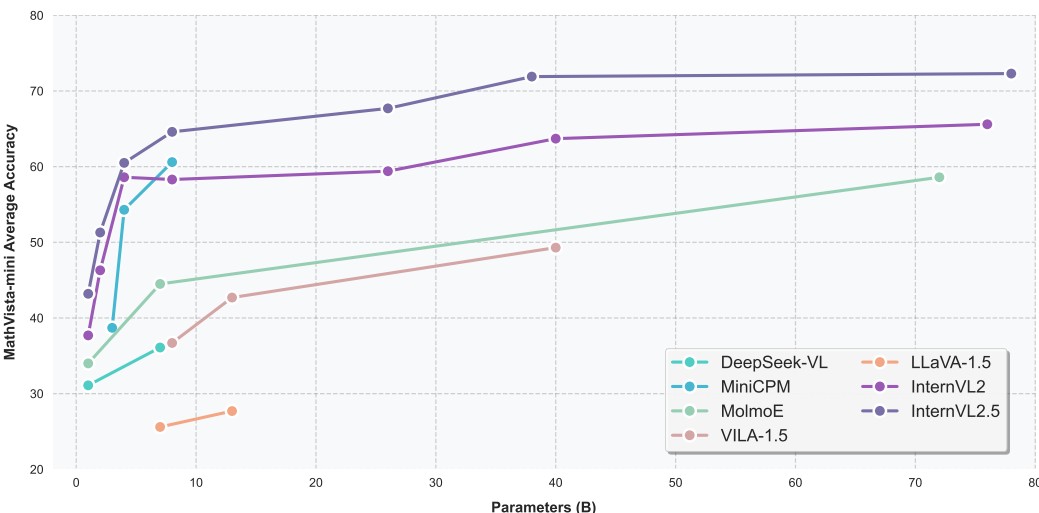

Figure 112: The performance of models of different sizes on MathVista (Lu et al., 2024b).

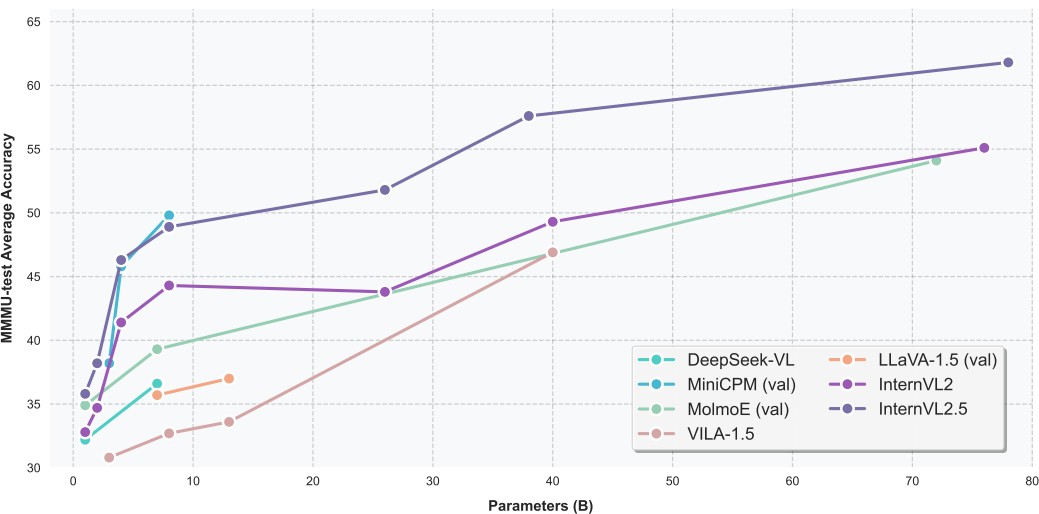

Figure 113: The performance of models of different sizes on MMMU (Yue et al., 2024).

| | Size | Format | ✿Property | ➤Relationships | ♠Scene | △Dynamics | Avg |
|---|---|---|---|---|---|---|---|
| | | | Image VLM | | | | |
| MolmoE-1B Deitke et al. (2024) | 1B | merge | 43.17 | 58.06 | 31.12 | 41.58 | 40.63 |
| MolmoE-7B Deitke et al. (2024) | 7B | merge | 42.35 | 63.33 | 28.86 | 42.22 | 40.41 |
| MolmoE-7B-D Deitke et al. (2024) | 7B | merge | 49.48 | 67.78 | 30.81 | 47.18 | 45.18 |
| MolmoE-72B Deitke et al. (2024) | 72B | merge | 55.13 | 72.08 | 36.61 | 46.20 | 48.67 |
| MiniCPM2 (Yao et al., 2024) | 3B | merge | 45.62 | 44.17 | 27.53 | 39.92 | 38.66 |
| MiniCPM2.5 (Yao et al., 2024) | 4B | merge | 44.26 | 47.08 | 25.68 | 37.95 | 37.35 |
| MiniCPM2.6 (Yao et al., 2024) | 8B | merge | 51.43 | 67.92 | 29.44 | 45.44 | 44.89 |
| Xinyuan-VL (Group, 2024) | 2B | merge | 47.97 | 57.50 | 33.91 | 42.22 | 43.09 |
| Aquila-VL (Gu et al., 2024) | 2B | merge | 47.04 | 62.50 | 30.15 | 45.49 | 43.22 |
| DeepSeek-VL-1B (Lu et al., 2024a) | 1B | merge | 38.37 | 40.28 | 31.65 | 38.12 | 36.58 |
| DeepSeek-VL-7B (Lu et al., 2024a) | 7B | merge | 43.79 | 55.97 | 30.10 | 38.71 | 39.46 |
| PaliGemma2-3B (Steiner et al., 2024) | 3B | merge | 22.03 | 23.33 | 24.75 | 31.73 | 25.94 |
| PaliGemma2-10B (Steiner et al., 2024) | 10B | merge | 35.31 | 41.53 | 28.73 | 39.69 | 35.40 |
| | | | General VLM + Interleaved data | | | | |
| Phi-3.5V (AzureML, 2024) | 4B | seq | 45.72 | 40.15 | 33.02 | 39.40 | 39.75 |
| NVILA-8B (Liu et al., 2024f) | 8B | seq | 55.79 | 40.29 | 33.95 | 43.43 | 43.82 |
| NVILA-15B (Liu et al., 2024f) | 15B | seq | 59.16 | 42.34 | 38.78 | 45.72 | 46.91 |
| NVILA-Lite-8B (Liu et al., 2024f) | 8B | seq | 53.81 | 39.25 | 34.62 | 41.17 | 42.55 |
| NVILA-Lite-15B (Liu et al., 2024f) | 15B | seq | 55.44 | 40.15 | 38.11 | 44.38 | 44.93 |
| mPLUG-Owl3-1B (Ye et al., 2024) | 1B | seq | 38.02 | 31.54 | 21.87 | 33.68 | 31.68 |
| mPLUG-Owl3-2B (Ye et al., 2024) | 2B | seq | 40.92 | 35.11 | 26.69 | 35.64 | 34.87 |
| mPLUG-Owl3-7B (Ye et al., 2024) | 7B | seq | 49.25 | 45.62 | 35.90 | 40.61 | 42.83 |
| InternVL2-1B (Wang et al., 2024e) | 1B | seq | 37.05 | 33.06 | 22.84 | 34.92 | 32.35 |
| InternVL2-2B (Wang et al., 2024e) | 2B | seq | 44.17 | 35.06 | 30.54 | 35.64 | 36.57 |
| InternVL2-4B (Wang et al., 2024e) | 4B | seq | 47.12 | 39.96 | 30.94 | 39.76 | 39.71 |
| InternVL2-8B (Wang et al., 2024e) | 8B | seq | 49.05 | 43.58 | 27.05 | 39.47 | 40.00 |
| InternVL2-26B (Wang et al., 2024e) | 26B | merge | 51.92 | 45.20 | 37.94 | 39.34 | 43.50 |
| InternVL2-40B (Wang et al., 2024e) | 40B | merge | 55.79 | 50.05 | 35.86 | 41.33 | 45.66 |
| InternVL2-76B (Wang et al., 2024e) | 76B | merge | 57.65 | 52.43 | 38.07 | 40.12 | 46.77 |
| InternVL2.5-1B (Gao et al., 2024b) | 1B | seq | 44.25 | 33.30 | 26.87 | 38.13 | 36.15 |
| InternVL2.5-2B (Gao et al., 2024b) | 2B | seq | 49.63 | 38.15 | 29.44 | 38.39 | 39.22 |
| InternVL2.5-4B (Gao et al., 2024b) | 4B | seq | 51.03 | 44.77 | 31.34 | 41.79 | 42.44 |
| InternVL2.5-8B (Gao et al., 2024b) | 8B | seq | 55.87 | 48.67 | 29.35 | 41.20 | 43.88 |
| InternVL2.5-26B (Gao et al., 2024b) | 26B | merge | 59.08 | 58.33 | 36.61 | 41.79 | 48.56 |
| InternVL2.5-38B (Gao et al., 2024b) | 38B | merge | 58.77 | 67.51 | 39.04 | 45.00 | 51.94 |
| InternVL2.5-78B (Gao et al., 2024b) | 78B | merge | 60.32 | 62.13 | 37.32 | 46.11 | 51.16 |
| o1 (Jaech et al., 2024) | - | merge | 59.27 | 73.79 | 40.95 | 49.22 | 55.11 |

Table 28: **Evaluation results for latest 36 VLMs.** The evaluation of General VLMs is based on the data from Video and Image VLM evaluations, with the addition of interleaved data. "Seq" refers to sequential input of frames of videos, while "merge" refers to merging video frames into a single image. Results of original 39 VLMs can be seen in Table 3.

| Model | Size | Identify | Comparison | Static | Dynamic | Perception | Prediction | Judgment | Reasoning | Avg |
|---|---|---|---|---|---|---|---|---|---|---|
| | | | | Image VLM | | | | | | |
| MolmoE-1B Deitke et al. (2024) | 1B | 56.76 | 32.42 | 64.74 | 39.07 | 48.22 | 37.84 | 41.88 | 29.22 | 40.63 |
| MolmoE-7B Deitke et al. (2024) | 7B | 54.84 | 32.49 | 72.01 | 37.75 | 49.85 | 39.22 | 45.01 | 25.20 | 40.41 |
| MolmoE-7B-D Deitke et al. (2024) | 7B | 67.04 | 35.63 | 74.44 | 49.67 | 54.74 | 43.82 | 49.00 | 27.19 | 45.18 |
| MolmoE-72B Deitke et al. (2024) | 72B | 68.70 | 44.51 | 76.49 | 58.94 | 54.33 | 43.13 | 50.71 | 33.00 | 48.67 |
| MiniCPM2 (Yao et al., 2024) | 3B | 62.41 | 32.49 | 45.71 | 38.41 | 47.40 | 36.33 | 44.86 | 24.12 | 38.66 |
| MiniCPM2.5 (Yao et al., 2024) | 4B | 49.48 | 40.07 | 45.90 | 51.66 | 40.88 | 36.06 | 41.43 | 23.62 | 37.35 |
| MiniCPM2.6 (Yao et al., 2024) | 8B | 65.82 | 40.07 | 72.20 | 56.95 | 52.50 | 42.03 | 48.57 | 26.01 | 44.89 |
| Xinyuan-VL (Group, 2024) | 2B | 63.12 | 35.77 | 63.43 | 41.06 | 49.34 | 39.56 | 45.30 | 30.86 | 43.09 |
| Aquila-VL (Gu et al., 2024) | 2B | 62.51 | 35.02 | 66.04 | 51.66 | 50.97 | 41.00 | 46.72 | 28.68 | 43.22 |
| DeepSeek-VL-1B (Lu et al., 2024a) | 1B | 48.82 | 29.90 | 40.49 | 43.05 | 41.59 | 35.78 | 40.17 | 30.47 | 36.58 |
| DeepSeek-VL-7B (Lu et al., 2024a) | 7B | 55.36 | 35.22 | 59.51 | 41.72 | 45.77 | 35.23 | 40.74 | 27.73 | 39.46 |
| PaliGemma2-3B (Steiner et al., 2024) | 3B | 18.31 | 25.19 | 23.69 | 21.19 | 24.57 | 37.29 | 20.51 | 25.20 | 25.94 |
| PaliGemma2-10B (Steiner et al., 2024) | 10B | 46.21 | 27.03 | 42.35 | 39.07 | 44.04 | 37.57 | 40.17 | 26.59 | 35.40 |
| | | | | General VLM + Interleaved data | | | | | | |
| Phi-3.5-Vision-Instruct (AzureML, 2024) | 4B | 58.59 | 35.84 | 39.09 | 50.33 | 48.32 | 39.55 | 32.99 | 30.64 | 39.75 |
| NVILA-8B (Liu et al., 2024f) | 8B | 70.71 | 44.10 | 38.05 | 64.71 | 55.76 | 44.18 | 34.02 | 30.45 | 43.82 |
| NVILA-15B (Liu et al., 2024f) | 15B | 75.85 | 46.28 | 40.03 | 64.71 | 57.70 | 48.54 | 33.12 | 35.42 | 46.91 |
| NVILA-Lite-8B (Liu et al., 2024f) | 8B | 69.05 | 41.77 | 37.47 | 59.48 | 53.82 | 41.18 | 33.12 | 31.16 | 42.55 |
| NVILA-Lite-15B (Liu et al., 2024f) | 15B | 68.09 | 45.60 | 38.31 | 58.17 | 59.53 | 45.41 | 31.46 | 34.71 | 44.93 |
| mPLUG-Owl3-1B (Ye et al., 2024) | 1B | 46.56 | 31.26 | 30.64 | 41.83 | 41.28 | 36.22 | 25.83 | 18.53 | 31.68 |
| mPLUG-Owl3-2B (Ye et al., 2024) | 2B | 54.23 | 30.65 | 34.24 | 41.83 | 40.77 | 37.58 | 31.46 | 23.89 | 34.87 |
| mPLUG-Owl3-7B (Ye et al., 2024) | 7B | 65.39 | 36.59 | 44.68 | 56.21 | 53.62 | 41.59 | 30.69 | 32.50 | 42.83 |
| InternVL2-1B (Wang et al., 2024e) | 1B | 48.04 | 28.67 | 32.36 | 39.22 | 40.06 | 35.60 | 30.31 | 20.68 | 32.35 |
| InternVL2-2B (Wang et al., 2024e) | 2B | 60.77 | 31.19 | 34.24 | 43.79 | 43.12 | 35.47 | 33.25 | 27.62 | 36.57 |
| InternVL2-4B (Wang et al., 2024e) | 4B | 61.20 | 36.18 | 38.31 | 56.86 | 53.62 | 42.95 | 27.24 | 26.28 | 39.71 |
| InternVL2-8B (Wang et al., 2024e) | 8B | 64.95 | 36.79 | 42.28 | 56.86 | 52.19 | 39.96 | 32.35 | 22.40 | 40.00 |
| InternVL2-26B (Wang et al., 2024e) | 26B | 66.35 | 40.55 | 44.02 | 58.56 | 52.31 | 47.11 | 38.18 | 32.18 | 44.18 |
| InternVL2-40B (Wang et al., 2024e) | 40B | 69.83 | 44.91 | 49.01 | 60.13 | 55.66 | 43.50 | 28.13 | 32.17 | 45.66 |
| InternVL2-76B (Wang et al., 2024e) | 76B | 71.23 | 47.37 | 51.10 | 64.05 | 52.91 | 40.16 | 33.38 | 34.47 | 46.77 |
| InternVL2.5-1B (Gao et al., 2024b) | 1B | 63.12 | 29.56 | 32.41 | 41.18 | 47.09 | 41.39 | 25.96 | 24.03 | 36.15 |
| InternVL2.5-2B (Gao et al., 2024b) | 2B | 68.35 | 35.02 | 37.58 | 42.48 | 47.50 | 39.28 | 29.92 | 27.00 | 39.22 |
| InternVL2.5-4B (Gao et al., 2024b) | 4B | 66.52 | 38.84 | 43.37 | 60.78 | 54.43 | 43.02 | 30.69 | 27.91 | 42.44 |
| InternVL2.5-8B (Gao et al., 2024b) | 8B | 72.62 | 43.00 | 47.39 | 60.78 | 51.58 | 42.82 | 31.33 | 26.09 | 43.88 |
| InternVL2.5-26B (Gao et al., 2024b) | 26B | 75.94 | 45.94 | 58.04 | 60.78 | 55.15 | 40.84 | 33.89 | 33.60 | 48.56 |
| InternVL2.5-38B (Gao et al., 2024b) | 38B | 74.02 | 46.89 | 67.38 | 69.28 | 58.92 | 41.93 | 38.87 | 36.57 | 51.94 |
| InternVL2.5-78B (Gao et al., 2024b) | 38B | 78.47 | 46.21 | 61.17 | 71.90 | 60.35 | 42.61 | 39.77 | 34.90 | 51.16 |
| o1 (Jaech et al., 2024) | - | 78.73 | 44.23 | 73.38 | 78.43 | 55.56 | 43.29 | 52.56 | 40.35 | 55.11 |

Table 29: **Evaluation Results for 36 new VLMs Categorized by Ability Dimensions**. PhysBench includes two evaluation dimensions: ability and task. Table 28 presents results categorized by the task-type dimension.

| Model | Size | Number | Mass | Color | Attribute |
|---|---|---|---|---|---|
| Image VLM | | | | | |
| MolmoE-1B Deitke et al. (2024) | 1B | 56.85 | 23.89 | 52.67 | 39.39 |
| MolmoE-7B Deitke et al. (2024) | 7B | 48.01 | 30.03 | 56.67 | 39.46 |
| MolmoE-7B-D Deitke et al. (2024) | 7B | 65.16 | 36.52 | 63.33 | 42.73 |
| MolmoE-72B Deitke et al. (2024) | 72B | 63.95 | 51.19 | 64.00 | 50.25 |
| MiniCPM2 (Yao et al., 2024) | 3B | 56.52 | 42.66 | 66.56 | 37.26 |
| MiniCPM2.5 (Yao et al., 2024) | 4B | 46.26 | 50.17 | 45.82 | 41.80 |
| MiniCPM2.6 (Yao et al., 2024) | 8B | 54.09 | 58.70 | 70.23 | 44.71 |
| Xinyuan-VL (Group, 2024) | 2B | 54.25 | 47.44 | 71.67 | 40.38 |
| Aquila-VL (Gu et al., 2024) | 2B | 55.63 | 43.69 | 64.33 | 40.38 |
| DeepSeek-VL-1B (Lu et al., 2024a) | 1B | 48.18 | 32.76 | 49.33 | 33.29 |
| DeepSeek-VL-7B (Lu et al., 2024a) | 7B | 51.99 | 39.59 | 52.33 | 39.46 |
| PaliGemma2-3B (Steiner et al., 2024) | 3B | 20.80 | 17.75 | 17.67 | 24.34 |
| PaliGemma2-10B (Steiner et al., 2024) | 10B | 40.03 | 29.01 | 45.67 | 32.51 |
| General VLM + Interleaved data | | | | | |
| Phi-3.5-Vision-Instruct (AzureML, 2024) | 4B | 49.05 | 44.37 | 65.00 | 40.45 |
| NVILA-8B (Liu et al., 2024f) | 8B | 61.35 | 57.34 | 75.67 | 48.90 |
| NVILA-15B (Liu et al., 2024f) | 15B | 774.18 | 55.29 | 73.67 | 50.53 |
| NVILA-Lite-8B (Liu et al., 2024f) | 8B | 60.14 | 59.04 | 75.00 | 45.42 |
| NVILA-Lite-15B (Liu et al., 2024f) | 15B | 61.18 | 55.97 | 72.33 | 49.25 |
| mPLUG-Owl3-1B (Ye et al., 2024) | 1B | 37.95 | 44.37 | 50.33 | 34.07 |
| mPLUG-Owl3-2B (Ye et al., 2024) | 2B | 44.19 | 37.54 | 61.00 | 35.84 |
| mPLUG-Owl3-7B (Ye et al., 2024) | 7B | 57.02 | 36.52 | 71.67 | 43.79 |
| InternVL2-1B (Wang et al., 2024e) | 1B | 38.99 | 34.47 | 61.67 | 31.44 |
| InternVL2-2B (Wang et al., 2024e) | 2B | 57.89 | 34.47 | 69.33 | 35.13 |
| InternVL2-4B (Wang et al., 2024e) | 4B | 49.39 | 39.93 | 71.00 | 42.51 |
| InternVL2-8B (Wang et al., 2024e) | 8B | 58.75 | 36.18 | 74.67 | 42.09 |
| InternVL2-26B (Wang et al., 2024e) | 26B | 57.19 | 44.71 | 68.00 | 47.69 |
| InternVL2-40B (Wang et al., 2024e) | 40B | 64.30 | 55.29 | 67.00 | 49.82 |
| InternVL2-76B (Wang et al., 2024e) | 76B | 67.76 | 60.41 | 64.67 | 51.24 |
| InternVL2.5-1B (Gao et al., 2024b) | 1B | 58.58 | 35.49 | 71.00 | 34.49 |
| InternVL2.5-2B (Gao et al., 2024b) | 2B | 63.95 | 42.32 | 72.67 | 40.38 |
| InternVL2.5-4B (Gao et al., 2024b) | 4B | 56.15 | 52.90 | 75.00 | 43.29 |
| InternVL2.5-8B (Gao et al., 2024b) | 8B | 67.94 | 50.85 | 75.33 | 47.69 |
| InternVL2.5-26B (Gao et al., 2024b) | 26B | 73.31 | 57.00 | 72.67 | 50.67 |
| InternVL2.5-38B (Gao et al., 2024b) | 38B | 68.11 | 51.88 | 73.00 | 53.16 |
| InternVL2.5-78B (Gao et al., 2024b) | 38B | 5.22 | 55.63 | 73.67 | 52.24 |
| o1 (Jaech et al., 2024) | - | 75.22 | 49.83 | 74.67 | 51.31 |

Table 30: Evaluation Results for 36 new VLMs in PhysBench Physical Object ✿Property Sub-task (the fifth last column of Table 28).

| Model | Model Size | Size | Location | Depth | Distance | Motion |
|---|---|---|---|---|---|---|
| Image VLM | | | | | | |
| MolmoE-1B Deitke et al. (2024) | 1B | 39.29 | 51.13 | 72.28 | 48.33 | 50.56 |
| MolmoE-7B Deitke et al. (2024) | 7B | 42.86 | 61.65 | 75.44 | 53.33 | 50.56 |
| MolmoE-7B-D Deitke et al. (2024) | 7B | 58.93 | 60.15 | 79.30 | 66.67 | 59.55 |
| MolmoE-72B Deitke et al. (2024) | 72B | 76.79 | 67.29 | 76.49 | 68.33 | 74.16 |
| MiniCPM2 (Yao et al., 2024) | 3B | 50.00 | 45.49 | 39.65 | 46.67 | 59.55 |
| MiniCPM2.5 (Yao et al., 2024) | 4B | 66.07 | 36.84 | 51.23 | 43.33 | 67.42 |
| MiniCPM2.6 (Yao et al., 2024) | 8B | 71.43 | 67.67 | 68.77 | 56.67 | 73.03 |
| Xinyuan-VL (Group, 2024) | 2B | 55.36 | 54.14 | 62.46 | 50.00 | 61.80 |
| Aquila-VL (Gu et al., 2024) | 2B | 69.64 | 60.90 | 64.56 | 61.67 | 59.55 |
| DeepSeek-VL-1B (Lu et al., 2024a) | 1B | 41.07 | 43.23 | 31.93 | 50.00 | 51.69 |
| DeepSeek-VL-7B (Lu et al., 2024a) | 7B | 37.50 | 51.13 | 61.40 | 61.67 | 59.55 |
| PaliGemma2-3B (Steiner et al., 2024) | 3B | 23.21 | 21.05 | 27.02 | 26.67 | 13.48 |
| PaliGemma2-10B (Steiner et al., 2024) | 10B | 48.21 | 45.11 | 36.14 | 41.67 | 50.56 |
| General VLM + Interleaved data | | | | | | |
| Phi-3.5-Vision-Instruct (AzureML, 2024) | 4B | 57.14 | 59.45 | 66.67 | 55.00 | 30.36 |
| NVILA-8B (Liu et al., 2024f) | 8B | 76.79 | 67.35 | 70.88 | 73.33 | 26.83 |
| NVILA-15B (Liu et al., 2024f) | 15B | 76.79 | 69.07 | 80.35 | 75.00 | 27.59 |
| NVILA-Lite-8B (Liu et al., 2024f) | 8B | 73.21 | 61.86 | 71.23 | 60.00 | 27.18 |
| NVILA-Lite-15B (Liu et al., 2024f) | 15B | 62.50 | 67.35 | 73.33 | 70.00 | 26.97 |
| mPLUG-Owl3-1B (Ye et al., 2024) | 1B | 46.43 | 39.52 | 31.93 | 35.00 | 29.46 |
| mPLUG-Owl3-2B (Ye et al., 2024) | 2B | 60.71 | 41.58 | 30.88 | 43.33 | 34.09 |
| mPLUG-Owl3-7B (Ye et al., 2024) | 7B | 69.64 | 60.14 | 69.82 | 58.33 | 37.34 |
| InternVL2-1B (Wang et al., 2024e) | 1B | 44.64 | 53.26 | 35.79 | 50.00 | 27.66 |
| InternVL2-2B (Wang et al., 2024e) | 2B | 78.57 | 64.95 | 73.68 | 63.33 | 34.30 |
| InternVL2-4B (Wang et al., 2024e) | 4B | 69.64 | 59.11 | 75.44 | 55.00 | 28.08 |
| InternVL2-8B (Wang et al., 2024e) | 8B | 75.00 | 67.01 | 77.89 | 56.67 | 31.05 |
| InternVL2-26B (Wang et al., 2024e) | 26B | 78.57 | 64.95 | 73.68 | 63.33 | 34.30 |
| InternVL2-40B (Wang et al., 2024e) | 40B | 80.36 | 64.26 | 77.54 | 73.33 | 40.18 |
| InternVL2-76B (Wang et al., 2024e) | 76B | 80.36 | 67.01 | 82.81 | 76.67 | 41.91 |
| InternVL2.5-1B (Gao et al., 2024b) | 1B | 48.21 | 53.26 | 47.72 | 50.00 | 25.73 |
| InternVL2.5-2B (Gao et al., 2024b) | 2B | 48.21 | 57.73 | 68.07 | 50.00 | 28.08 |
| InternVL2.5-4B (Gao et al., 2024b) | 4B | 75.00 | 62.54 | 72.63 | 55.00 | 34.92 |
| InternVL2.5-8B (Gao et al., 2024b) | 8B | 67.86 | 72.16 | 78.25 | 75.00 | 37.00 |
| InternVL2.5-26B (Gao et al., 2024b) | 26B | 75.00 | 65.64 | 79.65 | 76.67 | 51.66 |
| InternVL2.5-38B (Gao et al., 2024b) | 38B | 85.71 | 68.73 | 75.79 | 80.00 | 64.59 |
| InternVL2.5-78B (Gao et al., 2024b) | 38B | 80.36 | 69.42 | 77.54 | 86.67 | 56.43 |
| o1 (Jaech et al., 2024) | - | 83.93 | 76.63 | 78.25 | 80.00 | 71.51 |

Table 31: Evaluation Results for 36 new VLMs in PhysBench Object ⬈Relationships Sub-task (the forth last column of Table 28).

| Model | Size | Temperature | Viewpoint | Air | Light |
|---|---|---|---|---|---|
| Image VLM | | | | | |
| MolmoE-1B Deitke et al. (2024) | 1B | 50.00 | 30.73 | 26.00 | 30.21 |
| MolmoE-7B Deitke et al. (2024) | 7B | 64.71 | 21.11 | 36.00 | 33.76 |
| MolmoE-7B-D Deitke et al. (2024) | 7B | 79.41 | 42.70 | 86.00 | 26.84 |
| MolmoE-72B Deitke et al. (2024) | 72B | 76.79 | 67.29 | 76.49 | 68.33 |
| MiniCPM2 (Yao et al., 2024) | 3B | 66.18 | 20.83 | 40.00 | 31.12 |
| MiniCPM2.5 (Yao et al., 2024) | 4B | 48.53 | 13.85 | 42.00 | 35.21 |
| MiniCPM2.6 (Yao et al., 2024) | 8B | 67.65 | 19.51 | 42.00 | 36.21 |
| Xinyuan-VL (Group, 2024) | 2B | 64.71 | 33.46 | 48.00 | 32.21 |
| Aquila-VL (Gu et al., 2024) | 2B | 61.76 | 24.79 | 28.00 | 33.48 |
| DeepSeek-VL-1B (Lu et al., 2024a) | 1B | 50.00 | 32.33 | 32.00 | 29.75 |
| DeepSeek-VL-7B (Lu et al., 2024a) | 7B | 58.82 | 25.64 | 28.00 | 32.85 |
| PaliGemma2-3B (Steiner et al., 2024) | 3B | 17.65 | 26.20 | 28.00 | 23.66 |
| PaliGemma2-10B (Steiner et al., 2024) | 10B | 48.53 | 27.14 | 40.00 | 28.57 |
| General VLM + Interleaved data | | | | | |
| Phi-3.5-Vision-Instruct (AzureML, 2024) | 4B | 51.47 | 33.65 | 50.91 | 30.85 |
| NVILA-8B (Liu et al., 2024f) | 8B | 72.06 | 31.29 | 50.91 | 33.58 |
| NVILA-15B (Liu et al., 2024f) | 15B | 83.82 | 38.36 | 52.73 | 35.85 |
| NVILA-Lite-8B (Liu et al., 2024f) | 8B | 76.47 | 31.39 | 25.45 | 35.30 |
| NVILA-Lite-15B (Liu et al., 2024f) | 15B | 82.35 | 38.55 | 61.82 | 34.03 |
| mPLUG-Owl3-1B (Ye et al., 2024) | 1B | 57.35 | 10.84 | 50.91 | 29.48 |
| mPLUG-Owl3-2B (Ye et al., 2024) | 2B | 63.24 | 25.35 | 29.09 | 25.75 |
| mPLUG-Owl3-7B (Ye et al., 2024) | 7B | 76.47 | 37.51 | 78.18 | 30.66 |
| InternVL2-1B (Wang et al., 2024e) | 1B | 45.59 | 18.47 | 18.18 | 25.75 |
| InternVL2-2B (Wang et al., 2024e) | 2B | 50.00 | 28.46 | 21.82 | 31.48 |
| InternVL2-4B (Wang et al., 2024e) | 4B | 52.94 | 28.37 | 54.55 | 31.39 |
| InternVL2-8B (Wang et al., 2024e) | 8B | 73.53 | 17.91 | 36.36 | 32.67 |
| InternVL2-26B (Wang et al., 2024e) | 26B | 76.47 | 40.25 | 29.09 | 33.30 |
| InternVL2-40B (Wang et al., 2024e) | 40B | 94.12 | 37.89 | 58.18 | 29.39 |
| InternVL2-76B (Wang et al., 2024e) | 76B | 91.18 | 39.87 | 52.73 | 32.58 |
| InternVL2.5-1B (Gao et al., 2024b) | 1B | 52.94 | 20.17 | 14.55 | 32.03 |
| InternVL2.5-2B (Gao et al., 2024b) | 2B | 58.82 | 21.58 | 16.36 | 35.58 |
| InternVL2.5-4B (Gao et al., 2024b) | 4B | 67.65 | 28.09 | 29.09 | 32.30 |
| InternVL2.5-8B (Gao et al., 2024b) | 8B | 69.12 | 18.94 | 47.27 | 36.49 |
| InternVL2.5-26B (Gao et al., 2024b) | 26B | 88.24 | 34.31 | 74.55 | 34.58 |
| InternVL2.5-38B (Gao et al., 2024b) | 38B | 88.24 | 41.47 | 78.18 | 32.48 |
| InternVL2.5-78B (Gao et al., 2024b) | 38B | 89.71 | 40.62 | 90.91 | 29.30 |
| o1 (Jaech et al., 2024) | - | 95.59 | 47.31 | 81.82 | 30.30 |

Table 32: Evaluation Results for 36 new VLMs in PhysBench Physical 🌲 Scene Understanding Sub-task (the third last column of Table 28.

| Model | Size | Collision | Throwing | Manipulation | Fluid | Chemistry | Others |
|---|---|---|---|---|---|---|---|
| Image VLM | | | | | | | |
| MolmoE-1B Deitke et al. (2024) | 1B | 32.72 | 33.02 | 43.47 | 44.75 | 71.62 | 57.48 |
| MolmoE-7B Deitke et al. (2024) | 7B | 28.26 | 28.74 | 36.68 | 53.55 | 67.57 | 67.01 |
| MolmoE-7B-D Deitke et al. (2024) | 7B | 38.56 | 39.43 | 39.45 | 50.15 | 68.92 | 72.11 |
| MolmoE-72B Deitke et al. (2024) | 72B | 37.02 | 40.38 | 30.90 | 48.30 | 63.51 | 80.61 |
| MiniCPM2 (Yao et al., 2024) | 3B | 34.82 | 34.20 | 34.92 | 34.88 | 67.57 | 67.69 |
| MiniCPM2.5 (Yao et al., 2024) | 4B | 433.44 | 36.82 | 27.89 | 35.03 | 54.05 | 60.88 |
| MiniCPM2.6 (Yao et al., 2024) | 8B | 35.29 | 45.37 | 38.19 | 43.06 | 64.86 | 76.19 |
| Xinyuan-VL (Group, 2024) | 2B | 35.94 | 39.67 | 38.44 | 39.81 | 50.00 | 64.63 |
| Aquila-VL (Gu et al., 2024) | 2B | 38.10 | 40.14 | 42.21 | 42.75 | 59.46 | 75.17 |
| DeepSeek-VL-1B (Lu et al., 2024a) | 1B | 32.10 | 33.49 | 38.44 | 39.35 | 48.65 | 52.04 |
| DeepSeek-VL-7B (Lu et al., 2024a) | 7B | 31.64 | 35.39 | 35.43 | 37.04 | 54.05 | 61.90 |
| PaliGemma2-3B (Steiner et al., 2024) | 3B | 37.63 | 34.68 | 25.13 | 34.57 | 36.49 | 18.03 |
| PaliGemma2-10B (Steiner et al., 2024) | 10B | 37.17 | 40.62 | 33.92 | 35.80 | 56.76 | 54.08 |
| General VLM + Interleaved data | | | | | | | |
| Phi-3.5-Vision-Instruct (AzureML, 2024) | 4B | 33.94 | 38.72 | 29.19 | 43.67 | 59.46 | 58.58 |
| NVILA-8B (Liu et al., 2024f) | 8B | 38.76 | 45.13 | 31.24 | 43.21 | 70.27 | 69.75 |
| NVILA-15B (Liu et al., 2024f) | 15B | 40.42 | 43.23 | 29.31 | 54.17 | 77.03 | 71.93 |
| NVILA-Lite-8B (Liu et al., 2024f) | 8B | 38.31 | 46.08 | 28.59 | 38.43 | 77.03 | 65.40 |
| NVILA-Lite-15B (Liu et al., 2024f) | 15B | 41.48 | 40.38 | 31.60 | 45.83 | 70.27 | 72.21 |
| mPLUG-Owl3-1B (Ye et al., 2024) | 1B | 29.41 | 32.07 | 23.16 | 42.59 | 51.35 | 45.78 |
| mPLUG-Owl3-2B (Ye et al., 2024) | 2B | 34.69 | 35.63 | 24.85 | 38.58 | 56.76 | 50.14 |
| mPLUG-Owl3-7B (Ye et al., 2024) | 7B | 37.10 | 36.34 | 28.35 | 43.06 | 67.57 | 64.03 |
| InternVL2-1B (Wang et al., 2024e) | 1B | 31.83 | 29.22 | 30.16 | 41.82 | 48.65 | 43.32 |
| InternVL2-2B (Wang et al., 2024e) | 2B | 37.10 | 32.54 | 29.43 | 35.65 | 50.00 | 47.96 |
| InternVL2-4B (Wang et al., 2024e) | 4B | 35.29 | 41.09 | 27.50 | 47.53 | 63.51 | 52.04 |
| InternVL2-8B (Wang et al., 2024e) | 8B | 38.46 | 38.95 | 29.19 | 38.43 | 58.11 | 61.31 |
| InternVL2-26B (Wang et al., 2024e) | 26B | 38.91 | 41.09 | 26.42 | 37.96 | 58.11 | 64.58 |
| InternVL2-40B (Wang et al., 2024e) | 40B | 43.14 | 47.51 | 20.39 | 39.20 | 67.57 | 74.39 |
| InternVL2-76B (Wang et al., 2024e) | 76B | 40.27 | 42.99 | 22.07 | 37.04 | 64.86 | 74.93 |
| InternVL2.5-1B (Gao et al., 2024b) | 1B | 40.27 | 37.05 | 26.42 | 43.36 | 52.70 | 49.86 |
| InternVL2.5-2B (Gao et al., 2024b) | 2B | 33.48 | 40.38 | 28.11 | 42.13 | 62.16 | 55.86 |
| InternVL2.5-4B (Gao et al., 2024b) | 4B | 38.76 | 42.52 | 28.83 | 45.37 | 60.81 | 63.49 |
| InternVL2.5-8B (Gao et al., 2024b) | 8B | 39.52 | 36.82 | 24.25 | 46.14 | 67.57 | 70.03 |
| InternVL2.5-26B (Gao et al., 2024b) | 26B | 38.16 | 43.23 | 25.69 | 38.58 | 67.57 | 78.75 |
| InternVL2.5-38B (Gao et al., 2024b) | 38B | 43.89 | 41.81 | 31.85 | 36.88 | 78.38 | 83.92 |
| InternVL2.5-78B (Gao et al., 2024b) | 38B | 42.08 | 46.32 | 33.53 | 37.35 | 75.68 | 85.29 |
| o1 (Jaech et al., 2024) | - | 48.27 | 41.09 | 37.03 | 43.52 | 71.62 | 89.92 |

Table 33: Evaluation Results for 36 new VLMs in PhysBench Physics-based ⚗Dynamics Sub-task (the second last column of Table 28).

