# OpenReview forum: "PhysBench: Benchmarking and Enhancing Vision-Language Models for Physical World Understanding"
_ICLR.cc/2025/Conference — ICLR 2025 Oral_

### Official Review · Reviewer_eQW6 · 2024-10-28

**Soundness:** 4
**Presentation:** 2
**Contribution:** 4
**Rating:** 8
**Confidence:** 4

**Summary:**

The work proposes PhysBench, which serves as a comprehensive benchmark to evaluate VLM's understanding of physics. PhysBench mainly includes four types of physical questions: Property, Relations, Scene, and Dynamics, and each type has many sub-classes, resulting in 19 different tasks. The work also proposes PhysAgent to improve VLM's ability of physical reasoning.

**Strengths:**

- Compared to all previous works on physical reasoning benchmarks, this dataset is the most comprehensive one to my knowledge. PhysBench includes many different types, which can well evaluate VLM's general physical understanding ability.

- The data sources for videos are various, instead of mainly simulation-based generation. This can further diversify the video data if the dataset has been well-cleaned and filtered.

- Experiments are also extensive, which are conducted across 39 VLMs. The authors also well analyze the reasons why VLMs are still struggling with physical reasoning.

- Further experiments on the transferability of real-world tasks are sound.

**Weaknesses:**

- The article is quite lengthy, so it is really important to improve its logical flow to enhance readability. Using appropriate formatting elements, such as bold text for key points, can help guide readers through the content and make it easier for them to locate the information they are looking for. It is difficult for me to find my questions when reading this paper.

- There is a lack of consistency in the article's formatting. For example, task names are referred to in multiple ways, and similar tables or images are styled differently across various sections, which may confuse readers. Specifically, Table 9a labels the four main types as Property/Spatial/Envir./Phe., whereas other sections refer to them as Property/Relations/Scene/Dynamics. Are they describing the same thing? Also, styles for Tab 16-20 are different, and what is the last column of Tab 20? The labels above Figure 4 are misaligned with the ticks, and extra ticks appear at the bottom. I believe these ticks should likely be placed on the left side instead. It is better to re-organize the whole paper.

- How do questions design for each video? The paper mentions, "For videos annotated with physical principles, we generate physics-related questions using both manual design and GPT-4o, following predefined rules.". But I cannot find further details on it. What are the question templates? How do humans design questions? What is the procedure of question generation by GPT-4o? The authors may comment on more details on this matter.

- How is the human evaluation in Tab 3 (or other possible places) conducted? Also, how do human annotators contribute to the dataset? I can only find the GUI description for this matter. The authors may comment on more details, such as how many people are engaging in each part and so on.

- What is the detailed performance on 19 tasks among PhysAgent and baselines of Fig 9a, since the distribution of the dataset is not balanced.

**Questions:**

- Are there any reasoning steps included in each question?

---

> ### Author Response · Authors · 2024-11-22
> **Official Response to Reviewer eQW6 (1/2)**
>
> We sincerely thank you for the valuable comments and we will explain your concern as follows.
>
> > **Q1:** The article is quite lengthy, so it is really important to improve its logical flow to enhance readability. Using appropriate formatting elements, such as bold text for key points, can help guide readers through the content and make it easier for them to locate the information they are looking for. It is difficult for me to find my questions when reading this paper.
> >
>
> **A1**: Thank you for your feedback! We are dedicated to making our paper more concise and have added further clarifications to improve its readability. Additionally, we have included a table of contents for the appendix on *page 24* to facilitate your review. We have already implemented some revisions to the original manuscript and will continue to refine it further to enhance its overall clarity and coherence.
>
> > **Q2:** There is a lack of consistency in the article's formatting. For example, task names are referred to in multiple ways, and similar tables or images are styled differently across various sections, which may confuse readers. Specifically, Table 9a labels the four main types as Property/Spatial/Envir./Phe., whereas other sections refer to them as Property/Relations/Scene/Dynamics. Are they describing the same thing? Also, styles for Tab 16-20 are different, and what is the last column of Tab 20? The labels above Figure 4 are misaligned with the ticks, and extra ticks appear at the bottom. I believe these ticks should likely be placed on the left side instead. It is better to re-organize the whole paper.
> >
>
> **A2**: Thank for your careful suggestions! we have meticulously revised the manuscript, with all modifications highlighted in blue. We remain committed to further enhancing the readability of our work in future iterations. Once again, we deeply appreciate your thorough and insightful feedback.
>
> > **Q3:** How do questions design for each video? The paper mentions, "For videos annotated with physical principles, we generate physics-related questions using both manual design and GPT-4o, following predefined rules.". But I cannot find further details on it. What are the question templates? How do humans design questions? What is the procedure of question generation by GPT-4o? The authors may comment on more details on this matter.
> >
>
> **A3**: Thank you for your query! We will address your concerns point by point, and additional details are provided in the revised PDF.
>
> **(3.1)** To ensure data diversity, we collected data through three approaches: web searches, simulations, and real-world captures. The data collection team ensured that the collected topics were aligned with real-world understanding, guided by the objectives outlined in *Appendix B*. Additionally, annotators provided relevant physical descriptions, such as the direction of shadow movement or causes of events. Captions for the videos were supplemented with sample outputs generated by GPT; however, **these outputs were only used as references. It is important to emphasize that all data underwent rigorous human annotation and review**. Specific details can be found in *Appendix B.9* and *Figure 13*.
>
> **(3.2)** All questions were manually designed and subsequently verified and refined by a separate team. GPT was only used as a reference to enhance the diversity of the questions. Given the **complexity of collecting such data**, we **employed LLM as heuristic strategy to broaden the search scope** and implemented a **five-step annotation procedure**, which required **a total of 4,000 hours**.
>
> For more details on the data annotation process, please refer to *Appendix E.8*, and for additional templates, see *Appendix E.1*. Thank you again for your insightful comments.
>
> > **Q4:** How is the human evaluation in Tab 3 (or other possible places) conducted? Also, how do human annotators contribute to the dataset? I can only find the GUI description for this matter. The authors may comment on more details, such as how many people are engaging in each part and so on.
> >
>
> **A4:** Thank you for your inquiry.
>
> **(4.1)** The human evaluation process involved **12 graduate students from STEM fields**, with the final results reflecting the average performance of each participant. Detailed information can be found in *Appendix E.8*.
>
> **(4.2)** Overall, we divided the annotators into **3 groups, each consisting of 6 individuals**, with each group responsible for different parts of the annotation process. Given the **complexity of collecting such data**—where expressing specific properties often requires multiple images—we implemented a **five-step annotation procedure**, which required **a total of 4,000 hours**. Additional details can be found in *Appendix B.9* and *Figure 13* of the paper.

---

> ### Author Response · Authors · 2024-11-22
> **Official Response to Reviewer eQW6 (2/2)**
>
> > **Q5:** What is the detailed performance on 19 tasks among PhysAgent and baselines of Fig 9a, since the distribution of the dataset is not balanced.
> >
>
> **A5:** Thank you for your thoughtful suggestions. The details for *Figure 9(a)* are as follows, with additional explanations provided in *Appendix F.8 of the revised PDF*. In the first table, the first five rows correspond to *Property*, while the last four rows correspond to *Relationships*. In the second table, the first four rows correspond to *Scene*, while the last six rows correspond to *Dynamics*.
>
> |  | **number** | **mass** | **color** | **attribute** | **size** | **location** | **depth** | **distance** | **movement** |
> | --- | --- | --- | --- | --- | --- | --- | --- | --- | --- |
> | ContPhy | 62.22 | 47.10 | 62.67 | 48.33 | 67.86 | 59.79 | 67.72 | 70.00 | 46.06 |
> | Phi-3V | 43.15 | 37.88 | 63.00 | 40.81 | 66.07 | 53.61 | 74.39 | 48.33 | 26.69 |
> | + CoT | 44.89 | 36.18 | 60.33 | 39.11 | 39.29 | 48.45 | 60.35 | 50.00 | 26.42 |
> | + Desp-CoT | 49.22 | 29.01 | 67.00 | 37.62 | 66.07 | 53.61 | 62.46 | 55.00 | 24.69 |
> | + PLR | 43.67 | 26.62 | 59.67 | 34.63 | 58.93 | 41.92 | 30.53 | 43.33 | 30.43 |
> | + PhysAgent | 45.10 | 40.27 | 64.67 | 40.53 | 67.86 | 54.98 | 74.74 | 66.67 | 26.76 |
> | GPT-4o | 61.18 | 54.27 | 71.00 | 52.52 | 85.71 | 70.45 | 73.33 | 83.33 | 60.58 |
> | + CoT | 63.43 | 56.66 | 71.67 | 54.22 | 82.14 | 74.23 | 75.44 | 83.33 | 67.91 |
> | + Desp-CoT | 65.16 | 55.29 | 71.00 | 52.16 | 80.36 | 68.38 | 70.18 | 71.67 | 61.20 |
> | + PLR | 68.63 | 41.98 | 66.00 | 47.27 | 66.07 | 57.73 | 59.30 | 53.33 | 43.91 |
> | + PhysAgent | 67.07 | 54.27 | 73.33 | 55.86 | 87.50 | 78.35 | 74.04 | 95.00 | 71.99 |
>
> |  | **temperature** | **camera** | **gas** | **light** | **collision** | **throwing** | **manipulation** | **fluid** | **chemistry** | **others** |
> | --- | --- | --- | --- | --- | --- | --- | --- | --- | --- | --- |
> | ContPhy | 73.53 | 33.08 | 74.55 | 36.21 | 39.67 | 42.52 | 36.07 | 50.15 | 40.54 | 58.04 |
> | Phi-3V | 52.94 | 38.93 | 65.45 | 29.48 | 33.18 | 35.39 | 28.95 | 35.34 | 59.46 | 56.68 |
> | + CoT | 55.88 | 30.73 | 50.91 | 26.93 | 31.83 | 31.35 | 31.72 | 40.43 | 47.30 | 50.41 |
> | + Desp-CoT | 58.82 | 40.15 | 67.27 | 30.30 | 34.99 | 32.07 | 28.47 | 31.94 | 56.76 | 50.68 |
> | + PLR | 67.65 | 39.96 | 67.27 | 30.94 | 32.13 | 28.03 | 25.69 | 35.03 | 56.76 | 51.50 |
> | + PhysAgent | 72.06 | 38.93 | 70.91 | 34.58 | 36.05 | 35.39 | 29.55 | 35.49 | 67.57 | 65.67 |
> | GPT-4o | 91.18 | 22.05 | 83.64 | 32.67 | 43.74 | 46.32 | 35.22 | 39.20 | 62.16 | 86.92 |
> | + CoT | 91.18 | 30.73 | 74.55 | 36.49 | 45.40 | 46.32 | 35.71 | 41.67 | 64.86 | 85.56 |
> | + Desp-CoT | 85.29 | 29.41 | 80.00 | 39.95 | 44.19 | 42.28 | 30.52 | 43.83 | 70.27 | 85.83 |
> | + PLR | 88.24 | 30.73 | 89.09 | 34.94 | 38.01 | 34.68 | 24.85 | 37.50 | 66.22 | 79.56 |
> | + PhysAgent | 95.59 | 41.19 | 85.45 | 43.49 | 45.25 | 47.27 | 52.11 | 44.91 | 67.57 | 87.47 |
>
>
> > **Q6:** Are there any reasoning steps included in each question?
> >
>
> **A6**: Each question does not explicitly include reasoning steps. However, answering the questions requires reasoning capabilities from the VLM.
>
> **(6.1)** To ensure **ease of testing**, PhysBench adopts a multiple-choice format for responses. However, despite its relatively straightforward format, **PhysBench remains highly challenging**, with even the strongest GPT-4o model achieving **less than 50% accuracy**.
>
> **(6.2)** Considering the challenges of using LLMs for evaluation, such as hallucinations and knowledge limitations, assessing open-ended formats is particularly difficult [1,2]. Additionally, automatically measuring the quality of reasoning processes is also challenging [3]. In our experiments (*Section 3.4*), we found that even with CoT prompting, GPT-4o exhibited **an error rate of 3%** in answer extraction (as discussed in *Appendix J.1*).
>
> **(6.3)** However, this does not suggest that our work disregards the importance of reasoning processes. In *Section 3.4,* **we manually reviewed the reasoning steps** of three representative models, categorized the error types, and observed that while reasoning errors are significant, the primary issues stem from a lack of relevant physical world knowledge and perception errors. Consequently, we further propose PhysBench to enhance physical world perception capabilities.
>
> We greatly value your suggestions and are committed to introducing more complex tasks and incorporating automated reasoning step validation in future work.
>
> *[1] MM-Vet: Evaluating Large Multimodal Models for Integrated Capabilities*
>
> *[2] Fine-tuning Multimodal LLMs to Follow Zero-shot Demonstrative Instructions*
>
> *[3] MathVista: Evaluating Math Reasoning in Visual Contexts*
>
> Thank you for your valuable comments. We have revised our manuscript based on your suggestions and are committed to further refining the paper to improve its readability. We sincerely appreciate your help once again!

---

> > ### Comment · Reviewer_eQW6 · 2024-11-22
> > **post-rebuttal**
> >
> > I would like to thank the authors for their detailed comments. All of my concerns have been solved. I really appreciate the contribution this paper offers. The benchmark is comprehensive and unique to other VLM benchmarks, as demonstrated by Fig 4. Thus, I would like to raise my score and recommend acceptance.

---

> > > ### Author Response · Authors · 2024-11-22
> > > **Official Comment by Authors**
> > >
> > > Thank you for increasing the score! Your valuable suggestions greatly contribute to the quality of our manuscript. Thank you again for your precious time and valuable suggestions!

---

### Official Review · Reviewer_xxa5 · 2024-10-29

**Soundness:** 3
**Presentation:** 3
**Contribution:** 3
**Rating:** 8
**Confidence:** 3

**Summary:**

This paper proposes a large-scale benchmark to evaluate VLMs' capability to understand the physical world. The benchmark covers diverse domains such as daily life images and videos, robot manipulation, and autonomous driving. The benchmark takes thousands of hours for data annotation and multi-round quality review, ensuring quality. The extensive experiments test 39 representative VLMs and reveal the limitations of such models' abilities to understand the physical world. To enhance VLMs' physical world understanding abilities, the authors propose a unified approach named PhysAgent to leverage visual foundation models into a chain-of-thought protocol. Since there are existing works that aim at evaluating VLMs' ability in spatial reasoning and physical common senses, this paper claims that the benchmark can enhance the VLMs' abilities on embodied tasks. However, the experiments on embodied tasks are very limited and unfair in demonstrating the role of this benchmark in the era of VLA models.
In summary, the paper proposes an important benchmark and proposes a large-scale, diverse, and high-quality benchmark to evaluate VLMs' abilities to understand the physical world. The authors also propose a new framework to enhance VLMs' performance on this benchmark.

**Strengths:**

1. Dataset Scale and Quality: The scale and quality of the dataset are promising. The paper provides thorough details regarding data annotation and quality control processes, which enhance the benchmark's credibility and reliability.
2. Scalability and Error Analysis: The paper offers valuable insights by analyzing the scalability of the model, data, and framework. Notably, it presents an in-depth examination of error types encountered by commercial VLMs on the benchmark, which is critical for advancing the development of VLMs.
3. Impact of Fine-tuning: The results demonstrate that fine-tuning on a subset of PhysBench significantly improves performance on embodied tasks compared to the zero-shot performance of the PhysAgent framework, highlighting the value of task-specific adaptation.

**Weaknesses:**

1. Discussion of Related Work: The paper lacks a comprehensive discussion of related works. The proposed benchmark aims to evaluate and enhance the ability of VLMs to understand the physical world at both the object and scene levels. However, it is crucial to emphasize that understanding 3D spatial relationships at the scene level is a fundamental capability for VLMs. To strengthen the discussion, the authors should compare this work with other 3D spatial VQA benchmarks such as SQA3D [1], OpenEQA [2], EmbodiedScan [3], MMScan [4], and MSQA [5]. These benchmarks offer direct evaluations of VLMs’ understanding of 3D scenes through interleaved 2D or 3D representations, providing valuable context and highlighting gaps or overlaps with the proposed approach.
2. Comparison of Embodied Tasks: The comparison between zero-shot models (such as existing VLMs and PhysAgent) and fine-tuned models trained on PhysBench raises concerns about fairness. PhysBench, in its current form, is relatively simplistic and may not provide a robust or balanced evaluation. To ensure fair comparisons, the authors should either (1) incorporate more challenging tasks or datasets in PhysBench to match the capabilities of fine-tuned models, or (2) highlight the limitations of comparing zero-shot performance with fine-tuned results to avoid misleading conclusions.

[1] SQA3D: Situated Question Answering in 3D Scenes

[2] OpenEQA: Embodied Question Answering in the Era of Foundation Models

[3] EmbodiedScan: A Holistic Multi-Modal 3D Perception Suite Towards Embodied AI

[4] MMScan: A Multi-Modal 3D Scene Dataset with Hierarchical Grounded Language Annotations

[5] Multi-modal Situated Reasoning in 3D Scenes

**Questions:**

I appreciate the authors' efforts in developing a benchmark for physical world understanding and evaluating its impact on embodied agents. Recently, several open-source VLA models have emerged, such as OpenVLA [1], Embodied-COT [2], and Octo [3]. Did the authors consider comparing the fine-tuned VLMs with other VLA models on the proposed manipulation tasks? Additionally, is there a way to assess the benchmark’s role in advancing the development of VLA models? Such comparisons and evaluations would further highlight the benchmark’s contribution to the field.

[1] OpenVLA: An Open-Source Vision-Language-Action Model

[2] Robotic Control via Embodied Chain-of-Thought Reasoning

[3] Octo: An Open-Source Generalist Robot Policy

**Details Of Ethics Concerns:**

The paper has discussed data privacy in the appendix.

---

> ### Author Response · Authors · 2024-11-22
> **Official Response to Reviewer xxa5 (1/3)**
>
> We sincerely thank you for your comprehensive comments and constructive advice. We will explain your concern as follows.
>
> > **Q1:** Discussion of Related Work: The paper lacks a comprehensive discussion of related works. The proposed benchmark aims to evaluate and enhance the ability of VLMs to understand the physical world at both the object and scene levels. However, it is crucial to emphasize that understanding 3D spatial relationships at the scene level is a fundamental capability for VLMs. To strengthen the discussion, the authors should compare this work with other 3D spatial VQA benchmarks such as SQA3D [1], OpenEQA [2], EmbodiedScan [3], MMScan [4], and MSQA [5]. These benchmarks offer direct evaluations of VLMs’ understanding of 3D scenes through interleaved 2D or 3D representations, providing valuable context and highlighting gaps or overlaps with the proposed approach.
> >
>
> **A1**: Thank you for your valuable suggestions! We appreciate your recognition of the importance of understanding spatial relationships, which is also a emphasis in our paper.
>
> **(2.1)** As noted in **L117 and L136 in related work section**, we previously discussed several **Spatial VQA datasets**, which focus on geometric relationships and represent only a subset of physical world understanding. **Our work emphasizes a more comprehensive evaluation of the physical world perception capabilities of VLMs**, encompassing four major task categories: *Physical Object Properties*, *Physical Object Relationships*, *Physical Scene Understanding*, and *Physics-based Dynamics*, while spatial reasoning can be viewed as a sub-category of our benchmark. However, the works you have mentioned are also highly valuable to us. **We have further discussed in detail and added the appropriate citations for these works you mentioned in Appendix G. Thanks for your suggestions!**
>
> **(2.2)** The spatial components in our dataset also **differ in focus** from those in Spatial VQA datasets. While these benchmarks typically consider inputs as 3D point cloud scenes (interleaved 2D images from multiple views), our dataset employs interleaved images to represent **physical world dynamics**, such as viewpoint rotation and the progression of physical phenomena.
>
> **(2.3)** **Our data and methods also enhance performance on Spatial VQA**. EmbSpatial is a comprehensive benchmark evaluating spatial reasoning in embodied environments, with source images derived from 3D scenes. Fine-tuning with PhysBench and zero-shot usage of PhysAgent both improve performance on EmbSpatial, increasing scores by 19.75% and 11.39%, respectively.
>
> *Experimental results and details are provided in Appendix F.7 of the PDF.*
>
> |  | **ContPhy-Property** | **ContPhy-Dynamics** | **Physion++** | **EmbSpatial** |
> | --- | --- | --- | --- | --- |
> | Phi-3V | 49.25 | 43.25 | 68.80 | 55.91 |
> | Phi-3V + finetune | 64.50 | 62.75 | 82.20 | 66.95 |
> | Phi-3V + PhysAgent | 52.00 | 44.75 | 78.40 | 62.28 |

---

> ### Author Response · Authors · 2024-11-22
> **Official Response to Reviewer xxa5 (2/3)**
>
> > Q2: Comparison of Embodied Tasks: The comparison between zero-shot models (such as existing VLMs and PhysAgent) and fine-tuned models trained on PhysBench raises concerns about fairness. PhysBench, in its current form, is relatively simplistic and may not provide a robust or balanced evaluation. To ensure fair comparisons, the authors should either (1) incorporate more challenging tasks or datasets in PhysBench to match the capabilities of fine-tuned models, or (2) highlight the limitations of comparing zero-shot performance with fine-tuned results to avoid misleading conclusions.
> >
>
> Thank you for your constructive suggestions. We believe there may be some misunderstandings here. The comparisons in the embodied evaluation are fair, and PhysBench demonstrates robust performance.
>
> **(2.1)** **PhysAgent** is a novel agent-based method that operates directly on VLMs **without any fine-tuning**. As stated in our paper, PhysAgent is used in a zero-shot manner throughout the experiments, with no fine-tuning involved.
>
> **(2.2)** First, it is important to clarify that the embodied tasks are not part of PhysBench and are unrelated to it. The embodied tasks presented in *Figure 9(c)* represent the **manipulation success rates in MuJoCo** simulation scenarios. The experimental settings, including the number of trials and methods, align with those used in prior simulation studies. The baseline in these experiments is MOKA. ***"+PhysAgent"*** indicates that MOKA's VLM inference is replaced by PhysAgent's inference in a zero-shot setting. ***"+Fine-tune"*** refers to VLMs fine-tuned on PhysBench being applied to MOKA, rather than fine-tuning being directly conducted on the embodied tasks themselves.
>
> **(2.3)** Thank you for your suggestion to *“incorporate more challenging tasks or datasets in PhysBench to match the capabilities of fine-tuned models”*. As noted in *Answer 1*, we have considered testing our methods and data on additional tasks. This has resulted in consistent performance improvements across other benchmarks related to physical-world understanding. To illustrate the effectiveness of our dataset and methods, we selected three benchmarks relevant to physical world understanding: *ContPhy*, which focuses on fluid dynamics comprehension; *Physion++*, emphasizing collision prediction; and *EmbSpatial*, evaluating spatial reasoning in embodied environments. Images in these benchmarks are not included in PhysBench, so they can be viewed as out-of-distribution images for evaluation.
>
> ***+Finetune*** indicates that we fine-tuned Phi-3V using the PhysBench dataset, while ***+PhysAgent*** refers to the zero-shot application of our proposed agent-based method to assist the VLM, without any fine-tuning. Experimental results demonstrate consistent performance improvements across all benchmarks under both settings. Notably, on Physion++, we observe significant gains of 19.50% and 9.6%, with fine-tuning achieving the most pronounced improvements. These findings highlight the **effectiveness of our dataset and approach in enhancing the physical world understanding capabilities of VLMs.**
>
> *Experimental results and details are provided in Appendix F.7 of the PDF.*
>
> |  | **ContPhy-Property** | **ContPhy-Dynamics** | **Physion++** | **EmbSpatial** |
> | --- | --- | --- | --- | --- |
> | Phi-3V | 49.25 | 43.25 | 68.80 | 55.91 |
> | Phi-3V + finetune | 64.50 | 62.75 | 82.20 | 66.95 |
> | Phi-3V + PhysAgent | 52.00 | 44.75 | 78.40 | 62.28 |
>
>
> **(2.4)** For ease of testing, PhysBench adopts a relatively simple format. However, **it remains highly challenging**, as even the strongest GPT-4o model achieves a performance of **less than 50%**. We have **acknowledged this limitation** in *Appendix J.1*, as space constraints prevented us from expanding on this in the main text. Moreover, the data and methods we propose **contribute to enhancing VLMs' ability to understand the physical world and support embodied tasks**, as demonstrated in the experiments outlined in *Figure 9* and *Answer 3*.
>
> If you have any additional concerns, please do not hesitate to let us know. We are more than willing to address them and sincerely appreciate your valuable feedback and support.

---

> ### Author Response · Authors · 2024-11-22
> **Official Response to Reviewer xxa5 (3/3)**
>
> > **Q3:** Recently, several open-source VLA models have emerged, such as OpenVLA [1], Embodied-COT [2], and Octo [3]. Did the authors consider comparing the fine-tuned VLMs with other VLA models on the proposed manipulation tasks? Additionally, is there a way to assess the benchmark’s role in advancing the development of VLA models? Such comparisons and evaluations would further highlight the benchmark’s contribution to the field.
> >
>
> **A3**: Thanks for your valuable comment.
>
> **(3.1)** **The scope of this paper differs from that of VLA models**. Currently, there are two main approaches for applying VLMs to robotic manipulation: **(a)** directly outputting actions (e.g., OpenVLA) and **(b)** using VLMs as agents (e.g., MOKA [1]). Our embodied experiments are conducted within the MOKA framework. Unfortunately, we cannot validate the effectiveness of our data or methods through simple fine-tuning or in-context approaches on models following approach (a). This is because our dataset is in a question-answering format, which is more suitable for approach (b), whereas fine-tuning data for approach (a) follows a question-action format. A more detailed discussion is provided in *Appendix G*.
>
> **(3.2)** **MOKA-based methods can also perform robotic manipulation similar to VLAs**. Both OpenVLA and Octo require fine-tuning to adapt output actions to specific robotic arms, a process that heavily depends on demonstration data. In contrast, MOKA requires less training but suffers from weaker perception capabilities in real-world settings, resulting in performance limitations [1].
>
> We report the **manipulation success rate** in comparison with VLAs, demonstrating that MOKA-based methods exhibit greater generalizability, as MOKA even outperforms Octo and OpenVLA . Additionally, using PhyBench for fine-tuning or applying PhysAgent in a zero-shot manner further improves manipulation performance, providing additional evidence for the effectiveness of our dataset and approach.
>
> |  | **Affordance** | **Force** | **Color** | **Location** | **Tool** |
> | --- | --- | --- | --- | --- | --- |
> | MOKA | 56% | 20% | 70% | 66% | 36% |
> | MOKA + finetune using PhysBench | 86% | 70% | 76% | 78% | 72% |
> | MOKA + PhysAgent | 74% | 56% | 74% | 68% | 40% |
> | Octo + finetune using emobodied tasks | 24% | 12% | 24% | 16% | 10% |
> | OpenVLA + finetune using emobodied tasks | 54% | 28% | 64% | 32% | 28% |
>
> *[1] MOKA: Open-World Robotic Manipulation through Mark-Based Visual Prompting*
>
> Thank you once again for your recognition and constructive suggestions, which have been instrumental in enhancing the quality of our research!

---

> ### Author Response · Authors · 2024-11-24
> **Reminder: Discussion Phase Closing Soon**
>
> We sincerely appreciate your invaluable feedback, which has significantly contributed to the improvement of our work. In response, we have incorporated specific experiments to address your concerns. As the rebuttal period draws to a close, we kindly request your prompt review of our responses to ensure that we have thoroughly addressed all your comments. If our responses have satisfactorily resolved your concerns, we would be deeply grateful for a reconsideration of our score. Should any issues remain, we are more than willing to provide further clarifications to address them comprehensively.

---

> ### Comment · Reviewer_xxa5 · 2024-11-25
> **Post-rebuttal**
>
> Thank you for the thoughtful rebuttal, which addressed many of my concerns. However, I still have some questions regarding the experiments. The newly added robot manipulation experiments effectively demonstrate the benefits of integrating PhysBench with MOKA. That said, a more compelling evaluation would involve fine-tuning OpenVLA or Octo on both embodied tasks and PhysBench. This approach would provide direct evidence of the dataset's ability to enhance other VLA models.

---

> ### Author Response · Authors · 2024-11-25
> **Official Response to Reviewer xxa5**
>
> We wish to clarify that **the scope of our paper** is to benchmark vision-language models (VLMs) for physical world understanding. To explore the potential application of VLMs to embodied tasks, we additionally employed a MOKA-based pipeline that is compatible with VLM outputs for robotic manipulation.
>
> We agree that vision-language-action models (VLAs) also have the great potential for embodied tasks such as robotic manipulation, and **we have provided relevant baselines in our paper (as well as in *Answer 3)***. However, **VLMs and VLAs are essentially different.** Specifically, they utilize different decoders and tokenizers—VLAs decode specialized action tokens rather than natural language as in VLMs—and VLAs are trained using robotic data (action in digital number) instead of images and text, which are used for training VLMs. Our proposed PhysBench is a image-text question-answering (QA) based benchmark rather than a robotic dataset; therefore, **it is not suitable for training or fine-tuning VLAs as we do not have action labels in this dataset**. Moreover, as the main focus of our paper is to benchmark VLMs for physical world understanding, the **discussion about VLAs is out of the scope of this paper**.
>
> To summarize, **the technical contributions of our paper focus on physical understanding**. While the development of a novel dataset for VLM training holds significant potential, it is beyond the scope of this paper. In this work, we introduce a comprehensive benchmark aimed at evaluating the physical world understanding capabilities of VLMs, an area that has been under-explored in existing literature. Furthermore, we identify potential causes of performance limitations and propose a novel framework to enhance physical understanding capabilities based on our analysis. Our results demonstrate that when equipped with the MOKA pipeline, both VLMs fine-tuned on PhysBench and the PhysAgent achieve improved performance in embodied tasks such as robotic manipulation. Feel free to let us know if you have more questions about the PhysBench and PhysAgent. We strive to address any outstanding concerns of our paper. Thank you!

---

> > ### Comment · Reviewer_xxa5 · 2024-11-25
> >
> > How can you evaluate the performance of OpenVLA + fine-tuning on embodied tasks on the table without access to action labels? In my view, if you claim the benchmark is beneficial for VLAs, the most direct way is to fine-tune VLAs based on your data and VLA data and evaluate them on manipulation tasks.

---

> > > ### Author Response · Authors · 2024-11-25
> > > **Official Response to Reviewer xxa5**
> > >
> > > Dear Reviewer xxa5,
> > >
> > > Thank you for your additional comments!
> > >
> > > First, OpenVLA/Octo requires fine-tuning using additional action labels. We follow the setting of OpenVLA and fine-tune all model parameters using 25 samples (5 per task) of simulated robotics tasks, where each task contains 7-dimensional action labels. These target simulation robotic tasks are not part of PhysBench, instead, are created to address your Question 3 regarding embodied tasks. **Our method, strictly following MOKA, does not have an action decoder nor require fine-tuning on these robotic tasks. Our method, combined with MOKA, is only fine-tuned on the PhysBench dataset.** Further details can be found in *Answer 3* and *Appendix F.5, F.8*.
> > >
> > > Second, our paper **does not focus on demonstrating benefits of our benchmark to VLA**. Instead, our claim is ***"help the deployment of embodied agents" (L028)***, which underscores the importance of physical understanding for VLMs. As shown in *Figure 9(c)* and elaborated in *Answer 3*, our experiments demonstrate that our approach and dataset are effective in a MOKA-based pipeline. This pipeline is designed to integrate seamlessly with VLM outputs for robotic manipulation tasks performed by embodied agents. MOKA converts motion generation into a sequence of visual question-answering tasks that can be addressed by the VLM, thereby **eliminating the need for action labels in the process**.
> > >
> > > Lastly, since our primary focus is on benchmarking VLMs for physical world understanding, we believe a focus on **VLAs falls outside the scope of this work**. The VLA experiments mentioned in *Answer 3* were included to address questions raised in the rebuttal process and are not part of the main discussion in our paper. As noted in our previous response, VLMs and VLAs have distinct characteristics, and our dataset **is not designed for training or fine-tuning VLAs.**
> > >
> > > Thank you again for your time and thoughtful consideration. We deeply appreciate your help to improve the submission. If you have any further questions, please don’t hesitate to let us know.
> > >
> > > Best regards,
> > > The Authors

---

> ### Comment · Reviewer_xxa5 · 2024-11-25
>
> > We follow the OpenVLA setup, fine-tuning all model parameters on 25 samples (5 per task) of simulated robotics tasks, with each task providing 7-dimensional action labels.
>
> Does this imply you trained OpenVLA on MOKA's action tasks? If so, 1) co-training PhysBench with action data or 2) pertaining on PhysBench data + fine-tuning on action data would be straightforward. For instance, EmbodiedCOT [1] demonstrates this by training OpenVLA with interleaved QA and action data. Co-training your benchmark with action data and comparing its performance to action-only training would provide a more robust and compelling evaluation. Relying solely on comparisons with zero-shot agent frameworks offers limited value to the embodied AI community. If the authors encounter challenges in implementing such experiments or obtain negative results, it may be worth reconsidering the claim "help the deployment of embodied agents". However, I do not believe this poses any significant obstacle.
>
> [1] *Robotic Control via Embodied Chain-of-Thought Reasoning*

---

> ### Author Response · Authors · 2024-11-25
> **Official Response to Reviewer xxa5**
>
> Thank you for your additional question! We want to clarify that OpenVLA is not our method, instead is a strong baseline we compare to. For MOKA, it doesn’t involve action and is a pure affordance prediction framework.
>
> We show that MOKA (that is trained on our benchmark and action agnostic) has already outperformed a OpenVLA (that is fine-tuned on target robotic tasks and action aware). We believe fine-tuning MOKA using a combination of our benchmark and target robotic data (although this requires significant architecture change) will further improve the performance.
>
> We appreciate your suggestion of EmbodiedCOT. We did not notice this very recent paper and per [ICLR code](https://iclr.cc/Conferences/2025/FAQ), it's not necessary to compare to very recent works. We can certainly add a discussion of this paper. However, due to time limit, training a OpenVLA within two days before the conclusion of the rebuttal period is almost impossible given our compute resources and data volume required by OpenVLA in the EmbodiedCOT setting. Once again, the primary goal of our benchmark is to investigate VLM’s physical world understanding capabilities. We included the experiments of MOKA as a way to show our benchmark could potentially help the development of embodied agents. We can remove this claim from the paper per your suggestion, although we believe our experiments with MOKA have already justified this claim.
>
> If you have any further comments that can help us improve the paper, please let us know! Thank you so much for all these valuable discussions.

---

> > ### Comment · Reviewer_xxa5 · 2024-11-25
> >
> > > We show that MOKA (that is trained on our benchmark and action agnostic) has already outperformed a OpenVLA (that is fine-tuned on target robotic tasks and action aware). We believe fine-tuning MOKA using a combination of our benchmark and target robotic data (although this requires significant architecture change) will further improve the performance.
> >
> > My initial suggestion is to directly integrate PhysBench into OpenVLA, which has been fine-tuned for target robotic tasks and is action-aware, rather than conducting a comparison with EmbodiedCOT.
> >
> > >  However, due to time limit, training a OpenVLA within two days before the conclusion of the rebuttal period is almost impossible given our compute resources and data volume required by OpenVLA in the EmbodiedCOT setting.
> >
> > It is unfortunate to remove such an important experiment. However, I believe it is straightforward and requires minimal effort.
> >
> > > We can remove this claim from the paper per your suggestion, although we believe our experiments with MOKA have already justified this claim.
> >
> > The authors have addressed several of my concerns and committed to making their claim more concise based on the existing experimental evidence. As a result, I have decided to slightly increase my score to 6.

---

> ### Author Response · Authors · 2024-11-27
>
> Thank you for your constructive comments!
>
> As the rebuttal period has been extended, we are working on additional experiments to address all your concerns. We will post the response as soon as possible. Thank you!

---

> ### Author Response · Authors · 2024-12-02
> **Official Response to Reviewer xxa5**
>
> Dear Reviewer,
>
> We would like to express our sincere gratitude for the time and effort you have devoted to reviewing our work and providing valuable feedback. Due to the scale of the dataset and time constraints, we selected randomly 500 items from PhysBench, covering various attributes such as size, location, distance, color, mass, and number. Given the limited size of the demonstration action dataset, we supplemented it with a portion of the TableTop Frank Panda dataset from Open X-Embodiment, specifically the NYU Franka Play Dataset, along with data from Austin Sailor, CMU Play Fusion and Austin Sirius, focusing on data relevant to downstream tasks, in order to balance the dataset scale.
>
> *OpenVLA+PhysBench*: We initially performed joint fine-tuning using the Frank Panda dataset from Open X-Embodiment and a subset of the PhysBench data, followed by fine-tuning on the target robotic data.
>
> *OpenVLA baseline*: We first fine-tuned using the Frank Panda dataset from Open X-Embodiment, then proceeded with fine-tuning on the target robotic data.
>
> The experimental results indicate that incorporating PhysBench during the initial stage of fine-tuning enhances performance, particularly in the Tool category, with an improvement of nearly 90%.
>
> |  | Affordance | Force | Color | Location | Tool |
> | --- | --- | --- | --- | --- | --- |
> | OpenVLA baseline | 0.58±0.04 | 0.26±0.05 | 0.62±0.05 | 0.32±0.04 | 0.22±0.04 |
> | OpenVLA+PhysBench | 0.60±0.08 | 0.32±0.01 | 0.68±0.01 | 0.38±0.04 | 0.42±0.08 |
>
> Since the current version of the PDF is no longer editable, we will include the additional experiments and corresponding details in the next version.
>
> We welcome any further discussions and value the opportunity to continually improve our work.
>
> Best regards,
>
> Authors

---

> > ### Comment · Reviewer_xxa5 · 2024-12-03
> > **Final decision**
> >
> > Thank you for conducting additional experiments! The results clearly demonstrate that incorporating knowledge from PhysBench significantly enhances OpenVLA's performance in the target robotic task. This conclusion presents a promising advancement for the embodied AI community. With all my concerns now addressed, I have decided to raise my score to 8.

---

> > > ### Author Response · Authors · 2024-12-03
> > > **Official Comment by Authors**
> > >
> > > Dear Reviewer,
> > >
> > > We sincerely appreciate your support for our work and the thoughtful suggestions you have provided. We are truly grateful for the opportunity to engage with you regarding our research. Your valuable insights have greatly contributed to enhancing the quality of our work.
> > >
> > > Best regards,
> > > The Authors

---

### Official Review · Reviewer_52XN · 2024-11-03

**Soundness:** 3
**Presentation:** 2
**Contribution:** 3
**Rating:** 8
**Confidence:** 4

**Summary:**

The authors collect a human-annotated dataset to test physical reasoning in MLMs. They demonstrate that MLMs struggle to reason about physics in images by showing a gap between human and MLM performance. They further collect annotations to analyze why MLMs fail and identify perceptual errors and a lack of knowledge as the main reasons why they fail. Then, they incorporate some specialist vision models to infer depth, objects, etc., to further help the MLMs and show improved performance. They finally show that fine-tuning on their benchmark also helps an object manipulation task in simulation.

**Strengths:**

- The topic is very relevant and interesting since numerous works also show that MLMs lack spatial and physical reasoning and these are crucial skills for embodied AI.
- Their dataset seems varied and tests a lot of fundamental physical skills.
- The annotation of why MLMs are failing is interesting.
- The fact that PhysBench fine-tuning helps an object manipulation task is a good way to show that physical reasoning helps downstream applications.

**Weaknesses:**

- I am a bit confused about the correlation matrix in Figure 4a.  I am assuming the last 4 rows are PhysBench splits? In that case, the bottom right half seems to indicate that that performance on their benchmark doesn't correlate on the other existing VLM benchmarks (since the left part of the bottom 4 rows are blue). This can imply improving on their benchmark doesn't improve on other vision-language benchmarks? I am also not sure what part of that figure to look at for their claim in Section 3,4 that their benchmark is correlated to MMMU.

- More details on the PhysAgent would have been helpful. What is the base model for PhysAgent - is it an MLM? If so, which one? They outline a 3-step pipeline:
    1. asking task-specific prompts: How does it learn to ask these questions? Is it tuned to do so or in-context? If so, how is the tuning done?
    2. Incorporating VLM predictions: How exactly do they incorporate depth, grounding etc? The figure 8 and supplementary makes it seem like a  JSON-like format or a list. Where does the knowledge memory come from? Can the PhysAgent interpret such a format without any fine-tuning?
    3. How is PhysAgent prompted to do COT? Does it do it zero-shot or, is it taught to do the chain of thought using some training examples? If so, how are the COT training examples made?

- The results in Table 9 are confusing - what the individual columns? They do not match the names with the splits in PhysBench in Table 3. PhysAgent improves on PhysBench, but does it at least retain performance on some other similar benchmarks?

**Questions:**

See weaknesses.

I like the direction of the paper and believe the benchmark would be impactful. However, the details presented are not clear. Hence, I will be willing to increase my score if the authors can clarify my doubts above.

---

> ### Author Response · Authors · 2024-11-22
> **Official Response to Reviewer 52XN (1/2)**
>
> Overall, we are encouraged by the recognition and deeply appreciate your valuable suggestions. Moving forward, we will address each of your concerns point by point.
>
> > **Q1:** I am a bit confused about the correlation matrix in Figure 4a. I am assuming the last 4 rows are PhysBench splits? In that case, the bottom right half seems to indicate that that performance on their benchmark doesn't correlate on the other existing VLM benchmarks (since the left part of the bottom 4 rows are blue). This can imply improving on their benchmark doesn't improve on other vision-language benchmarks? I am also not sure what part of that figure to look at for their claim in Section 3,4 that their benchmark is correlated to MMMU.
> >
>
> **A1**: Thank you very much for your insightful question!
>
> **(1.1)** Your understanding is correct. The last four rows correspond to the PhysBench splits.  As highlighted in the results, PhysBench exhibits relatively low correlation with conventional VQA benchmarks, further emphasizing the **distinctiveness and uniqueness** of our task compared to traditional vision-language benchmarks.
>
> **(1.2)** While the large differences between individual tasks **do not directly lead to** the conclusion that *"improving on one benchmark does not enhance performance on other vision-language benchmarks,"* the experiments in *Section 3.4* provide critical insights. Specifically, increasing model size and data scale significantly improves performance on conventional VQA tasks (detailed results can be found in *Appendix F.6*). However, this trend does not extend to PhysBench. We attribute this discrepancy to the **lack of physical world-related knowledge** in the pretraining data, as identified in the error analysis in *Section 3.4*. We further validate this through fine-tuning VLMs on PhysBench's held-in data.
>
> **(1.3)** In relation to *Figure 4(a)*, the tasks most closely associated with MMMU are the Property" and Relationship subcategories of PhysBench. The task most similar to PhysBench is POPE, which is designed for hallucination detection, while also demonstrating that performance does not consistently improve with larger datasets or model scales. We have clarified this point in the revised PDF.
>
> In summary, the analysis presented in *Figure 4(a)* partially illustrates the **uniqueness of PhysBench** compared to conventional VQA tasks, shedding light on the distinctive challenges it poses for VLMs.
>
>
> > **Q2:** More details on the PhysAgent would have been helpful. What is the base model for PhysAgent - is it an MLM? If so, which one? They outline a 3-step pipeline:
> >
> > 1. asking task-specific prompts: How does it learn to ask these questions? Is it tuned to do so or in-context? If so, how is the tuning done?
> > 2. Incorporating VLM predictions: How exactly do they incorporate depth, grounding etc? The figure 8 and supplementary makes it seem like a JSON-like format or a list. Where does the knowledge memory come from? Can the PhysAgent interpret such a format without any fine-tuning?
> > 3. How is PhysAgent prompted to do COT? Does it do it zero-shot or, is it taught to do the chain of thought using some training examples? If so, how are the COT training examples made?
>
> **A2:** Thank you for your suggestion. PhysAgent is centered around a VLM. In our paper, we conducted experiments using the open-source Phi-3V and GPT-4o models.
>
> **(2.1)** We predefined task categories and required the VLMs to classify input questions and activate corresponding physical world knowledge to assist with **in-context question answering**. PhysAgent and tuning are not contradictory; PhysAgent essentially builds a series of processes based on a VLM at its core, making full use of prior physical knowledge and the perceptual capabilities of powerful foundational vision models. The lack of these two factors is the primary reason for poor performance in VLMs as dicussed in *Section 3.4*.
>
> **(2.2)** The three models are processed concurrently, with their outputs **converted into textual format and concatenated with the question as input to the LLMs**. This design facilitates the incorporation of advanced visual perception capabilities into VLMs, effectively addressing the key challenge discussed in *Section 3.4*. **Knowledge memory is predefined by us** and consists of two parts: one part contains general properties of objects, such as the high ductility of gold, while the other part includes physical inferences, such as the direction of shadow movement relative to the light source.
>
> **(2.3)** **The CoT in PhysAgent is zero-shot**. PhysAgent uses CoT in two parts: when providing an answer, PhysAgent is required to explain it step by step. After the answer is given, we prompt PhysAgent a second time to verify the reasonableness of its own answer.

---

> ### Author Response · Authors · 2024-11-22
> **Official Response to Reviewer 52XN (2/2)**
>
> > **Q3:** The results in Table 9 are confusing - what the individual columns? They do not match the names with the splits in PhysBench in Table 3. PhysAgent improves on PhysBench, but does it at least retain performance on some other similar benchmarks?
> >
>
> **A3:** Thank you for your suggestion!
>
> **(3.1)** The details for *Figure 9(a)* are as follows, and we have updated *Figure 9* included more information in the revised version's *Appendix F.8 of the PDF*. (We noticed that *Table 9* is the hyperparameter table, and we believe you may be referring to *Figure 9(a)*. Please let us know if you have any further questions.)
>
> |  | **number** | **mass** | **color** | **attribute** | **size** | **location** | **depth** | **distance** | **movement** |
> | --- | --- | --- | --- | --- | --- | --- | --- | --- | --- |
> | ContPhy | 62.22 | 47.10 | 62.67 | 48.33 | 67.86 | 59.79 | 67.72 | 70.00 | 46.06 |
> | Phi-3V | 43.15 | 37.88 | 63.00 | 40.81 | 66.07 | 53.61 | 74.39 | 48.33 | 26.69 |
> | + CoT | 44.89 | 36.18 | 60.33 | 39.11 | 39.29 | 48.45 | 60.35 | 50.00 | 26.42 |
> | + Desp-CoT | 49.22 | 29.01 | 67.00 | 37.62 | 66.07 | 53.61 | 62.46 | 55.00 | 24.69 |
> | + PLR | 43.67 | 26.62 | 59.67 | 34.63 | 58.93 | 41.92 | 30.53 | 43.33 | 30.43 |
> | + PhysAgent | 45.10 | 40.27 | 64.67 | 40.53 | 67.86 | 54.98 | 74.74 | 66.67 | 26.76 |
> | GPT-4o | 61.18 | 54.27 | 71.00 | 52.52 | 85.71 | 70.45 | 73.33 | 83.33 | 60.58 |
> | + CoT | 63.43 | 56.66 | 71.67 | 54.22 | 82.14 | 74.23 | 75.44 | 83.33 | 67.91 |
> | + Desp-CoT | 65.16 | 55.29 | 71.00 | 52.16 | 80.36 | 68.38 | 70.18 | 71.67 | 61.20 |
> | + PLR | 68.63 | 41.98 | 66.00 | 47.27 | 66.07 | 57.73 | 59.30 | 53.33 | 43.91 |
> | + PhysAgent | 67.07 | 54.27 | 73.33 | 55.86 | 87.50 | 78.35 | 74.04 | 95.00 | 71.99 |
>
> |  | **temperature** | **camera** | **gas** | **light** | **collision** | **throwing** | **manipulation** | **fluid** | **chemistry** | **others** |
> | --- | --- | --- | --- | --- | --- | --- | --- | --- | --- | --- |
> | ContPhy | 73.53 | 33.08 | 74.55 | 36.21 | 39.67 | 42.52 | 36.07 | 50.15 | 40.54 | 58.04 |
> | Phi-3V | 52.94 | 38.93 | 65.45 | 29.48 | 33.18 | 35.39 | 28.95 | 35.34 | 59.46 | 56.68 |
> | + CoT | 55.88 | 30.73 | 50.91 | 26.93 | 31.83 | 31.35 | 31.72 | 40.43 | 47.30 | 50.41 |
> | + Desp-CoT | 58.82 | 40.15 | 67.27 | 30.30 | 34.99 | 32.07 | 28.47 | 31.94 | 56.76 | 50.68 |
> | + PLR | 67.65 | 39.96 | 67.27 | 30.94 | 32.13 | 28.03 | 25.69 | 35.03 | 56.76 | 51.50 |
> | + PhysAgent | 72.06 | 38.93 | 70.91 | 34.58 | 36.05 | 35.39 | 29.55 | 35.49 | 67.57 | 65.67 |
> | GPT-4o | 91.18 | 22.05 | 83.64 | 32.67 | 43.74 | 46.32 | 35.22 | 39.20 | 62.16 | 86.92 |
> | + CoT | 91.18 | 30.73 | 74.55 | 36.49 | 45.40 | 46.32 | 35.71 | 41.67 | 64.86 | 85.56 |
> | + Desp-CoT | 85.29 | 29.41 | 80.00 | 39.95 | 44.19 | 42.28 | 30.52 | 43.83 | 70.27 | 85.83 |
> | + PLR | 88.24 | 30.73 | 89.09 | 34.94 | 38.01 | 34.68 | 24.85 | 37.50 | 66.22 | 79.56 |
> | + PhysAgent | 95.59 | 41.19 | 85.45 | 43.49 | 45.25 | 47.27 | 52.11 | 44.91 | 67.57 | 87.47 |
>
> **(3.2)** To illustrate the effectiveness of our dataset and methods, we selected three benchmarks relevant to physical world understanding: *ContPhy*, which focuses on fluid dynamics comprehension; *Physion++*, emphasizing collision prediction; and *EmbSpatial*, evaluating spatial reasoning in embodied environments. While these benchmarks target specific aspects of physical world understanding, our **PhysBench offers a more comprehensive evaluation for VLMs**, as summarized in *Table 1* of our paper.
>
> Our experiments demonstrate that both **fine-tuning with PhysBench data** and employing **PhysAgent in a zero-shot** setting result in consistent performance improvements across all benchmarks. Notably, on *Physion++*, we observe substantial gains of **19.50%** and **9.6%**, with fine-tuning yielding the most significant enhancements. These results underscore **the effectiveness of our dataset and approach in advancing the physical world understanding capabilities of VLMs**.
>
> *Experimental results and details are provided in Appendix F.7 of the PDF.*
>
> |  | **ContPhy-Property** | **ContPhy-Dynamics** | **Physion++** | **EmbSpatial** |
> | --- | --- | --- | --- | --- |
> | Phi-3V | 49.25 | 43.25 | 68.80 | 55.91 |
> | Phi-3V + finetune | 64.50 | 62.75 | 82.20 | 66.95 |
> | Phi-3V + PhysAgent | 52.00 | 44.75 | 78.40 | 62.28 |
>
>
> We hope this clarifies your concerns. We are committed to thoroughly incorporating your suggestions in the next version of the paper. Thank you once again for your excellent feedback.

---

> ### Author Response · Authors · 2024-11-24
> **Reminder: Discussion Phase Closing Soon**
>
> We would like to take this opportunity to once again express our gratitude for your valuable feedback, which has greatly contributed to improving our paper. As the rebuttal period is nearing its conclusion, we will greatly appreciate an increased score if you find we have addressed your concerns. Otherwise if you have any additional questions, please do not hesitate to let us know. We would be more than willing to provide further clarification. Thank you once again for helping us improve this paper.

---

> ### Comment · Reviewer_52XN · 2024-11-25
>
> I see, now the correlation matrix makes sense.
>
> One concern may be whether MLMs forget other pre-training commonsense (VQA, GQA etc) when finetuned on PhysBench. However, I believe this concern can be mitigated by large-scale mixed training. Reading the other reviews and responses, I also believe that a strong statement could have been showing the benefit of PhysBench fine-tuning when mixed in with other target robotic tasks or OpenVLA data. However, I agree with the authors that such a large-scale tuning is currently out of scope given the time limits and that can be future work.
>
> However, while tuning with robotic action tokens may be out of scope, it might be good to compare other kinds of action/spatial/physical QA data available to compare to. For instance compare PhysBench fine-tuning to finetuning with https://arxiv.org/abs/2309.02561, which also has physics information about objects. Do we still see improvements? I recommend authors to add some baselines of similar data fine-tuning to show their benefit.
>
> I believe the benchmark is still impactful to the community since multiple works show that VLMs lack spatial skills, and this paper focuses on a related impactful dimension of physics understanding, which will be important for several real-world tasks. Further, their correlation matrix shows that it is not yet another VLM dataset, but evaluates a unique perspective.
>
> Hence, considering the strengths of the benchmark, I increase my score to a 6.

---

> ### Author Response · Authors · 2024-11-27
>
> Thank you for your constructive comments!
>
> As the rebuttal period has been extended, we are working on additional experiments to address all your concerns. We will post the response as soon as possible. Thank you!

---

> ### Author Response · Authors · 2024-12-02
> **Official Response to Reviewer 52XN**
>
> Dear Reviewer,
>
> Thank you for your constructive feedback. We have conducted additional experiments based on your suggestion:
>
> > **Q1**: Reading the other reviews and responses, I also believe that a strong statement could have been showing the benefit of PhysBench fine-tuning when mixed in with other target robotic tasks or OpenVLA data. However, I agree with the authors that such a large-scale tuning is currently out of scope given the time limits and that can be future work.
> >
>
> **(1)** Due to the scale of the dataset and time constraints, we randomly selected 500 items from PhysBench, covering various attributes such as size, location, distance, color, mass, and number. Given the limited size of the demonstration action dataset, we supplemented it with a portion of the TableTop Frank Panda dataset from Open X-Embodiment, specifically the NYU Franka Play Dataset, along with data from Austin Sailor, CMU Play Fusion and Austin Sirius, focusing on data relevant to downstream tasks, in order to balance the dataset scale. We conducted the following experiments: *OpenVLA+PhysBench*: We first performed joint fine-tuning using the Frank Panda dataset from Open X-Embodiment and a subset of the PhysBench data, followed by fine-tuning on the target robotic data. *OpenVLA baseline*: We first fine-tuned using the Frank Panda dataset from Open X, then fine-tuned on the target robotic data.
>
> The experimental results suggest that using PhysBench during the first stage of fine-tuning improves performance, particularly in the Tool category.
>
> |  | Affordance | Force | Color | Location | Tool |
> | --- | --- | --- | --- | --- | --- |
> | OpenVLA baseline | 0.58±0.04 | 0.26±0.05 | 0.62±0.05 | 0.32±0.04 | 0.22±0.04 |
> | OpenVLA+PhysBench | 0.60±0.08 | 0.32±0.01 | 0.68±0.01 | 0.38±0.04 | 0.42±0.08 |
>
> > **Q2**: However, while tuning with robotic action tokens may be out of scope, it might be good to compare other kinds of action/spatial/physical QA data available to compare to. For instance compare PhysBench fine-tuning to finetuning with https://arxiv.org/abs/2309.02561, which also has physics information about objects. Do we still see improvements? I recommend authors to add some baselines of similar data fine-tuning to show their benefit.
> >
>
> **(2)** We ensured that the data volume for PhysObjects and PhysBench was comparable and fine-tuned MOKA accordingly. The results indicate that PhysObjects is also effective, particularly for Affordance and Force, though the improvement in Location was less significant. Overall, the gains were not as substantial as those achieved with PhysBench data.
>
> |  | Affordance | Force | Color | Location | Tool |
> | --- | --- | --- | --- | --- | --- |
> | MOKA | 0.56±0.05 | 0.20±0.07 | 0.70±0.07 | 0.66±0.05 | 0.36±0.08 |
> | MOKA + finetune using PhysBench | 0.86±0.09 | 0.70±0.04 | 0.76±0.05 | 0.78±0.08 | 0.72±0.11 |
> | MOKA + finetune using PhysObjects | 0.84±0.05 | 0.72±0.05 | 0.70±0.04 | 0.68±0.01 | 0.54±0.08 |
>
> Since the current version of the PDF is no longer editable, we will incorporate the experiments and details discussed in the next version. We hope this explanation helps alleviate any doubts you may have.
>
> Best regards,
>
> Authors

---

> > ### Comment · Reviewer_52XN · 2024-12-02
> >
> > I thank the authors for these additional experiments. This shows the benefit of their physics-based data. Hence, I will increase my score to recommend acceptance.

---

> ### Author Response · Authors · 2024-12-03
> **Official Comment by Authors**
>
> Dear Reviewer,
>
> We sincerely appreciate your valuable comment with our work! Your suggestions have significantly improved the quality of our research.
>
> Best regards,
> The Authors

---

### Official Review · Reviewer_khH4 · 2024-11-08

**Soundness:** 3
**Presentation:** 4
**Contribution:** 3
**Rating:** 8
**Confidence:** 4

**Summary:**

The paper introduces a comprehensive benchmark for evaluating VLMs for physical world understanding. The authors identify four major dimensions comprising of Object Property, Object Relationships, Scene Understanding, and Dynamics and create 100k entries covering these categories. Further, the paper introduces PhysAgent as a solution to aid VLMs in gaining physical world understanding which improves performance for Embodied AI applications as well.

**Strengths:**

- Paper identifies key issue of missing physical world understanding in VLMs and release a comprehensive benchmark for evaluation and training (Table-1).
- Fig.4 experiment explains the difference of physical world understanding against the common VQA benchmarks which is very insightful.
- The method PhysAgent reuses existing foundation models and gains a shows a significant performance boost on the benchmark.
- The cross-task ability of PhysBench for embodied AI makes the benchmark very useful.

**Weaknesses:**

- The test set of PhysBench is derived from the similar set of images as train. This ensures that any fine-tuning would result in performance improvement. Did the authors test PhysAgent against out of distribution images that are labeled for physical undersatnding?
- L443: 'PhysAgent first classifies the question and activaets task-specific prompts'. Does that mean that PhysAgent is dataset dependent? Also, how does it support the claim made at L90 "PhysAgent retrains the strong generalization abilities of VLMs and their capacity to solve open-ended problems"? It would be good to see an experiment supporting the L90 claim in the paper.
- Nit: The appendix should have been kept concise. I appreciate the details but a concise appendix makes it more approachable.

**Questions:**

- Is the PhysAgent using the DepthAnything, SAM and DINO outputs as input to LLMs or these individual model's outputs are used to get a textual insight which is the sole input to VLMs?
- Please refer to weakness section. I am open for discussion and changing my score.

**Details Of Ethics Concerns:**

Since, the images/videos in the paper are from internet. Please make sure that there are freely available for distribution and reuse or any copyright laws are not breached.

---

> ### Author Response · Authors · 2024-11-22
> **Official Response to Reviewer khH4**
>
> We sincerely appreciate your constructive and insightful comments. We will explain your concerns point by point.
>
> > **Q1:** The test set of PhysBench is derived from the similar set of images as train. This ensures that any fine-tuning would result in performance improvement. Did the authors test PhysAgent against out of distribution images that are labeled for physical undersatnding?
> >
>
> **A1**: Thank you for your insightful suggestions! To illustrate the effectiveness of our dataset and methods, we selected three benchmarks relevant to physical world understanding: *ContPhy*, which focuses on fluid dynamics comprehension; *Physion++*, emphasizing collision prediction; and *EmbSpatial*, evaluating spatial reasoning in embodied environments. Images in these benchmarks are not included in PhysBench, so they can be viewed as out-of-distribution images for evaluation.
>
> ***+Finetune*** indicates that we fine-tuned Phi-3V using the PhysBench dataset, while ***+PhysAgent*** refers to the zero-shot application of our proposed agent-based method to assist the VLM, without any fine-tuning. Experimental results demonstrate consistent performance improvements across all benchmarks under both settings. Notably, on Physion++, we observe significant gains of 19.50% and 9.6%, with fine-tuning achieving the most pronounced improvements. These findings highlight the **effectiveness of our dataset and approach in enhancing the physical world understanding capabilities of VLMs.**
>
> *Experimental results and details are provided in Appendix F.7 of the PDF.*
>
> |  | **ContPhy-Property** | **ContPhy-Dynamics** | **Physion++** | **EmbSpatial** |
> | --- | --- | --- | --- | --- |
> | Phi-3V | 49.25 | 43.25 | 68.80 | 55.91 |
> | Phi-3V + finetune | 64.50 | 62.75 | 82.20 | 66.95 |
> | Phi-3V + PhysAgent | 52.00 | 44.75 | 78.40 | 62.28 |
>
> > **Q2:** L443: 'PhysAgent first classifies the question and activates task-specific prompts'. Does that mean that PhysAgent is dataset dependent? Also, how does it support the claim made at L90 "PhysAgent retrains the strong generalization abilities of VLMs and their capacity to solve open-ended problems"? It would be good to see an experiment supporting the L90 claim in the paper
> >
>
> **A2:** Thank you for raising an important concern. We will address your question from the following three aspects:
>
> (2.1) **PhysAgent is dataset-independent**. We have defined distinct prompts and knowledge, with PhysAgent autonomously incorporating this information as context to assist VLMs in providing answers. For instance, in tasks involving the design of a light source, we prepend relevant optical knowledge to the query, such as "The movement direction of a shadow is opposite to the direction of the light source."
>
> **(2.2)** To further demonstrate the generalizability of PhysAgent across a broader range of tasks, we selected three benchmark datasets related to physical world understanding for additional experimentation. The experimental results are presented in ***A1**.*
>
> > **Q3:** Is the PhysAgent using the DepthAnything, SAM and DINO outputs as input to LLMs or these individual model's outputs are used to get a textual insight which is the sole input to VLMs?
> >
>
> **A3:** We sincerely appreciate your question. The three models are processed simultaneously, and their outputs are **transformed into text and concatenated with the question as input to the LLMs**. This design allows for the integration of advanced visual perception capabilities into VLMs, addressing the critical issue highlighted in *Section 3.4*.
>
> We commit to making the appendix more concise and adding clarifications to improve the readability of the text. We have already made some revisions to the original manuscript and will continue to refine it to enhance its readability.
>
> We hope the above discussion resolves your concerns. The discussion remains open, and we welcome your further review.

---

> ### Author Response · Authors · 2024-11-24
> **Reminder: Discussion Phase Closing Soon**
>
> As the discussion phase is coming to a close, we kindly ask if our responses have fully addressed your questions and concerns. We truly appreciate your time and effort in reviewing our work and look forward to receiving your valuable feedback.

---

> > ### Comment · Reviewer_khH4 · 2024-11-25
> >
> > Thank you authors for their detailed response to my questions. The authors have satisfied my queries, hence, I am increasing my score.

---

> > > ### Author Response · Authors · 2024-11-25
> > > **Official Comment by Authors**
> > >
> > > We sincerely thank you for your support of our work. Your valuable feedback has significantly contributed to improving its quality!

---

### Author Response · Authors · 2024-11-22
**General Response**

We deeply appreciate the insightful and valuable comments provided by all reviewers.

We are grateful for the reviewers' recognition that this paper introduces a highly significant benchmark and dataset for **comprehensively evaluating the understanding of the physical world by Vision-Language Models**. Furthermore, we **identify potential causes of performance limitations** and propose **a novel framework** to enhance physical understanding capabilities based on our analysis. Additionally, our work investigates the **applicability of our dataset and approach in robotic manipulation tasks**, contributing to advancements in machine intelligence’s comprehension of the physical world with meaningful implications for future research.

Overall, we are encouraged by the reviewers' positive feedback, which highlights:

- The motivation and novelty of PhysBench are well-defined, reasonable, and impactful (as noted by all reviewers).
- The challenges faced by existing VLMs are clearly articulated, and a comprehensive and well-structured benchmark is provided (as noted by all reviewers).
- The thorough experiments effectively demonstrate the gap between VLMs’ physical world understanding and their performance on general VQA benchmarks, pinpoint the causes of their limitations, and suggest actionable approaches for improvement (as noted by all reviewers).
- The exploration of performance shortcomings, scalability, and implications for embodied AI is compelling and has the potential to guide future research directions (highlighted by Reviewers 52XN, eQW6, and khH4).

To address the reviewers' concerns, we have conducted several additional experiments and analyses, including:

- Further validation of our proposed dataset and methods across three related benchmarks, as presented in *Appendix F.7*.
- New experiments evaluating various VLMs on manipulation tasks, with findings detailed in *Appendix F.8*.
- Polishing and revising sentences for improved clarity throughout the manuscript.

All revisions in the paper are marked in *blue font*. We sincerely appreciate the reviewers' constructive suggestions and remain committed to further improving our work.

Below, we address each reviewer's specific concerns in detail. We extend our heartfelt thanks to all reviewers for their recognition and valuable feedback. We welcome further discussion and look forward to continued engagement. Thank you!

---

### Meta-Review · Area_Chair_etLk · 2024-12-20

**Metareview:**

This paper proposes a novel benchmark for evaluating physical reasoning for vision-language models (VLMs).

All reviewers agree that this is a solid paper that identifies shortcomings in current VLMs and proposes a novel benchmark / evaluation setting to measure performance on physical world understanding. The proposed PhysAgent approach, a heuristic to enrich the prompt used in visual reasoning tasks (e.g. by utilising other vision models to extract objects and descriptions) is also of interest and demonstrates that performance on this visual reasoning benchmark can be consistently improved via enriched prompts.

Reviewers highlighted that this paper successfully addresses an undoubtedly important problem (advancing VLMs to improve real-world understanding) and commended the thoroughness of the investigation. Remaining concerns are related to the transferability to real-world (especially robotics) settings of the proposed benchmark and dataset, as it partially relies on synthetic data.

Overall, this is a great addition to the literature on VLM evaluation, deserving a spotlight at ICLR.

**Additional Comments On Reviewer Discussion:**

Most remaining concerns were successfully addressed in the rebuttal and reviewer consensus did not change during discussion. The reviewers and AC converged on a Spotlight recommendation.

---

### Decision · Program_Chairs · 2025-01-22

Accept (Oral)